# On the Power of Context-Enhanced Learning in LLMs

Xingyu Zhu [1][*]   Abhishek Panigrahi [1][*]   Sanjeev Arora [1]

## Abstract

We formalize a new concept for LLMs, **context-enhanced learning**. It involves standard gradient-based learning on text except that the context is enhanced with additional data on which no auto-regressive gradients are computed. This setting is a gradient-based analog of usual in-context learning (ICL) and appears in some recent works.

Using a multi-step reasoning task, we prove in a simplified setting that context-enhanced learning can be **exponentially more sample-efficient** than standard learning when the model is capable of ICL. At a mechanistic level, we find that the benefit of context-enhancement arises from a more accurate gradient learning signal. We also experimentally demonstrate that it appears hard to detect or recover learning materials that were used in the context during training. This may have implications for data security as well as copyright.

## 1. Introduction

Pre-trained LLMs (Brown et al., 2020; Touvron et al., 2023; Team et al., 2023) show strong capability to learn new material at inference time, for instance via in-context-learning (ICL). There is also emerging evidence that gradient-based learning on a piece of text (say, math Q&A) can be enhanced if additional helpful text is placed in the context, even though no auto-regressive loss is computed on this helpful text  (Liao et al., 2024; Zou et al., 2024; Choi et al., 2025).  Such strategies have also been shown to benefit pre-training as prepending source URLs to documents can enhance the model's training efficiency and memorization capacity (Allen-Zhu & Li, 2024; Gao et al., 2025).

In this paper we seek to formally study this phenomenon, whereby LLMs' gradient-based learning is *enhanced* via

placement of additional helpful material in the context, but without actual auto-regressive gradient updates on this material. We will call this form of learning *context-enhanced learning*. Because the material used for context-enhancement can evolve over the course of training, this approach naturally aligns with the idea of a *curriculum*.

Context-enhanced learning intuitively mirrors how humans learn: when solving problems, they refer to textbooks or demonstrations for guidance, yet they do not seek to memorize these resources *per se*. An analogous concept *Learning using Privileged Information* (or LUPI), has been well-studied in the context of kernel SVMs (Vapnik & Vashist, 2009) and classification models. Our work adapts this concept for LLMs, and surfaces the following questions:

**Q1:** Even though autoregressive loss is computed on the same set of tokens, can context-enhanced learning be significantly more powerful than usual auto-regressive learning that has no additional in-context materials? If so, can we theoretically characterize and understand the mechanism behind such improvement?

**Q2:** Do models need a certain capability level to benefit from context-enhanced learning? This is a natural question, since leveraging in-context information (e.g., ICL) likely requires a minimum capability level or model size (Brown et al., 2020; Wei et al., 2022).

**Q3:** Is context-enhanced learning a viable way to use privileged/private information during learning? Providing such privileged information in the context could conceivably enhance the model's learning, but since no auto-regressive gradient updates happen on the privileged/private information, there might be a lower risk of leakage of such information via API calls.

**Paper Overview:** Section 2.1 formally defines context-enhanced learning. To allow rigorous understanding of the power of context-enhanced learning, Section 2.2 introduces a multi-step reasoning task called *Multi-layer translation*. This is a synthetic setting involving $d + 1$ languages $L_1, L_2, \ldots, L_{d+1}$ over finite alphabets. For each $i$, there is a simple *phrasebook* that describes how to translate from $L_i$ to $L_{i+1}$, and the mapping from $L_1$ to $L_{d+1}$ is a sequential application of the set of phrasebooks.

The goal is to learn how to translate text from $L_1$ into $L_{d+1}$

---

[*]Equal contribution with theoretical framework and setting up the problem, XZ undertook most of the experiments. [1]Princeton Language and Intelligence, Princeton University. Correspondence to:  <{xingyu.zhu, ap34}@princeton.edu>.

without explicitly writing down intermediate steps. The learner is provided excerpts from these phrasebooks as helpful information in the context during training, but allowed no auto-regressive gradient updates on these tokens.

If we train with auto-regressive loss on translation output conditioning on the phrasebooks' excerpts and the input, a model with certain ICL capacity level may quickly learn the translation task by leveraging the in-context phrasebooks. However, this learning could be brittle, that the model becomes reliant on having the phrasebooks' excerpts in context. This reliance can be weaned off by use of probabilistic *dropout* on phrasebooks tokens in context. Intuitively, this curriculum forces the model to not only read phrasebooks' excerpts, but also gradually internalize the phrasebooks' contents. Over time, the model's ability to translate from $L_1$ to $L_{d+1}$ will become robust to the dropout of phrasebooks' excerpts, and eventually, their complete removal.

Experiments show that this training strategy indeed works when the learner is a pre-trained LLM that is capable of ICL (but fails when LLM is incapable of ICL). Even when training with $20\%$ dropout rate, the model can perfectly translate strings from $L_1$ to $L_{d+1}$ without any phrasebooks' excerpts at test time. The rest of the paper is structured as follows:

• Section 3 details our experiments and the findings sketched above. Experiments show that an ICL-capable model follows an intuitive sequential processing of the phrasebooks provided in-context, whereby transformer layers approach stages of translation in an intuitive way; e.g., $L_3 \rightarrow L_4$ is done after $L_2 \rightarrow L_3$ (Section 3.3).

• Section 4, shows that after context-enhanced learning, the output probabilities of the model reveal little about the phrasebooks rules that were seen during training.

• In Section 5, we propose a theoretical framework using a surrogate/simplified model that represents an ideal LLM for the translation task (Section 5.1). This framework shows an exponential gap in sample complexity depending on whether the model is trained with or without in-context information on phrasebooks (Sections 5.2 and 5.3). Experiments reveal that the mechanism behind the increased sample efficiency of context-enhanced learning is an improved gradient signal, measured by gradient prediction accuracy (Section 5.4).

## 2. Setup

### 2.1. Context-Enhanced Learning

Let $X$ be the space of all possible text strings and let $\mathcal{Y}$ be the space of all possible distributions over texts. Let $g$ be a language task mapping inputs $x \in X_g \subset X$ to a distribution $Y \in \mathcal{Y}$. Let $f_\theta : X \rightarrow \mathcal{Y}$ be a general auto-regressive language model. We characterize $f_\theta$'s capability on task $g$ as follows:

---

**Definition 2.1** ($g$-capable model, informal). A language model $f_\theta$ is $g$-capable for a language task $g$ if $f_\theta$ is close to $g$, as measured by a suitable metric on $X_g$.

---

Vanilla supervised fine-tuning (SFT) aims to create a $g$-capable model by minimizing auto-regressive loss $\ell_{\text{auto}}$ on a supervised dataset $D_g = \{(x_i, y_i)\}_{i=1}^N$, where the label $y_i$ for each $x_i \in X_g$ is sampled from $g(x_i)$.

*Context-enhanced learning* involves augmenting the supervision with additional curriculum-text that depends on the task $g$, input $x$, and training step $t$. We denote curriculum-text as $\text{CURR}_g(x, t)$, which could be anything (helpful explanations, excerpts from textbooks, worked-out examples, etc.).

---

**Algorithm 1 Context-Enhanced Learning**
In contrast to standard SFT, it relies on curriculum-text in context on which **no auto-regressive loss is computed**.

**Input:** Supervised dataset $D_g$, curriculum-text $\text{CURR}_g$, initialization $\theta$, total steps $T$
**for** $t = 1$ **to** $T$ **do**
 Sample $(x, y) \sim D_g$
 Compute loss $l \leftarrow \ell_{\text{auto}}\left(f_\theta([\text{CURR}_g(x, t), x, y]), y\right)$.
 Update parameters $\theta$ with gradient $\nabla_\theta l$
**end for**
Return $\theta$

---

On sample $(x, y)$ drawn from the supervised dataset, we use auto-regressive loss for model's prediction on $y$ conditioned on $[\text{CURR}_g(x, t), x]$ to train our models. Note that *no loss is computed for curriculum-text tokens*. We denote this loss as $\ell_{\text{auto}}\left(f_\theta([\text{CURR}_g(x, t), x, y]), y\right)$.

### 2.2. Multi-level Translation (MLT)

To study the power of context-enhanced learning, we introduce a multi-step translation task that is easy to learn with a straightforward curriculum in the context, but very difficult to learn with just input-output examples.

The task is inspired by encryption methods[1] such as the Feistel cipher (Knudsen, 1993). The multi-level translation (**MLT**) task involves a bijective mapping from strings to strings that is a composition of $2d$ simpler bijections, each involving simple shift by $1$, or transforming bigrams (2-tuples of characters) via a bijection. The depth-$d$ translation can be described by $O(d)$ bits. But we will show that learning the task only from input-output pairs would require $e^{\Omega(d)}$ sample complexity in the SQ-learning framework (Kearns, 1998) (see Theorem 5.4).

---

[1]Note that there is no proof that current cryptographic tasks are fundamentally difficult for computers. Here the task makes sense as an example of multi-step reasoning and we are interested in sample-complexity lower bounds.

*Table 1.* Important notations for defining **MLT**

| | |
|---|---|
| $d$ | Depth of translation task |
| $n$ | Number of characters in each alphabet |
| $A$ | An alphabet set |
| $\pi$ | A phrasebook between 2-tuples in two alphabets |
| $\mathbf{\Pi}$ | A set of phrasebooks $\{\pi_i\}$ defining a translation task |
| $\mathbf{MLT_\Pi}$ | Translation task with a set of phrasebooks $\mathbf{\Pi}$ |
| $\mathbf{MLT}(d, n)$ | Family of translation tasks of depth $d$ and $n$ characters |

Concretely, let $A_1, \ldots, A_{d+1}$ be $d+1$ alphabets all of the same size with $n$ characters. For every consecutive pair of alphabets $A_i$ and $A_{i+1}$, we fix a *phrasebook* $\pi_i : A_i^2 \to A_{i+1}^2$ as a bijective mapping from 2-tuples in $A_i$ to 2-tuples in $A_{i+1}$. Each phrasebook $\pi_i$ can be represented by a binary stochastic matrix $\mathtt{Matrix}(\pi_i)$ with rules represented as one-hot columns (see Definition G.4).

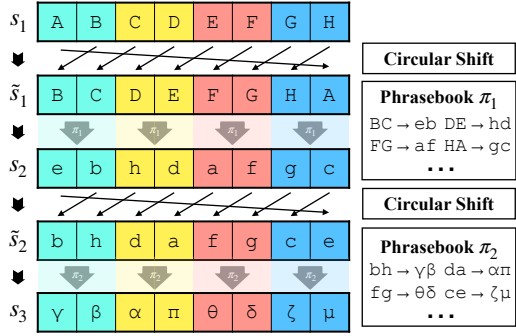

*Figure 1.* Illustration of an $\mathbf{MLT}(2, 8)$ instance with input sequence of 8 tokens. Input sequence $s_1$ went through 2 translation steps to $s_3$. Each output character depends on 4 input characters, for example, character $\mu$ is derived from $\mathtt{c}, \mathtt{e}$ in $s_2$, which, in turn, are computed from $\mathtt{A}, \mathtt{B}, \mathtt{C}, \mathtt{H}$ in $s_1$.

The input of the translation process is an even-length sequence, referred to as $s_1 \in A_1^L$ where $L$ is the sequence length. The translation process modifies $s_1$ recursively. For every $i \in [d]$, $s_i \in A_i^L$ will be transformed to $s_{i+1} \in A_{i+1}^L$ using *phrasebook* $\pi_i$ through the following 2 sub-processes:

• *Circular shift*: The characters in $s_i \in A_i^L$ are shifted by 1 character leftward (and wrapped around to the end if necessary) to give sequence $\tilde{s}_i \in A_i^L$. Formally, for each $j \in [1, L]$ we have $\tilde{s}_{i,j} = s_{i,(j+1)\%L}$.

• *Translate*: Using the *phrasebook* $\pi_i : A_i^2 \to A_{i+1}^2$, we translate 2-tuples (bigrams) of consecutive characters in sequence $\tilde{s}_i$ to create $s_{i+1}$. That is, for every odd $j \in [1, L]$, $(s_{i+1,j}, s_{i+1,j+1}) = \pi_i(\tilde{s}_{i,j}, \tilde{s}_{i,j+1})$.

We denote the mapping from $s_i$ to $s_{i+1}$ as $s_{i+1} = T_{\pi_i}(s_i)$. The $d$-step translation is defined as the composition $s_{d+1} = T_{\pi_d} \circ T_{\pi_{d-1}} \circ \cdots \circ T_{\pi_1}(s_1)$ which converts the input sequence $s_1$ to $s_{d+1}$ through $d$ translation steps. Please see Figure 1 for a visual illustration with $d = 2, n = 8$.

We denote $\mathbf{\Pi} = \{\pi_i\}_{i=1}^d$ the set of phrasebooks of all levels, and denote $\mathbf{MLT_\Pi} : s_1 \mapsto \mathbf{MLT_\Pi}(s_1) := T_{\pi_d} \circ \cdots \circ T_{\pi_1}(s_1)$ the mapping from input $s_1$ to output $s_{d+1}$ using $\mathbf{\Pi}$. We use $\mathbf{MLT}(d, n)$ to refer to the family of translation tasks involving $d$ step and $n$ characters in each alphabet.

We note that $\mathbf{MLT}(d, n)$ has two key properties: **1.** Once the phrasebooks are fixed, the translation task defines a bijection between input and output strings since *Circular shift* and *Translate* are invertible (see Lemma E.1). **2.** Each character in the output string depends on $2d$ characters from the input text string (see caption of Figure 1), making learning from input-output pairs very difficult (Theorem 5.4).

For each phrasebook $\pi_i$, we compose its textual representation $\mathtt{STR}(\pi_i)$ to be of the form `... a b -> C D; e d -> B A; ...` which lists (insensitive of ordering) phrasebook rules between 2-tuples in the previous alphabet and the next alphabet. Moreover, we denote the concatenation $[\mathtt{STR}(\pi_1), \ldots, \mathtt{STR}(\pi_d)]$ as $\mathtt{STR}(\mathbf{\Pi})$, and will be used to define curriculum-text.

### 2.3. Needed: Curriculum without Explicit CoT

To teach the model a particular translation task $\mathbf{MLT_\Pi}$ from input-output pairs of the form $(s_1, \mathbf{MLT_\Pi}(s_1))$, we can train it with relevant sections of the phrasebooks $\mathtt{STR}(\mathbf{\Pi})$ in context as curriculum, but at test time it would not have access to the phrasebook so it is important not to teach it explicit chain-of-thought (CoT) containing in-context information. (Another consideration is data privacy, with the phrasebook being considered privileged information.) However, a dual use of CoT is to provide the model extra compute at inference time (Goyal et al., 2023), which is needed here since the translation task has $d$ stages. To facilitate such silent computation we teach the model to output a fixed number of `<THINK>` tokens, sometimes refered to as *silent CoT* or *internalized CoT*.

### 2.4. ICL-capablity for $\mathbf{MLT}(d, n)$

To learn from books, one needs to know how to read. The analogous notion under study here is whether context-enhanced learning requires capability to sort-of "understand" the in-context material (**Q2**). In the context of **MLT**, we formalize such capability as being able to achieve low loss on the translation task when provided with the relevant phrasebook sections in context while allowing silent CoT.

**Definition 2.2** ($\mathbf{MLT}(d, n)$-ICL-capability, informal)**.** A language model $f_\theta$ is $\mathbf{MLT}(d, n)$-*ICL-capable* if for any set of phrasebooks $\mathbf{\Pi}$ in $\mathbf{MLT}(d, n)$, $f_\theta([\mathtt{STR}(\mathbf{\Pi}), s_1])$ is close to $s_{d+1} = \mathbf{MLT_\Pi}(s_1)$ disregarding the `<THINK>` tokens, when measured by a discrepancy metric over all valid input strings $s_1$.

# 3. Experiments and Observations

In this section, we fix a set of phrasebooks $\mathbf{\Pi}^*$ and study context-enhanced learning on $\mathbf{MLT}_{\mathbf{\Pi}^*}$.

We first introduce the preparation of an $\mathbf{MLT}(d, n)$-ICL-capable model. We then introduce a context-enhanced learning curriculum involving random dropping of phrasebook rules in context. We then present empirical evidence for significant sample efficiency of context-enhanced learning. We conclude the section with mechanistic insights into context-enhanced learning concerning internal representations and evolution of parameters.

We use the **Llama 3.2-3B instruction-tuned model** (Dubey et al., 2024) as the base model and fix $d = 5$ with $n = 8$ or 10. Detailed configurations are available in Appendix B.5.

## 3.1. Experimental Setup

**(i) Preparing an MLT**$(d, n)$**-ICL-Capable Model:**
The Llama 3.2B model is $\mathbf{MLT}(d, n)$-ICL-capable as it has not seen the task during training. To make it ICL-capable for our purpose, we use SFT on other random translation tasks with random phrasebooks $\mathbf{\Pi}_1, \ldots, \mathbf{\Pi}_M$, following common CoT internalization pipeline (Deng et al., 2024; Pfau et al., 2024; Hao et al., 2024). We use one training example per set of phrasebooks to prevent memorization of specific phrasebooks. At the end of training, given input $[\text{STR}(\mathbf{\Pi}), s_1]$ for any $s_1$ and $\mathbf{\Pi}$ the model can generate $\texttt{<THINK>}, \ldots, \mathbf{MLT}_{\mathbf{\Pi}}(s_1)$ correctly. Details on the first stage of training are described in Appendix B.4.

**(ii) Setting up context-enhanced learning for MLT**$_{\mathbf{\Pi}^*}$**:**
We use the $\mathbf{MLT}(d, n)$-ICL-capable model above as initialization and train for $\mathbf{MLT}_{\mathbf{\Pi}^*}$. Supervised dataset $D_{\mathbf{\Pi}^*}$ is curated with input-label pairs of the form $(s_1, [\texttt{<THINK>}, \ldots, \mathbf{MLT}_{\mathbf{\Pi}^*}(s_1)])$, where $s_1$ is a random string sampled from $A_1$, with length between 20 and 40.

We define curriculum-text $\text{CURR}_{\mathbf{\Pi}^*}(s_1, t)$ using excerpts from phrasebooks $\text{STR}(\mathbf{\Pi}^*)$ (selected based on $s_1$) with random dropout of rules (parameterized by training step $t$). We explore the following curriculum and study their impact:

• **No Context** (vanilla SFT): Empty curriculum-text.

• **Fixed Dropout**: A simple strategy independent of step $t$; given $s_1$, only curate rules in $\mathbf{\Pi}^*$ used in the translation of $s_1$, then randomly drop 20% of the curated rules.

• **Annealing Dropout**: A better strategy: for $s_1$, select the necessary rules from $\mathbf{\Pi}^*$ plus 25% unused rules. Apply random dropout on these rules, increasing linearly from 0% to 100% over the first 60% of training, then maintain 100%.

• **No Dropout** (ablation): Given $s_1$, always provide all rules in $\mathbf{\Pi}^*$ used in the translation of $s_1$ in curriculum-text.

• **Wrong Context** (ablation): Equivalent to **Annealing Dropout** but the rules in the curriculum are incorrect.

## 3.2. Experiment Results

To check the sample efficiency benefit of context-enhanced learning (**Q1**), we construct supervised datasets $D_{\mathbf{\Pi}^*}$ with $10^4$ to $10^6$ unique samples and train the models for one epoch on each.[2] We report the next-token prediction accuracy on the final answer tokens (ignoring thought tokens) for held-out samples when conditioning on no curriculum-text (100% dropped-out) and compare against the supervised dataset size. To check the necessity of proper ICL capability (**Q2**), we ablate with **Annealing Dropout** but starting from non-$\mathbf{MLT}(d, n)$-ICL-Capable 3B base model.

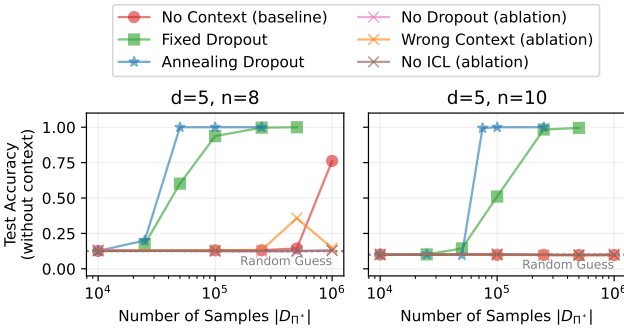

*Figure 2.* Context-enhanced learning of $\mathbf{MLT}_{\mathbf{\Pi}^*}$ when $n = 8$ and $n = 10$. **Annealing Dropout** learns the fastest followed by **Fixed Dropout**, requiring 10x less samples compared to vanilla SFT. Ablations show the necessity of correct context, proper dropout, and sufficient ICL capability as a good initialization. Note that there are multiple ablations overlapping around random guesses.

Figure 2 demonstrates the significant sample efficiency of context-enhanced learning. Moreover, models trained with subsets of phrasebooks and just 20% dropout give perfect heldout test-time accuracy with 100% dropout rate. Thus they are able to effectively use phrasebook rules from subsets of the phrasebook that did not co-occur in the same training sample. Clearly, the model has learned the phrasebook atomically, and can combine the rules as needed at test time.

In an ablation (Appendix C), we show that context-enhanced learning only internalizes the rules whose dropout from curriculum-text leads to an increase in loss on training data.

The experiment results can be summarized as follows:

> **(i)** Context-enhanced learning from an ICL-capable model greatly improves training sample efficiency.
> **(ii)** The phrasebook rules are internalized atomically, and only when missing them incurs an increased loss.

---

[2] We fix 1 epoch for fair comparison on sample efficiency.

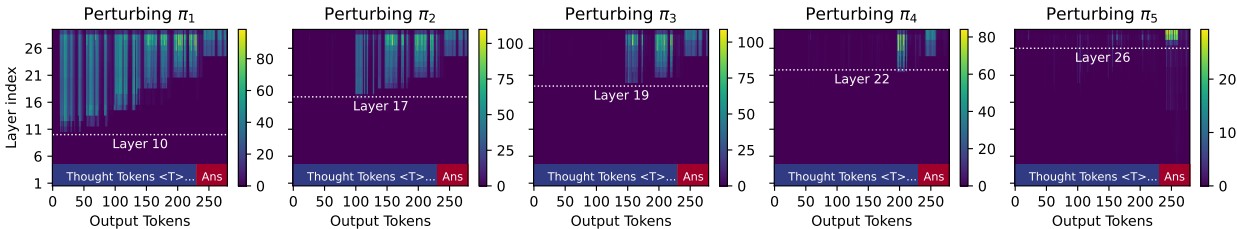

**Figure 3.** Evidence for sequential processing in a **MLT**(5, 8)-ICL-capable model. Each entry is the $l_2$ norm between the latent representation pre and post-perturbing $\pi_i$ (after certain layer at certain token position). Perturbing later phrasebooks in the context changes representations in the later layers, suggesting that later translation steps happens in later layers. To rule out the possibility that the affected depth is only dependent on position of the perturbation instead of the semantic content, we conduct the same set of experiment except that we only perturb rules that are *not* used when translation $s_1$, which yields negligible representation difference (see comparison in Figure 8).

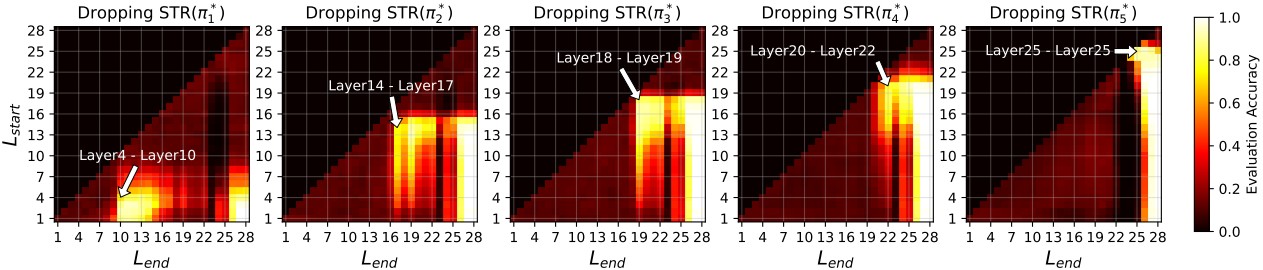

**Figure 4.** Starting from the **MLT**(5, 8)-ICL-capable model $f_\theta$ evaluated in Figure 3, we fix a set of phrasebooks $\Pi^*$, train with **Annealing Dropout** curriculum on 100k samples, and obtain an **MLT$_{\Pi^*}$**-capable model $f_{\theta^*}$. We construct "stitched models" by selectively substituting layers from layer $L_{\text{start}}$ to layer $L_{\text{end}}$ in $f_\theta$ by $f_{\theta^*}$ and evaluate on **MLT$_{\Pi^*}$** with certain phrasebooks dropped from context. Bright colors close to the diagonal reflects that knowledge required to compensate the dropped phrasebook in context can be localized in a small subset of layers in $f_{\theta^*}$. It is particularly worth noting that for any level $i$, the end of the group of layers responsible for storing $\pi_i^*$ matches the start of the layers responsible for reading $\pi_i$ (see Figure 3) before context-enhanced learning.

### 3.3. Mechanistic Insights: Layer by Layer Mapping of the Translation Task

To understand the behavior of context-enhanced learning, we probe into the hidden representations and weights of models before and after context-enhanced learning.

**Sequential Processing in ICL-capable model**

First, we look into how phrasebooks in curriculum-text are used by an **MLT**$(d, n)$-ICL-capable model during the translation task via the following controlled experiment. Fix a random set of phrasebooks $\Pi = \{\pi_1, \ldots, \pi_d\}$ and an input sequence $s_1$, we pass $[\text{STR}(\Pi), s_1, \texttt{<THINK>}, \ldots, s_{d+1}]$ into the ICL-capable model, and record the model's hidden representations (output of each transformer block) for the tokens $\texttt{<THINK>}, \ldots, s_{d+1}$.

Next, we apply independent perturbations to each phrasebook $\pi_i$ in the curriculum-text and measure the change in the model's representations. That is, for each level $1 \leq i \leq d$, we randomly sample 10 rules in $\pi_i$ used in the translation of $s_1$ and replace their image by other randomly chosen tuples. We denote the perturbed phrasebook $\text{STR}(\hat{\pi}_i)$, and compute the norm difference between the model's representations at tokens $\texttt{<THINK>}, \ldots, s_{d+1}$ before and after replacing

$\text{STR}(\pi_i)$ with $\text{STR}(\hat{\pi}_i)$. The first layer showing a significant difference in its output representation is identified as the layer where the model begins processing the phrasebook.

In Figure 3, we show that an ICL-capable model processes phrasebooks in curriculum-text sequentially, with earlier phrasebooks read by earlier layers. Formal details are in Appendix B.2 and additional experiments are in Appendix C.2.

**Localized Storage after Context-Enhanced Learning**

Here, we verify whether a similar sequential pattern for internalizing phrasebooks is used in a **MLT$_{\Pi^*}$**-capable model. We denote the ICL-capable model as $f_\theta$ and its post context-enhanced learning counterpart as $f_{\theta^*}$. As $f_{\theta^*}$'s capability no longer depends on textual representation of the phrasebooks, our analysis focuses on the model's parameters.

We assess the importance of each layer in model $f_{\theta^*}$ for each phrasebook by constructing "stitched" models and measuring their output behavior. For every $1 \leq L_{\text{start}} \leq L_{\text{end}} \leq 28$ (total layers in Llama3.2 3B), we replace layers $L_{\text{start}}$ to $L_{\text{end}}$ of the ICL model $f_\theta$ with corresponding layers from $f_{\theta^*}$ to create a stitched model. This strategy has been used in prior works on localizing information after SFT (Gong et al., 2022; Panigrahi et al., 2023; Wei et al., 2024).

For each level $1 \leq i \leq d$, we evaluate the stitched models on $\textbf{MLT}_{\Pi^*}$ using the following in-context information: $[\text{STR}(\pi_1^*), \cdots, \text{STR}(\pi_{i-1}^*), \text{STR}(\pi_{i+1}^*), \cdots, \text{STR}(\pi_d^*)]$. Note here the textual representation of $\pi_i^*$ has been dropped. If a stitched model shows high accuracy on inputs that use $\pi_i^*$ for translation but contain only the above in-context information, then the layers selected from $f_{\theta^*}$ to create the stitched model can be deemed responsible for storing information on $\pi_i^*$ in their parameter space.

Figure 4 demonstrates that information of all phrasebooks can be localized to a few mutually disjoint layers in $f_{\theta^*}$. Moreover, the end of the group of layers where $\pi_i^*$ is localized in $f_{\theta^*}$ marks the start of the layers that begin processing level $i$ phrasebook in the ICL-capable model $f_\theta$ (e.g. compare the role of layer 17 in Figures 3 and 4). Formal details and additional experiments are in Appendices B.3 and C.2.

This suggests that instead of storing phrasebooks as a single chunk in its parameters, the model re-learns each translation step locally to compensate for missing information when rules are dropped. Thus, we conjecture that context-enhanced learning leverages curriculum-text to improve training by localizing learning in the parameter space. We build a surrogate model in Section 5 to show how such localized learning can prove beneficial for faster training.

## 4. Curriculum-text: Detectable from Queries?

Context-enhanced learning for MLT uses the phrasebooks in its curriculum-text. Can rules of the phrasebooks be recovered post-hoc querying the model (**Q3**) using likelihood-based methods for detecting training data?

We take an $\textbf{MLT}_{\Pi^*}$-capable model (with $d = 5, n = 10$) trained with context-enhanced learning from Section 3.2. Let $\text{STR}(\Pi^*)$ denote the concatenation of all phrasebooks. For each rule in $\text{STR}(\Pi^*)$, which are of the form "`a b -> C D`"[3], we measure the probability of the model generating the ground truth tokens "`C D`" after observing "`a b ->`". We compute (1) the fraction of cases where the model's top-1 prediction matches the ground truth (greedy decoding) and (2) the probability of generating ground truth tokens via random sampling (with temperature set as 1). These are standard tests on textual description, and we would expect high probabilities for the correct tokens.

We also test two adversarial strategies with additional *token filtering*[4] in generation: (i) setting probability of `<THINK>` tokens to zero (ii) when querying for a rule in $\pi_i$ with output alphabet $A_{i+1}$, setting probability of all tokens outside $A_{i+1}$ to zero[5]. Note that this corresponds

---
[3]`a,b,C,D` are generic tokens for presentation

[4]See detailed characterization in Appendix C.3.

[5]We did not explore the full spectrum of attacks (e.g. adversarial prompt engineering) and leave that as interesting future work.

*Table 2.* Recovery Success Rate (Rounded to 2 decimals)

| Token Filter | Greedy Decoding | | Sampling | |
|---|---|---|---|---|
| | $\pi_1^* - \pi_4^*$ | $\pi_5^*$ | $\pi_1^* - \pi_4^*$ | $\pi_5^*$ |
| None | 0.00% | 0.20% | 0.00% | 0.89% |
| No `<THINK>` | 0.00% | 0.20% | 0.00% | 0.90% |
| Only from $A_{i+1}$ | 1.66% | 0.20% | 1.28% | 0.94% |

to a strong adversary which knows the alphabet set of the intermediate phrasebooks, but not the rules.

As shown in Table 2, recovering rules from intermediate phrasebooks $(\pi_1, \ldots, \pi_4)$ is nearly impossible without token filtering. For the final phrasebook $(\pi_5)$, results remain near-random, where random guess probability is $1\%$ for a 2-tuple when $n = 10$. Even with filtering (ii), recovery success rates remain only slightly above random. Setup details and more results are available in Appendix C.3.

## 5. Mathematical Analysis

Having empirically demonstrated the sample efficiency benefits of context-enhanced learning, we now formalize them through a mathematical lens. We first define the sample complexity of an algorithm for learning the task.

**Definition 5.1** (informal). Sample complexity for an algorithm to learn $\textbf{MLT}_\Pi$ is defined as the minimum total length of all (possibly repeated) input sequences required by the algorithm to return an $\textbf{MLT}_\Pi$-capable model $f_\theta$.

Analysis of gradient-based learning on a multilayer transformer (let alone a 28-layer model like Llama 3.2-3B) is an open mathematical question. We use our mechanistic findings (i.e., translation layers map onto transformer layers) to propose a surrogate model to think about how transformers learn $\textbf{MLT}(d, n)$. In this surrogate model we demonstrate that learning a task $\textbf{MLT}_\Pi$ via vanilla SFT will require $n^{\Omega(d)}$ samples. Then, we prove that, with context-enhanced learning, the surrogate model can learn $\textbf{MLT}_{\Pi^*}$ with a sample complexity of $\mathcal{O}(\text{poly}(n)d \log d)$.

### 5.1. Surrogate Model (SURR-MLT)

Formalization of the surrogate model, in short SURR-MLT, relies on observations from Section 3.3, which reveal that an ICL-capable model (Definition 2.2) performs the translation task step-by-step, with earlier phrasebook processed by lower layers. This aligns with how the model stores internalized knowledge in layers after context-enhanced learning. Our SURR-MLT represents an idealized and simplified transformer that has already been "pre-conditioned" to solve $\textbf{MLT}$ in this sequential fashion. Using SURR-MLT, we will show the benefits of context-enhanced learning.

Without loss of generality we assume the alphabet sets as $A_1, \ldots, A_{d+1} := A = \{1, 2, \ldots, n\}$. SURR-MLT will represent a length-$L$ sequence $s_i = (s_{i,1}, \ldots, s_{i,L})$ as an embedding matrix $V_i \in \mathbb{R}^{n^2 \times L/2}$ that uses $\{v(s_{i,1}, s_{i,2}), \cdots, v(s_{i,L-1}, s_{i,L})\}$ as columns. Here, for any 2-tuple $(a, b)$, $v(a, b) \in \mathbb{R}^{n^2}$ represents a one-hot vector with 1 at dimension $an + b$. SURR-MLT operates on embedding matrices, transforming $V_1 \to V_2 \to \cdots \to V_{d+1}$. Each layer $i$ will be primarily defined by two matrices, $C_i, W_i \in \mathbb{R}^{n^2 \times n^2}$. $C_i$ presents a (possibly partial or completely dropped) phrasebook that is provided in-context, and $W_i$ is a trainable parameter storing phrasebook information during context-enhanced learning[6].

---

**Definition 5.2.** SURR-MLT with trainable parameters $\{W_i\}_{i=1}^d$ and in-context representation $\{C_i\}_{i=1}^d$, is represented by its operation on an input $s_1$ as

$$V_{d+1} = \text{SURR-MLT}_{\{W_i\}_{i=1}^d}\left(\{C_i\}_{i=1}^d, V_1\right), \quad \text{where}$$

$$V_{i+1} = \text{HardMax}(C_i + W_i)\,\text{Shift}(V_i), \quad \text{for } i \geq 1,$$

and $V_1$ is the embedding matrix for the input string $s_1$.

---

Here `Shift` represents *Circular shift* operation and is defined as a Hadamard product on the embedding matrices (details in Definition G.2). `HardMax` represents hard-max function converting $C_i + W_i$ to a binary column stochastic matrix. In the following discussion, we show 2 examples where the surrogate model can perfectly represent an ICL-capable model and $\mathbf{MLT_{\Pi^*}}$-capable model.

**Case 1 (Representing $\mathbf{MLT}(d, n)$-ICL-capable):** The trainable matrices are all 0's as the model hasn't undergone context-enhanced learning. To produce the output for a task $\mathbf{MLT_\Pi}$, SURR-MLT takes phrasebooks into in-context representations by setting each $C_i$ as a stochastic matrix `Matrix`$(\pi_i)$, which represents rules of $\pi_i$ as one-hot columns (please see Definition G.4 and Lemma G.5).

**Case 2 (Representing $\mathbf{MLT_{\Pi^*}}$-capable):** For a model that has performed context-enhanced learning on a translation task $\mathbf{MLT_{\Pi^*}}$, no in-context information will be provided to the surrogate model and so $\{C_i\}_{i=1}^d$ will be all 0s. A $\mathbf{MLT_{\Pi^*}}$-capable-model should contain the phrasebooks as $\{\text{Matrix}(\pi_i^*)\}_{i=1}^d$ in its trainable parameters $\{W_i\}_{i=1}^d$.

**SURR-MLT as an Ideal Transformer for MLT:** The following theorem constructs a transformer that can simulate SURR-MLT. Thus, while our discussions and proofs focus on the surrogate model for simplicity, they remain fully applicable to the transformer architecture.

---

[6] For simplicity of presentation $C_i$'s are provided directly to the corresponding layers. We note that SURR-MLT can be exactly reparameterized that $C_i$'s are provided in context (i.e. concatenated with $V_1$). Please see Definition G.8 for more discussions.

**Theorem 5.3** (cf Lemma H.4). *There exists a transformer that can simulate SURR-MLT with $2d$ self-attention and $2d$ MLP layers with embedding dimension $2n^2 + 2d + 4$.*

### 5.2. Sample Complexity for Vanilla SFT

For the surrogate model, vanilla SFT corresponds to always setting in-context representations $\{C_i\}_{i=1}^d$ to 0s when training for $\mathbf{MLT_{\Pi^*}}$. While we use the surrogate model for consistency in our discussion, the argument generalizes to any model learning $\mathbf{MLT_{\Pi^*}}$ with vanilla SFT.

Our analysis is built on the Statistical Query (SQ) framework (Kearns, 1998), which measures the difficulty of learning tasks using algorithms that rely on expectation estimates of specific functions to approximate the true solution. Gradient-based methods, such as Stochastic Gradient Descent (SGD), fall under this framework as they compute gradients by estimating expectations of loss functions and their derivatives.

The complexity of learning a task is quantified by the SQ dimension, which measures the number of candidate functions that are pairwise uncorrelated under the input distribution and difficult to distinguish with limited samples. A higher SQ dimension implies a richer hypothesis class, that requires more samples to identify the correct function. We show in the following theorem that the SQ dimension of $\mathbf{MLT}(d, n)$ grows exponentially with task parameters.

**Theorem 5.4.** *SQ dimension of $\mathbf{MLT}(d, n)$ under uniform input distribution is at least $n^{\Omega(d)}$.*

---

Informally this implies that any algorithm that tries to learn a $\mathbf{MLT_{\Pi^*}}$-capable SURR-MLT with trainable parameters $\{W_i\}_{i=1}^d$, and in-context information $\{C_i\}_{i=1}^d$ always fixed at $\mathbf{0}$s, by minimizing loss across samples will require at least $n^{\Omega(d)}$ sample complexity.

---

A corollary is that for vanilla SFT with SGD, sample complexity to learn $\mathbf{MLT_{\Pi^*}}$ can be at least $n^{\Omega(d)}$. This is adapted from Edelman et al. (2023), who analyse for sparse parity that has similar SQ dimension (Corollary F.6).

**Informal proof for SQ dimension:** Our proof extensively analyses the case where number of characters is 2 (lem. F.10), on which we will build proof for general $n$ (lem. F.20). The proof for $n = 2$ leverages uncorrelations between two randomly selected translation tasks; i.e. we show that for two random set of phrasebooks $\Pi^\alpha, \Pi^\beta$, the translation tasks $\mathbf{MLT_{\Pi^\alpha}}, \mathbf{MLT_{\Pi^\beta}}$ will have 0 output correlation with probability at least $1 - 2^{-\Omega(d)}$ w.r.t. random choice of $\Pi^\alpha, \Pi^\beta$ (Lemma F.13). We then show that we can pick exponentially many such random set of phrasebooks for which the translation tasks will be pairwise uncorrelated. This translates to a high SQ dimension.

## 5.3. Sample Complexity of Context-Enhanced Learning

Here, we show that context-enhanced learning substantially improves the sample complexity of learning in SURR-MLT. An $\mathbf{MLT}(d, n)$-ICL-capable model gets a set of phrasebooks $\mathbf{\Pi}^*$ by setting $\{\texttt{Matrix}(\pi_i^*)\}_{i=1}^d$ as in-context representations $\{C_i\}_{i=1}^d$. When a curriculum is followed such that a translation rule is dropped from a phrasebook, say $\pi_i^*$, SURR-MLT will set the corresponding column in its in-context representation $C_i$ as $\mathbf{0}$'s. Denote the zero-ed out column by $C_i^{(j)}$ for some generic column index $j$, we note that the loss will be low if and only if the corresponding column in the learnable parameters $W_i^{(j)}$ exactly matches $\texttt{Matrix}(\pi_i^*)^{(j)}$.

A heuristic search algorithm, that searches among $n^2$ possibilities for the dropped rule and stores its one-hot representation in the corresponding column in $W_i$, can be used to minimize the loss. By sequentially dropping rules, followed by a search and store process, the algorithm achieves polynomial sample complexity.

**Theorem 5.5** (Informal; cf Corollary G.19). *For any task* $\mathbf{MLT_{\Pi^*}}$*, there is a heuristic search algorithm paired with a curriculum of iteratively dropping rules from phrasebooks, that can learn a* $\mathbf{MLT_{\Pi^*}}$*-capable* SURR-MLT *with sample complexity* $\mathcal{O}(n^6 d \log d)$ *with high probability.*

The enumerative step in the heuristic search algorithm requires $\Theta(n^2)$ steps on average, as the algorithm needs to search over $\Theta(n^2)$ possibilities when a rule is dropped from a phrasebook. Instead, we show that gradient descent requires only a few steps per dropped rule. Dropping a rule sets the respective column in the in-context representation to 0, causing the gradient for the corresponding column in the trainable parameters to strongly align with the missing column. We formally present results for $d = 2$; due to exponentially growing number of terms to analyse with higher $d$, we keep the result for general $d$ as a conjecture.

**Theorem 5.6** (Informal; cf Corollary G.25). *When* $d = 2$*, there is a gradient descent based algorithm, paired with a curriculum of iteratively dropping a random rule from phrasebooks, that can return* $\mathbf{MLT_{\Pi^*}}$*-capable* SURR-MLT *with sample complexity* $\mathcal{O}(n^4)$ *with high probability.*

While theoretical analysis of GD dynamics beyond $d = 2$ is challenging, empirically we show that when training with SGD, the trainable parameters for much deeper SURR-MLT can quickly learn the set of phrasebooks in $\mathbf{\Pi}^*$. In Figure 5, we show context-enhanced learning results for a SURR-MLT on a randomly selected set of phrasebooks $\mathbf{\Pi}^*$ in $\mathbf{MLT}(10, 10)$. Regardless of whether we learn each layer separately with layer-wise SGD or all layers simultaneously with SGD in the surrogate model, the trainable parameters learn the phrasebooks in $\mathbf{\Pi}^*$ very quickly. More discussions are deferred to Appendix G.5.

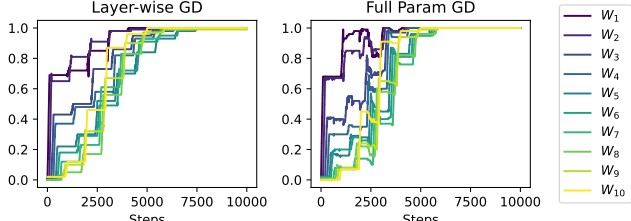

*Figure 5.* Percentage of columns in $\texttt{HardMax}(W_i)$ matching $\texttt{Matrix}(\pi_i^*)$ when optimizing a SURR-MLT for $\mathbf{MLT}(10, 10)$. Learnable parameters of all layers quickly converge to the correct values, regardless of whether layer-wise updates are imposed.

## 5.4. Insight from SURR-MLT: Gradient Quality

In this section, we show the major difference between context-enhanced learning and vanilla SFT to be the amount of predictive information in gradients for the trainable parameters. Our theoretical analysis in Theorem 5.6 on SURR-MLT primarily shows that when a single translation rule in a phrasebook $\pi_i^*$ is dropped, equivalently a column in the in-context representation, say $C_i^{(j)}$, is zero-ed out, the gradient of the corresponding column in trainable parameter $W_i^{(j)}$ aligns strongly with the one-hot vector representation of the dropped rule, $\texttt{Matrix}(\pi_i^*)^{(j)}$. However, this strong alignment heavily relies on the presence of other rules in-context. When more rules are dropped, the gradient signal gets increasingly "inaccurate". We quantitatively characterize such degradation as follows:

Consider an ICL-capable SURR-MLT, we focus on a column (with index $j$, denoted as superscript) $W_1^{(j)}$ in the first layer with ground truth $\texttt{Matrix}(\pi_1^*)^{(j)}$. We define **gradient prediction accuracy** (see formal definition in Definition B.1) for $W_1^{(j)}$ as the probability that the argmax entry of the negative stochastic batch gradient on $W_1^{(j)}$ matches the argmax entry of $\texttt{Matrix}(\pi_1^*)^{(j)}$. Intuitively, a higher gradient prediction accuracy will make learning $\texttt{Matrix}(\pi_1^*)$ easier. We track the accuracy metric when only $C_i^{(j)}$ is zero-ed out as well as when taking more aggressive context dropout schemes.

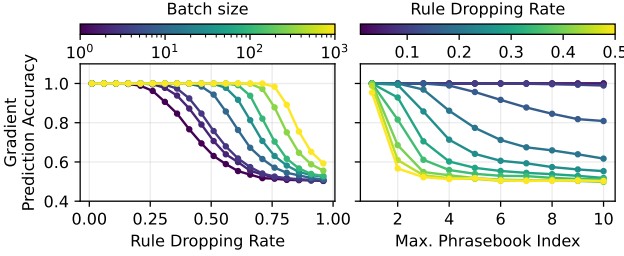

*Figure 6.* Gradient prediction accuracy when (left) varying dropping rates for rules in $\pi_1$ and batch sizes of gradient computation, and (right) varying dropping rates and max phrasebook index ($k$) for rules dropped from $\pi_1, \ldots, \pi_k$. Higher dropping rates always lead to lower gradient prediction accuracy.

In Figure 6, we report the gradient prediction accuracy with dropout schemes involving (1) multiple rules dropped out from the first phrasebook and (2) rules dropped from multiple phrasebooks. We can see that higher dropping rates significantly degrade the gradient prediction accuracy, highlighting the necessity of proper in-context information in order to obtain the optimization benefit. More details on the metric and experiments are deferred to Appendix B.1. We note that finding above is based on observations on the simplified surrogate model. A quantitative characterization of the optimization benefit for context-enhanced learning in the LLM regime is left as an interesting future work.

## 6. Related Works

In this section, we discuss related works to this paper. More related works regarding (1) compositional and OOD generalization and (2) mechanistic understanding of transformers are discussed more extensively in Appendix D.

### Learning using Privileged Information (LUPI)

LUPI was formally introduced by Vapnik & Vashist (2009) in kernel SVMs, and it is heavily related to concepts of Learning with Side Information (Kuusela & Ocone, 2002; Jonschkowski et al., 2015; Zhang et al., 2018). Primarily, both concepts refer to training a model with additional information that could help training but may not be available at test time. This framework has been extended theoretically for classification tasks (e.g. Pechyony & Vapnik (2010); Momeni et al. (2018)) and has been used to explain benefits of knowledge distillation (Vapnik et al., 2015; Lopez-Paz et al., 2015). To name a few applications, this concept has been heavily studied for improving boosting for classification tasks (Chen et al., 2012), visual and video encoders (Hoffman et al., 2016; Cheng et al., 2020; Xu et al., 2017), human preference predictions (Farias & Li, 2019), multiagent games (Sessa et al., 2020), speech recognition (Synnaeve et al., 2014), and medical recognition (Ceccarelli & Maratea, 2008; Sabeti et al., 2020).

While LUPI has rich application in classification, extending LUPI to LLMs introduces unique challenges due to their auto-regressive training. Such a concept raises questions on whether applying auto-regressive loss on the additional information is necessary to get the benefits of additional supervision information in context, and how the additional information changes the training behavior of LLMs. Hence, our work is a nontrivial generalization of this framework to LLMs that connects to their in-context learning strengths.

### In-context learning and memorization

Recent works have studied emergence of ICL and competition with in-weights learning (IWL) (equivalent to knowledge memorization) during pre-training of language models (Chan et al., 2022; Reddy, 2024; Singh et al., 2023; 2024; Nguyen & Reddy, 2024). These works show that data distribution properties affect the behavior of the model during training. Our work can be thought of as contemporary to the works above, where we show that strong ICL capabilities can be utilized for improving knowledge on a task by context-enhanced learning.

**Benefits of in-context learning:** In-context learning has been primarily studied in the context of few-shot prompting of large language models. Better supervision with in-context supervision can help in improved performance (e.g. some representative works (Arora et al., 2022; Si et al., 2022; Wu et al., 2022; Lu et al., 2021; Su et al., 2022)), OOD generalization and factuality (reduced hallucination) (Yang et al., 2023; Dhuliawala et al., 2023; Chen et al., 2023; Didolkar et al., 2024), and more structured latent representations (Park et al., 2024) for large language models. On the other hand, we show that improved supervision with in-context supervision can also help a model learn faster in SFT, while seemingly not leaking the in-context information in its output probabilities.

## 7. Discussion, Limitations, and Future work

Some experimental works have implicitly used the notion of context-enhanced learning but the current paper formalized this notion for auto-regressive models and showed, using **MLT**, that this form of learning can be exponentially more sample-efficient than standard SFT. At the end of training it is hard to recover the in-context information seen during training from the model's output probabilities. We note that this finding appears to have implications about copyright law (e.g., whether or not LLM training amounts to "transformative use" of text (Carlini et al., 2021; 2022; Karamolegkou et al., 2023)) whose further study is left for future work.

Our experiments focus on a synthetic **MLT** task for a few reasons: (1) to ensure that the task is absent from LLM pre-training, which allows precise quantification of benefits of context-enhanced learning, including not revealing the curriculum text at inference time. (2) the task is too difficult (at least for Llama 3.2 3B model) to learn via vanilla SFT, but is learnable via context-enhanced learning. Extending these findings to real-world complicated tasks (e.g., in math and coding) is left for future work.

Our convergence analysis for context-enhanced learning relies on a surrogate model, and extending it to an actual transformer remains an open challenge for theory of deep learning. Extending formalization of context-enhanced learning to explore LLM training in multi-agent settings–where models collaborate and learn from each other to discover novel concepts–would be an exciting avenue for future research.

## Acknowledgment

We thank Yun Cheng, Simon Park, Tianyu Gao, Yihe Dong, Zixuan Wang, Haoyu Zhao, and Bingbin Liu for discussions, suggestions, and proof-reading at various stages of the paper. AP, SA acknowledge funding from NSF, PLI, DARPA, ONR, and OpenAI. XZ is additionally supported by a Gordon Y.S. Wu Fellowship in Engineering.

## Impact statement

We formulate a basic notion "context-enhanced learning", and study how it is different from usual learning. One consequence of our study is that enhancing the context with good quality data can enhance the quality of the learner.

This enhancement could be used with privileged information and it appears that the training can be done in such a way that the model does not leak this privileged information. This can be seen as enhancing privacy, since there is a lower chance of leaking privileged training data.

The flip side is that it suggests — albeit in very toy and synthetic setting, with full study left for future work — that model training could use off-limits data (albeit with no gradient updates on it) and this use might not be detectable from querying the trained model. But this is hypothetical at this point since the paper concerns a very toy setting.

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

# Appendix

## A. Overview of the Appendix and Common Notations

Here, we outline the structure for the appendix for easier readability. Due to space constraints, we had to defer a lot of details from the main paper. Appendix B.1 contains details on the experiments on gradient prediction accuracy from Section 5.4. Appendix B.4 shows the details on CoT internalization pipeline that we followed to get an **MLT**$(d, n)$-ICL-capable model. Appendix B.5 gives more details on the context-enhanced learning experiments conducted in Section 3.

Appendix C shows mechanistic experiments, designed in Section 3.3, for additional experimental settings. Appendix C.3 present additional details and results for information recovery through querying experiments conducted in Section 4. Appendix D presents additional relevant related works, including discussion on OOD generalization of language models and mechanistic understanding of large transformer architectures. Appendix E discusses few properties on the **MLT**. Appendix F presents the formal theorems on the computational hardness of learning the translation task and their proofs, which were informally outlined in Section 5.2 in the main paper. Appendix G then presents the formal statements and proofs for context-enhanced learning in the surrogate model, which were informally outlined in Section 5.1 in the main paper. Finally, Appendix H presents the theoretical construction of an ideal transformer that can simulate the surrogate model. We present extensive details on prompts, and examples of training sequences, that we used for context-enhanced learning in Appendix I.

*Table 3.* Important notations

| Scope of Notation | Symbol | Description |
|---|---|---|
| General Notations | $A$ | A set |
| | $\boldsymbol{A}$ | A matrix |
| | $\boldsymbol{A}^{(j)}$ | The $j$-th column of matrix $\boldsymbol{A}$ |
| | $\mathbf{e}_k$ | One-hot embedding vector with 1 at dimension $k$ |
| Language Modeling | $X$ | All possible text strings (input space for a causal LM) |
| | $\mathcal{Y}$ | All possible distribution over texts (output space for a causal LM) |
| | $f_\theta$ | Causal LM parameterized by $\theta$ |
| | $g$ | Language task mapping inputs $x \in X_g \subset X$ to a distribution $Y \in \mathcal{Y}$ |
| | $\ell_{\mathrm{auto}}$ | Auto-regressive cross entropy loss |
| | $\mathrm{CURR}_g(x, t)$ | In-context curriculum for learning task $g$ with input $x$ at step $t$ |
| MLT Translation Task | $d$ | Depth of translation task |
| | $n$ | Number of characters in each alphabet |
| | $A$ | An alphabet set |
| | $\boldsymbol{s}$ | A sequence in alphabet $A$ |
| | $\pi$ | A phrasebook between 2-tuples in two alphabets, e.g. $(\pi_1 : A_1^2 \to A_2^2)$. |
| | $\mathcal{B}^\pi$ | Space of all possible phrasebook on $A_1^2 \to A_2^2$ |
| | $\boldsymbol{\Pi}$ | A set of phrasebooks $\{\pi_i\}$ defining a translation task |
| | $\mathbf{MLT}_{\boldsymbol{\Pi}}$ | Translation task with a set of phrasebooks $\boldsymbol{\Pi}$ |
| | $\mathbf{MLT}(d, n)$ | Family of translation tasks of depth $d$ and $n$ characters |
| | $\mathrm{STR}(\pi)$ | Descriptive text for a phrasebook $\pi$ (see Section 2.2) |
| Surrogate Model | $\boldsymbol{V}_i$ | Embedding matrix representing sequence $\boldsymbol{s}_i$ (Definition G.1) |
| | $\boldsymbol{C}_i$ | In-context information matrix for level $i$ |
| | $\boldsymbol{W}_i$ | Trainable parameter matrix for level $i$ |
| | $\boldsymbol{P}_i$ | Effective translation matrix for level $i$ (Definition G.6) |
| | HardMax | Column-wise hard-max function |
| | Matrix$(\pi)$ | Matrix representation of a phrasebook $\pi$ (Definition G.4) |
| | $\mathrm{SURR\text{-}MLT}_{\{\boldsymbol{W}_i\}_{i=1}^d}$ | Surrogate model parameterized by $\boldsymbol{W}_i$'s (Definition G.7) |

# B. Deferred definitions, and experimental details from the main paper

## B.1. Gradient Prediction Accuracy

Our theoretical analysis in Theorem 5.6 was built on the fact that when a translation rule in a phrasebook is dropped by zeroing out a column in an in-context representation $C_i$, the gradient for the corresponding column in $W_i$ points to the direction of the dropped rule.

On the other hand, we show that when multiple rules are simultaneously dropped from the phrasebooks, the gradients for the trainable parameters become increasingly noisy. To quantify this degradation, we compute gradient for each column of the trainable parameters for an ICL-capable model, when the corresponding rule is dropped from phrasebooks by zeroing out the relevant column in the in-context representations. We then compute whether the computed gradient points to the right rule. By progressively increasing the number of simultaneously dropped rules, we measure the resulting degradation in the accuracy of the gradient's predictions.

More formally, denote RANDOM-DROP as an operation that takes in column dropping rates per layer $p_1, \cdots, p_d$, set of phrasebooks $\mathbf{\Pi}^*$, and returns in-context representations $\{C_i\}_{i=1}^d$, such that $C_i^{(j)} = \mathtt{Matrix}\left(\pi_i^*\right)^{(j)}$ ($j$th columns of $C_i$ and $\mathtt{Matrix}\left(\pi_i^*\right)$ are equal) with probability $1 - p_i$ and 0 otherwise. Then,

---

**Definition B.1.** For a set of column dropping rates per layer $p_1, \cdots, p_d$, set of phrasebooks $\mathbf{\Pi}^*$ and $\mathbf{MLT}(d, n)$-ICL-capable model, predictive accuracy of gradients is defined as

$$
\mathbb{E}_{\{C_i\}_{i=1}^d = \text{RANDOM-DROP}(p_1, \cdots, p_d, \mathbf{\Pi}^*)}
$$
$$
\mathbb{E}_{j \in [1, n^2] | C_1^{(j)} = 0} \mathbb{I}\left[ \mathtt{HardMax}\left( -\nabla_{\boldsymbol{W}_1^{(j)}} \mathcal{L} \right) = \left(\mathtt{Matrix}\left(\pi_i^*\right)\right)^{(j)} \right]
$$
$$
\mathcal{L} = \mathbb{E}_{\boldsymbol{s}_1} \ell\left( \text{SURR-MLT}_{\{\boldsymbol{W}_i\}_{i=1}^d}\left( \{C_i\}_{i=1}^d, \boldsymbol{V}_1 \right), \mathbf{MLT}_{\mathbf{\Pi}^*}(\boldsymbol{s}_1) \right),
$$

where $\ell$, adapted from Section 2.1, computes cross-entropy loss on the predicted output embeddings of surrogate model using true output string and $\mathbb{I}$ denotes the indicator function.

---

The above definition computes gradients on the expected loss of the model. We primarily focus on predicting the trainable parameters of the first layer, i.e. $W_1$, as that is the deepest layer in the surrogate model and intuitively should suffer the most with noise accumulation from dropped rules. On the other hand, we can further adapt the definition to compute the accuracy for batched gradients, where the gradients are computed using average loss on a randomly sampled batch of input sequences.

In Figure 6, we report the predictive accuracy of gradient for an $\mathbf{MLT}(d, n)$-ICL-capable model, and its behavior with varying batch size and the column dropping rates. We report for two cases, one where column dropping rates is non-zero only for the first phrasebook, and one where we increase the number of phrasebooks for which rules are independently and uniformly dropped. In both cases,

- Increased column dropping rates leads to noisier gradients and reduced prediction accuracy.

- Larger batch sizes improve gradient accuracy but cannot fully compensate for high dropout rates.

- Dropping rules from multiple phrasebooks significantly degrades gradient prediction accuracy.

## B.2. Hidden Representations of a Model

A transformer $f_\theta$ with embedding dimension $p$ and $K$ layers takes any input sequence $\boldsymbol{x}$, say of length $L$, converts to an embedding matrix $\boldsymbol{H}_1 \in \mathbb{R}^{L \times p}$, and modifies the embeddings using a succession of $K$ transformer layers; which we will denote by $f_\theta^{(1)}, f_\theta^{(2)}, \cdots, f_\theta^{(K)}$. We refer to the hidden representations for the input $\boldsymbol{x}$, with embedding matrix $\boldsymbol{H}_1$, as the output of the model after every layer. We will denote them as $\boldsymbol{H}_{i+1} \in \mathbb{R}^{L \times p}$ for the output of layer $f_\theta^{(i)}$. That is,

$$\boldsymbol{H}_{i+1} = f_\theta^{(i)} \circ \cdots \circ f_\theta^{(2)} \circ f_\theta^{(1)} (\boldsymbol{H}_1), \quad \text{for all } i \geq 1.$$

$\ell_2$**-norm in change in hidden representation with perturbation in in-context information**   For an $\mathbf{MLT}(d, n)$-ICL-capable model, we supply in-context information for the textual description of a set of phrasebooks $\boldsymbol{\Pi}$ as $\mathrm{STR}(\boldsymbol{\Pi}) = [\mathrm{STR}(\pi_1), \cdots, \mathrm{STR}(\pi_d)]$. Our inputs to the transformer for an input string $\boldsymbol{s}_1$ will be of the form $[\mathrm{STR}(\boldsymbol{\Pi}), \boldsymbol{s}_1, \texttt{<THINK>}, \ldots, \boldsymbol{s}_{d+1}]$, where $\boldsymbol{s}_{d+1} = \mathbf{MLT}_{\boldsymbol{\Pi}}(\boldsymbol{s}_1)$. By the definition of hidden representations, $\boldsymbol{H}_2, \cdots, \boldsymbol{H}_{K+1}$ will denote the output of the transformer layers for this input string. However, we will be only interested in the hidden representations for the tokens involved in the tokens for $\texttt{<THINK>}, \ldots, \boldsymbol{s}_{d+1}$; and we will refer to the corresponding subsets of $\boldsymbol{H}_2, \cdots, \boldsymbol{H}_{K+1}$ that represent these specific tokens as $\boldsymbol{V}_2, \cdots, \boldsymbol{V}_{K+1}$.

Now, suppose we randomly take a phrasebook $\pi_i$ in $\boldsymbol{\Pi}$ and change to a random phrasebook $\tilde{\pi}_i$. The corresponding textual description that will augment the context for an input string will then be $[\mathrm{STR}(\pi_1), \cdots, \mathrm{STR}(\pi_{i-1}), \mathrm{STR}(\tilde{\pi}_i), \mathrm{STR}(\pi_{i+1}), \cdots, \mathrm{STR}(\pi_d)]$. If $\tilde{\boldsymbol{V}}_2, \cdots, \tilde{\boldsymbol{V}}_{K+1}$ now denote the hidden representations that represent the tokens for $\texttt{<THINK>}, \ldots, \boldsymbol{s}_{d+1}$, then the $\ell_2$-norm in the change of the hidden representation after layer $j$ (for any $1 \leq j \leq K$) with the perturbation in $\boldsymbol{\Pi}$ will be given by $\left\| \tilde{\boldsymbol{V}}_j - \boldsymbol{V}_j \right\|_2$.

## B.3. Definition of a "stitched" model

We reuse notations from Appendix B.2. Suppose we have an $\mathbf{MLT}(d, n)$-ICL-capable model $f_\theta$ and an $\mathbf{MLT}_{\boldsymbol{\Pi}^*}$-capable model $f_{\theta^*}$. Their corresponding transformer layers are denoted by $f_\theta^{(1)}, f_\theta^{(2)}, \cdots, f_\theta^{(K)}$ and $f_{\theta^*}^{(1)}, f_{\theta^*}^{(2)}, \cdots, f_{\theta^*}^{(K)}$. Each model takes in an input sequence and processes them with their $K$ transformer layers.

Formally, we will write for the ICL-capable model. It takes in input sequence $\boldsymbol{x}$, and converts to an embedding matrix, say $\boldsymbol{H}_1$, and the output after the $K$ layers are given by:

$$\boldsymbol{H}_{K+1} = f_\theta^{(d)} \circ \cdots \circ f_\theta^{(2)} \circ f_\theta^{(1)} (\boldsymbol{H}_1).$$

**Process of "stitching":**   The process of stitching takes in two parameters $L_{\text{start}}$ and $L_{\text{end}}$ and replaces all layers from $L_{\text{start}}$ to $L_{\text{end}}$ in $f_\theta$ with the corresponding layers in $f_{\theta^*}$ to give a "stitched" model, say $f_{\theta, \theta^*, L_{\text{start}}, L_{\text{end}}}$. The output of the "stitched" model $f_{\theta, \theta^*, L_{\text{start}}, L_{\text{end}}}$ on an input sequence $\boldsymbol{x}$ will be given by

$$\boldsymbol{H}_{K+1} = f_\theta^{(d)} \circ \cdots \circ f_\theta^{(L_{\text{end}}+1)} \circ \underbrace{f_{\theta^*}^{(L_{\text{end}})} \circ \cdots f_{\theta^*}^{(L_{\text{start}})}}_{\text{Layers are replaced by layers from } f_{\theta^*}} \circ f_\theta^{(L_{\text{start}}-1)} \circ f_\theta^{(2)} \circ f_\theta^{(1)} (\boldsymbol{H}_1).$$

## B.4. Pipeline on CoT Internalization

We randomly sample $M$ sets of phrasebooks $\pi_1, \ldots, \pi_M$ not equal to $\boldsymbol{\Pi}^*$. For each set of phrasebooks $\pi_i$, we randomly sample a single input sequence $\boldsymbol{s}_1$ and compute all the intermediate translation steps $\boldsymbol{s}_2, \ldots, \boldsymbol{s}_{d+1}$. We first train the model to do robust explicit CoT by auto-regressive training on sequences $[\mathrm{STR}(\pi_i), \boldsymbol{s}_1, \boldsymbol{s}_2, \ldots, \boldsymbol{s}_d, \boldsymbol{s}_{d+1}]$ with loss computed over $\boldsymbol{s}_2, \ldots, \boldsymbol{s}_{d+1}$. Then we follow common CoT internalization strategies (Deng et al., 2024; Hao et al., 2024; Yu et al., 2024; Su et al., 2024) and gradually replace the intermediate sequences by $\texttt{<THINK>}$ tokens in training. After all intermediate sequences have been replaced, the model has low loss on $\boldsymbol{s}_{d+1}$ with input $[\mathrm{STR}(\boldsymbol{\Pi}_1), \boldsymbol{s}_1, \texttt{<THINK>}, \ldots, \texttt{<THINK>}, \boldsymbol{s}_{d+1}]$, satisfying Definition 2.2. Since we only sample on sequence per set of phrasebooks, there is little memorization on particular phrasebooks.

**Details on training hyperparameters:** We use $M = 3 \times 10^5$ random sets of phrasebooks with length between 20 and 40, for getting $\mathbf{MLT}(5, 8)$-ICL-capable model, and $M = 10^6$ random sets of phrasebooks with length between 20 and 40, for getting $\mathbf{MLT}(5, 10)$-ICL-capable model. We use cosine learning rate schedule (Loshchilov & Hutter, 2016), with peak

learning rate $10^{-4}$ and a $6\%$ warmup phase, where learning rate is linearly increased from $0$ to the peak. We use AdamW optimizer (Loshchilov & Hutter, 2019) with weight decay fixed at $10^{-4}$. We use a batch size of $64$ for training.

**CoT internalization curriculum:** For the first $10\%$ fraction of training, we train the model with explicit CoT tokens that contain the intermediate steps in translation. Then between $10\%$ to $60\%$ fractions of training, CoT tokens are gradually replaced by <THINK> tokens, with the rate of replacement increasing linearly from $0\%$ to $100\%$. We follow a deterministic first-to-last order for replacing CoT tokens; earlier CoT tokens are replaced first with <THINK> tokens. After that, the model is trained with the <THINK> CoT tokens till the end of training.

### B.5. Experiment configuration for context-enhanced learning

For context-enhanced learning, we create supervised datasets $D_{\Pi^*}$ of different sizes; each containing between $10^4$ to $10^6$ samples. When performing **Annealing Dropout** or **Fixed Dropout**, at each step of training, we randomly perform dropout on the rules of all phrasebooks or apply dropout to the rules of a randomly sampled phrasebook to define curriculum-text. That is, if $\pi_1^*, \ldots, \pi_5^*$ represent the phrasebooks, then at each step of training, we either randomly drop rules uniformly from all of $\pi_1^*, \ldots, \pi_5^*$, or just drop from one of the phrasebooks randomly selected from $\pi_1^*, \ldots, \pi_5^*$, while keeping the rules of all other phrasebooks intact, to create curriculum-text.

**Hyperparameters:** Training hyperparameters are set equal to the optimization hyperparameters used in preparation of ICL-capable training phase (Appendix B.4), except we set weight decay to $0$ in all experiments. We report the performance of the trained model after single epoch of training on each $D_{\Pi^*}$ and plot against the size of the dataset in Figure 2.

# C. Additional Experiments Results

## C.1. Selective Internalization of Context

In this experiment, we test whether the model can internalize rules that don't incur an increase in loss when dropped during training. To do so, we ablate on **Annealing Dropout**. We select a phrasebook and create 2 splits of rules in the phrasebook; one set of rules will be utilized by training samples for performing the translation task (which we call the training split), while other set of rules will appear in curriculum-text during training but never utilized for the translation task (which we call the heldout split).

At test time, we measure the performance of the trained model on 2 sets of evaluation examples, one that only use rules from the training split for their translation (equal to training distribution), and other that uses rules only from the heldout split for their translation (different from training distribution). The model is being measured without any phrasebooks information at evaluation. We conduct the above experiment for each phrasebook, i.e. we create 5 sets of experiments where we only create heldout split for one specific level of phrasebook. In Figure 7, we show that the model fails to perform any translation that use the rules from the held-out split in all settings. This shows that the model only internalizes those rules that are important for the translation task for the training samples, and which incurs an increase in training loss when dropped.

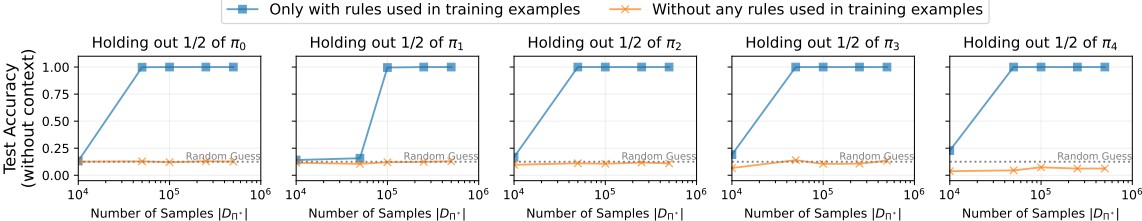

*Figure 7.* Ablations on **Annealing Dropout** for $\mathbf{MLT}(5, 10)$ assess whether models internalize rules not used in training samples. (Left to right) $1 \leq i \leq 5$: Five independent experiments where a held-out split is created for phrasebook $\pi_i^*$, and training excludes samples that use the translation rules in the heldout split of $\pi_i^*$. Evaluation is conducted on two sets: one where samples do not use held-out rules from $\pi_i^*$ and one where only held-out rules from $\pi_i^*$ are used. The model's random performance on the latter indicates that it internalizes only rules used during training, particularly those whose removal increases loss.

## C.2. Mechanistic Insights

Here we provide additional figures corresponding to Figure 3 and Figure 4, but in more settings ($n = 10$ vs $n = 8$, **Annealing Dropout** vs **Fixed Dropout**).

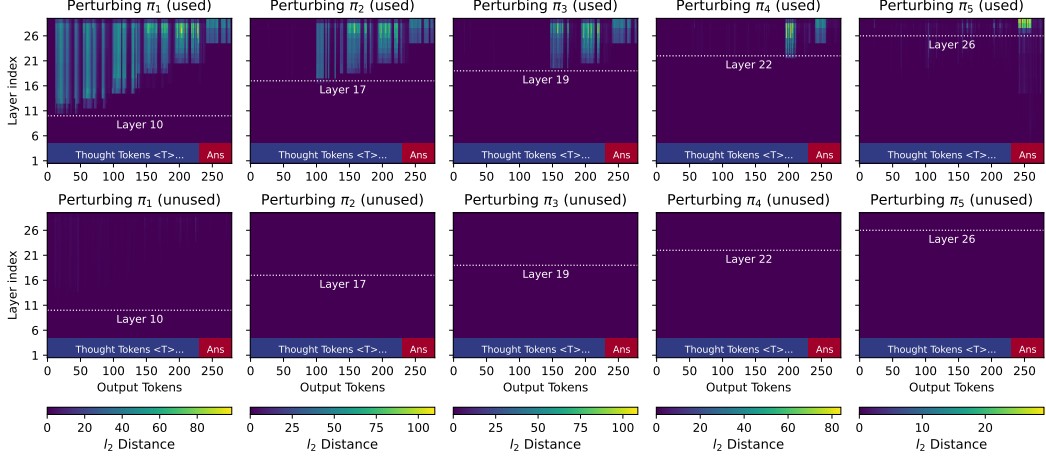

*Figure 8.* Comparison between perturbing used rules (top row, identical to Figure 3) and unused rules (bottom row) in context. When perturbing rules used in the translation at a later phrasebook, representation changes at later layers. However only perturbing unused rules leads to negligible representation changes. This experiment rules out the possibility that the affected depth is only dependent on position of the perturbation instead of the semantic content.

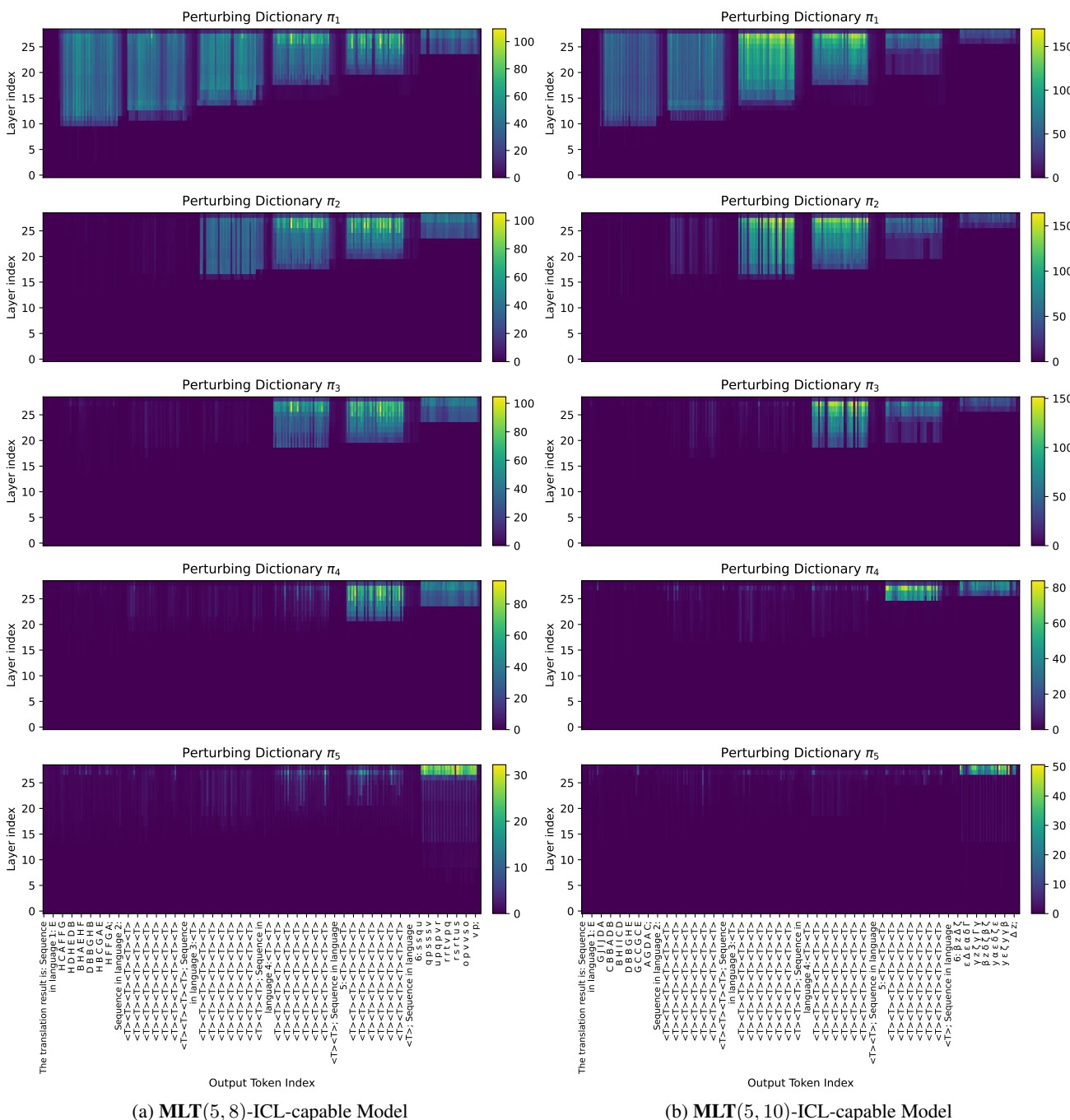

(a) **MLT**$(5, 8)$-ICL-capable Model

(b) **MLT**$(5, 10)$-ICL-capable Model

*Figure 9.* Evidence for sequential processing in **MLT**-ICL capable models ($n = 8, n = 10$). The left figure is identical to Figure 3, except we substitute the entire phrasebook $\pi_i$ with another random phrasebook of the same level $\hat{\pi}_i$. We observe the same behavior for **MLT**$(5, 10)$-ICL-capable model: perturbing later phrasebooks in the context changes output representations in the later layers.

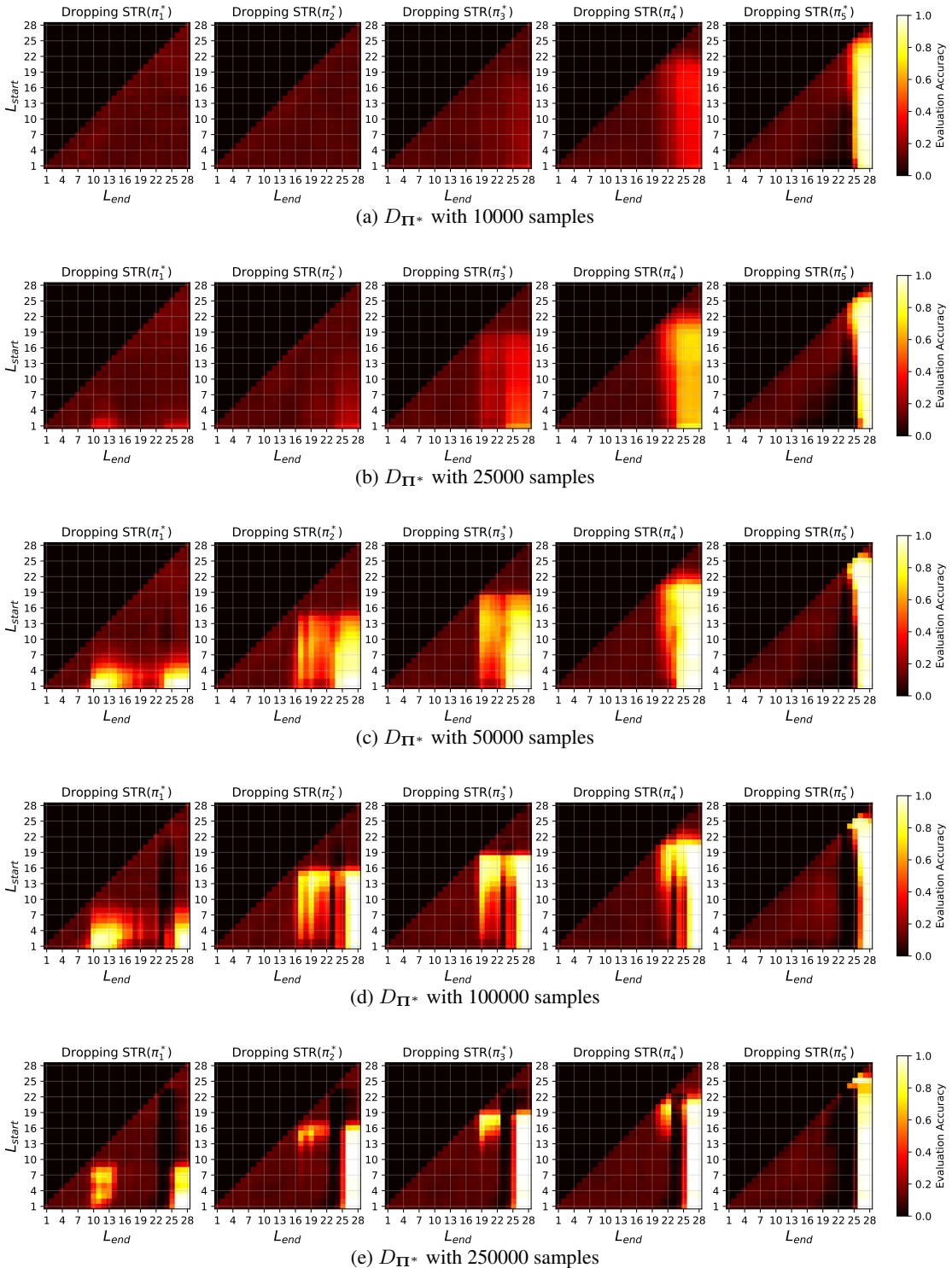

Figure 10. Repeated experiments from Figure 4 for models trained with context-enhanced learning at varying supervised dataset sizes (Plots for row (d) are identical to the plots in Figure 4). We observe that phrasebooks are progressively internalized with the number of training samples available during context-enhanced learning; later phrasebooks are internalized with fewer samples than the earlier ones. We observe similar localization patterns across layers from different phrasebooks across the models, however, we also observe that the localization patterns get increasingly sparser as training continues.

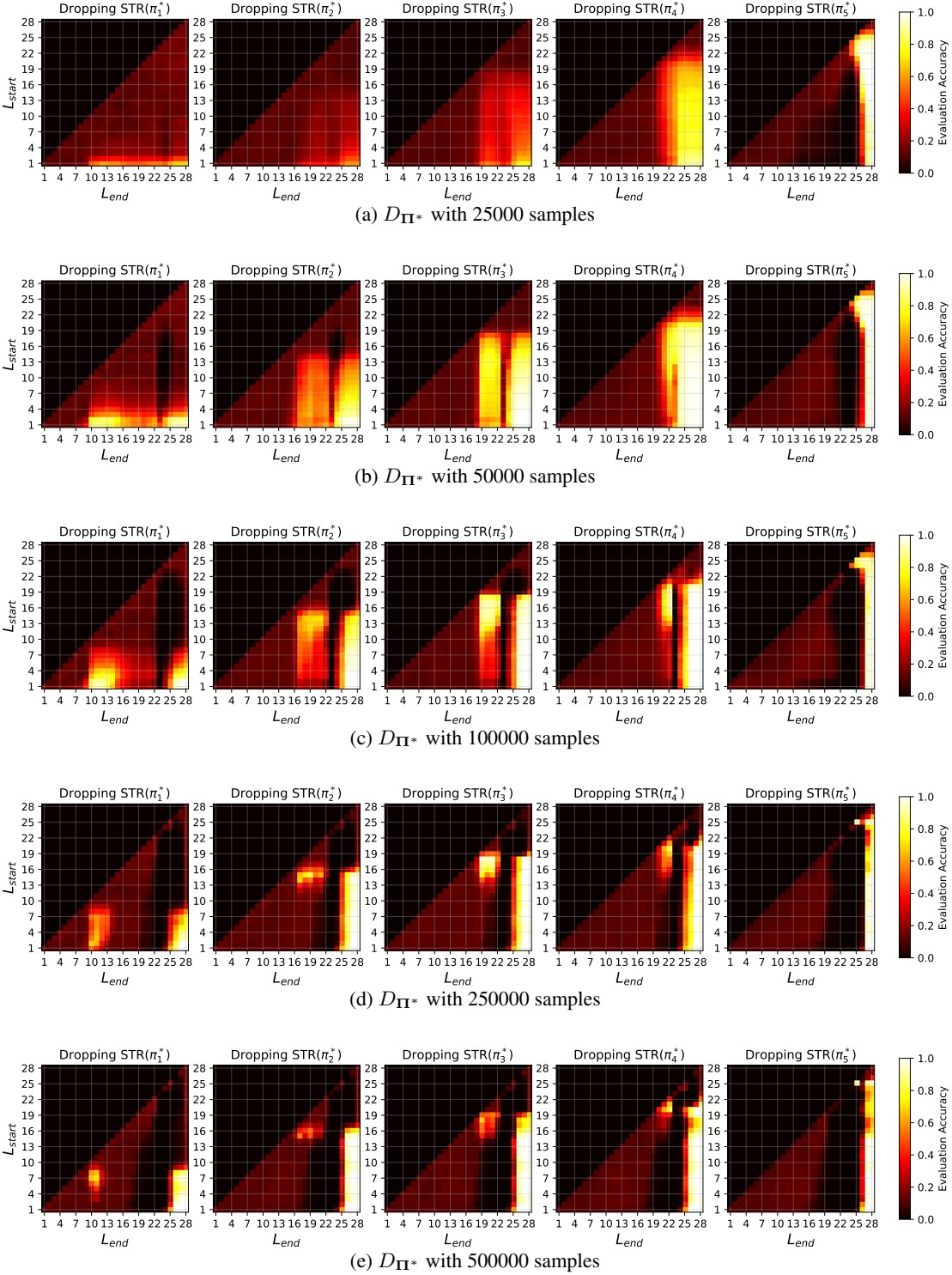

(a) $D_{\mathbf{\Pi}^*}$ with 25000 samples

(b) $D_{\mathbf{\Pi}^*}$ with 50000 samples

(c) $D_{\mathbf{\Pi}^*}$ with 100000 samples

(d) $D_{\mathbf{\Pi}^*}$ with 250000 samples

(e) $D_{\mathbf{\Pi}^*}$ with 500000 samples

*Figure 11.* Repeated experiments from Figure 10 for models trained with **Fixed Dropout** instead of **Annealing Dropout**. Observations remain similar. This suggests that the position of phrasebook internalization is primarily decided by the **MLT**$(d, n)$-ICL-capable initialization, and less dependent on the specific dropout curriculum we use for context-enhanced learning.

## C.3. (Non)Verbatim Memorization of Phrasebook Rules

In this subsection, we provide a more detailed explanation for the evaluation on the feasibility of recovering phrasebook rules from models trained with context-enhanced learning with phrasebook excerpts in context.

Given a $\textbf{MLT}_{\Pi^*}$-capable model $f_{\theta^*}$ trained with context-enhanced learning where phrasebook rules from $\text{STR}(\Pi^*)$ are provided within the context during training, we test if the model retains explicit memory of the textual format of rules. Namely, for each of the phrasebook rule of the form "a b -> C D" in $\text{STR}(\Pi^*)$, whether it can complete the ground truth output tokens "C D" providing "a b ->" and additional context of the phrasebook $\text{STR}(\Pi^*)$ before "a b -> C D". We conduct the test by computing the forward pass through $f_{\theta^*}$ with $\text{STR}(\Pi^*)$ as the input (see exact input format in Figure 17).

Now we formally define the query strategies and metrics we used for creating Table 2. Suppose there are $h$ unique tokens and $\text{STR}(\Pi^*)$ contains $L$ tokens, let $\boldsymbol{S} \in \mathbb{R}^{h \times L}$ denote the corresponding output logit score matrix when we pass $\text{STR}(\Pi^*)$ into $f_{\theta^*}$. For a translation rule with ground truth output tokens of index $a_1, a_2 \in A_{i+1} \subset [h]$, let $\boldsymbol{S}^{(k)}$ and $\boldsymbol{S}^{(k+1)} \in \mathbb{R}^h$ denote the logit vector corresponding to predicting these two entries.

If we are doing greedy decoding, then we will recover the correct phrasebook rule if and only if we greedily select both $a_1$ from $\boldsymbol{S}^{(k)}$ and $a_2$ from $\boldsymbol{S}^{(k+1)}$, so the probability of correctly recovering $(a_1, a_2)$ conditioned on $\boldsymbol{S}$ is

$$\mathbb{P}\left[\text{Greedy-Recover}(a_1, a_2)\right] = \mathbb{1}\left[\arg\max_{i \in [h]} \boldsymbol{S}^{(k)} = a_1\right] \cdot \mathbb{1}\left[\arg\max_{i \in [h]} \boldsymbol{S}^{(k+1)} = a_2\right].$$

Meanwhile, if we apply random sampling with softmax temperature of 1, the probability of correctly recovering $(a_1, a_2)$ conditioned on $\boldsymbol{S}$ is just then

$$\mathbb{P}\left[\text{Sampling-Recover}(a_1, a_2)\right] = \left(\frac{\exp\left(\boldsymbol{S}^{(k)}_{a_1}\right)}{\sum_{i \in [h]} \exp\left(\boldsymbol{S}^{(k)}_i\right)}\right) \cdot \left(\frac{\exp\left(\boldsymbol{S}^{(k+1)}_{a_1}\right)}{\sum_{i \in [h]} \exp\left(\boldsymbol{S}^{(k+1)}_i\right)}\right).$$

Recall from Section 4 that we have also introduced two stronger adversaries with additional *token filtering* when doing decoding: (1) setting probability of <THINK> tokens to zero (2) when querying for a rule in $\pi_i$ with output alphabet $A_{i+1}$, setting probability of all tokens outside $A_{i+1}$ to zero.

Denoting the token index of <THINK> as $a_{\text{<THINK>}}$, then filter 1 corresponds to

$$\mathbb{P}\left[\text{Greedy-Filter1-Recover}(a_1, a_2)\right] = \mathbb{1}\left[\arg\max_{i \in [h] \setminus \{a_{\text{<THINK>}}\}} \boldsymbol{S}^{(k)} = a_1\right] \cdot \mathbb{1}\left[\arg\max_{i \in [h] \setminus \{a_{\text{<THINK>}}\}} \boldsymbol{S}^{(k+1)} = a_2\right],$$

$$\mathbb{P}\left[\text{Sampling-Filter1-Recover}(a_1, a_2)\right] = \left(\frac{\exp\left(\boldsymbol{S}^{(k)}_{a_1}\right)}{\sum_{i \in [h] \setminus \{a_{\text{<THINK>}}\}} \exp\left(\boldsymbol{S}^{(k+1)}_i\right)}\right) \cdot \left(\frac{\exp\left(\boldsymbol{S}^{(k)}_{a_1}\right)}{\sum_{i \in [h] \setminus \{a_{\text{<THINK>}}\}} \exp\left(\boldsymbol{S}^{(k+1)}_i\right)}\right).$$

Similarly, for filter 2, the probabilities are

$$\mathbb{P}\left[\text{Greedy-Filter2-Recover}(a_1, a_2)\right] = \mathbb{1}\left[\arg\max_{i \in A_{i+1}} \boldsymbol{S}^{(k)} = a_1\right] \cdot \mathbb{1}\left[\arg\max_{i \in A_{i+1}} \boldsymbol{S}^{(k+1)} = a_2\right],$$

$$\mathbb{P}\left[\text{Sampling-Filter2-Recover}(a_1, a_2)\right] = \left(\frac{\exp\left(\boldsymbol{S}^{(k)}_{a_1}\right)}{\sum_{i \in A_{i+1}} \exp\left(\boldsymbol{S}^{(k+1)}_i\right)}\right) \cdot \left(\frac{\exp\left(\boldsymbol{S}^{(k)}_{a_1}\right)}{\sum_{i \in A_{i+1}} \exp\left(\boldsymbol{S}^{(k+1)}_i\right)}\right).$$

Note that the second filter is a very strong adversarial assumption which assumes the user already has side information on the set of tokens contained in alphabet $A_i$ and $A_{i+1}$. To compute the final statistics, we sample 20 permutations of $\text{STR}(\Pi^*)$ for forward passes and compute the mean of the above statistics over all atomic phrasebook rules appearing in the context. That is 1280 entries for each phrasebook in the case of $n = 8$ and 2000 entries for each phrasebook in the case of $n = 10$.

Table 4. Recovery Success Rate For $n = 8, d = 5$ Cases (Rounded to 2 decimals) Random Guess Baseline: 1.56%

| Curriculum | # Training Samples | Query Method | Greedy Decoding | | Sampling ($T = 1$) | |
|---|---|---|---|---|---|---|
| | | | $\pi_1 - \pi_4$ | $\pi_5$ | $\pi_1 - \pi_4$ | $\pi_5$ |
| **Annealing Dropout** | 50000 | Base | 0.00% | 0.00% | 0.00% | 0.01% |
| | | Rule out <THINK> | 0.06% | 1.95% | 0.00% | 0.54% |
| | | Only keeping $A_i$ | 3.18% | 1.95% | 1.82% | 1.49% |
| **Annealing Dropout** | 100000 | Base | 0.00% | 0.00% | 0.00% | 0.64% |
| | | Rule out <THINK> | 0.00% | 0.08% | 0.00% | 1.25% |
| | | Only keeping $A_i$ | 1.37% | 0.08% | 1.69% | 1.80% |
| **Annealing Dropout** | 250000 | Base | 0.00% | 0.23% | 0.00% | 1.25% |
| | | Rule out <THINK> | 0.00% | 0.23% | 0.00% | 1.28% |
| | | Only keeping $A_i$ | 2.95% | 0.23% | 2.53% | 1.38% |
| **Fixed Dropout** | 50000 | Base | 0.00% | 0.00% | 0.00% | 0.00% |
| | | Rule out <THINK> | 0.08% | 0.78% | 0.00% | 0.09% |
| | | Only keeping $A_i$ | 0.82% | 0.86% | 1.48% | 1.43% |
| **Fixed Dropout** | 100000 | Base | 0.00% | 0.00% | 0.00% | 0.21% |
| | | Rule out <THINK> | 0.43% | 0.00% | 0.03% | 0.80% |
| | | Only keeping $A_i$ | 2.36% | 0.00% | 1.95% | 1.32% |
| **Fixed Dropout** | 250000 | Base | 0.00% | 0.47% | 0.02% | 0.51% |
| | | Rule out <THINK> | 0.68% | 1.09% | 0.11% | 0.73% |
| | | Only keeping $A_i$ | 2.23% | 1.09% | 2.19% | 1.10% |

Table 5. Recovery Success Rate For $n = 10, d = 5$ Cases (Rounded to 2 decimals) Random Guess Baseline: 1%

| Curriculum | # Training Samples | Query Method | Greedy Decoding | | Sampling ($T = 1$) | |
|---|---|---|---|---|---|---|
| | | | $\pi_1 - \pi_4$ | $\pi_5$ | $\pi_1 - \pi_4$ | $\pi_5$ |
| **Annealing Dropout** | 100000 | Base | 0.00% | 0.20% | 0.00% | 0.89% |
| | | Rule out <THINK> | 0.00% | 0.20% | 0.00% | 0.90% |
| | | Only keeping $A_i$ | 1.66% | 0.20% | 1.28% | 0.94% |
| **Annealing Dropout** | 250000 | Base | 0.00% | 1.80% | 0.00% | 1.12% |
| | | Rule out <THINK> | 0.00% | 1.80% | 0.00% | 1.13% |
| | | Only keeping $A_i$ | 3.05% | 1.80% | 1.72% | 1.23% |
| **Fixed Dropout** | 250000 | Base | 0.00% | 1.60% | 0.00% | 1.07% |
| | | Rule out <THINK> | 0.05% | 1.60% | 0.01% | 1.07% |
| | | Only keeping $A_i$ | 2.46% | 2.15% | 1.99% | 1.39% |
| **Fixed Dropout** | 500000 | Base | 0.26% | 2.05% | 0.05% | 1.13% |
| | | Rule out <THINK> | 0.33% | 2.05% | 0.07% | 1.13% |
| | | Only keeping $A_i$ | 2.11% | 2.05% | 1.91% | 1.34% |

Here we report the query success rate for a variaty of models trained with context-enhanced learning using different curriculum on different datasets sizes. All models reaches nearly perfect test accuracy when no context is provided (see Figure 2), i.e., the in context phrasebook rules played significant role in the learning of the models.

For all runs, we can see that it is nearly impossible (with recovery probability $< 0.1\%$ in most cases) to recover the correct phrasebook rules for hidden steps ($\pi_1, \ldots, \pi_4$) even when we provide the correct partial phrasebooks in context (see columns corresponding to Query Method "Base"). Ruling out <THINK> token when sampling also did not significantly increase the recovery rate. We note that the phrasebook knowledge **are not memorized in an completely undetectable manner**, as the strongest token filtering gives non-random probability of outputting the correct target 2-tuple. However the probability is still very low ($< 3\%$), which can be considered as negligible to recover the full correct phrasebooks $\text{STR}(\mathbf{\Pi}^*)$.

# D. Additional Related Works

**Differences with Masked Language Modeling (MLM) and Language Infilling**

MLM models like BERT, RoBERTa, and T5 (Kenton & Toutanova, 2019; Liu et al., 2019; Raffel et al., 2020) train models by either masking or removing tokens from a sequence and compute loss on the model's prediction on the missing tokens. This concept has been adapted for training auto-regressive models via language infilling task (Bavarian et al., 2022; Li et al., 2022; Donahue et al., 2020; Li et al., 2022). The primary difference from these works is that context-enhanced learning does not take loss on the context tokens when they are removed from our curriculum-text.

**Compositional and OOD generalization**

Measuring generalization for a transformer beyond training distribution has been a study of interest in many prior works. OOD generalization is measured by training a transformer on simpler examples and measuring its performance on harder ones. Prominent studies include length generalization, informally defined as the ability of the model to reason longer than what it has been trained on (Zhou et al., 2023; Anil et al., 2022), and compositional generalization on concepts, defined as the ability of the model to reason on composition of the concepts that it has seen during training (Press et al., 2022; Allen-Zhu & Li, 2023b; Ramesh et al., 2023; Yu et al., 2023; Zhao et al., 2024; Wang et al., 2024; Yang et al., 2024). Our experiments on **Fixed Dropout** in Section 3, where we train with $20\%$ dropout on the phrasebooks information but measure performance with $100\%$ dropout at test time, measures compositional OOD generalization behavior of the language model. The results show that the model internalizes the rules from the phrasebooks in an atomic way, and re-compose them together as necessary at test time.

**Mechanistic behavior of transformers with synthetic datasets:**

Our work builds on a growing body of research exploring the behavior of transformers trained on synthetic datasets. Prior studies have examined tasks such as modular addition (Nanda et al., 2023; Zhong et al., 2023), context-free grammars (Zhao et al., 2023; Allen-Zhu & Li, 2023a), regular and $n$-gram languages (Bhattamishra et al., 2020; Yao et al., 2021; Akyürek et al., 2024; Li et al., 2023), and synthetic article-style datasets (Allen-Zhu & Li, 2023b; 2024; Eldan & Li, 2023). While our work is structurally similar to these studies, it investigates mechanistic study on context-enhanced learning that has not been explored in previous works.

# E. Properties of MLT

**MLT**$(d, n)$ is defined by the phrasebooks $\pi_1, \cdots, \pi_d$ at each of its translation layers. We use $\mathcal{B}^\pi$ as all possible set of bijective maps that can be used to define the phrasebooks. We will use variable $\pi$ to refer to an arbitrary bijective map from the set $\mathcal{B}^\pi$. For simplicity of proof, we will refer to any alphabet set $A$ of size $n$ as $\{0, 1, \cdots, n-1\}$.

Here, we formally mention some of the properties of **MLT**$(d, n)$.

**Lemma E.1** (Invertibility of sequence translation). *For any level $i \in [d]$, fixing $\{\pi_i, \pi_{i+1}, \ldots, \pi_d\}$ gives a bijection between $s_i$ and $s_{d+1}$.*

*Proof.* All of the four operations involved in the mapping from $s_i$ to $s_{i+1}$ are invertible. $\qquad\square$

**Structure of the mappings:** The mappings $\pi_i$ are selected as random bijective maps between 2-tuples of characters in $A_i^2$ and 2-tuples of characters in $A_{i+1}^2$. A combinatorial argument can then give the number of such possible phrasebooks to be $n^2!$.

**Lemma E.2** (Number of mappings in each level). *The number of possible bijective maps between 2-tuples of characters from alpbhabet sets $A_i, A_{i+1}$, each being of size $n$, is $n^2!$, i.e. $|\mathcal{B}^\pi| = n^2!$.*

*Proof.* Each alphabet set contains $n^2$ possible 2-tuples of characters. If we fix an order in which the 2-tuples appear in $A_{i+1}^2$, then the number of bijective maps between $A_i^2$ and $A_{i+1}^2$ can be reduced to the number of possible ordering of the 2-tuples in $A_i^2$. The number of possible orderings is $n^2!$. $\qquad\square$

**Importance of *Circular shift*:** The composition of the $d$ random bijective maps can be demonstrated to result in another random bijective map. Consequently, without the *Circular shift*, each character in the output sequence $s_{d+1}$ depends on only two characters from the input sequence $s_1$ via a shared random map across all the 2-tuples.

Incorporating *Circular shift* on the other hand enables each character in the output sequence to depend on $2d$ characters from the input sequence. This is because the character positions are shifted to the right at each step, causing the input 2-tuples to the bijective map to also shift to the right at each stage. As we will elaborate later, incorporating *Circular shift* increases the required number of training samples to learn the set of phrasebooks from input and label pairs to $n^{\Omega(d)}$, whereas without *Circular shift*, this requirement is only $\mathcal{O}(n^2)$.

**Representing the mappings on 2-tuples in-context:** Each map can be defined using $\mathcal{O}(n^2)$ characters, as it can be simply defined by the $n^2$ relations each connecting 2 unique random 2-tuples from their corresponding alphabet sets. Thus, defining $d$ maps in-context will require $\mathcal{O}(n^2 d)$ characters in curriculum-text. In contrast, describing a completely random bijective mapping that maps $d$-tuples of characters in input sequence to a character in output sequence will require $\Omega(d\binom{n}{d})$ bits. Thus, **MLT** with $d$ translation steps involving random mappings on 2-tuples and *Circular shift* helps define a mapping where each character in the output sequence can depend on $d$ character in the input sequence, and the set of phrasebooks can be described using $\mathcal{O}(n^2 d)$ characters in curriculum-text.

# F. Lower Bound: Hardness of Learning MLT$(d, n)$ without Context

### F.1. Brief introduction to SQ framework

**Statistical query (SQ) bounds** : The statistical query (SQ) framework measures the computational hardness of learning a task in the presence of noise. It measures the hardness of learning a task by the number of statistical queries needed by a learning algorithm to learn the true function. Statistical queries are defined by some polynomially-computable property $Q$ of labeled instances and a tolerance parameter $\tau \in [0, 1]$ over $(x, y) \sim D$ where D is the data distribution. For a query, the algorithm receives a response from the oracle within $\tau$ error of the true value. The statistical dimension, or SQ-dim, is measured in terms of the number of functions in the hypothesis class that the learning algorithm needs to distinguish and the number of queries necessary to do the same. Correlation between two functions is used to define the statistical query dimension.

**Definition F.1.** Correlation of two functions $f_1, f_2$ on a domain $\mathcal{X}$ with respect to a distribution $\mathcal{D}$ is given by

$$\text{Correlation}(f_1, f_2, \mathcal{D}) := \left| \Pr_{x \in \mathcal{D}}[f_1(x) = f_2(x)] - \Pr_{x \in \mathcal{D}}[f_1(x) \neq f_2(x)] \right|.$$

For functions $f_1, f_2 : \mathcal{X} \to \{0, 1\}$, the above definition is also equivalent to

$$\text{Correlation}(f_1, f_2, \mathcal{D}) := |1 - 2\mathbb{E}_{x \in \mathcal{D}}[f_1(x) \oplus f_2(x)]|.$$

*Remark* F.2. Two functions $f_1, f_2 : \mathcal{X} \to \{0, 1\}$ are said to be uncorrelated w.r.t. $\mathcal{D}$ if

$$\Pr_{x \in \mathcal{D}}[f_1(x) = f_2(x)] = \Pr_{x \in \mathcal{D}}[f_1(x) \neq f_2(x)]$$

$$\text{(alternately)} \ \mathbb{E}_{x \in \mathcal{D}}[f_1(x) \oplus f_2(x)] = \frac{1}{2}.$$

On the other hand, if

$$Pr_{x \in \mathcal{D}}[f_1(x) = f_2(x)] = 1 \quad (\text{or } 0)$$

$$\text{(alternately)} \ \mathbb{E}_{x \in \mathcal{D}}[f_1(x) \oplus f_2(x)] = 0 \quad (\text{or } 1),$$

then $\text{Correlation}(f_1, f_2, \mathcal{D}) = 1$.

We take the following formal definitions of SQ-dim and its relation to computational hardness in the SQ framework from (Blum et al., 1994).

**Definition F.3** (Definition 2 in Blum et al. (1994)). For a function class $\mathcal{F}$ of boolean functions over $\{0, 1\}^n$ and $\mathcal{D}$ a distribution over $\{0, 1\}^n$, SQ-dim$(\mathcal{F}, \mathcal{D})$, the statistical query dimension of $\mathcal{F}$ with respect to $\mathcal{D}$, is defined to be the largest natural number $\mu$ such that $\mathcal{F}$ contains $\mu$ functions $f_1, \cdots, f_\mu$ with the property that for all $i \neq j$ we have:

$$\text{Correlation}(f_i, f_j, \mathcal{D}) := \left| \Pr_{x \in \mathcal{D}} [f_i = f_j] - \Pr_{x \in \mathcal{D}} [f_i \neq f_j] \right| \leq \frac{1}{\mu^3}.$$

**Theorem F.4** (Theorem 12 in Blum et al. (1994)). *Let $\mathcal{F}$ be a class of functions $\{0, 1\}^n$ and $\mathcal{D}$ a distribution such that SQ-dim$(\mathcal{F}, \mathcal{D}) \geq \mu \geq 16$. Then if all queries are made with a tolerance of atleast $\frac{1}{\mu^{1/3}}$, at least $\mu^{1/3}/2$ queries are required to learn $\mathcal{F}$ with error less than $1/2 - 1/\mu^3$ in the statistical query model.*

### F.2. Lower bound lemma

Here, we mention the 2 main theorems that study the SQ dimension of the **MLT** task at hand. The first theorem shows the SQ dimension bound when number of characters $n = 2$, which is then adapted to get the SQ dimension bound for general $n$. Proofs of both the theorems are given in Appendix F.4 and Appendix F.5 respectively.

**Theorem F.10** (SQ dimension for $n = 2$). *The family of translation task **MLT**$(d, 2)$ on input distribution $\mathcal{U}(\{0, 1\}^{2d})$ has statistical query dimension SQ-dim(**MLT**$(d, 2)$) atleast $2^{\Omega(d)}$.*

**Theorem F.20** (SQ dimension for general $n$). *For the translation task **MLT**$(d, n)$ that has depth $d$ and $n$ characters per level, the statistical query dimension SQ-dim(**MLT**$(d, n)$) is atleast $n^{\Omega(d)}$.*

**Notations:** We will require the following notations for proving the above theorems. First, we will denote a 2-tuple that contains arbitrary characters $a$ and $b$ as $(a, b)$. For a bijective map $\pi$, $\pi((a, b))_i$ will represent $i$th character in the output of $\pi$ on any 2-tuple input $(a, b)$ and any $i \in \{1, 2\}$. Similarly, for any set of phrasebooks $\mathbf{\Pi}$, we will use **MLT**$_{\mathbf{\Pi}}(s_1)_i$ to denote the $i$th character in the output of **MLT**$_{\mathbf{\Pi}}$ on any $L$ length input sequence $s_1$ and any $i \in [1, L]$.

**Recall that** for any input sequence $s_1$, the output of the $i$th level of a **MLT** task will be denoted as $s_i$. Furthermore, to denote the $j$th character (arbitrary) in the sequence $s_i$, we will use $s_{i,j}$.

#### F.2.1. BOUNDS FOR SGD

The following corollary measures the sample complexity needed to learn **MLT**$_{\mathbf{\Pi}^*}$ by SGD. It has been adapted from proposition 3 in Edelman et al. (2023), who study the sample complexity for learning $d$-sparse parity task on $n$ length sequences, whose SQ dimension is $\binom{n}{d} = \Theta(n^d)$. We simply state the corollary without specifying the proof.

**Loss function and SGD updates:** Consider training of a model $f_\theta$ with $r$ parameters that is trained with mean squared error, i.e. $\ell_{\text{auto}}(f_\theta([\text{CURR}_g(x, t), x, y]), y) = \|f_\theta([\text{CURR}_g(x, t), x, y]) - y\|^2$ [7]. The empirical loss on a batch $S$ of batch size $B$ from the supervised dataset $D_{\mathbf{\Pi}^*}$ will be denoted by $L_S(f_\theta, \mathbf{MLT}_{\mathbf{\Pi}^*}) = \mathbb{E}_{(x,y) \sim S} \ell_{\text{auto}}(f_\theta([\text{CURR}_g(x, t), x, y]), y) = \|f_\theta([\text{CURR}_g(x, t), x, y]) - y\|^2$; the population loss will be denoted by $L_{\mathcal{U}(\{0,1\}^L)}(f_\theta, \mathbf{MLT}_{\mathbf{\Pi}^*})$. SGD updates are of the form:

$$\theta_{t+1} = \theta_t - \eta_t (\nabla_\theta L_{S_t}(f_{\theta_t}, \mathbf{MLT}_{\mathbf{\Pi}^*}) + R(\theta_t) + \zeta_t).$$

for some sample $S_t$, step size $\eta_t$, regularizer $R(\cdot)$, and adversarial noise $\zeta_t \in [-\tau, \tau]^r$. For simplicity, we assume the gradient $\nabla_\theta L_{S_t}(\theta_t)$ is bounded on all parameters $\theta$ in parameter space of the model.

**Fake trajectory:** Suppose $\mathbf{0}$ denote the constant function that maps all inputs to 0. Then, with the contemporary losses $L_S(f_\theta, \mathbf{0})$ and $L_{\mathcal{U}(\{0,1\}^L)}(f_\theta, \mathbf{0})$ that computes difference from this constant function, consider the following trajectory $\tilde{\theta}_1, \cdots, \tilde{\theta}_t, \cdots$ starting from the same initiation $\theta_0$:

$$\tilde{\theta}_{t+1} = \tilde{\theta}_t - \eta_t (\nabla_\theta L_{S_t}(f_{\tilde{\theta}_t}, \mathbf{0}) + R(\tilde{\theta}_t)).$$

**Assumption F.5.** For all $t$, suppose $\left\| \nabla_\theta L_{\mathcal{U}(\{0,1\}^L)}(f_{\tilde{\theta}_t}, \mathbf{0}) - \nabla_\theta L_{S_t}(f_{\tilde{\theta}_t}, \mathbf{0}) \right\|_2 \leq \tau/2$.

**Corollary F.6** (Sample complexity bounds for SGD). *Fix an initialization $\theta_0$ such that $f_{\theta_0}$ is statistically independent (correlation 0) from all possible **MLT**$_{\mathbf{\Pi}^*}$. Under this assumption, if $\frac{LTB}{\tau^2} \leq n^{\Omega(d)}/r$, then there exists at least one task*

---

[7]The results holds for any loss that satisfies $\frac{\partial \ell_{\text{auto}}(y', y)}{\partial y'} = -y + \frac{\partial \ell_{\text{auto}}(y', 0)}{\partial y'}$ .

$\mathbf{MLT_{\Pi^*}} \in \mathbf{MLT}(d, n)$ *for which the functions obtained after the first $T$ SGD updates, $f_{\theta_1}, \ldots, f_{\theta_T}$, remain statistically independent (correlation 0) from* $\mathbf{MLT_{\Pi^*}}$ *despite training on it.*

$LTB$ is the product of input length, total training steps, and batch size, which represents sample complexity used by the SGD algorithm.

### F.3. Useful lemmas for $n = 2$

Here, we mention some useful lemmas for charaterizing the bijective maps on $\{0, 1\}^2 \rightarrow \{0, 1\}^2$ that we will regularly use for the proof of Theorem F.10. The first lemma will show that each bijective map can be represented by three operations on input characters, **copy**, **xor**, and **not** operations. The second lemma measures correlation on the outputs of two randomly sampled bijective maps. The proofs are given in Appendix F.6.

We define the following necessary operations to describe the random bijective maps on $\{0, 1\}^2 \rightarrow \{0, 1\}^2$:

1. **copy**: Given a tuple $(x_1, x_2)$, and a position argument $i \in \{1, 2\}$, the operation returns value of $x_i$ as output. We will use **copy**$_1$ and **copy**$_2$ to indicate the **copy** operation on positions 1 and 2 respectively.

2. **not**: Given a variable $x_i$, this operation returns flipped value of $x_i$. That is, if $x_i = 1$, then it returns 0 and vice-versa.

3. **xor**: Given a tuple $(x_1, x_2)$, this operation returns $x_1 \oplus x_2$.

**Lemma F.21** (Formulation of bijective maps for $n = 2$). *Any bijective map $\pi : \{0, 1\}^2 \rightarrow \{0, 1\}^2$ can be expressed using* **copy**, **not**, *and* **xor** *operations. Furthermore, from* 24 *possible maps for $\pi$,*

1. *There are 6 maps $\mathbf{\Delta}_1, \mathbf{\Delta}_2, \cdots, \mathbf{\Delta}_6$ for which characters in the output tuple can be defined by* **copy** *and* **xor** *operations on the characters in the input tuple.*

2. *__Mirror maps:__ For each map $\pi \in \{\mathbf{\Delta}_1, \mathbf{\Delta}_2, \cdots, \mathbf{\Delta}_6\}$, there exist mirror maps $\pi_{(1)}, \pi_{(2)}, \pi_{(3)}$ whose output on each input tuple can be defined by selective* **not** *operations on either or both characters of the output tuple of $\pi$. We call $\{\pi, \pi_{(1)}, \pi_{(2)}, \pi_{(3)}\}$ as a __mirror map set__ of $\pi$, in short, Mirrorset($\pi$).*

*Remark* F.7. Mirror map set definition is general and isn't restricted to maps $\pi \in \{\mathbf{\Delta}_1, \mathbf{\Delta}_2, \cdots, \mathbf{\Delta}_6\}$. Because mirror maps are defined in terms of **not** operation, a mirror map set Mirrorset($\pi$) for $\pi \in \{\mathbf{\Delta}_1, \mathbf{\Delta}_2, \cdots, \mathbf{\Delta}_6\}$ is also equivalent to Mirrorset($\pi_{(i)}$) for $i \in \{1, 2, 3\}$.

> *Remark* F.8. Lemma F.21 can be re-stated as follows: the 24 possible bijective maps $\{0, 1\}^2 \rightarrow \{0, 1\}^2$ can be grouped into 6 family of maps, each containing 4 maps and represented by a unique map from $\{\mathbf{\Delta}_1, \mathbf{\Delta}_2, \cdots, \mathbf{\Delta}_6\}$. Each family is defined by **mirror map set** of their corresponding representative map.

**Lemma F.22** (Correlation of bijective maps for $n = 2$). *For two randomly selected maps $\pi^\alpha, \pi^\beta : \{0, 1\}^2 \rightarrow \{0, 1\}^2$, the following hold true.*

1. *__Correlation at atleast one output character pair:__ Fix an $i, j \in \{0, 1\}$. With probability $\frac{1}{3}$ w.r.t. the random selection of the maps, the $i$th character in the output of $\pi^\alpha$ has perfect correlation to $j$th character in the output of $\pi^\beta$, i.e.*

$$Correlation(\pi^\alpha(\cdot)_i, \pi^\beta(\cdot)_j, \mathcal{U}(\{0, 1\}^2)) := \left| \Pr_{x \sim \mathcal{U}(\{0,1\}^2)} [\pi^\alpha(x)_i \neq \pi^\beta(x)_j] - \Pr_{x \sim \mathcal{U}(\{0,1\}^2)} [\pi^\alpha(x)_i = \pi^\beta(x)_j] \right| = 1.$$

2. *__Correlation at both output character pairs:__ With probability $\frac{1}{3}$ w.r.t. the random selection of the maps, both the characters in the outputs of $\pi^\alpha, \pi^\beta$ have perfect correlation, i.e. either one of the cases hold true*

$$Correlation(\pi^\alpha(\cdot)_1, \pi^\beta(\cdot)_1, \mathcal{U}(\{0, 1\}^2)) = 1.$$
$$Correlation(\pi^\alpha(\cdot)_2, \pi^\beta(\cdot)_2, \mathcal{U}(\{0, 1\}^2)) = 1.$$

*or*

$$Correlation(\pi^\alpha(\cdot)_1, \pi^\beta(\cdot)_2, \mathcal{U}(\{0,1\}^2)) = 1.$$
$$Correlation(\pi^\alpha(\cdot)_2, \pi^\beta(\cdot)_1, \mathcal{U}(\{0,1\}^2)) = 1.$$

*Other cases are not possible, i.e. for any $i \in \{1,2\}$, both $Correlation(\pi^\alpha(\cdot)_i, \pi^\beta(\cdot)_1, \mathcal{U}(\{0,1\}^2)) = 1$ and $Correlation(\pi^\alpha(\cdot)_i, \pi^\beta(\cdot)_2, \mathcal{U}(\{0,1\}^2)) = 1$ can't hold true.*

*Remark* F.9. Implication of Lemma F.22 is as follows: With probability at most $\frac{1}{3}$, output of two random maps can stay correlated at either one output character pair or both pairs of output characters. This would suggest that the probability of the output of **MLT** under two random set of phrasebooks staying correlated should decay exponentially with the depth of the task.

**F.4. Proof for Statistical dimension lower bound for $n = 2$**

We repeat the lemma of interest for presentation.

**Theorem F.10** (SQ dimension for $n = 2$). *The family of translation task **MLT**$(d, 2)$ on input distribution $\mathcal{U}(\{0,1\}^{2d})$ has statistical query dimension SQ-dim(**MLT**$(d, 2)$) atleast $2^{\Omega(d)}$.*

*Proof.* We narrow our argument to the first character output of the task. The proof goes through 2 major steps.

1. First, we show that two random instances of **MLT**$(d, 2)$, defined by 2 set of phrasebooks $\{\pi_1^\alpha, \cdots, \pi_d^\alpha\}$ and $\{\pi_1^\beta, \cdots, \pi_d^\beta\}$, will be uncorrelated with probability at least $1 - (1/3)(7/9)^{d-1}$ w.r.t. the selection of the random set of phrasebooks. The lemma is formally given in Lemma F.13.

2. Then, we can apply Lovász local lemma (Theorem F.11) to show that we can create $2^{\Omega(d)}$ instances of **MLT**$(d, 2)$ that will have zero pairwise correlation of their output (Corollary F.12).

By the definition of SQ-dim from Definition F.3, the above observations suggest that SQ-dim(**MLT**$(d, 2)$) $\geq 2^{\Omega(d)}$. $\qquad\square$

**Theorem F.11** (Lovász local lemma, theorem 1.5 in Spencer (1977)). *Let $A_1, \cdots, A_k$ be events in some probability space with $\Pr(A_i) \leq p$, $1 \leq i \leq k$, such that each event is dependent on atmost $k_0$ other events. If $ep(k_0 + 1) < 1$, then $\Pr(\neg A_1 \wedge \neg A_2 \wedge \neg A_3 \cdots \wedge \neg A_k) > 0$.*

**Corollary F.12** (Number of uncorrelated instances of **MLT**$(d, 2)$). *There exists a set of $2^{\Omega(d)}$ instances in **MLT**$(d, 2)$ that are pairwise uncorrelated to each other on input distribution $\mathcal{U}(\{0,1\}^{2d})$.*

*Proof.* Suppose $\mathbf{\Pi}_1, \cdots, \mathbf{\Pi}_k$ represent $k$ randomly sampled set of phrasebooks. For each $1 \leq i < j \leq k$, we will denote event $A_{ij}$ as the event that the outputs of **MLT**$_{\mathbf{\Pi}_i}$ and **MLT**$_{\mathbf{\Pi}_j}$ are uncorrelated on input distribution $\mathcal{U}(\{0,1\}^{2d})$. This happens with probability $p = 1 - (1/3)(7/9)^{d-1}$ from Lemma F.13. Because each event can atmost depend on atmost $k(k+1)/2 < k^2$ events (total number of events), by Theorem F.11, if

$$ep(k^2 + 1) < 1,$$

then there exists a list of such set of phrasebooks which are pairwise uncorrelated w.r.t. the outputs of their corresponding translation tasks on input distribution $\mathcal{U}(\{0,1\}^{2d})$. Solving the above for $k$, we can set $k = ((ep)^{-1} - 1)^{1/2} = 2^{\Omega(d)}$. $\quad\square$

F.4.1. AUXILIARY LEMMAS

Here, we will prove the following primary lemma, that is used to prove Theorem F.10.

---

**Lemma F.13.** *With probability at least* $1 - \frac{1}{3}\left(\frac{7}{9}\right)^{d-1}$ *w.r.t. random map selection, the following holds true for* 2 *sets of random phrasebooks* $\mathbf{\Pi}^{\alpha} = \{\pi_1^{\alpha}, \cdots, \pi_d^{\alpha}\}$ *and* $\mathbf{\Pi}^{\beta} = \{\pi_1^{\beta}, \cdots, \pi_d^{\beta}\}$:

$$Correlation(\mathbf{MLT}_{\mathbf{\Pi}^{\alpha}=\{\pi_{\ell}^{\alpha}\}_{\ell=1}^d}(\cdot)_1, \mathbf{MLT}_{\mathbf{\Pi}^{\beta}=\{\pi_{\ell}^{\beta}\}_{\ell=1}^d}(\cdot)_1, \mathcal{U}(\{0,1\}^{2d})) = 0.$$

---

*Proof.* We first dive into the dependencies between characters in input and output sequences in the translation process. In Lemma F.15, we show that at any level $1 \leq \ell \leq d$ of $\mathbf{MLT}_{\mathbf{\Pi}}$ with phrasebooks $\mathbf{\Pi} = \{\pi_{\ell}\}_{\ell=1}^d$, the following relation holds true on an input $s_1 \in \{0,1\}^{2d}$:

$$(s_{\ell+1,1}, s_{\ell+1,2}) = \pi_{\ell}((s_{\ell+1,2}, s_{\ell+1,3})).$$

We will use superscripts $\alpha$ and $\beta$ to differentiate the intermediate outputs of $\mathbf{MLT}$ when the phrasebooks are set as $\mathbf{\Pi}^{\alpha} = \{\pi_1^{\alpha}, \cdots, \pi_d^{\alpha}\}$ and $\mathbf{\Pi}^{\beta} = \{\pi_1^{\beta}, \cdots, \pi_d^{\beta}\}$ respectively. We will use an induction strategy to find the correlation of $s_{d+1,1}^{\alpha}$ and $s_{d+1,1}^{\beta}$. To do so, we will require the following variables:

1. $p_{\ell,i;j}$: For one pair of $i,j \in \{2,3\}$, this represents the probability at a level $2 \leq \ell \leq 1+d$ w.r.t. the randomness of $\{\pi_1^{\alpha}, \cdots, \pi_{\ell-1}^{\alpha}\}$ and $\{\pi_1^{\beta}, \cdots, \pi_{\ell-1}^{\beta}\}$, that $s_{\ell,i}^{\alpha}$ and $s_{\ell,j}^{\beta}$ are perfectly correlated. Additionally, we define $p_{\ell,1;1}$ that represents the probability at a level $1 \leq \ell \leq 1+d$ w.r.t. the randomness of $\{\pi_1^{\alpha}, \cdots, \pi_{\ell-1}^{\alpha}\}$ and $\{\pi_1^{\beta}, \cdots, \pi_{\ell-1}^{\beta}\}$, that $s_{\ell,1}^{\alpha}$ and $s_{\ell,1}^{\beta}$ are perfectly correlated.

2. $p_{\ell,\text{both}}$: This represents the probability at a level $2 \leq \ell \leq 1+d$ w.r.t. the randomness of $\{\pi_1^{\alpha}, \cdots, \pi_{\ell-1}^{\alpha}\}$ and $\{\pi_1^{\beta}, \cdots, \pi_{\ell-1}^{\beta}\}$, that either $s_{\ell,2}^{\alpha}, s_{\ell,2}^{\beta}$ are correlated and $s_{\ell,3}^{\alpha}, s_{\ell,3}^{\beta}$ are correlated, or $s_{\ell,2}^{\alpha}, s_{\ell,3}^{\beta}$ are correlated and $s_{\ell,3}^{\alpha}, s_{\ell,2}^{\beta}$ are perfectly correlated.

There are three relations that we need to keep in mind, before we proceed with the induction proof.

1. First, the correlations are circular in nature, i.e., for $j \in \{2,3\}$ and any integer $k$, $p_{\ell,j;j} = p_{\ell,j;(j+2k)\%2d}$ (Lemma F.17). We will use this relation to connect $p_{\ell,1;1}$ with $p_{\ell,3;3}$, i.e.

$$p_{\ell,1;1} = p_{\ell,3;3}. \tag{1}$$

2. Second, $p_{\ell,\text{both}} \leq \max_{ij} p_{\ell,i;j}$. Intuitively, this is because getting correlations at both positions must be at most as probable as getting correlations at one of the positions.

$$p_{\ell,\text{both}} \leq \max_{i,j} p_{\ell,i;j}. \tag{2}$$

3. Third, cross-correlation probabilities given by $p_{\ell,2;3}$ and $p_{\ell,3;2}$ must be $\leq \frac{1}{2}(p_{\ell,2;2} + p_{\ell,3;3})$ (Lemma F.19).

$$p_{\ell,i;j} \leq \frac{1}{2}(p_{\ell,i;i} + p_{\ell,j;j}), \text{ for any } i,j \in \{2,3\}, i \neq j. \tag{3}$$

**Base condition:** We will use the values of the variables at $\ell = 2$ as our base condition. The input for first layer of translation is same to both $\mathbf{MLT}_{\mathbf{\Pi}^{\alpha}}, \mathbf{MLT}_{\mathbf{\Pi}^{\beta}}$. Then, for an input $s_1$,

$$(s_{2,1}^{\alpha}, s_{2,2}^{\alpha}) = \pi_1^{\alpha}((s_{1,2}, s_{1,3}))$$
$$(s_{2,1}^{\beta}, s_{2,2}^{\beta}) = \pi_1^{\beta}((s_{1,2}, s_{1,3})).$$

We can then use Lemma F.22 to show that

$$p_{2,i;j} = \frac{1}{3}, \text{for all } i,j \in \{1,2\},$$
$$p_{2,\text{both}} = \frac{1}{3}.$$

**Connecting the variables at $\ell$ and $\ell + 1$:** We connect $p_{\ell+1,2;2}$ to $p_{\ell,2;3}, p_{\ell,3;2}, p_{\ell,3;3}$ and $p_{\ell,\text{both}}$. This will undergo a case by case analysis.

1. First, if under phrasebooks $\pi_1^\alpha, \cdots, \pi_{\ell-1}^\alpha$ and $\pi_1^\beta, \cdots, \pi_{\ell-1}^\beta$, if

    (either) $s_{\ell,2}^\alpha$ and $s_{\ell,2}^\beta$ are perfectly correlated , and $s_{\ell,3}^\alpha$ and $s_{\ell,3}^\beta$ are perfectly correlated

    (or) $s_{\ell,2}^\alpha$ and $s_{\ell,3}^\beta$ are perfectly correlated , and $s_{\ell,3}^\alpha$ and $s_{\ell,2}^\beta$ are perfectly correlated , $\quad\quad\quad$ (4)

    we can use Lemma F.14 to show that with probability $\frac{1}{3}$ w.r.t. the selection of $\pi_\ell^\alpha$ and $\pi_\ell^\beta$, $s_{\ell+1,2}^\alpha$ and $s_{\ell+1,2}^\beta$ will be correlated. Conditions necessary for Lemma F.14, i.e. uniform distribution of the characters in the input sequence, are shown to hold true in Lemma F.16. The probability w.r.t. the selection of phrasebooks $\pi_1^\alpha, \cdots, \pi_{\ell-1}^\alpha$ and $\pi_1^\beta, \cdots, \pi_{\ell-1}^\beta$ such that Equation (4) holds true is given by the variable $p_{\ell,\text{both}}$.

2. On the other hand, if there exists a pair $i, j \in \{2, 3\}$, such that under phrasebooks $\pi_1^\alpha, \cdots, \pi_{\ell-1}^\alpha$ and $\pi_1^\beta, \cdots, \pi_{\ell-1}^\beta$, $s_{\ell,i}^\alpha$ and $s_{\ell,j}^\beta$ are perfectly correlated, then with probability $\frac{1}{9}$ w.r.t. the selection of $\pi_\ell^\alpha$ and $\pi_\ell^\beta$, $s_{\ell+1,2}^\alpha$ and $s_{\ell+1,2}^\beta$ will be correlated. We again refer to Lemma F.14 for this statement. Probability that this condition happens under phrasebooks $\pi_1^\alpha, \cdots, \pi_{\ell-1}^\alpha$ and $\pi_1^\beta, \cdots, \pi_{\ell-1}^\beta$ is given by the variable $p_{\ell,i;j}$.

3. If none of the above situation occurs, then for all pairs $i, j \in \{2, 3\}$, $s_{\ell,i}^\alpha$ and $s_{\ell,j}^\beta$ are uncorrelated, and so, by Lemma F.22, correlation between $s_{\ell+1,2}^\alpha$ and $s_{\ell+1,2}^\beta$ will stay 0 for any choice of $\pi_\ell^\alpha$ and $\pi_\ell^\beta$.

Thus, combining the 3 cases, we must have

$$p_{\ell+1,2;2} \le \frac{1}{3}p_{\ell,\text{both}} + \frac{1}{9}\sum_{i,j\in\{2,3\}} p_{\ell,i;j} \quad\quad\quad (5)$$

$$\le \frac{1}{3}p_{\ell,\text{both}} + \frac{4}{9}\max_{i,j\in\{2,3\}} p_{\ell,i;j}$$

$$\le \frac{1}{3}\max_{i,j\in\{2,3\}} p_{\ell,i;j} + \frac{4}{9}\max_{i,j\in\{2,3\}} p_{\ell,i;j} = \frac{7}{9}\max_{i,j\in\{2,3\}} p_{\ell,i;j} \quad\quad\quad (6)$$

Reasoning for each step is as follows:

1. Equation (5) follows from conditional probability computations using the 3 cases that we discussed before.

2. Equation (6) uses Equation (2) to connect $p_{\ell,\text{both}}$ to $p_{\ell,i;j}$.

We can give the same inequality for $p_{\ell+1,1;1}$. As $p_{\ell+1,1;1} = p_{\ell+1,3;3}$ from Equation (1) and $p_{\ell,2;3}$ and $p_{\ell,3;2}$ must be atmost the average of $p_{\ell,2;2}$ and $p_{\ell,3;3}$ (Equation (3)), we can write

$$\max_{i,j\in\{2,3\}} p_{\ell+1,i;j} \le \frac{7}{9}\max_{i,j\in\{2,3\}} p_{\ell,i;j}.$$

This implies that the probability of correlation at any output character under the two random set of phrasebooks decays at the rate of $\frac{7}{9}$. Solving the recurrence will give:

$$\max_{i,j\in\{2,3\}} p_{d+1,i;j} \le \left(\frac{7}{9}\right)^{d-1} p_{2,i;j} = \left(\frac{7}{9}\right)^{d-1} \frac{1}{3}.$$

As $p_{d+1,3;3}$ should be equal to $p_{d+1,1;1}$, this implies that with probability at least $1 - p_{d+1,3;3} = 1 - \left(\frac{7}{9}\right)^{d-1}\frac{1}{3}$, the following must hold true:

$$\text{Correlation}(\mathbf{MLT}_{\mathbf{\Pi}^\alpha}(\cdot)_1, \mathbf{MLT}_{\mathbf{\Pi}^\beta}(\cdot)_1, \mathcal{U}(\{0,1\}^{2d})) = 0.$$

$\square$

**Lemma F.14.** *Suppose $f, g : \{0,1\}^k \to \{0,1\}^2$ denote 2 functions for some arbitrary $k$, satisfying conditions for uniformity in output distribution, i.e.*

$$\mathbb{E}_{x \sim \mathcal{U}(\{0,1\}^k)} f(x)_j = 1/2, \quad \mathbb{E}_{x \sim \mathcal{U}(\{0,1\}^k)} g(x)_j = 1/2$$

$$\mathbb{E}_{x \sim \mathcal{U}(\{0,1\}^k)} f(x)_1 \oplus f(x)_2 = 1/2, \quad \mathbb{E}_{x \sim \mathcal{U}(\{0,1\}^k)} g(x)_1 \oplus g(x)_2 = 1/2$$

*for all $j \in \{1,2\}$. Then, the following holds true:*

1. **Both output character pairs have correlations between $f$ and $g$:** *If*

   *(either) $Correlation(f(\cdot)_1, g(\cdot)_1, \mathcal{U}(\{0,1\}^k)) = 1$ and $Correlation(f(\cdot)_2, g(\cdot)_2, \mathcal{U}(\{0,1\}^k)) = 1$*

   *(or) $Correlation(f(\cdot)_1, g(\cdot)_2, \mathcal{U}(\{0,1\}^k)) = 1$ and $Correlation(f(\cdot)_2, g(\cdot)_1, \mathcal{U}(\{0,1\}^k)) = 1$,*

   *then for two randomly picked phrasebooks $\pi^\alpha, \pi^\beta$,*

   (a) **Correlation of an output character pair of $\pi^\alpha(f(\cdot))$ and $\pi^\beta(g(\cdot))$:** *Fix an $i, j \in \{1,2\}$. With probability $1/3$ w.r.t. the random selections of $\pi^\alpha, \pi^\beta$,*

   $$Correlation(\pi^\alpha(f(\cdot))_i, \pi^\alpha(g(\cdot))_j, \mathcal{U}(\{0,1\}^k)) = 1$$

   (b) **Correlation of both character pairs in output of $\pi^\alpha(f(\cdot))$ and $\pi^\beta(g(\cdot))$:** *With probability $1/3$ w.r.t. the random selections of $\pi^\alpha, \pi^\beta$, one of the following two conditions hold true.*

   *(either) $Correlation(\pi^\alpha(f(\cdot))_1, \pi^\alpha(g(\cdot))_1, \mathcal{U}(\{0,1\}^k)) = 1$ and $Correlation(\pi^\alpha(f(\cdot))_2, \pi^\alpha(g(\cdot))_2, \mathcal{U}(\{0,1\}^k)) = 1$*

   *(or) $Correlation(\pi^\alpha(f(\cdot))_1, \pi^\alpha(g(\cdot))_2, \mathcal{U}(\{0,1\}^k)) = 1$ and $Correlation(\pi^\alpha(f(\cdot))_2, \pi^\alpha(g(\cdot))_1, \mathcal{U}(\{0,1\}^k)) = 1$*

2. **Only a pair of characters have correlations between $f$ and $g$:** *If there exists only one pair $i, j \in \{0,1\}$ such that*

   $$Correlation(f(\cdot)_i, g(\cdot)_j, \mathcal{U}(\{0,1\}^k)) = 1,$$

   *and for all other $i', j'$, $Correlation(f(\cdot)_{i'}, g(\cdot)_{j'}, \mathcal{U}(\{0,1\}^k)) = 0$, then*

   (a) *Fix any $i', j' \in \{1,2\}$. With probability $\frac{1}{9}$ w.r.t. the random selections of $\pi^\alpha, \pi^\beta$,*

   $$Correlation(\pi^\alpha(f(\cdot))_{i'}, \pi^\beta(g(\cdot))_{j'}, \mathcal{U}(\{0,1\}^k)) = 1.$$

   *If the above condition holds true, for all other $\bar{i}, \bar{j}$ pairs,*

   $$Correlation(\pi^\alpha(f(\cdot))_{\bar{i}}, \pi^\beta(g(\cdot))_{\bar{j}}, \mathcal{U}(\{0,1\}^k)) = 0.$$

3. **No correlations between $f$ and $g$:** *If for all pairs $i, j \in \{1,2\}$, $Correlation(f(\cdot)_i, g(\cdot)_j, \mathcal{U}(\{0,1\}^k)) = 0$, then for all pairs $i', j' \in \{1,2\}$,*

   $$Correlation(\pi^\alpha(f(\cdot))_{i'}, \pi^\beta(g(\cdot))_{j'}, \mathcal{U}(\{0,1\}^k)) = 0.$$

*Proof.* We will prove each case separately.

1. **Both output character pairs have correlations between $f$ and $g$:** We will consider the case when $Correlation(f(\cdot)_1, g(\cdot)_1, \mathcal{U}(\{0,1\}^k)) = 1$ and $Correlation(f(\cdot)_2, g(\cdot)_2, \mathcal{U}(\{0,1\}^k)) = 1$, proof for the other case is similar. Then, there are 4 cases possible.

   Subcase 1: $f(x)_1 = g(x)_1, f(x)_2 = g(x)_2 \quad$ for all $x \in \{0,1\}^k$

   Subcase 2: $f(x)_1 = \textbf{not}(g(x)_1), f(x)_2 = g(x)_2 \quad$ for all $x \in \{0,1\}^k$

   Subcase 3: $f(x)_1 = g(x)_1, f(x)_2 = \textbf{not}(g(x)_2) \quad$ for all $x \in \{0,1\}^k$

   Subcase 4: $f(x)_1 = \textbf{not}(g(x)_1), f(x)_2 = \textbf{not}(g(x)_2) \quad$ for all $x \in \{0,1\}^k$

Because $f(\cdot)$ and $g(\cdot)$ are inputs to the phrasebooks $\pi^\alpha$ and $\pi^\beta$ respectively and $f(\cdot)$ and $g(\cdot)$ are related by selective **not** operations in their outputs in each of the possible 4 subcases, the output of $\pi^\beta(g(\cdot))$ can be replaced by $\tilde{\pi}^\beta(f(\cdot))$ for a mirror map $\tilde{\pi}^\beta$ of $\pi^\beta$. We showcase this formally for one subcase, say subcase 2; argument for other subcases are similar.

**Suppose Subcase 2 is true:** Then, for any two phrasebooks $\pi^\alpha$ and $\pi^\beta$, suppose $\tilde{\pi}^\beta$ denotes a mirror map of $\pi^\beta$ such that

$$\tilde{\pi}^\beta(x)_1 = \textbf{not}(\pi^\alpha(x)_1), \; \tilde{\pi}^\beta(x)_2 = \pi^\alpha(x)_2, \quad \text{for all } x \in \{0,1\}^2.$$

Such a map exists by Lemma F.21. Then, for a pair $i, j \in \{1, 2\}$,

$$\begin{aligned}
&\text{Correlation}(\pi^\alpha(f(\cdot))_i, \pi^\beta(g(\cdot))_j, \mathcal{U}(\{0,1\}^k)) \\
&= \text{Correlation}(\pi^\alpha((f(\cdot)_1, f(\cdot)_2))_i, \pi^\beta((g(\cdot)_1, g(\cdot)_2))_j, \mathcal{U}(\{0,1\}^k)) && (7) \\
&= \text{Correlation}(\pi^\alpha((f(\cdot)_1, f(\cdot)_2))_i, \pi^\beta((\textbf{not}(f(\cdot)_1), f(\cdot)_2))_j, \mathcal{U}(\{0,1\}^k)) && (8) \\
&= \text{Correlation}(\pi^\alpha((f(\cdot)_1, f(\cdot)_2))_i, \tilde{\pi}^\beta((f(\cdot)_1, f(\cdot)_2))_j, \mathcal{U}(\{0,1\}^k)) && (9) \\
&= \text{Correlation}(\pi^\alpha(\cdot)_i, \tilde{\pi}^\beta(\cdot)_j, \mathcal{U}(\{0,1\}^2)). && (10)
\end{aligned}$$

Reasoning for each step is as follows.

- Because the outputs of $f$ and $g$ are a 2-tuple on any input, Equation (7) simply replaces $f(x)$ (similarly $g(x)$) as $(f(x)_1, f(x)_2)$ for any input $x$.
- Equation (8) uses the relation between the outputs of $f$ and $g$ when subcase 2 is true.
- Equation (9) simply replaces $\pi^\beta$ with $\tilde{\pi}^\beta$ due to their relation as mirror phrasebooks.
- Finally, because of the assumption on the output of $f$, i.e. $\mathbb{E}_{x \sim \mathcal{U}(\{0,1\}^k)} f(x)_j = 1/2, \mathbb{E}_{x \sim \mathcal{U}(\{0,1\}^k)} f(x)_1 \oplus f(x)_2 = 1/2$, the distribution of outputs of $f$ is identical to $\mathcal{U}(\{0,1\}^2)$.

Hence, $\text{Correlation}(\pi^\alpha(f(\cdot))_i, \pi^\beta(g(\cdot))_j, \mathcal{U}(\{0,1\}^k))$ boils down to $\text{Correlation}(\pi^\alpha(\cdot)_i, \tilde{\pi}^\beta(\cdot)_j, \mathcal{U}(\{0,1\}^2))$. We can then use Lemma F.22 to show the probabilities of each of the desired conditions.

2. **Only a pair of characters in output have correlations between $f$ and $g$:** For typographical simplicity, consider the case when

$$\text{Correlation}(f(\cdot)_1, g(\cdot)_1, \mathcal{U}(\{0,1\}^k)) = 1,$$

and for all pairs $i', j'$ with atleast one of them being not equal to 1, $\text{Correlation}(f(\cdot)_{i'}, g(\cdot)_{j'}, \mathcal{U}(\{0,1\}^k)) = 0$. The argument when the condition holds for other $i, j$ pairs can be similarly handled. Then, there are 2 cases possible.

$$\begin{aligned}
&\text{Subcase 1: } f(x)_1 = g(x)_1, \text{ for all } x \in \{0,1\}^k \\
&\text{Subcase 2: } f(x)_1 = \textbf{not}(g(x)_1), \text{ for all } x \in \{0,1\}^k,
\end{aligned}$$

while in each of these pairs, $(f(\cdot)_2, g(\cdot)_2)$, $(f(\cdot)_2, g(\cdot)_1)$, and $(f(\cdot)_1, g(\cdot)_2)$, the output bits of $f$ and $g$ are completely independent of each other. We only consider subcase 1; subcase 2 can be handled using mirror phrasebooks similar to the proof for case 1 (in particular with Equation (9)).

For simplicity, we will argue for $i', j' = 1, 1$; the argument about other $i', j'$ pairs are similar. The proof will follow by two steps,

- **Step (a):** We first argue about the probability with which $\text{Correlation}(\pi^\alpha(f(\cdot))_1, \pi^\beta(g(\cdot))_1, \mathcal{U}(\{0,1\}^k))$ is equal to 1,
- **Step (b):** We then argue that if the condition in (a) holds true, then $\text{Correlation}(\pi^\alpha(f(\cdot))_{i'}, \pi^\beta(g(\cdot))_{j'}, \mathcal{U}(\{0,1\}^k))$ must be 0 for any other pair $i', j'$ where atleast one of them is not equal to 1.

**Step (a):** For simplicity, we will assume that $\pi^\alpha, \pi^\beta \in \{\boldsymbol{\Delta}_1, \boldsymbol{\Delta}_2, \cdots, \boldsymbol{\Delta}_6\}$ defined in Table 7; other cases can be handled similarly as Lemma F.23.

$$\text{Correlation}(\pi^{\alpha}(f(\cdot))_1, \pi^{\beta}(g(\cdot))_1, \mathcal{U}(\{0,1\}^k))$$

$$= \text{Correlation}(\pi^{\alpha}((f(\cdot)_1, f(\cdot)_2))_1, \pi^{\beta}((g(\cdot)_1, g(\cdot)_2))_1, \mathcal{U}(\{0,1\}^k)) \tag{11}$$

$$= \text{Correlation}(\pi^{\alpha}((f(\cdot)_1, f(\cdot)_2))_1, \pi^{\beta}((f(\cdot)_1, g(\cdot)_2))_1, \mathcal{U}(\{0,1\}^k)) \tag{12}$$

$$= \left| \Pr_{x \sim \mathcal{U}(\{0,1\}^k)} \left[ \pi^{\alpha}((f(x)_1, f(x)_2))_1 = \pi^{\beta}((f(x)_1, g(x)_2))_1 \right] \right.$$

$$\left. - \Pr_{x \sim \mathcal{U}(\{0,1\}^k)} \left[ \pi^{\alpha}((f(x)_1, f(x)_2))_1 \neq \pi^{\beta}((f(x)_1, g(x)_2))_1 \right] \right| \tag{13}$$

$$= \left| \Pr_{x \sim \mathcal{U}(\{0,1\})} \left[ \Pr_{y,y' \sim \mathcal{U}(\{0,1\}^2)} \left[ \pi^{\alpha}((x,y))_1 = \pi^{\beta}((x,y'))_1 \right] - \Pr_{y,y' \sim \mathcal{U}(\{0,1\}^2)} \left[ \pi^{\alpha}((x,y))_1 = \pi^{\beta}((x,y'))_1 \right] \right] \right| \tag{14}$$

The reasoning for each step is as follows:

- Because the outputs of $f$ and $g$ are a 2-tuple on any input, Equation (11) simply replaces $f(x)$ (similarly $g(x)$) as $(f(x)_1, f(x)_2)$ for any input $x$.
- Equation (12) uses the relation between the outputs of $f$ and $g$ when subcase 1 is true.
- Equation (9) simply writes the definition of correlation.
- Finally, because of the assumption on the output of $f$, i.e. $\mathbb{E}_{x \sim \mathcal{U}(\{0,1\}^k)} f(x)_j = 1/2, \mathbb{E}_{x \sim \mathcal{U}(\{0,1\}^k)} f(x)_1 \oplus f(x)_2 = 1/2$, the distribution of outputs of $f$ can be shown to be identical to $\mathcal{U}(\{0,1\}^2)$.

From the definitions of $\pi^{\alpha}, \pi^{\beta} \in \{\boldsymbol{\Delta}_1, \boldsymbol{\Delta}_2, \cdots, \boldsymbol{\Delta}_6\}$ in Table 7, the first character in the outputs of $\pi^{\alpha}$ and $\pi^{\beta}$ can be expressed using 3 operators on the input characters, which are $\mathbf{copy}_1$, $\mathbf{copy}_2$, and $\mathbf{xor}$. These operators are independently selected for $\pi^{\alpha}$ and $\pi^{\beta}$, as they are randomly picked from these 6 possibilities. Formally, for any tuple of variables $(x,y)$ and $(x,y')$,

$$\mathbf{copy}_1(x,y) = x, \quad \mathbf{copy}_2(x,y) = y, \quad \mathbf{xor}(x,y) = x \oplus y,$$
$$\mathbf{copy}_1(x,y') = x, \quad \mathbf{copy}_2(x,y') = y', \quad \mathbf{xor}(x,y') = x \oplus y'.$$

However, because $x, y, y'$ are independent variables, only operations $\mathbf{copy}_1(x,y)$ and $\mathbf{copy}_1(x,y')$ for $(x,y,y') \sim \mathcal{U}(\{0,1\}^3)$ will be correlated. This can be verified by writing the definition of correlation and using $\mathbf{copy}_1$ for the first character output of $\pi^{\alpha}$ and $\pi^{\beta}$:

$$\left| \Pr_{x \sim \mathcal{U}(\{0,1\})} \left[ \Pr_{y,y' \sim \mathcal{U}(\{0,1\}^2)} \left[ \pi^{\alpha}((x,y))_1 = \pi^{\beta}((x,y'))_1 \right] - \Pr_{y,y' \sim \mathcal{U}(\{0,1\}^2)} \left[ \pi^{\alpha}((x,y))_1 = \pi^{\beta}((x,y'))_1 \right] \right] \right|$$

$$= \left| \Pr_{x \sim \mathcal{U}(\{0,1\})} \left[ \Pr_{y,y' \sim \mathcal{U}(\{0,1\}^2)} \left[ \mathbf{copy}_1(x,y) = \mathbf{copy}_1(x,y') \right] - \Pr_{y,y' \sim \mathcal{U}(\{0,1\}^2)} \left[ \mathbf{copy}_1(x,y) \neq \mathbf{copy}_1(x,y') \right] \right] \right| = 1,$$

as the first term is 1 and the second term is 0. However, if you pick any other pair of operations, the correlation will be 0. We demonstrate by using $\mathbf{copy}_1$ and $\mathbf{xor}$ for $\pi^{\alpha}$ and $\pi^{\beta}$ respectively.

$$\left| \Pr_{x \sim \mathcal{U}(\{0,1\})} \left[ \Pr_{y,y' \sim \mathcal{U}(\{0,1\}^2)} \left[ \pi^{\alpha}((x,y))_1 = \pi^{\beta}((x,y'))_1 \right] - \Pr_{y,y' \sim \mathcal{U}(\{0,1\}^2)} \left[ \pi^{\alpha}((x,y))_1 = \pi^{\beta}((x,y'))_1 \right] \right] \right|$$

$$= \left| \Pr_{x \sim \mathcal{U}(\{0,1\})} \left[ \Pr_{y,y' \sim \mathcal{U}(\{0,1\}^2)} \left[ \mathbf{copy}_1(x,y) = \mathbf{xor}(x,y') \right] - \Pr_{y,y' \sim \mathcal{U}(\{0,1\}^2)} \left[ \mathbf{copy}_1(x,y) \neq \mathbf{xor}(x,y') \right] \right] \right|$$

$$= \left| \Pr_{x \sim \mathcal{U}(\{0,1\})} \left[ \Pr_{y,y' \sim \mathcal{U}(\{0,1\}^2)} \left[ x = x \oplus y' \right] - \Pr_{y,y' \sim \mathcal{U}(\{0,1\}^2)} \left[ x \neq x \oplus y' \right] \right] \right| = 0,$$

as both terms are equal to $\frac{1}{2}$ in the final step. By a counting argument, one can reason that the probability of $corr(\pi^{\alpha}(f(\cdot))_1, \pi^{\beta}(g(\cdot))_1, \mathcal{U}(\{0,1\}^k))$ being 1 is equal to the probability of $\mathbf{copy}_1$ being selected to define the first characters of both $\pi^{\alpha}$ and $\pi^{\beta}$, which will be equal to $\frac{1}{9}$.

**Step (b):** Now, say we have selected a pair $\pi^{\alpha}$ and $\pi^{\beta}$ such that $\text{Correlation}(\pi^{\alpha}(f(\cdot))_1, \pi^{\beta}(g(\cdot))_1, \mathcal{U}(\{0,1\}^k)) = 1$. From the proof of step (a), this is only possible when $\mathbf{copy}_1$ was selected to define the first characters in the outputs of

$\pi^\alpha$ and $\pi^\beta$. Any other operation pairs for defining the first characters in the outputs of $\pi^\alpha$ and $\pi^\beta$ would have meant correlation to be $0$.

We can use this same argument to show that for any other pair $i', j'$ where atleast one of them is not equal to $1$. Correlation$(\pi^\alpha(f(\cdot))_{i'}, \pi^\beta(g(\cdot))_{j'}, \mathcal{U}(\{0,1\}^k))$ must be $0$. This is because once $\mathbf{copy}_1$ has been used to define the operation for the first character, it can't be used to define the operation for the second character (please refer at the truth tables for $\{\boldsymbol{\Delta}_1, \cdots, \boldsymbol{\Delta}_6\}$ in Table 7). Following a similar argument, we can then show that correlation between output characters $\pi^\alpha(f(\cdot))_{i'}, \pi^\beta(g(\cdot))_{j'}$ will be $0$, as $\mathbf{copy}_1$ can't be used to define at least one of these characters.

3. The third case is when none of the above conditions hold true. That is, for any pair $i, j \in \{1, 2\}$,

$$\text{Correlation}(f(\cdot)_i, g(\cdot)_j, \mathcal{U}(\{0,1\}^k)) = 0.$$

In such case, following similar arguments as case 1 and 2, one can show that for any $i, j \in \{1, 2\}$, for any choice of $\pi^\alpha$ and $\pi^\beta$:

$$\text{Correlation}(\pi^\alpha(f(\cdot))_i, \pi^\beta(g(\cdot))_j, \mathcal{U}(\{0,1\}^k))$$
$$= \left| \Pr_{x,y \sim \mathcal{U}(\{0,1\}^2)} \Pr_{x',y' \sim \mathcal{U}(\{0,1\}^2)} \left[ \pi^\alpha((x,y))_i = \pi^\beta((x',y'))_j \right] \right.$$
$$\left. - \Pr_{x,y \sim \mathcal{U}(\{0,1\}^2)} \Pr_{x',y' \sim \mathcal{U}(\{0,1\}^2)} \left[ \pi^\alpha((x,y))_i \neq \pi^\beta((x',y'))_j \right] \right|$$
$$= 0,$$

due to independence of inputs to $\pi^\alpha$ and $\pi^\beta$.

$\square$

**Lemma F.15.** *At any step $i$ of **MLT**$(d, 2)$ with phrasebooks $\{\pi_\ell\}_{\ell=1}^d$, the following relation holds true for the intermediate outputs $\{s_i\}_{i=2}^{d+1}$ on an input $s_1 \in \{0, 1\}^L$:*

$$(s_{i+1,1}, s_{i+1,2}) = \pi_i((s_{i,2}, s_{i,3})).$$

*Proof.* **MLT**$(d, 2)$ has 2 primary steps: *Circular shift* and *Translate*. By *Circular shift*, first, we first get sequence $\tilde{s}_i$, where for any $j \in [L]$ we have $\tilde{s}_{i,j} = s_{i,(j+1)\%L}$. After *Translate* step,

$$(s_{i+1,1}, s_{i+1,2}) = \pi_i((\tilde{s}_{i,1}, \tilde{s}_{i,2})) = \pi_i((s_{i,2}, s_{i,3})).$$

$\square$

**Lemma F.16.** *At any step $i$ of **MLT**$(d, 2)$ with phrasebooks $\{\pi_\ell\}_{\ell=1}^d$, the following conditions hold true for the intermediate output $s_i$.*

$$\mathbb{E}_{s_1 \sim \mathcal{U}(\{0,1\}^L)} s_{i,j} = \frac{1}{2}, \text{ for all } 1 \leq j \leq L$$
$$\mathbb{E}_{s_1 \sim \mathcal{U}(\{0,1\}^L)} s_{i,j} \oplus s_{i,j'} = \frac{1}{2}, \text{ for all } j \neq j'.$$

*The above conditions are equivalent to showing that $s_{i,j}$ behaves like a uniformly random boolean variable, independent of any other character $s_{i,j'}$ for all coordinates $j' \neq j$.*

*Proof.* The proof will follow by induction on the output of the translation task at each step. We will show the result for coordinate $j = 1$ in $s_i$; similar argument holds for other coordinates $j$.

**Base condition:** At layer $i = 1$, the $s_1$ represents the input sequence from $\mathcal{U}(\{0,1\}^L)$. By definition of uniform distribution, the conditions hold true for the input.

| Operator | Expected value | Expected $\oplus$ value with operator | | |
|---|---|---|---|---|
| | | **copy$_1$** | **copy$_2$** | **xor** |
| **copy$_1$** | $\mathbb{E}_{\boldsymbol{s}_1}\boldsymbol{s}_{i-1,2}=1/2$ | - | $\mathbb{E}_{\boldsymbol{s}_1}\boldsymbol{s}_{i-1,2}\oplus\boldsymbol{s}_{i-1,3}=1/2$ | $\mathbb{E}_{\boldsymbol{s}_1}\boldsymbol{s}_{i-1,3}=1/2$ |
| **copy$_2$** | $\mathbb{E}_{\boldsymbol{s}_1}\boldsymbol{s}_{i-1,3}=1/2$ | $\mathbb{E}_{\boldsymbol{s}_1}\boldsymbol{s}_{i-1,2}\oplus\boldsymbol{s}_{i-1,3}=1/2$ | - | $\mathbb{E}_{\boldsymbol{s}_1}\boldsymbol{s}_{i-1,2}=1/2$ |
| **xor** | $\mathbb{E}_{\boldsymbol{s}_1}\boldsymbol{s}_{i-1,2}\oplus\boldsymbol{s}_{i-1,3}=1/2$ | $\mathbb{E}_{\boldsymbol{s}_1}\boldsymbol{s}_{i-1,3}=1/2$ | $\mathbb{E}_{\boldsymbol{s}_1}\boldsymbol{s}_{i-1,2}=1/2$ | - |

*Table 6.* $\mathbb{E}\boldsymbol{s}_{i,1}$ (similarly $\mathbb{E}\boldsymbol{s}_{i,2}$) and $\mathbb{E}\boldsymbol{s}_{i,1}\oplus\boldsymbol{s}_{i,2}$ under different $\pi_i$ phrasebooks, defined by **copy** and **xor** operations on $\boldsymbol{s}_{i-1,2}$ and $\boldsymbol{s}_{i-1,3}$.

**Induction step: Argument for general $i > 1$:**  Suppose the conditions are true for all layers $1 \leq \ell < i$. Then for layer $i$, we will provide an argument for the condition to hold true for $j = 1$ and $j = 2$, arguments for other $j$s will extend similarly. By Lemma F.15,

$$(\boldsymbol{s}_{i,1}, \boldsymbol{s}_{i,2}) = \pi_i((\boldsymbol{s}_{i-1,2}, \boldsymbol{s}_{i-1,3})).$$

From Lemma F.21, we have each output character can be defined in terms of **copy**, **xor**, and **not** operations on input characters. Then, the relations for $\boldsymbol{s}_{i,1}, \boldsymbol{s}_{i,2}$ can be computed as follows:

- If a map $\pi^\alpha$ is selected from Mirrorset($\pi$) for some $\pi \in \{\boldsymbol{\Delta}_1, \cdots, \boldsymbol{\Delta}_6\}$, then one can show that the conditions hold true for $\pi_i = \pi^\alpha$ if the conditions hold true for $\pi_i = \pi$. This follows because the output of $\pi^\alpha$ follows from the output of $\pi$ by selective **not** operations to the output of $\pi$. As **not** operation won't change the expected values of a variable which behaves like a random boolean variable, the argument follows.

- Now, we show that for $\pi_i = \pi$ for some $\pi \in \{\boldsymbol{\Delta}_1, \cdots, \boldsymbol{\Delta}_6\}$, the conditions hold true. For these phrasebooks, the output characters are defined by **copy** and **xor** operations. We use the definitions of these phrasebooks from Table 7 and show the expected values $\boldsymbol{s}_{i,1}$ in terms of expected values of $\boldsymbol{s}_{i-1,1}$ and $\boldsymbol{s}_{i-1,2}$, and the expected values $\boldsymbol{s}_{i,1} \oplus \boldsymbol{s}_{i,2}$ in terms of expected values of $\boldsymbol{s}_{i-1,1}$ and $\boldsymbol{s}_{i-1,2}$ in Table 6.

Both of the above arguments then can be combined to show that

$$\mathbb{E}_{\boldsymbol{s}_1 \sim \mathcal{U}(\{0,1\}^L)}\boldsymbol{s}_{i,j} = \frac{1}{2}, \text{ for } j \in \{1, 2\}$$

$$\mathbb{E}_{\boldsymbol{s}_1 \sim \mathcal{U}(\{0,1\}^L)}\boldsymbol{s}_{i,1} \oplus \boldsymbol{s}_{i,2} = \frac{1}{2}.$$

We can similarly extend the argument for expectation of each individual character to other positions $j > 2$, i.e. $\mathbb{E}_{\boldsymbol{s}_1 \sim \mathcal{U}(\{0,1\}^L)}\boldsymbol{s}_{i,j} = \frac{1}{2}$ for all other possible $j$s. We can also similarly extend the argument for joint expectation of two consecutive characters $\boldsymbol{s}_{i,2k+1} \oplus \boldsymbol{s}_{i,2k+2}$ for any general $k$.

The remaining argument will be to show that characters that don't form a consecutive 2-tuple are going to be independent of each other as well. Consider any two 2-tuples $(\boldsymbol{s}_{i,2k+1}, \boldsymbol{s}_{i,2k+2})$ and $(\boldsymbol{s}_{i,2k'+1}, \boldsymbol{s}_{i,2k'+2})$, with $k \neq k'$. By adapting Lemma F.15, one can show that

$$(\boldsymbol{s}_{i,2k+1}, \boldsymbol{s}_{i,2k+2}) = \pi_i((\boldsymbol{s}_{i-1,2k+2}, \boldsymbol{s}_{i,2k+3}))$$
$$(\boldsymbol{s}_{i,2k'+1}, \boldsymbol{s}_{i,2k'+2}) = \pi_i((\boldsymbol{s}_{i-1,2k'+2}, \boldsymbol{s}_{i,2k'+3}))$$

The primary thing to note here is that both the tuples depend on two distinct tuples in layer $i - 1$. By induction assumption, characters across these two 2-tuples are independent of each other. As $\pi_i$ is a bijective map, this will also suggest that characters across the 2-tuples in the resulting output must also be independent of each other. This will give the final argument.

$\square$

**Lemma F.17.** *Under the assumption that sequence lengths $L$ are even, for any $2 \leq i \leq d + 1$, where $\boldsymbol{s}_{i,j}^\alpha, \boldsymbol{s}_{i,j}^\beta$ denote the output after $i - 1$st translation step for a random sequence $\boldsymbol{s}_1 \sim \mathcal{U}(\{0,1\}^L)$ at any position $1 \leq j \leq L$ under the two set of phrasebooks $\boldsymbol{\Pi}_{:i}^\alpha := \{\pi_\ell^\alpha\}_{\ell=1}^{i-1}$ and $\boldsymbol{\Pi}_{:i}^\beta := \{\pi_\ell^\beta\}_{\ell=1}^{i-1}$, the following holds true for all positions $j$:*

$$Correlation(\boldsymbol{s}_{i,j}^\alpha, \boldsymbol{s}_{i,j}^\beta, \mathcal{U}(\{0,1\}^L)) = Correlation(\boldsymbol{s}_{i,(j+2k)\%L}^\alpha, \boldsymbol{s}_{i,(j+2k)\%L}^\beta, \mathcal{U}(\{0,1\}^L)), \text{ for all integer } k.$$

*Proof.* Fix an integer $k$. By Definition F.1, the necessary condition would be to show

$$\left| 1 - 2\mathbb{E}_{s_1 \sim \{0,1\}^L} s_{i,j}^\alpha \oplus s_{i,j}^\beta \right| = \left| 1 - 2\mathbb{E}_{s_1 \sim \{0,1\}^L} s_{i,(j+2k)\%L}^\alpha \oplus s_{i,(j+2k)\%L}^\beta \right| .$$

We will look at the behavior of the function $h(s_1) = s_{i,j}^\alpha \oplus s_{i,j}^\beta$. Denote $\mathcal{C}_k$ as a circular operation that takes a sequence $s_1$ and returns a shifted sequence ${}^{\circlearrowleft}s_1$, i.e. if ${}^{\circlearrowleft}s_1 = \mathcal{C}_k(s_1)$ then for all $j$, ${}^{\circlearrowleft}s_{1,j} = s_{1,(j+2k)\%L}$. Note that, we can also define an inverse function $\mathcal{C}_k^{-1}$ and $s_1 = \mathcal{C}_k^{-1}(\mathcal{C}_k(s_1))$.

From the definition of $\mathbf{MLT_\Pi}$ for any phrasebook $\mathbf{\Pi}$, we have for any input $s_1$:

$$s_i = T_{\pi_{i-1}} \circ \cdots \circ T_{\pi_1}(s_1),$$

where for any level $i$, $T_{\pi_i}$ denotes the translation step at level $i$ that includes *Circular shift* and *Translate* using $\pi_i$.

In Lemma F.18, we show that for any translation task $\mathbf{MLT_\Pi}$ and any input $s_1$,

$$T_{\pi_{i-1}} \circ \cdots \circ T_{\pi_1}(s_1) = \mathcal{C}_k^{-1}\left( T_{\pi_{i-1}} \circ \cdots \circ T_{\pi_1} \circ \mathcal{C}_k(s_1) \right).$$

On input ${}^{\circlearrowleft}s_1 = \mathcal{C}_k(s_1)$, if ${}^{\circlearrowleft}s_i^\alpha$ and ${}^{\circlearrowleft}s_i^\beta$ represent the output of translations using $\mathbf{\Pi}_{:i}^\alpha := \{\pi_\ell^\alpha\}_{\ell=1}^{i-1}$ and $\mathbf{\Pi}_{:i}^\beta := \{\pi_\ell^\beta\}_{\ell=1}^{i-1}$ respectively, then the above statement says that

$$s_i^\alpha = \mathcal{C}_k^{-1}\left( {}^{\circlearrowleft}s_i^\alpha \right), \quad s_i^\beta = \mathcal{C}_k^{-1}\left( {}^{\circlearrowleft}s_i^\beta \right)$$

$$(\text{or}) \ \mathcal{C}_k\left( s_i^\alpha \right) = {}^{\circlearrowleft}s_i^\alpha, \quad \mathcal{C}_k\left( s_i^\beta \right) = {}^{\circlearrowleft}s_i^\beta$$

Thus, for any position $j$, we will have $\mathcal{C}_k\left( s_i^\alpha \right)_j = {}^{\circlearrowleft}s_{i,j}^\alpha$ (and similarly for ${}^{\circlearrowleft}s_{i,j}^\beta$). We can then compute the value of $h$ on input ${}^{\circlearrowleft}s_1$ as follows:

$$\begin{aligned} h({}^{\circlearrowleft}s_1) &= {}^{\circlearrowleft}s_{i,j}^\alpha \oplus {}^{\circlearrowleft}s_{i,j}^\beta \\ &= \mathcal{C}_k\left( s_i^\alpha \right)_j \oplus \mathcal{C}_k\left( s_i^\beta \right)_j \\ &= s_{i,(j+2k)\%L}^\alpha \oplus s_{i,(j+2k)\%L}^\beta \end{aligned}$$

The last step follows from using the definition of $\mathcal{C}_k$. On the other hand,

$$\mathbb{E}_{{}^{\circlearrowleft}s_1 \sim \mathcal{U}(\{0,1\}^L)} h({}^{\circlearrowleft}s_1) = \mathbb{E}_{s_1 \sim \mathcal{U}(\{0,1\}^L)} h(s_1),$$

as the uniform distribution can be shown to not change under circular function $\mathcal{C}_k$. This will imply:

$$\mathbb{E}_{s_1 \sim \mathcal{U}(\{0,1\}^L)} s_{i,(j+2k)\%L}^\alpha \oplus s_{i,(j+2k)\%L}^\beta = \mathbb{E}_{s_1 \sim \{0,1\}^L} s_{i,j}^\alpha \oplus s_{i,j}^\beta$$

$$\implies \left| 1 - 2\mathbb{E}_{s_1 \sim \{0,1\}^L} s_{i,j}^\alpha \oplus s_{i,j}^\beta \right| = \left| 1 - 2\mathbb{E}_{s_1 \sim \{0,1\}^L} s_{i,(j+2k)\%L}^\alpha \oplus s_{i,(j+2k)\%L}^\beta \right|$$

$$\implies \text{Correlation}(s_{i,j}^\alpha, s_{i,j}^\beta, \mathcal{U}(\{0,1\}^L)) = \text{Correlation}(s_{i,(j+2k)\%L}^\alpha, s_{i,(j+2k)\%L}^\beta, \mathcal{U}(\{0,1\}^L)).$$

$\square$

**Lemma F.18.** *For any translation task $\mathbf{MLT_\Pi}$, at any level $i \leq d$ and any input $s_1$,*

$$T_{\pi_i} \circ \cdots \circ T_{\pi_1}(s_1) = \mathcal{C}_k^{-1}\left( T_{\pi_i} \circ \cdots \circ T_{\pi_1} \circ \mathcal{C}_k(s_1) \right).$$

*Proof.* Denote $\mathcal{C}_k$ as a circular operation that takes a sequence $s_1$ and returns a shifted sequence ${}^{\circlearrowleft}s_1$, i.e. if ${}^{\circlearrowleft}s_1 = \mathcal{C}_k(s_1)$ then for all $j$, ${}^{\circlearrowleft}s_{1,j} = s_{1,(j+2k)\%L}$. Note that, we can also define an inverse function $\mathcal{C}_k^{-1}$ and $s_1 = \mathcal{C}_k^{-1}(\mathcal{C}_k(s_1))$.

Recall that for any level $i$, $T_{\pi_i}$ denotes the translation step at level $i$ that includes *Circular shift*, and *Translate* with $\pi_i$. The argument will again follow by an induction step.

**Base condition:** $i = 1$: We need to show that

$$T_{\pi_1}(s_1) = \mathcal{C}_k^{-1}(T_{\pi_1} \circ \mathcal{C}_k(s_1)).$$

To do so, we will look at the behavior of the first 2 characters. Argument for others can be extended. If ${}^{\circlearrowleft}s_1 = \mathcal{C}_k(s_1)$, $s_2 = T_{\pi_1}(s_1)$, ${}^{\circlearrowleft}s_2 = T_{\pi_1}({}^{\circlearrowleft}s_1)$, then by Lemma F.15,

$$(s_{2,1}, s_{2,2}) = \pi_1((s_{1,2}, s_{1,3}))$$
$$({}^{\circlearrowleft}s_{2,1}, {}^{\circlearrowleft}s_{2,2}) = \pi_1(({}^{\circlearrowleft}s_{1,2}, {}^{\circlearrowleft}s_{1,3})).$$

But by definition of $\mathcal{C}_k$,

$$({}^{\circlearrowleft}s_{1,2}, {}^{\circlearrowleft}s_{1,3}) = (s_{1,(2+2k)\%L}, s_{1,(3+2k)\%L}).$$

On the other hand, Lemma F.15 can be adapted to give

$$(s_{2,(1+2k)\%L}, s_{2,(2+2k)\%L}) = \pi_1(s_{1,(2+2k)\%L}, s_{1,(3+2k)\%L}).$$

Thus, we can show by combining the above 3 steps that

$$\begin{aligned}
({}^{\circlearrowleft}s_{2,1}, {}^{\circlearrowleft}s_{2,2}) &= \pi_1(({}^{\circlearrowleft}s_{1,2}, {}^{\circlearrowleft}s_{1,3})) \\
&= \pi_1(s_{1,(2+2k)\%L}, s_{1,(3+2k)\%L}) \\
&= (s_{2,(1+2k)\%L}, s_{2,(2+2k)\%L}).
\end{aligned}$$

We can extend the above argument to show that for any position $j$,

$${}^{\circlearrowleft}s_{2,j} = s_{2,(j+2k)\%L},$$

which by definition of $\mathcal{C}_k$, implies

$${}^{\circlearrowleft}s_2 = \mathcal{C}_k(s_2) \quad \text{or } s_2 = \mathcal{C}_k^{-1}({}^{\circlearrowleft}s_2),$$

which can be further simplified (using the notations: ${}^{\circlearrowleft}s_1 = \mathcal{C}_k(s_1)$, $s_2 = T_{\pi_1}(s_1)$, ${}^{\circlearrowleft}s_2 = T_{\pi_1}({}^{\circlearrowleft}s_1)$)

$$\begin{aligned}
s_2 &= \mathcal{C}_k^{-1}({}^{\circlearrowleft}s_2) \\
&= \mathcal{C}_k^{-1}(T_{\pi_1}({}^{\circlearrowleft}s_1)) \\
&= \mathcal{C}_k^{-1}(T_{\pi_1}(\mathcal{C}_k(s_1))) := \mathcal{C}_k^{-1}(T_{\pi_1} \circ \mathcal{C}_k(s_1)).
\end{aligned}$$

**General argument for $i$:** Suppose the induction condition holds true for all layers $\ell < i$. Then,

$$T_{\pi_i} \circ \cdots \circ T_{\pi_1}(s_1) = T_{\pi_i}\left(\mathcal{C}_k^{-1}\left(T_{\pi_{i-1}} \cdots \circ T_{\pi_1} \circ \mathcal{C}_k(s_1)\right)\right).$$

We can then follow the same argument as the base condition, and show that

$$T_{\pi_i}\left(\mathcal{C}_k^{-1}\left(T_{\pi_{i-1}} \cdots \circ T_{\pi_1} \circ \mathcal{C}_k(s_1)\right)\right) = \mathcal{C}_k^{-1}\left(T_{\pi_i} \cdots \circ T_{\pi_1} \circ \mathcal{C}_k(s_1)\right).$$

$\square$

**Lemma F.19.** *For any $1 \leq i \leq d$, if $s_{i,j}^\alpha$, $s_{i,j}^\beta$ denote the output after $i-1$st translation step for a random sequence $s_1 \sim \mathcal{U}(\{0,1\}^L)$ at any position $1 \leq j \leq L$ under the two sets of random phrasebooks $\mathbf{\Pi}_{:i}^\alpha := \{\pi_\ell^\alpha\}_{\ell=1}^{i-1}$ and $\mathbf{\Pi}_{:i}^\beta := \{\pi_\ell^\beta\}_{\ell=1}^{i-1}$, the following holds true for all positions $j$:*

$$\Pr_{\mathbf{\Pi}_{:i}^\alpha, \mathbf{\Pi}_{:i}^\beta}\left[Correlation\left(s_{i,j}^\alpha, s_{i,j+1}^\beta, \mathcal{U}(\{0,1\}^L)\right) = 1\right]$$

$$\leq \frac{1}{2}\left(\Pr_{\mathbf{\Pi}_{:i}^\alpha, \mathbf{\Pi}_{:i}^\beta}\left[Correlation\left(s_{i,j}^\alpha, s_{i,j}^\beta, \mathcal{U}(\{0,1\}^L)\right) = 1\right] + \Pr_{\mathbf{\Pi}_{:i}^\alpha, \mathbf{\Pi}_{:i}^\beta}\left[Correlation\left(s_{i,j+1}^\alpha, s_{i,j+1}^\beta, \mathcal{U}(\{0,1\}^L)\right) = 1\right]\right).$$

*Proof.* We will prove the required result with a counting argument. We will create a family of phrasebooks: let FAMILY$(j)$ and FAMILY$(j + 1)$ denote two sets of phrasebooks, such that for every phrasebook $\mathbf{\Pi}_{:i}^{\alpha}$ in FAMILY$(j)$, there exists at least one phrasebook $\mathbf{\Pi}_{:i}^{\beta}$ in FAMILY$(j + 1)$ such that

$$\text{Correlation}\left(\boldsymbol{s}_{i,j}^{\alpha}, \boldsymbol{s}_{i,j+1}^{\beta}, \mathcal{U}(\{0,1\}^L)\right) = 1,$$

(Equiv. to saying translations for $\mathbf{\Pi}_{:i}^{\alpha}$ and $\mathbf{\Pi}_{:i}^{\beta}$ have correlations at position $j$ and $j + 1$)

where $\boldsymbol{s}_{i,j}^{\alpha}, \boldsymbol{s}_{i,j+1}^{\beta}$ are outputs on a random sequence $\boldsymbol{s}_1 \sim \mathcal{U}(\{0,1\}^L)$ corresponding to using $\mathbf{\Pi}_{:i}^{\alpha}$ and $\mathbf{\Pi}_{:i}^{\beta}$ respectively.

We then apply a grouping algorithm GROUP to group correlated phrasebooks together in each family. That is, in FAMILY$(j)$, we create groups of phrasebooks $\{S_1, S_2, \cdots\}$ such that for any two phrasebooks $\mathbf{\Pi}_{:i}^{\alpha}$ and $\mathbf{\Pi}_{:i}^{\alpha'}$ that belong to a group $S$,

$$\text{Correlation}\left(\boldsymbol{s}_{i,j}^{\alpha}, \boldsymbol{s}_{i,j}^{\alpha'}, \mathcal{U}(\{0,1\}^L)\right) = 1,$$

(Equiv. to saying translations for $\mathbf{\Pi}_{:i}^{\alpha}$ and $\mathbf{\Pi}_{:i}^{\beta}$ have correlations at position $j$)

where $\boldsymbol{s}_{i,j}^{\alpha}, \boldsymbol{s}_{i,j+1}^{\alpha'}$ are outputs on a random sequence $\boldsymbol{s}_1 \sim \mathcal{U}(\{0,1\}^L)$ corresponding to using $\mathbf{\Pi}_{:i}^{\alpha'}$ and $\mathbf{\Pi}_{:i}^{\alpha'}$ respectively. We call the resulting output of this operation as GROUP(FAMILY$(j)$). Similarly, we compute GROUP(FAMILY$(j + 1)$).

We can observe the following two characteristics of GROUP(FAMILY$(j)$) and GROUP(FAMILY$(j + 1)$):

1. For every set $S_1 \in$ GROUP(FAMILY$(j)$) there will exist one set $S_2 \in$ GROUP(FAMILY$(j + 1)$), such that for all phrasebooks $\mathbf{\Pi}_{:i}^{\alpha} \in S_1$ and $\mathbf{\Pi}_{:i}^{\beta} \in S_2$, correlation will be 1 for output at positions $j$ and $j + 1$.

2. For every set $S_1 \in$ GROUP(FAMILY$(j)$) there can't exist two sets $S_2, S_2^* \in$ GROUP(FAMILY$(j + 1)$), such that the phrasebooks in $S_1$ are correlated to phrasebooks from both $S_2, S_2^*$ for output at positions $j$ and $j + 1$ respectively. Otherwise, we could have merged $S_2$ and $S_2^*$ under the GROUP operation.

Let CORRELATION-MAP denote the map between GROUP(FAMILY$(j)$) and GROUP(FAMILY$(j + 1)$), which connects sets $S_1 \in$ GROUP(FAMILY$(j)$) to a set $S_2 \in$ GROUP(FAMILY$(j + 1)$) such that any two phrasebooks in $S_1$ and $S_2$ have correlations for output at positions $j$ and $j + 1$ respectively.

The result will then follow from a counting argument. The number of possible pairs (can be identical phrasebooks) that can give correlations for output at position $j$ are given by: $\sum_{S_1 \in \text{GROUP}(\text{FAMILY}(j))} |S_1|^2$. Similarly, the number of possible pairs that can give correlations for output at position $j + 1$ are given by: $\sum_{S_2 \in \text{GROUP}(\text{FAMILY}(j+1))} |S_2|^2$. On the other hand, the number of possible pairs that can give correlations for output at position $j$ and $j + 1$ respectively are given by: $\sum_{S_1 \in \text{GROUP}(j); S_2 = \text{CORRELATION-MAP}(S_1)} |S_1||S_2|$. Applying the AM-GM inequality, we can show that the average of the number of pairs for which correlation is 1 for output characters at either position $j$ or $j + 1$ is higher than the number of pairs for which correlation is 1 for output at positions $j$ and $j + 1$.

$\square$

## F.5. Proof for Statistical query lower bound for general $n$

We present the main theorem statement again for readability.

> **Theorem F.20** (SQ dimension for general $n$). *For the translation task $\boldsymbol{MLT}(d, n)$ that has depth $d$ and $n$ characters per level, the statistical query dimension SQ-dim($\boldsymbol{MLT}(d, n)$) is atleast $n^{\Omega(d)}$.*

*Proof.* We adapt the SQ-dimension proof for $\boldsymbol{MLT}(d, 2)$ to show the SQ-dimension proof for $\boldsymbol{MLT}(d, n)$. We will design a family of set of phrasebooks $\mathbf{\Pi} = \{\pi_i : \{0, 1, \cdots, n - 1\}^2 \to \{0, 1, \cdots, n - 1\}^2\}_{i=1}^d$, where phrasebooks in $\mathbf{\Pi}$ are built on top of a translation task in $\boldsymbol{MLT}(\log_2 n, 2)$.

For a character $a \in \{0, 1, \cdots, n - 1\}$, suppose BIT$(a) \in \{0, 1\}^{\log_2 n}$ indicates its binary representation, and NUMERIC represents the map from binary representation to its corresponding numeric representation. Then, we design each phrasebook $\pi$ using a random translation task $\nu \in \boldsymbol{MLT}(\log_2 n, 2)$. For any tuple $(a, b) \in \{0, 1, \cdots, n - 1\}^2$, output of $\pi$ is given as

$$\pi(a, b) = (o_1, o_2), \text{ where}$$

$$o_1 = \text{NUMERIC}\left(\nu\left(\{\tilde{\boldsymbol{a}}_i \oplus \tilde{\boldsymbol{b}}_i\}_{i=1}^{\log_2 n}\right)\right)$$

$$o_2 = \text{NUMERIC}\left(\nu\left(\tilde{\boldsymbol{b}}\right)\right)$$

$$\tilde{\boldsymbol{a}} = \text{BIT}(a)$$

$$\tilde{\boldsymbol{b}} = \text{BIT}(b)$$

Primarily, the phrasebooks are defined as follows:

1. On a 2-tuple of characters $(a, b)$, we first compute their binary representations $(\text{BIT}(a), \text{BIT}(b))$. We then compute two intermediate outputs, one where a **xor** operation is applied on $\text{BIT}(a), \text{BIT}(b)$ at each bit, and another where $\text{BIT}(b)$ is simply copied. This operation is equivalent to applying a deterministic map on 2-tuples of binary characters (identical to $\boldsymbol{\Delta}_6$ from Table 7), where 2-tuples are created by pairing binary bits in binary representation of $a$ and $b$.

2. We then apply a random **MLT** task of depth $\log_2 n$ on each of the intermediate outputs. This applies a random bijective map on the sequence of bits in the intermediate outputs, using a translation task in $\mathbf{MLT}(\log_2 n, 2)$ (Lemma E.1).

3. The final tuple of characters is returned by applying a NUMERIC operation on the binary representations.

Thus, we have narrowed our focus on a special group of tasks from $\mathbf{MLT}(d, n)$ that applies $\mathbf{MLT}(\log_2 n, 2)$ on the binary representations of the characters at each level. By Theorem F.10, at any level $1 \leq i \leq d$, we can create $2^{\Omega(\log_2 n)}$ phrasebooks, the output of which are pairwise uncorrelated . Now, we can compose these uncorrelated phrasebooks to give multiple set of phrasebooks that are uncorrelated. That is, following a similar proof as Theorem F.10, we can show that we can create $\left(2^{\Omega(\log_2 n)}\right)^{\Omega(d)} = n^{\Omega(d)}$ set of phrasebooks, using the above restriction, that are pairwise uncorrelated on any bit in the binary representation of the output of their corresponding translation tasks. This will translate to the output of the translation task in numeric form as well, as the mapping between binary representation and numeric form of a digit is bijective.

$\square$

### F.6. Proofs of Useful lemmas

Here we give the proofs for the useful lemmas necessary to prove Theorem F.10. We repeat the lemma statements for easier readability.

**Lemma F.21** (Formulation of bijective maps for $n = 2$). *Any bijective map $\pi : \{0, 1\}^2 \to \{0, 1\}^2$ can be expressed using* **copy**, **not**, *and* **xor** *operations. Furthermore, from $24$ possible maps for $\pi$,*

1. *There are $6$ maps $\boldsymbol{\Delta}_1, \boldsymbol{\Delta}_2, \cdots, \boldsymbol{\Delta}_6$ for which characters in the output tuple can be defined by* **copy** *and* **xor** *operations on the characters in the input tuple.*

2. ***Mirror maps:** For each map $\pi \in \{\boldsymbol{\Delta}_1, \boldsymbol{\Delta}_2, \cdots, \boldsymbol{\Delta}_6\}$, there exist mirror maps $\pi_{(1)}, \pi_{(2)}, \pi_{(3)}$ whose output on each input tuple can be defined by selective **not** operations on either or both characters of the output tuple of $\pi$. We call $\{\pi, \pi_{(1)}, \pi_{(2)}, \pi_{(3)}\}$ as a **mirror map set** of $\pi$, in short, Mirrorset$(\pi)$.*

*Proof.* There are $4$ possible tuples $(0, 0), (0, 1), (1, 0), (1, 1)$. By Lemma E.2, the number of possible bijective maps (phrasebooks) that connect 2-tuples are $4! = 24$. We will show that the output tuple of each map can be represented by $3$ operations.

**Operation not creates mirror maps:** For each map $\pi$, there exists $3$ alternative maps $\pi_{(1)}, \pi_{(2)}, \pi_{(3)}$ such that if $\pi(a, b)_i$ represents the $i$th character in the output tuple for input tuple $(a, b)$:

| Map name ($\pi$) | Output for corresponding input tuple $\pi(a,b)$ | | | | General formulation on output for map $\pi$ | |
|---|---|---|---|---|---|---|
| | $(0,0)$ | $(0,1)$ | $(1,0)$ | $(1,1)$ | $\pi(a,b)_1$ | $\pi(a,b)_2$ |
| $\boldsymbol{\Delta}_1$ | $(0,0)$ | $(0,1)$ | $(1,0)$ | $(1,1)$ | $\textbf{copy}_1(a,b)$ | $\textbf{copy}_2(a,b)$ |
| $\boldsymbol{\Delta}_2$ | $(0,0)$ | $(0,1)$ | $(0,1)$ | $(1,0)$ | $\textbf{copy}_1(a,b)$ | $\textbf{xor}(a,b)$ |
| $\boldsymbol{\Delta}_3$ | $(0,0)$ | $(1,0)$ | $(0,1)$ | $(1,1)$ | $\textbf{copy}_2(a,b)$ | $\textbf{copy}_1(a,b)$ |
| $\boldsymbol{\Delta}_4$ | $(0,0)$ | $(1,1)$ | $(0,1)$ | $(1,0)$ | $\textbf{copy}_2(a,b)$ | $\textbf{xor}(a,b)$ |
| $\boldsymbol{\Delta}_5$ | $(0,0)$ | $(1,0)$ | $(1,1)$ | $(0,1)$ | $\textbf{xor}(a,b)$ | $\textbf{copy}_1(a,b)$ |
| $\boldsymbol{\Delta}_6$ | $(0,0)$ | $(1,1)$ | $(1,0)$ | $(0,1)$ | $\textbf{xor}(a,b)$ | $\textbf{copy}_2(a,b)$ |

*Table 7.* The table captures the definition of 6 bijective maps (phrasebooks) on 2-tuples $\{0,1\}^2 \to \{0,1\}^2$, whose output characters can be defined in terms of **copy** and **xor** operations on the input characters. Any other bijective map can be shown to belong to Mirrorset of one of these maps.

- $\pi_{(1)}$ selectively applies the **not** operation to the first character in the output tuple of $\pi$ on any input tuple, i.e.

$$\pi_{(1)}(a,b) = (\textbf{not}(\pi(a,b)_1), \pi(a,b))$$

  for all tuples $(a,b) \in \{0,1\}^2$.

- $\pi_{(2)}$ selectively applies the **not** operation to the second character in the output tuple of $\pi$ on any input tuple, i.e.

$$\pi_{(2)}(a,b) = (\pi(a,b)_1, \textbf{not}(\pi(a,b)))$$

  for all tuples $(a,b) \in \{0,1\}^2$.

- $\pi_{(3)}$ selectively applies the **not** operation to both characters in the output tuple of $\pi$ on any input tuple, i.e.

$$\pi_{(3)}(a,b) = (\textbf{not}(\pi(a,b)_1), \textbf{not}(\pi(a,b)))$$

  for all tuples $(a,b) \in \{0,1\}^2$.

Thus, for each map $\pi$, there exist 3 other alternative maps that simply modify the output of map $\pi$ with the **not** operation.

**After removing the mirror maps:** We now show that there 6 possible maps that apply either a **xor** or a **copy** on the input characters to get the output characters. We name them $\boldsymbol{\Delta}_1, \boldsymbol{\Delta}_2, \cdots, \boldsymbol{\Delta}_6$. We give the output of each map on the 4 tuples in Table 7 and show that the each character in the output tuple can be represented using **copy** and **xor** operations.

$\square$

**Lemma F.22** (Correlation of bijective maps for $n = 2$). *For two randomly selected maps $\pi^\alpha, \pi^\beta : \{0,1\}^2 \to \{0,1\}^2$, the following hold true.*

1. ***Correlation at atleast one output character pair:*** *Fix an $i,j \in \{0,1\}$. With probability $\frac{1}{3}$ w.r.t. the random selection of the maps, the $i$th character in the output of $\pi^\alpha$ has perfect correlation to $j$th character in the output of $\pi^\beta$, i.e.*

$$Correlation(\pi^\alpha(\cdot)_i, \pi^\beta(\cdot)_j, \mathcal{U}(\{0,1\}^2)) := \left| \Pr_{x \sim \mathcal{U}(\{0,1\}^2)} [\pi^\alpha(x)_i \neq \pi^\beta(x)_j] - \Pr_{x \sim \mathcal{U}(\{0,1\}^2)} [\pi^\alpha(x)_i = \pi^\beta(x)_j] \right| = 1.$$

2. ***Correlation at both output character pairs:*** *With probability $\frac{1}{3}$ w.r.t. the random selection of the maps, both the characters in the outputs of $\pi^\alpha, \pi^\beta$ have perfect correlation, i.e. either one of the cases hold true*

$$Correlation(\pi^\alpha(\cdot)_1, \pi^\beta(\cdot)_1, \mathcal{U}(\{0,1\}^2)) = 1.$$
$$Correlation(\pi^\alpha(\cdot)_2, \pi^\beta(\cdot)_2, \mathcal{U}(\{0,1\}^2)) = 1.$$

*or*

$$Correlation(\pi^\alpha(\cdot)_1, \pi^\beta(\cdot)_2, \mathcal{U}(\{0,1\}^2)) = 1.$$
$$Correlation(\pi^\alpha(\cdot)_2, \pi^\beta(\cdot)_1, \mathcal{U}(\{0,1\}^2)) = 1.$$

*Other cases are not possible, i.e. for any $i \in \{1,2\}$, both $Correlation(\pi^\alpha(\cdot)_i, \pi^\beta(\cdot)_1, \mathcal{U}(\{0,1\}^2)) = 1$ and $Correlation(\pi^\alpha(\cdot)_i, \pi^\beta(\cdot)_2, \mathcal{U}(\{0,1\}^2)) = 1$ can't hold true.*

*Proof.* We prove the lemma for case 1, cases 2 and 3 can be similarly proved. From Lemma E.2, $\pi^\alpha$ and $\pi^\beta$ can be randomly selected from a set of 24 possible candidates. On the other hand, Lemma F.21 shows that there are 6 maps $\{\Delta_1, \cdots, \Delta_6\}$ whose output can be defined in terms of **copy** and **xor** operations of characters in the input tuple. For each $\pi$ in this set, there are mirror maps $\pi_{(1)}, \pi_{(2)}, \pi_{(3)}$ whose output are defined by selective **not** operations on the output of $\pi$, and the set of 4 maps is represented by Mirrorset($\pi$).

The proof will follow from 2 steps: first, we argue about correlations when $\pi^\alpha$ and $\pi^\beta$ are selected from $\{\Delta_1, \cdots, \Delta_6\}$, and then we argue about the general case when $\pi^\alpha$ and $\pi^\beta$ are selected from the general set of bijective maps.

- In Lemma F.24, we show that for two maps that are randomly selected from $\{\Delta_1, \cdots, \Delta_6\}$, the correlation is 1 with probability $1/3$.

- The remaining possibility is when $\pi^\alpha$ and $\pi^\beta$ belong to Mirrorset($\pi_i$) and Mirrorset($\pi_j$) for some $\pi_i, \pi_j \in \{\Delta_1, \cdots, \Delta_6\}$. We show in Lemma F.23, correlation of $\pi^\alpha$ and $\pi^\beta$ will be equal to correlation of $\pi_i, \pi_j$.

Thus, we can combine all the observations to show that for two random maps $\pi^\alpha, \pi^\beta$ that belong to Mirrorset($\pi_i$) and Mirrorset($\pi_j$) for some $\pi_i, \pi_j \in \{\Delta_1, \cdots, \Delta_6\}$,

$$\text{Correlation}(\pi^\alpha(\cdot)_1, \pi^\beta(\cdot)_1, \mathcal{U}(\{0,1\}^2))$$

$$= \text{Correlation}(\pi_i(\cdot)_1, \pi_j(\cdot)_1, \mathcal{U}(\{0,1\}^2)) = \begin{cases} 1, & \text{w.p. } 1/3 \text{ w.r.t. randomness in } \pi_i, \pi_j \\ 0, & \text{otherwise.} \end{cases}$$

$\square$

**Lemma F.23.** *The following holds true for any maps $\pi^\alpha$ and $\pi^\beta$ with $\pi^\alpha, \pi^\beta \in \{\Delta_1, \cdots, \Delta_6\}$ and for all $\tilde{\pi}^\alpha \in$ Mirrorset($\pi^\alpha$), $\tilde{\pi}^\beta \in$ Mirrorset($\pi^\beta$):*

$$Correlation(\pi^\alpha(\cdot)_1, \pi^\beta(\cdot)_1, \mathcal{U}(\{0,1\}^2)) = Correlation(\tilde{\pi}^\alpha(\cdot)_1, \tilde{\pi}^\beta(\cdot)_1, \mathcal{U}(\{0,1\}^2)),$$

*Proof.* We prove as follows:

$$\text{Correlation}(\pi^\alpha(\cdot)_1, \pi^\beta(\cdot)_1, \mathcal{U}(\{0,1\}^2)) = \left| \Pr_{x \sim \mathcal{U}(\{0,1\}^2)}[\pi^\alpha(x)_1 \neq \pi^\beta(x)_1] - \Pr_{x \sim \mathcal{U}(\{0,1\}^2)}[\pi^\alpha(x)_1 = \pi^\beta(x)_1] \right|$$

$$= \left| 2 \Pr_{x \sim \mathcal{U}(\{0,1\}^2)}[\pi^\alpha(x)_1 \neq \pi^\beta(x)_1] - 1 \right| \tag{15}$$

$$= \begin{cases} \left| 2 \Pr_{x \sim \mathcal{U}(\{0,1\}^2)}[\tilde{\pi}^\alpha(x)_1 \neq \tilde{\pi}^\beta(x)_1] - 1 \right|, \text{if condition "c1" is true} \\ \left| 1 - 2 \Pr_{x \sim \mathcal{U}(\{0,1\}^2)}[\tilde{\pi}^\alpha(x)_1 = \tilde{\pi}^\beta(x)_1] \right|, \text{if condition "c2" is true} \end{cases} \tag{16}$$

$$= \left| \Pr_{x \sim \mathcal{U}(\{0,1\}^2)}[\tilde{\pi}^\alpha(x)_1 \neq \tilde{\pi}^\beta(x)_1] - \Pr_{x \sim \mathcal{U}(\{0,1\}^2)}[\tilde{\pi}^\alpha(x)_1 = \tilde{\pi}^\beta(x)_1] \right|$$

$$= \text{Correlation}(\tilde{\pi}^\alpha(\cdot)_1, \tilde{\pi}^\beta(\cdot)_1, \mathcal{U}(\{0,1\}^2)), \tag{17}$$

where the second and the penultimate steps follow from the law of total probability. Here, condition "c1" holds when either case is true,

$$\tilde{\pi}^\alpha(x)_1 = \mathbf{not}(\pi^\alpha(x)_1), \quad \tilde{\pi}^\beta(x)_1 = \mathbf{not}(\pi^\beta(x)_1), \quad \text{for all } x \in \{0,1\}^2$$
$$\tilde{\pi}^\alpha(x)_1 = \pi^\alpha(x)_1, \quad \tilde{\pi}^\beta(x)_1 = \pi^\beta(x)_1, \quad \text{for all } x \in \{0,1\}^2.$$

and condition "c2" holds when either case is true,

$$\tilde{\pi}^{\alpha}(x)_1 = \pi^{\alpha}(x)_1, \quad \tilde{\pi}^{\beta}(x)_1 = \mathbf{not}(\pi^{\beta}(x)_1), \quad \text{for all } x \in \{0,1\}^2$$
$$\tilde{\pi}^{\alpha}(x)_1 = \mathbf{not}(\pi^{\alpha}(x)_1), \quad \tilde{\pi}^{\beta}(x)_1 = \pi^{\beta}(x)_1, \quad \text{for all } x \in \{0,1\}^2.$$

One of condition "c1" or condition "c2" is true because $\tilde{\pi}^{\alpha}$ and $\pi^{\alpha}$ (similarly, $\tilde{\pi}^{\beta}$ and $\pi^{\beta}$) are mirror maps.

$\square$

**Lemma F.24.** *The following holds true for two randomly selected maps $\pi^{\alpha}$ and $\pi^{\beta}$ with $\pi^{\alpha}, \pi^{\beta} \in \{\Delta_1, \cdots, \Delta_6\}$ :*

- *With probability $1/3$ w.r.t. random selection,*

$$Correlation(\pi^{\alpha}(\cdot)_1, \pi^{\beta}(\cdot)_1, \mathcal{U}(\{0,1\}^2)) > 0 \quad (= 1).$$

- *With probability $1/3$ w.r.t. random selection,*

$$Correlation(\pi^{\alpha}(\cdot)_2, \pi^{\beta}(\cdot)_2, \mathcal{U}(\{0,1\}^2)) > 0 \quad (= 1).$$

*Proof.* We prove for case 1, proof for case 2 is analogous.

Among $\{\Delta_1, \cdots, \Delta_6\}$, we can create three family of maps: $F_{\mathbf{copy}_1} : \{\Delta_1, \Delta_2\}$, $F_{\mathbf{copy}_2} : \{\Delta_3, \Delta_4\}$, $F_{\mathbf{xor}} : \{\Delta_5, \Delta_6\}$ that are identical in operation ($\mathbf{copy}_1, \mathbf{copy}_2, \mathbf{xor}$ respectively) at the first character in output tuple. This would imply, if the $\pi^{\alpha}$ and $\pi^{\beta}$ both belong to one of these families, $Correlation(\pi^{\alpha}(\cdot)_1, \pi^{\beta}(\cdot)_1, \mathcal{U}(\{0,1\}^2))$ will be 1.

On the other hand, one can show that for any operation $f_1, f_2 \in \{\mathbf{copy}_1, \mathbf{copy}_2, \mathbf{xor}\}$ with $f_1 \neq f_2$ will have $Correlation(f_1, f_2, \mathcal{U}(\{0,1\}^2)) = 0$. That would then suggest that if $\pi^{\alpha}$ and $\pi^{\beta}$ belong to different families among $F_{\mathbf{copy}_1}, F_{\mathbf{copy}_2}, F_{\mathbf{xor}}$, then $Correlation(\pi^{\alpha}(\cdot)_1, \pi^{\beta}(\cdot)_1, \mathcal{U}(\{0,1\}^2))$ will be 0.

By a simple counting argument, with probability $\frac{1}{3}$, two maps $\pi^{\alpha}$ and $\pi^{\beta}$ randomly selected from $\{\Delta_1, \cdots, \Delta_6\}$ will have non-zero correlation. $\square$

# G. Upper Bound: Context-Enhanced Learning of $\text{MLT}(d, n)$ with Simple Surrogate Model

## G.1. Setup of Surrogate Model with In-context Capability

Given $d + 1$ alphabets $A_1, \ldots, A_{d+1}$ of size $n$ and $d$ bijective phrasebooks $\pi_i : A_i^2 \to A_{i+1}^2$ The input of the translation process is an even-length sequence in the first alphabet, which we denote as $s_1 \in A_1^L$ where $L$ is the sequence length. The translation process modifies the input string recursively from $s_i$ to $s_{i+1}$ through the following 2 sub-processes:

1. *Circular shift*: The characters in $s_i \in A_i^L$ are shifted by 1 character leftward (and wrapped around to the end if necessary) to give sequence $\tilde{s}_i \in A_i^L$. Formally, for each $j \in [1, L]$ we have $\tilde{s}_{i,j} = s_{i,(j+1)\%L}$.

2. *Translate*: Using the *phrasebook* $\pi_i : A_i^2 \to A_{i+1}^2$, we translate 2-tuples (bigrams) of consecutive characters in sequence $\tilde{s}_i$ to create $s_{i+1}$. That is, for every odd $j \in [1, L]$, $(s_{i+1,j}, s_{i+1,j+1}) = \pi_i(\tilde{s}_{i,j}, \tilde{s}_{i,j+1})$.

Now let us revisit the surrogate model introduced in Section 5.1. Without loss of generality let $A_1 = A_2 = \cdots = A_{d+1} := A = \{1, 2, \ldots, n\}$. For any single character $a \in A$, let its vector representation be a one-hot vector $e_a \in \mathbb{R}^n$ such that $(e_a)_a = 1$. For any 2-tuple $(a, b) \in A^2$, let its vector representation be a $n^2$-dimensional vector $v(a, b) \triangleq e_a \otimes e_b$. Note that $v(a, b)$ is also a one-hot vector where

$$v(a, b)_i = \begin{cases} 1 & \text{if } i = an + b \\ 0 & \text{elsewhere} \end{cases}.$$

We use the notation $\bar{e}_a$ (long one-hot) to denote a one-hot vector in $\mathbb{R}^{n^2}$ with $a$-th position being 1 to avoid confusion.

---

**Definition G.1** (Matrix Representation of Sequence).
For a length-$L$ input sequence $s_i = (s_{i,1}, \ldots, s_{i,L})$, let its matrix representation be $\mathbf{Mat}(s_i) \triangleq V_i \in \mathbb{R}^{n^2 \times L/2}$ that

$$V_i = \begin{bmatrix} | & | & \cdots & | \\ v(s_{i,1}, s_{i,2}) & v(s_{i,3}, s_{i,4}) & \cdots & v(s_{i,L-1}, s_{i,L}) \\ | & | & \cdots & | \end{bmatrix}$$

For each $j \in [L/2]$, we use $V_i^{(j)}$ to denote the $j$-th column of $V_i$. We also denote the above conversion from a sequence $s_i$ to its matrix form as $V_i = \mathbf{Mat}(s_i)$ and assume that $V_1$ serves as the input to the surrogate model.

---

Note that the matricization operation is invertible by construction: for each column $V_i^{(j)}$, let $x = \arg\max V_i^{(j)}$, we may read off the two characters in the original alphabet by computing $\mathbf{Mat}^{-1}(V_i^{(j)}) = (\lceil x/n \rceil, x\%n)$.

At each level of translation, we assume the surrogate model will perform the following operations to $V_i$:

**(a) Circular shift from $V_i$ to $\tilde{V}_i$**

---

**Definition G.2** (Circular Shifting Operator `Shift`).
Given a matrix representation $V \in \mathbb{R}^{n^2 \times L/2}$ of a sequence $s$, the circular shifting operator `Shift` acts on $V$ as `Shift`$(V) := \tilde{V} \in \mathbb{R}^{n^2 \times L/2}$ where for all $j \in [L/2]$,

$$\tilde{V}^{(j)} = QV^{(j)} \odot Q^\top V^{((j+1)\%L)}$$

where $Q = (I_n \otimes \mathbf{1}_n)(\mathbf{1}_n \otimes I_n)^\top$, $\mathbf{1}_n \in \mathbb{R}^{n \times 1}$ is the all-ones vector, and $\odot$ is the Hadamard product.

---

**Lemma G.3** (Equivalence of `Shift` and circular shift).
*For any sequence $s \in A^L$, let $\tilde{s}$ be the circular shifted $s$, then*

$$\mathbf{Mat}(\tilde{s}) = \textit{Shift}(\mathbf{Mat}(s)).$$

*Proof of Lemma G.3.* In this proof we will show that $Mat(\tilde{s})$ and $\text{Shift}(Mat(s))$ agrees on every column.

Fix a column $j \in [n/2]$, without loss of generality let the input sequence $s$ be $(a, b, c, d) \in A$ starting from the $(2j-1)$-th position to the $(2j+2)$-th position (wrapped around when necessary). By construction each output column of $\tilde{V}^{(j)}$ is dependent on at most $V^{(j)}$ and $V^{(j+1)}$ which corresponds to 4 characters in the sequence $s$.

By definition of $V$ we then have $V^{(j)} = v(a, b) = e_a \otimes e_b$ and $V^{(j+1)\%L} = v(c, d) = e_c \otimes e_d$. It follows that

$$
\begin{aligned}
\tilde{V}^{(j)} &= QV^{(j)} \odot Q^\top V^{((j+1)\%L)} \\
&= \left((I_n \otimes \mathbf{1}_n)\left(\mathbf{1}_n^\top \otimes I_n^\top\right)(e_a \otimes e_b)\right) \odot \left((\mathbf{1}_n \otimes I_n)\left(I_n^\top \otimes \mathbf{1}_n^\top\right)(e_c \otimes e_d)\right) \\
&= \left((I_n \otimes \mathbf{1}_n)\left(\mathbf{1}_n^\top e_a \otimes I_n^\top e_b\right)\right) \odot \left((\mathbf{1}_n \otimes I_n)\left(I_n^\top e_c \otimes \mathbf{1}_n^\top e_d\right)\right) \\
&= \left((I_n \otimes \mathbf{1}_n)(1 \otimes e_b)\right) \odot \left((\mathbf{1}_n \otimes I_n)(e_c \otimes 1)\right) & (e_a, e_d \text{ are one-hot, } \mathbf{1}_n^\top e_a = \mathbf{1}_n^\top e_d = 1) \\
&= \left((I_n \otimes \mathbf{1}_n)(e_b \otimes 1)\right) \odot \left((\mathbf{1}_n \otimes I_n)(1 \otimes e_c)\right) \\
&= (e_b \otimes \mathbf{1}_n) \odot (\mathbf{1}_n \otimes e_c) \\
&= e_b \otimes e_c. & \text{(by definition of Kronecker product)}
\end{aligned}
$$

Note that $e_b \otimes e_c$ is just $Mat(\tilde{s})^{(j)}$ since the $(2j-1)$-th and the $2j$-th character of the shifted sequence $\tilde{s}$ is now $(b, c)$.

This concludes the proof. $\qquad\square$

### (b) Translation from $\tilde{V}_i$ to $V_{i+1}$

With $\text{Shift}$ effectively completing *Merge*, *Circular shift*, and *Split*, what remains is the translation leveraging $\pi_i$. Since $A_i = A_{i+1} = A = [n]$, there is a natural bijection between the space of binary column-stochastic matrix and the space of all possible (not necessarily bijective) mappings between 2-tuples from $A$. Concretely

> **Definition G.4** (Column-Stochastic Matrix Representation of phrasebook).
> Given phrasebook $\pi : A^2 \to A^2$, its matrix representation is defined to be $\text{Matrix}(\pi) \in \mathbb{R}^{n^2 \times n^2}$ that for $i, j \in [n^2]$,
> $$
> \text{Matrix}(\pi)_{j,i} = \begin{cases} 1 & \text{if } \pi(\lceil i/n \rceil, i\%n) = (\lceil j/n \rceil, j\%n) \\ 0 & \text{elsewhere} \end{cases}.
> $$

Let $V_{i+1} = \text{Matrix}\tilde{V}_i$ where $\text{Matrix}$ is a binary column-stochastic matrix, we can show that, a complete translation process for one level can be formally expressed as follows:

> **Lemma G.5** (Equivalence of $\text{Matrix}$ and *Translate*).
> *For any sequence $s_i \in A^L$, let $\pi_i$ be a bijective phrasebook $A^2 \to A^2$ defined in Section 2.2, then*
> $$
> V_{i+1} = \text{Matrix}(\pi_i)\,\text{Shift}\left(\mathbf{Mat}(s_i)\right) = \mathbf{Mat}\left(T_{\pi_i}(s_i)\right) = \mathbf{Mat}(s_{i+1}).
> $$

*Proof of Lemma G.5.* Fix any column $j \in [L/2]$, let $(a, b)$ be the $(2j-1)$-th and the $2j$-th character of the shifted sequence $\tilde{s}_i$. Let $(c, d) = \pi_i(a, b)$, by the translation construction we know the $(2j-1)$-th and the $2j$-th character of $s_{i+1}$ is just $c$ and $d$. Hence $\mathbf{Mat}(s_{i+1})^{(j)} = v(c, d)$.

From Lemma G.3 we know that $\text{Shift}(\mathbf{Mat}(s_i))^{(j)} = v(a, b)$. By construction of $\text{Matrix}(\pi_i)$ above, we know that $\text{Matrix}(\pi_i)\,\text{Shift}(\mathbf{Mat}(s_i))^{(j)}$ is a one-hot vector at the $(cn + d)$-th position, i.e. $V_{i+1}^{(j)} = \text{Matrix}(\pi_i)\,\text{Shift}(\mathbf{Mat}(s_i))^{(j)} = e_c \otimes e_c = v(c, d)$. Since $\mathbf{Mat}(s_{i+1})^{(j)} = V_{i+1}^{(j)}$ for all $j$, we have $V_{i+1} = \mathbf{Mat}(s_{i+1})$. $\qquad\square$

**Parameterization of the Translation Matrix $P$.**

With the two key operations in place, now we can introduce the surrogate model in its full detail. Since the multi-level translation task is a naturally sequential operation, we model the surrogate operations as a multi-layer network as well (which also matches with the solution found by Llama-based models when trained on real data as in Section 3.3).

For each level, the surrogate model needs to present both in-context learning capability at the initialization and in-weight capability toward the end of context-enhanced learning on a certain set of phrasebooks $\mathbf{\Pi}^*$. The in-weight capability requires certain parameter to store $\mathbf{\Pi}^*$ on its own that is independent of the context.

To capture both capabilities at the same time, we parameterize the translation matrix $P_i$ as a combination of in-context information $C_i \in \mathbb{R}^{n^2 \times n^2}$ and in-weight memory $W_i \in \mathbb{R}^{n^2 \times n^2}$.

---

**Definition G.6** (Effective Translation Matrix).
For in-context information $C_i \in \mathbb{R}^{n^2 \times n^2}$ and in-weight memory $W_i \in \mathbb{R}^{n^2 \times n^2}$ at level $i$, the corresponding effective translation matrix $P_i$ is defined as

$$P_i = \texttt{HardMax}\left(C_i + W_i\right)$$

where $\texttt{HardMax}$ is the column-wise hard-max function converting $C_i + W_i$ to a binary column stochastic matrix.

---

Note that column $k$ in $P_i$, which we denote as $P_i^{(k)}$, is equal to the one-hot vector at $\arg\max(C_i^{(k)} + W_i^{(k)})$.

**Surrogate Model with In-Context Capability**

Now we can formally introduce the surrogate model.

---

**Definition G.7** (Surrogate Model for **MLT**).
The surrogate model for $\textbf{MLT}(d, n)$ can be represented by the recursive expression

$$V_{i+1} = \texttt{HardMax}(C_i + W_i)\,\texttt{Shift}(V_i) \tag{18}$$

The learnable parameters for the surrogate model are the weight matrices $\{W_i\}_{i=1}^{d} := \{W_1, W_2, \ldots, W_d\}$. We denote the surrogate model parameterized by $\{W_i\}_{i=1}^{d}$ as $\textsc{Surr-MLT}_{\{W_i\}_{i=1}^{d}}(\cdot)$ which maps the input and the in-context information to the output as

$$\begin{aligned}
V_{d+1} &= \textsc{Surr-MLT}_{\{W_i\}_{i=1}^{d}}(C_1, C_2, \ldots, C_d, V_1) \\
&\triangleq \texttt{HardMax}(C_d + W_d) \\
&\quad \texttt{Shift}\left(\texttt{HardMax}(C_{d-1} + W_{d-1})\,\texttt{Shift}\left(\cdots \texttt{HardMax}(C_1 + W_1)\,\texttt{Shift}(V_1)\cdots\right)\right).
\end{aligned} \tag{19}$$

---

In this surrogate model, the in-context descriptive text $\textsc{desc}$ is just $\{C_1, \ldots, C_d\}$, where each matrix is directly being passed into the corresponding layer. When providing information about a set of phrasebooks $\mathbf{\Pi} = \{\pi_i\}_{i=1}^{d}$, we have $C_i = \texttt{Matrix}(\pi_i)$ where $\texttt{Matrix}(\pi_i)$ is the column-stochastic matrix representation of $\pi_i$ as defined in Definition G.4.

When partial phrasebook information is provided (corresponding to dropping certain $\texttt{ab->CD}$ entries in the language model context), we zero-out the corresponding column in $C_i$. When no in-context information is provided for level $i$, we just have $C_i$ be the all-zero matrix containing no information (assuming zero as prior).

For simplicity of presentation, in Definition G.7 the in-context information $C_i$'s are provided directly to the corresponding layers. We note that the equivalent operations can be exactly re-parameterized such that $C_i$'s are provided in-context (in concatenation with $V_1$). The reparameterization of the surrogate model is provided below:

**Definition G.8** (Context-Augmented Surrogate Model for **MLT**).
Let the context-augmented input be

$$\mathbf{X}_1 = [\boldsymbol{C}_1, \ldots, \boldsymbol{C}_d, \boldsymbol{V}_1] \in \mathbb{R}^{n^2 \times (dn^2 + L)},$$

then we can rewrite the same surrogate model as

$$
\begin{aligned}
\mathbf{X}_{i+1} &= \mathbf{X}_i \begin{bmatrix} I_{dn^2} & 0 \\ 0 & 0_{L \times L} \end{bmatrix} + \left( \left( \mathbf{X}_i \begin{bmatrix} e_i \otimes I_{n^2} \\ 0_{L \times n^2} \end{bmatrix} + \boldsymbol{W}_i \right) \mathtt{Shift}\left( \mathbf{X}_i \begin{bmatrix} 0_{n^2 d \times L} \\ I_L \end{bmatrix} \right) \begin{bmatrix} 0_{dn^2 \times dn^2} & 0 \\ 0 & I_L \end{bmatrix} \right) \\
&= [\boldsymbol{C}_1, \ldots, \boldsymbol{C}_d, 0_{n^2 \times L}] + [0, \ldots, 0, (\boldsymbol{C}_i + \boldsymbol{W}_i)\, \mathtt{Shift}\,(\boldsymbol{V}_i)] \\
&= [\boldsymbol{C}_1, \ldots, \boldsymbol{C}_d, \boldsymbol{V}_{i+1}]
\end{aligned}
\tag{20}
$$

With the reparameterization, the model is capable of generating $[\boldsymbol{C}_1, \ldots, \boldsymbol{C}_d, \boldsymbol{V}_{d+1}]$ with input $[\boldsymbol{C}_1, \ldots, \boldsymbol{C}_d, \boldsymbol{V}_1]$, with all information provided in-context in the input.

Now let us check what does Definition 2.2 (ICL-capable) and Definition 2.1 (specific task-capable) mean in the context of the surrogate model. To make things more rigorous we introduce two stronger notions of capabilities:

**Definition G.9** (Strongly **MLT**$(d, n)$-ICL-capable surrogate model).
We say a surrogate model $\textsc{Surr-MLT}_{\{\boldsymbol{W}_i\}_{i=1}^d}(\cdot)$ is strongly **MLT**$(d, n)$-ICL-capable if for any set of phrasebooks $\boldsymbol{\Pi} = \{\pi_i\}_{i=1}^d$ in **MLT**$(n, d)$, for any input sequence $\boldsymbol{s}_1 \in A^L$ where $L$ is even, we have

$$\textsc{Surr-MLT}_{\{\boldsymbol{W}_i\}_{i=1}^d}(\mathtt{Matrix}(\pi_1), \ldots, \mathtt{Matrix}(\pi_d), \mathbf{Mat}(\boldsymbol{s}_1)) = \mathbf{Mat}(\mathbf{MLT}_{\boldsymbol{\Pi}}(\boldsymbol{s}_1)).$$

**Definition G.10** (Strongly **MLT**$_{\boldsymbol{\Pi}^*}$-capable surrogate model).
For a fixed set of phrasebooks $\boldsymbol{\Pi}^* = \{\pi_i^*\}_{i=1}^d$ in **MLT**$(n, d)$, we say a surrogate model $\textsc{Surr-MLT}_{\{\boldsymbol{W}_i\}_{i=1}^d}(\cdot)$ is strongly **MLT**$_{\boldsymbol{\Pi}^*}$-capable if for any input sequence $\boldsymbol{s}_1 \in A^L$ where $L$ is even, we have

$$\textsc{Surr-MLT}_{\{\boldsymbol{W}_i\}_{i=1}^d}(\boldsymbol{0}, \ldots, \boldsymbol{0}, \mathbf{Mat}(\boldsymbol{s}_1)) = \mathbf{Mat}(\mathbf{MLT}_{\boldsymbol{\Pi}^*}(\boldsymbol{s}_1)).$$

Now we can show the following properties of the surrogate model $\textsc{Surr-MLT}_{\{\boldsymbol{W}_i\}_{i=1}^d}(\cdot)$:

**Lemma G.11.** *When $\|\boldsymbol{W}_i\|_0 < \frac{1}{2}$ for all $i \in [d]$, $\textsc{Surr-MLT}_{\{\boldsymbol{W}_i\}_{i=1}^d}$ is strongly* **MLT**$(d, n)$-*ICL-capable.*

*Proof.* Fix any set of phrasebooks $\boldsymbol{\Pi} = \{\pi_i\}_{i=1}^d$ and its corresponding matrix representations $\{\mathtt{Matrix}(\pi_i)\}_{i=1}^d$. Since $\|\boldsymbol{W}_i\|_0 < \frac{1}{2}$, for any column $k$, no entries in $\boldsymbol{W}_i^{(k)}$ can flip the argmax of $\mathtt{Matrix}(\pi_i)^{(k)} + \boldsymbol{W}_i^{(k)}$ away from being $\arg\max \mathtt{Matrix}(\pi_i)^{(k)}$. Therefore we have $\boldsymbol{P}_i = \mathtt{HardMax}(\mathtt{Matrix}(\pi_i) + \boldsymbol{W}_i) = \mathtt{Matrix}(\pi_i)$ for all layers $i \in [d]$.

By Lemma G.5, for all $i \in [d]$ we have $\boldsymbol{P}_i = \mathtt{Matrix}(\pi_i)$ recovering $T_{\pi_i}$. Hence for any input sequence $\boldsymbol{s}_1 \in A^L$, we have $\textsc{Surr-MLT}_{\{\boldsymbol{W}_i\}_{i=1}^d}(\{(\pi_1)\}, \ldots, \{(\pi_d)\}, \mathbf{Mat}(\boldsymbol{s}_1)) = \mathbf{Mat}(\mathbf{MLT}_{\boldsymbol{\Pi}}(\boldsymbol{s}_1))$. $\quad\square$

**Lemma G.12.** *Fix a target set of phrasebooks $\boldsymbol{\Pi}^* = \{\pi_i^*\}_{i=1}^d$, when $\mathtt{HardMax}(\boldsymbol{W}_i) = \mathtt{Matrix}(\pi_i^*)$ for all $i \in [d]$, $\textsc{Surr-MLT}_{\{\boldsymbol{W}_i\}_{i=1}^d}(\cdot)$ is strongly* **MLT**$_{\boldsymbol{\Pi}^*}$-*capable.*

*Proof.* By Lemma G.5, for all $i \in [d]$ we have $\boldsymbol{P}_i = \mathtt{HardMax}(\boldsymbol{W}_i + \boldsymbol{0}) = \mathtt{Matrix}(\pi_i^*)$ recovering $T_{\pi_i^*}$. Hence for any input sequence $\boldsymbol{s}_1 \in A^L$, we have $\textsc{Surr-MLT}_{\{\boldsymbol{W}_i\}_{i=1}^d}(\boldsymbol{0}, \ldots, \boldsymbol{0}, \mathbf{Mat}(\boldsymbol{s}_1)) = \mathbf{Mat}(\mathbf{MLT}_{\boldsymbol{\Pi}^*}(\boldsymbol{s}_1))$. $\quad\square$

Lemma G.11 suggests that when the weight matrices have small initializations, the model has perfect ICL capability. Meanwhile Lemma G.12 suggests that when the weights $\boldsymbol{W}_i$ recover $\texttt{Matrix}(\pi_i^*)$ in the column-wise hard-max sense, then the surrogate model can perform $\mathbf{MLT}_{\boldsymbol{\Pi}^*}$ when no context is being provided ($\boldsymbol{C}_i = \boldsymbol{0}$).

### G.2. Learning $\boldsymbol{\Pi}^*$ in $\mathbf{MLT}(d, n)$ with Heuristics Search

In this section, we provide a brute-force algorithm that can learn any target set of phrasebooks $\boldsymbol{\Pi}^*$ in $\mathbf{MLT}(d, n)$ using a single "short" sequence whose length is not exponentially dependent on $d$.

Before proceeding to the details, let us first investigate more on the nature of $\mathbf{MLT}$ and the surrogate model. For simplicity of notations, given $\boldsymbol{\Pi}^* = \{\pi_1^*, \ldots, \pi_d^*\}$, we denote the general translation operator $\texttt{Matrix}$ as $\boldsymbol{P}$, and denote the translation operator $\texttt{Matrix}^{(\pi_1^*)}$ as $\boldsymbol{W}_i^*$. Also, with slight abuse of notations we use $\mathbf{MLT}_{\boldsymbol{\Pi}^*}(\boldsymbol{V}_1)$ to denote $\mathbf{Mat}\left(\mathbf{MLT}_{\boldsymbol{\Pi}^*}(\mathbf{Mat}^{-1}(\boldsymbol{V}_1))\right)$.

First, we characterize the input sequence that is good for providing learning signals.

> **Definition G.13** ($\boldsymbol{\Pi}^*$-coverable input).
> Fix a target set of phrasebooks $\boldsymbol{\Pi}^* = \{\pi_1^*, \ldots, \pi_d^*\}$ in $\mathbf{MLT}(d, n)$ and an input matrix $\boldsymbol{V}_1 \in \mathbb{R}^{n^2 \times L}$, let $\tilde{\boldsymbol{V}}_1^*, \boldsymbol{V}_2^*, \tilde{\boldsymbol{V}}_2^*, \ldots, \tilde{\boldsymbol{V}}_d^*, \boldsymbol{V}_{d+1}^* \in \mathbb{R}^{n^2 \times L}$ be the intermediate outputs when applying $\mathbf{MLT}_{\boldsymbol{\Pi}^*}$ on $\boldsymbol{V}_1$. We say $\boldsymbol{V}_1$ is $\boldsymbol{\Pi}^*$-coverable if for all levels $i \in [d]$, $\tilde{\boldsymbol{V}}_i^*$ is of rank-$n^2$.

Note that as a matrix with only one-hot columns, $\tilde{\boldsymbol{V}}_i^*$ being rank-$n^2$ suggests that for all $k \in [n^2]$, there exists some column $j \in [L]$ such that $\tilde{\boldsymbol{V}}_i^{*(j)} = \boldsymbol{e}_k$. In the context of the translation process, it means that the correct translation process of a $\boldsymbol{\Pi}^*$-coverable input $\boldsymbol{V}_1$ would require all entries of all phrasebooks in $\boldsymbol{\Pi}^*$.

Next we will show that if we use the context to condition all translation operators $\boldsymbol{P}_l$ of the surrogate model to be $\boldsymbol{P}^{(\pi_i^*)}$ for all but one level $l \in [d] \setminus \{i\}$ ($i$ as the unconditioned level), then for the surrogate model to correctly perform $\mathbf{MLT}_{\boldsymbol{\Pi}^*}$, the operator for the unconditioned level $i$ must also be equal to $\boldsymbol{P}^{(\pi_i^*)}$.

**Lemma G.14** (Uniqueness of a single $\boldsymbol{P}_i$ when conditioning all other levels).

*Fix a target set of phrasebooks $\boldsymbol{\Pi}^* = \{\pi_1^*, \ldots, \pi_d^*\}$ in $\mathbf{MLT}(d, n)$ and a $\boldsymbol{\Pi}^*$-coverable input $\boldsymbol{V}_1$. For any level $i \in [d]$, consider a surrogate model $\textsc{Surr-MLT}_{\{\boldsymbol{W}_i\}_{i=1}^d}(\boldsymbol{C}_1, \boldsymbol{C}_2, \ldots, \boldsymbol{C}_d, \cdot)$ with certain context $\boldsymbol{C}_1, \boldsymbol{C}_2, \ldots, \boldsymbol{C}_d$ such that $\boldsymbol{P}_l = \boldsymbol{W}_l^*$ for all $l \in [d]$ except $l = i$. Then $\textsc{Surr-MLT}_{\{\boldsymbol{W}_i\}_{i=1}^d}(\boldsymbol{C}_1, \boldsymbol{C}_2, \ldots, \boldsymbol{C}_d, \boldsymbol{V}_1) = \mathbf{MLT}_{\boldsymbol{\Pi}^*}(\boldsymbol{V}_1)$ if and only if*

$$\boldsymbol{P}_i = \texttt{HardMax}\left(\boldsymbol{C}_i + \boldsymbol{W}_i\right) = \boldsymbol{W}_i^*.$$

*Proof.* This lemma is a direct consequence of the bijective property of the translation process shown in Lemma E.1. Let $\tilde{\boldsymbol{V}}_1^*, \boldsymbol{V}_2^*, \tilde{\boldsymbol{V}}_2^*, \ldots, \tilde{\boldsymbol{V}}_d^*, \boldsymbol{V}_{d+1}^* \in \mathbb{R}^{n^2 \times L}$ be the intermediate outputs when applying $\mathbf{MLT}_{\boldsymbol{\Pi}^*}$ on $\boldsymbol{V}_1$, and let $\tilde{\boldsymbol{V}}_1, \boldsymbol{V}_2, \tilde{\boldsymbol{V}}_2, \ldots, \tilde{\boldsymbol{V}}_d, \boldsymbol{V}_{d+1} \in \mathbb{R}^{n^2 \times L}$ be the intermediate outputs when applying $\textsc{Surr-MLT}_{\{\boldsymbol{W}_i\}_{i=1}^d}(\boldsymbol{C}_1, \boldsymbol{C}_2, \ldots, \boldsymbol{C}_d, \cdot)$ as described. Since we assume $\boldsymbol{P}_l = \boldsymbol{P}^{(\pi_l^*)}$ for all $l < i$, we have $\boldsymbol{V}_i = \boldsymbol{V}_i^*$ and therefore $\tilde{\boldsymbol{V}}_i = \tilde{\boldsymbol{V}}_i^*$.

On the other end, since $\textsc{Surr-MLT}_{\{\boldsymbol{W}_i\}_{i=1}^d}(\boldsymbol{C}_1, \boldsymbol{C}_2, \ldots, \boldsymbol{C}_d, \boldsymbol{V}_1) = \mathbf{MLT}_{\boldsymbol{\Pi}^*}(\boldsymbol{V}_1) = \boldsymbol{V}_{d+1}^*$ and $\boldsymbol{P}_l = \boldsymbol{P}^{(\pi_l^*)}$ for all $l > i$, by the invertible property of the translation process (Lemma E.1) we must have $\tilde{\boldsymbol{V}}_{i+1} = \tilde{\boldsymbol{V}}_{i+1}^*$ and thus $\boldsymbol{V}_{i+1} = \boldsymbol{V}_{i+1}^*$. Combining both ends we know that

$$\boldsymbol{P}_i \tilde{\boldsymbol{V}}_i = \boldsymbol{V}_{i+1} = \boldsymbol{V}_{i+1}^* = \boldsymbol{W}_i^* \tilde{\boldsymbol{V}}_i^* = \boldsymbol{W}_i^* \tilde{\boldsymbol{V}}_i. \tag{21}$$

Since $\tilde{\boldsymbol{V}}_i = \tilde{\boldsymbol{V}}_i^*$ is rank $n^2$ by the $\boldsymbol{\Pi}^*$-coverable assumption and $\boldsymbol{P}_i \in \mathbb{R}^{n^2 \times n^2}$, it must be so that $\boldsymbol{P}_i = \boldsymbol{W}_i^*$. $\qquad\square$

If we further condition on the held-out level $\boldsymbol{P}_i$ such that we only leave one column of $\boldsymbol{P}_i^{(k)}$ not necessarily equal to $\boldsymbol{W}_i^{*(k)}$, we have the following corollary

**Corollary G.15** (Uniqueness of a single $\boldsymbol{P}_i$ column when conditioning everything else).

*Fix a target set of phrasebooks $\boldsymbol{\Pi}^* = \{\pi_1^*, \ldots, \pi_d^*\}$ in $\mathbf{MLT}(d, n)$ and a $\boldsymbol{\Pi}^*$-coverable input $\boldsymbol{V}_1$. For any level $i \in [d]$ and translation entry $k \in [n^2]$, consider a surrogate model $\textsc{Surr-MLT}_{\{\boldsymbol{W}_i\}_{i=1}^d}(\boldsymbol{C}_1, \boldsymbol{C}_2, \ldots, \boldsymbol{C}_d, \cdot)$*

*with certain context $C_1, C_2, \ldots, C_d$ such that $P_l^{(j)} = W_l^{*(j)}$ for all $(l, j) \in [d] \times [n^2] \backslash \{(i, k)\}$. Then* $\text{SURR-MLT}_{\{W_i\}_{i=1}^d}(C_1, C_2, \ldots, C_d, V_1) = \text{MLT}_{\Pi^*}(V_1)$ *if and only if*

$$P_i^{(k)} = \text{HardMax}\left(C_i^{(j)} + W_i^{(j)}\right) = W_i^{*(k)}.$$

This suggests that if we condition everything else except for one column of the translation operator, then to match the final output on a $\Pi^*$-coverable sequence, the model must recover the held-out column to be the same as the ground truth in the set of phrasebooks.

Now we can introduce the search algorithm, which simply enumerate over all translation columns $W_i^{(j)}$ as learning target, generate a contextual information that only leaves that column unconditioned, and search over all possible one-hot vectors for $W_i^{(j)}$ until the output matches with $\text{MLT}_{\Pi^*}$. Once the output matches, by Corollary G.15 we know $\text{HardMax}(W_i^{(j)})$ recovers $W_i^{*(j)}$ and we move on to the next learning target. The algorithm can be formalized as follows:

---

**Algorithm 2** Context-Enhanced Searching Algorithm for $\text{MLT}(d, n)$

---

1: **Input:**
2: input $V_1 \in \mathbb{R}^{n^2 \times L}$, label $V_{d+1}^* \in \mathbb{R}^{n^2 \times L}$, descriptive text $W_1^*, \ldots, W_d^* \in \mathbb{R}^{n^2 \times n^2}$
3:
4: **Initialize** $W_1, \ldots, W_d \leftarrow \mathbf{0}$             # Start with zero initialization
5: **for** $i = 1$ **to** $d$ **do**
6:      **for** $k = 1$ **to** $n^2$ **do**
7:          **Initialize** $C_{i(k)} \leftarrow W_i^* (I_{n^2} - \text{diag}(\bar{e}_k))$          # Create masked context matrix
8:          # Search Loop
9:          **for** $a = 1$ **to** $n^2$ **do**
10:              $W_i^{(k)} \leftarrow \bar{e}_a$             # Search over one-hot columns
11:              $V_{d+1} \leftarrow \text{SURR-MLT}_{\{W_i\}_{i=1}^d}(W_1^* \ldots, W_{i-1}^*, C_{i(k)}, W_{i+1}^* \ldots, W_d^*, V_1)$
12:              **if** $V_{d+1} = V_{d+1}^*$ **then**
13:                  **break**             # Break when found the right column
14:              **end if**
15:          **end for**
16:      **end for**
17: **end for**
18: **Return** $W_1, \ldots, W_d$.

---

> **Theorem G.16** (Learning $\Pi^*$ with context-enhanced search with $\Pi^*$-coverable input)**.**
> *For any target set of phrasebooks $\Pi^* = \{\pi_1^*, \ldots, \pi_d^*\}$ in $\text{MLT}(d, n)$, given an $\Pi^*$-coverable input $V_1$ and the corresponding ground truth label $V_{d+1}^* = \text{MLT}_{\Pi^*}(V_1)$, Algorithm 2 terminates with $W_i = W_i^*$ for all $i \in [d]$ with $O(n^4 d)$ forward passes through the surrogate model.*

*Proof.* The statement can be proven by a simple induction.

Let the inductive hypothesis be such that when the enumeration goes to $k$-th column of the $i$-th layer, if $\text{HardMax}(W_l^{(j)} + W_l^{*(j)}) = W_l^{*(j)}$ for all $(l, j) \in [d] \times [n^2]$ and $W_l^{(j)} = W_l^{*(j)}$ for all $(l, j)$ such that $l < i$ or $l = i \wedge j < k$, then the search loop (Algorithm 2: line 13) breaks with $W_i^{(k)} = W_i^{*(k)}$ while $\text{HardMax}(W_l^{(j)} + W_l^{*(j)}) = W_l^{*(j)}$ for all $(l, j) \in [d] \times [n^2]$ is preserved.

The base case is satisfied as with zero initialization, we have $\text{HardMax}(W_l^{(j)} + W_l^{*(j)}) = \text{HardMax}(\mathbf{0} + W_l^{*(j)}) = W_l^{*(j)}$ for all $(l, j) \in [d] \times [n^2]$ and there are no requirements for $W_i^{(k)} = W_i^{*(k)}$ yet.

For the induction step, we note that with the condition of $\text{HardMax}(W_l^{(j)} + W_l^{*(j)}) = W_l^{*(j)}$ for all $(l, j) \in [d] \times [n^2]$,

$\boldsymbol{W}_1^* \ldots, \boldsymbol{W}_{i-1}^*, \boldsymbol{C}_{i(k)}, \boldsymbol{W}_{i+1}^* \ldots, \boldsymbol{W}_d^*$ will correctly condition all columns of $\boldsymbol{P}$'s except for the $\boldsymbol{P}_i^{(k)}$ since

$$\boldsymbol{C}_{i(k)}^{(k)} = \boldsymbol{W}_i^* \left(\boldsymbol{I}_{n^2} - \operatorname{diag}(\bar{e}_k)\right)^{(k)} = \boldsymbol{0}. \tag{22}$$

Thus by Corollary G.15, we know that the search loop will terminate when it finds $\boldsymbol{W}_i^{(k)} = \boldsymbol{W}_i^{*(k)}$. The newly added column provides the correct inductive hypothesis on $\boldsymbol{W}_l^{(j)} = \boldsymbol{W}_l^{*(j)}$ for the next enumeration step.

By induction to $i = d$ and $k = n^2$, we will be able to recover $\boldsymbol{W}_i = \boldsymbol{W}_i^*$ for all $i \in [d]$. $\qquad\square$

Given that we can learn $\boldsymbol{W}_i$ effectively with $\boldsymbol{\Pi}^*$-coverable input, how should we construct such inputs? It turned out that with high probability, short random strings suffices.

**Lemma G.17** (Distribution of intermediate sequences). *Fix a target set of phrasebooks $\boldsymbol{\Pi}^* = \{\pi_1^*, \ldots, \pi_d^*\}$ in $\mathbf{MLT}(d, n)$. Let $\boldsymbol{V}_1 \in \mathbb{R}^{n^2 \times L}$ be a random input matrix to $\mathbf{MLT}(d, n)$ such that each column $\boldsymbol{V}_1^{(j)}$ is i.i.d. sampled from $\mathcal{U}(\{\bar{e}_k\}_{k=1}^{n^2})$ (the uniform distribution over one-hot vectors $\{\bar{e}_k\}_{k=1}^{n^2}$), the columns of intermediate random sequences $\tilde{\boldsymbol{V}}_1^*, \boldsymbol{V}_2^*, \tilde{\boldsymbol{V}}_2^*, \ldots, \tilde{\boldsymbol{V}}_d^*, \boldsymbol{V}_{d+1}^* \in \mathbb{R}^{n^2 \times L}$ obtained by passing the input $\boldsymbol{V}_1$ through $\mathbf{MLT}_{\boldsymbol{\Pi}^*}$ also follow the same i.i.d. uniform distribution.*

*Proof.* We will prove the claim by induction on depth $i$. Let the inductive hypothesis be that columns in $\boldsymbol{V}_i$ independently follow an uniform distribution over the one-hot vectors $\{\bar{e}_k\}_{k=1}^{n^2}$. Note that the base case is just the assumption.

Now we prove for the inductive step. For any $j \in [L]$, we can write $\boldsymbol{V}_i^{(j)} = \boldsymbol{e}_{a_{(2j-1)}} \otimes \boldsymbol{e}_{a_{(2j)}}$ where $a_{(i)}$'s follows i.i.d. $\mathcal{U}([n])$. Intuitively this means each 2-tuple in the random input sequence is formed from two i.i.d. uniformly random characters, which is straightforward by construction.

Now by Lemma G.3 we have $\tilde{\boldsymbol{V}}_i^{*(j)} = \boldsymbol{e}_{a_{(2j)}} \otimes \boldsymbol{e}_{a_{(2j+1)}}$ also following $\mathcal{U}(\{\bar{e}_k\}_{k=1}^{n^2})$. Note that there is total independency of $\tilde{\boldsymbol{V}}_i^{*(j)}$ with respect to the set of any other columns of $\tilde{\boldsymbol{V}}_i$ since $a_{(2j)}$ and $a_{(2j+1)}$ are independent from the generative process of any other columns in $\tilde{\boldsymbol{V}}_i$.

With columns in $\tilde{\boldsymbol{V}}_i$ i.i.d. following $\mathcal{U}(\{\bar{e}_k\}_{k=1}^{n^2})$, permuting the indices via $\boldsymbol{P}^{(\pi_i^*)}$ does not change the distribution by symmetry of the uniform distribution. Therefore we have columns of $\boldsymbol{V}_{i+1} = \boldsymbol{P}^{(\pi_i^*)} \tilde{\boldsymbol{V}}_i$ also i.i.d. distributed as $\mathcal{U}(\{\bar{e}_k\}_{k=1}^{n^2})$ and this complete the inductive step. By induction on $i$ we have the desired statement proved. $\qquad\square$

Since each column in $\tilde{\boldsymbol{V}}_i$ i.i.d. follows $\mathcal{U}(\{\bar{e}_k\}_{k=1}^{n^2})$, sampling $\tilde{\boldsymbol{V}}_i$ to be rank $n^2$ becomes identical to the classic coupon collection problem (see Lemma G.29 from Motwani (1995)). Thus we have the following bound:

> **Lemma G.18** (Short $\boldsymbol{\Pi}^*$-coverable random sequence).
> *A random sequence $\boldsymbol{V}_1$ of length $L \geq 2n^2 \log \frac{nd}{\delta}$ is $\boldsymbol{\Pi}^*$-coverable with probability at least $1 - \delta$.*

*Proof.* Let the event $A_i$ denote that $\tilde{\boldsymbol{V}}_i$ is not rank $n^2$. By Lemma G.17, each column of $\tilde{\boldsymbol{V}}_i$ is i.i.d. distributed following $\mathcal{U}(\{\bar{e}_k\}_{k=1}^{n^2})$. Thus making $\tilde{\boldsymbol{V}}_i$ being rank $n^2$ is equivalent to a coupon collecting problem (Motwani, 1995) with set size $n^2$. By Lemma G.29 we know that with $L = 2n^2 \log \frac{nd}{\delta}$, $\mathbb{P}[A_i] \leq \frac{\delta}{d}$. Thus by a simple union bound the probability that $\tilde{\boldsymbol{V}}_i$ being not $\boldsymbol{\Pi}^*$-coverable is

$$\mathbb{P}\left[\bigcup_{i=1}^d A_i\right] \leq \sum_{i=1}^d \mathbb{P}[A_i] \leq d\frac{\delta}{d} = \delta. \tag{23}$$

$\qquad\square$

Now we can apply the above result and extend Theorem G.16.

**Corollary G.19** (Learning $\Pi^*$ with random input using heuristics search)**.**
*For any target set of phrasebooks $\Pi^* = \{\pi_1^*, \ldots, \pi_d^*\}$ in $\mathbf{MLT}(d, n)$, with probability at least $1 - \delta$ over a uniformly random input $V_1$ of length $L = 2n^2 \log \frac{nd}{\delta}$, Algorithm 2 provided with ground truth label $V_{d+1}^* = \mathbf{MLT}_{\Pi^*}(V_1)$ terminates with $W_i = W_i^*$ for all $i \in [d]$ with $O(n^4 d)$ forward passes through the surrogate model.*

### G.3. Learning $\Pi^*$ in $\mathbf{MLT}(2, n)$ with Surrogate Gradient Descent

In this section we take the analysis one step beyond the heuristics searching regime. We will show that any set of phrasebooks $\Pi^* = \{\pi_1^*, \pi_2^*\}$ can be sample-efficiently learned by a gradient-descent based algorithm. In this particular case, the surrogate model is parameterized by

$$V_3 = \text{SURR-MLT}_{\{W_i\}_{i=1}^d}(C_1, C_2, V_1) \triangleq \texttt{HardMax}(C_2 + W_2)\,\texttt{Shift}\,(\texttt{HardMax}(C_1 + W_1)\,\texttt{Shift}\,(V_1)). \quad (24)$$

We start with any weight initializations $W_1^{(0)}, W_2^{(0)} \in \mathbb{R}^{n^2 \times n^2}$ satisfying $\|W_1^{(0)}\|_1 < \frac{1}{2}, \|W_2^{(0)}\|_1 < \frac{1}{2}$, by Lemma G.11 the initialization is strongly $\mathbf{MLT}(d, n)$-ICL-capable.

For simplicity, we denote the ground truth permutation matrix induced by $\pi_1^*$ as $W_1^* \triangleq P^{(\pi_1^*)}$ and similarly the ground truth permutation matrix induced by $\pi_2^*$ as $W_2^* \triangleq P^{(\pi_2^*)}$. From Lemma G.12 we know that the learning is successful if we have $\texttt{HardMax}(W_1) = W_1^*$ and $\texttt{HardMax}(W_2) = W_2^*$ [8] (i.e. the maximum index of each weight column agrees with that of the ground truth).

We employ a layer-wise gradient descent algorithm for the learning process. The algorithm takes in a single fixed sequence $s_1$ with matrix representation $V_1$ and its corresponding ground truth label

$$V_3^* \triangleq \mathbf{Mat}(\mathbf{MLT}_{\Pi^*}(s_1)) = \text{SURR-MLT}_{(W_1^{(0)}, W_2^{(0)})}(W_1^*, W_2^*, V_1). \quad (25)$$

Given the input and label, we employ the following gradient descent based algorithm to update the weights:

The training happens in a layer-wise and column-wise fashion: We first freeze $W_2$ and set $W_1$ as the trainable parameter. For each entry $k \in [n^2]$, we create a context matrix $C_{1(k)} \triangleq W_1^*(I_{n^2} - \text{diag}(e_k))$ which essentially creates a copy of $W_1^*$ except of setting the $k$-th column to be zero. Then we take a forward pass through the surrogate model with the one-column dropped-out context and get output $V_{3(1,k)} \triangleq \text{SURR-MLT}_{(W_1, W_2)}(C_{1(k)}, W_2^*, V_1)$. Here the subscript $\cdot_{(1,k)}$ denotes the final output when the $i$-th column of the first context matrix is being dropped.

The weight update follows a surrogate gradient update scheme where we use the MSE loss: $\mathcal{L} = \|V_{3(1,k)} - V_3^*\|_2^2$. Since it is difficult to take gradient through the hardmax function, we instead compute the gradient of the loss with respect to the translation matrix $P_1 = \texttt{HardMax}\,(W_1 + C_{1(k)})$ and apply the update $W_1^{(k)} \leftarrow W_1^{(k)} - \frac{\partial \mathcal{L}}{\partial P_1^{(k)}}$. We apply such gradient update *twice* for each dropped column $k$.

For the second layer, we freeze the first layer $W_1$ and apply one-column dropouts to the $C_2$. Similarly we apply the surrogate gradient update $W_2 \leftarrow W_2 - \frac{\partial \mathcal{L}}{\partial P_2}$ but we only need one gradient step per column.

We claim the surrogate gradient descent update can correctly recover $P_1^*$ and $P_2^*$ similar to the heuristics search case.

**Theorem G.24** (Learning $\Pi^*$ with context-enhanced surrogate GD with $\Pi^*$-coverable input)**.**
*For any initialization $W_{1(0)}, W_{2(0)} \in \mathbb{R}^{n^2 \times n^2}$ such that $\|W_{1(0)}\|_0 \le \frac{1}{2}$ and $\|W_{2(0)}\|_0 \le \frac{1}{2}$, for any target set of phrasebooks $\Pi^* = \{\pi_1^*, \pi_2^*\}$ in $\mathbf{MLT}(2, n)$, given an $\Pi^*$-coverable input $V_1$ and the corresponding ground truth label $V_3^* = \mathbf{MLT}_{\Pi^*}(V_1)$, Algorithm 2 terminates with $\texttt{HardMax}(W_1) = W_1^*$ and $\texttt{HardMax}(W_2) = W_2^*$.*

To prove for Theorem G.24, we will carefully analyze the learning of the first layer and second layer respectively, and provide a similar induction argument as in the proof for the heuristics search case.

---

[8] Note that this is not the unique solution to attain strongly $\mathbf{MLT}_{\Pi^*}$-capable model

**Algorithm 3** Context-Enhanced Layerwise Gradient Descent

1: **Input:** input $V_1 \in \mathbb{R}^{n^2 \times L}$, label $V_3^* \in \mathbb{R}^{n^2 \times L}$, descriptive text $W_1^*, W_2^* \in \mathbb{R}^{n^2 \times n^2}$, init $W_1^{(0)}, W_2^{(0)} \in \mathbb{R}^{n^2 \times n^2}$
2:
3: # Train the first layer
4: **for** $k = 1$ **to** $n^2$ **do**
5:    $C_{1(k)} \triangleq W_1^* (I_{n^2} - \text{diag}(e_k))$                                   # Create context matrix with $k$-th column dropped.
6:    **for** $t = 1$ **to** $2$ **do**
7:       $V_{3(1,k)} \leftarrow \text{SURR-MLT}_{(W_1, W_2)}(C_{1(k)}, W_2^*, V_1)$        # Forward pass
8:       $\mathcal{L} \leftarrow \|V_{3(1,k)} - V_3^*\|_2^2$
9:       $W_1^{(k)} \leftarrow W_1^{(k)} - \frac{\partial l}{\partial P_1^{(k)}}$                              # Surrogate gradient update
10:    **end for**
11: **end for**
12:
13: # Train the second layer
14: **for** $k = 1$ **to** $n^2$ **do**
15:    $C_{2(k)} \triangleq W_1^* (I_{n^2} - \text{diag}(e_k))$                                   # Create context matrix with $k$-th column dropped.
16:    $V_{3(2,k)} \leftarrow \text{SURR-MLT}_{(W_1, W_2)}(W_1^*, C_{2(k)}, V_1)$       # Forward pass
17:    $\mathcal{L} \leftarrow \|V_{3(2,k)} - V_3^*\|_2^2$
18:    $W_2^{(k)} \leftarrow W_2^{(k)} - \frac{\partial l}{\partial P_2^{(k)}}$                              # Surrogate gradient update
19: **end for**
20: Return $W_1, W_2$

### G.3.1. LEARNING THE FIRST LAYER

To study the learning process we first need to compute the closed-form gradient $\frac{\partial \mathcal{L}}{\partial P_1^{(k)}}$, which requires the following lemma:

**Lemma G.20** (Gradient with respect to incorrect column in $P_1$).

*When only the $k$-th column of translation matrix $P_1^{(k)}$ is not equal to $P_1^{*(k)}$, let $P_1^{(k)} = e_a \otimes e_b$ and $P_1^{*(k)} = e_{a^*} \otimes e_{b^*}$ for some $a, b, a^*, b^* \in [n]$. If $P_1^{(k)}$ is used in the forward pass, there exists $\alpha \in \mathbb{Z}_+$ and $\beta \in \mathbb{N}$ such that the gradient of $\mathcal{L}$ with respect to $P_1^{(k)}$ is of the form*

$$\frac{\partial \mathcal{L}}{\partial P_1^{(k)}} = \begin{cases} (2\alpha + 2\beta)(\mathbf{1}_n \otimes e_b) - (2\alpha + 2\beta)(\mathbf{1}_n \otimes e_{b^*}) + 2\beta(e_{a^*} \otimes \mathbf{1}_n) & \text{if } a^* = a, b^* \neq b \\ (2\alpha + 2\beta)(e_a \otimes \mathbf{1}_n) - (2\alpha + 2\beta)(e_{a^*} \otimes \mathbf{1}_n) + 2\beta(\mathbf{1}_n \otimes e_{b^*}) & \text{if } a^* \neq a, b^* = b \\ (2\alpha + 2\beta)(e_a \otimes \mathbf{1}_n) + (2\alpha + 2\beta)(\mathbf{1}_n \otimes e_b) - 2\alpha(e_{a^*} \otimes \mathbf{1}_n) - 2\alpha(\mathbf{1}_n \otimes e_{b^*}) & \text{if } a^* \neq a, b^* \neq b. \end{cases}$$

*Proof of Lemma G.20.* In this proof we use $\bar{e}_a$ (long one-hot) to denote the one-hot vector in $\mathbb{R}^{n^2}$ with 1 on the $a$-th index and use $e_a$ (short one-hot) to denote the one-hot vector in $\mathbb{R}^n$ with 1 on the $a$-th index. We use $\tilde{V}_1, V_2, \tilde{V}_2$ and $V_3$ to denote the intermediate sequences attained with translation matrix $P_1^{(k)}$ and $\tilde{V}_1^*, V_2^*, \tilde{V}_2^*$ and $V_3^*$ to denote the counterfactual intermediate sequences should the forward pass is done with the ground truth translations $P_1^{*(k)} = W_1^*$.

Now we can proceed to the gradient calculations. First note that with $\mathcal{L}(P_1) = \|V_3 - V_3^*\|_2^2$, by chain rule we have

$$\frac{\partial \mathcal{L}}{\partial P_1} = \sum_{j=1}^L \left(\frac{\partial V_3^{(j)}}{\partial P_1}\right)^\top \left(\frac{\partial \|V_3^{(j)} - V_3^{*(j)}\|_2^2}{\partial V_3^{(j)}}\right) = 2\sum_{j=1}^L \left(\frac{\partial V_3^{(j)}}{\partial P_1}\right)^\top \left(V_3^{(j)} - V_3^{*(j)}\right). \tag{26}$$

Specifically for each column $l \in [n^2]$ of $P_1$ we have

$$\frac{\partial \mathcal{L}}{\partial P_1^{(l)}} = 2\sum_{j=1}^L \left(\frac{\partial V_3^{(j)}}{\partial P_1^{(l)}}\right)^\top \left(V_3^{(j)} - V_3^{*(j)}\right). \tag{27}$$

Let us fix a particular column $j \in [L]$ in the output $\boldsymbol{V}_3$ and compute the gradients. With $\boldsymbol{M}^{(j)}$ as the $j$-th column of the matrix $\boldsymbol{M}$, the computation graph for the forward pass is of the form:

$$
\begin{array}{ccccccccc}
\boldsymbol{V}_1^{(j)} & \rightarrow & \tilde{\boldsymbol{V}}_1^{(j)} & \xrightarrow{\boldsymbol{P}_1} & \boldsymbol{V}_2^{(j)} & \rightarrow & \tilde{\boldsymbol{V}}_2^{(j)} & \xrightarrow{\boldsymbol{P}_2} & \boldsymbol{V}_3^{(j)} \\
& \nearrow & & & & \nearrow & & & \\
\boldsymbol{V}_1^{(j+1)} & \rightarrow & \tilde{\boldsymbol{V}}_1^{(j+1)} & \xrightarrow{\boldsymbol{P}_1} & \boldsymbol{V}_2^{(j+1)} & & & & \\
& \nearrow & & & & & & & \\
\boldsymbol{V}_1^{(j+2)} & & & & & & & &
\end{array}
\tag{28}
$$

We can see that $\boldsymbol{V}_3^{(j)}$ is only affected by $\boldsymbol{P}_1$ through $\boldsymbol{V}_2^{(j)}$ and $\boldsymbol{V}_2^{(j+1)}$.

Let us first compute $\partial \boldsymbol{V}_3^{(j)}/\partial \boldsymbol{P}_1$. Assume $\boldsymbol{V}_2^{(j)} = \bar{\boldsymbol{e}}_p$ and $\boldsymbol{V}_2^{(j+1)} = \bar{\boldsymbol{e}}_q$ for some $p, q \in [n^2]$, $\boldsymbol{V}_3^{(j)}$ is computed as

$$
\boldsymbol{V}_3^{(j)} = \boldsymbol{W}_2^* \tilde{\boldsymbol{V}}_2^{(j)} = \boldsymbol{W}_2^* \left( \boldsymbol{Q} \boldsymbol{V}_2^{(j)} \odot \boldsymbol{Q}^\top \boldsymbol{V}_2^{(j+1)} \right) = \boldsymbol{W}_2^* \left( \boldsymbol{Q} \boldsymbol{P}_1^{(p)} \odot \boldsymbol{Q}^\top \boldsymbol{P}_1^{(q)} \right).
\tag{29}
$$

We may express the Hadamard product in the following two ways:

$$
\begin{aligned}
\boldsymbol{V}_3^{(j)} &= \boldsymbol{W}_2^* \left( \boldsymbol{Q} \boldsymbol{P}_1^{(p)} \odot \boldsymbol{Q}^\top \boldsymbol{P}_1^{(q)} \right) = \boldsymbol{W}_2^* \mathrm{diag} \left( \boldsymbol{Q} \boldsymbol{P}_1^{(p)} \right) \boldsymbol{Q}^\top \boldsymbol{P}_1^{(q)}; \\
&= \boldsymbol{W}_2^* \left( \boldsymbol{Q}^\top \boldsymbol{P}_1^{(q)} \odot \boldsymbol{Q} \boldsymbol{P}_1^{(p)} \right) = \boldsymbol{W}_2^* \mathrm{diag} \left( \boldsymbol{Q}^\top \boldsymbol{P}_1^{(q)} \right) \boldsymbol{Q} \boldsymbol{P}_1^{(p)}.
\end{aligned}
\tag{30}
$$

Therefore when $p \neq q$ we have

$$
\frac{\partial \boldsymbol{V}_3^{(j)}}{\partial \boldsymbol{P}_1^{(p)}} = \boldsymbol{W}_2^* \mathrm{diag} \left( \boldsymbol{Q}^\top \boldsymbol{P}_1^{(q)} \right) \boldsymbol{Q}; \quad \frac{\partial \boldsymbol{V}_3^{(j)}}{\partial \boldsymbol{P}_1^{(q)}} = \boldsymbol{W}_2^* \mathrm{diag} \left( \boldsymbol{Q} \boldsymbol{P}_1^{(p)} \right) \boldsymbol{Q}^\top; \quad \forall l \notin \{p, q\} : \frac{\partial \boldsymbol{V}_3^{(j)}}{\partial \boldsymbol{P}_1^{(l)}} = \boldsymbol{0}.
\tag{31}
$$

When $p = q$, the expression would be

$$
\frac{\partial \boldsymbol{V}_3^{(j)}}{\partial \boldsymbol{P}_1^{(p)}} = \boldsymbol{W}_2^* \mathrm{diag} \left( \boldsymbol{Q}^\top \boldsymbol{P}_1^{(p)} \right) \boldsymbol{Q} + \boldsymbol{W}_2^* \mathrm{diag} \left( \boldsymbol{Q} \boldsymbol{P}_1^{(p)} \right) \boldsymbol{Q}^\top; \quad \forall l \neq p : \frac{\partial \boldsymbol{V}_3^{(j)}}{\partial \boldsymbol{P}_1^{(l)}} = \boldsymbol{0}.
\tag{32}
$$

Now we move on to compute $(\boldsymbol{V}_3^{(j)} - \boldsymbol{V}_3^{*(j)})$. There are in total four cases to consider: depending on whether $\boldsymbol{P}_1^{(k)}$ is being used when computing $\tilde{\boldsymbol{V}}_2^{(j)}$ and $\tilde{\boldsymbol{V}}_2^{(j+1)}$. Let us go over these cases one-by-one.

1. **Case 1:** $\tilde{\boldsymbol{V}}_1^{(j)} \neq \bar{\boldsymbol{e}}_k$ and $\tilde{\boldsymbol{V}}_1^{(j+1)} \neq \bar{\boldsymbol{e}}_k$.

   Assume $\tilde{\boldsymbol{V}}_1^{(j)} = \bar{\boldsymbol{e}}_p$ and $\tilde{\boldsymbol{V}}_1^{(j+1)} = \bar{\boldsymbol{e}}_q$ for some $p, q \neq k$. Since $\boldsymbol{V}_2^{(j)} = \boldsymbol{P}_1 \tilde{\boldsymbol{V}}_1^{(j)}$ while $\boldsymbol{P}_1$ equals $\boldsymbol{P}_1^*$ for all columns not equal to $k$ by assumption, we have $\boldsymbol{V}_2^{(j)} = \boldsymbol{P}_1 \bar{\boldsymbol{e}}_p = \boldsymbol{P}_1^{(p)} = \boldsymbol{P}_1^{*(p)} = \boldsymbol{P}_1^* \bar{\boldsymbol{e}}_p = \boldsymbol{V}_2^{*(j)}$. With an identical argument we have $\boldsymbol{V}_2^{(j+1)} = \boldsymbol{V}_2^{*(j+1)}$. Thus in this case $\boldsymbol{V}_3^{(j)} = \boldsymbol{V}_3^{*(j)}$ and hence

   $$
   \left( \frac{\partial \boldsymbol{V}_3^{(j)}}{\partial \boldsymbol{P}_1} \right)^\top \left( \boldsymbol{V}_3^{(j)} - \boldsymbol{V}_3^{*(j)} \right) = \boldsymbol{0}.
   \tag{33}
   $$

2. **Case 2:** $\tilde{\boldsymbol{V}}_1^{(j)} = \bar{\boldsymbol{e}}_k$ and $\tilde{\boldsymbol{V}}_1^{(j+1)} \neq \bar{\boldsymbol{e}}_k$.

   Assume $\tilde{\boldsymbol{V}}_1^{(j+1)} = \bar{\boldsymbol{e}}_q$ for some $q \neq k$, from case 1 we know $\boldsymbol{V}_2^{(j+1)} = \boldsymbol{V}_2^{*(j+1)}$. However since $\tilde{\boldsymbol{V}}_1^{(j)} = \bar{\boldsymbol{e}}_k$ we have $\boldsymbol{V}_2^{(j)} = \boldsymbol{P}_1 \bar{\boldsymbol{e}}_k = \boldsymbol{P}_1^{(k)}$, which is not equal to $\boldsymbol{V}_2^{*(j)} = \boldsymbol{P}_1^{*(k)}$. Therefore

   $$
   \begin{aligned}
   \boldsymbol{V}_3^{(j)} - \boldsymbol{V}_3^{*(j)} &= \boldsymbol{W}_2^* \left( \boldsymbol{Q} \boldsymbol{P}_1^{(k)} \odot \boldsymbol{Q}^\top \boldsymbol{P}_1^{(q)} \right) - \boldsymbol{W}_2^* \left( \boldsymbol{Q} \boldsymbol{P}_1^{*(k)} \odot \boldsymbol{Q}^\top \boldsymbol{P}_1^{(q)} \right) \\
   &= \boldsymbol{W}_2^* \left( \boldsymbol{Q}^\top \boldsymbol{P}_1^{(q)} \odot \boldsymbol{Q} \left( \boldsymbol{P}_1^{(k)} - \boldsymbol{P}_1^{*(k)} \right) \right) \\
   &= \boldsymbol{W}_2^* \mathrm{diag} \left( \boldsymbol{Q}^\top \boldsymbol{P}_1^{(q)} \right) \boldsymbol{Q} \left( \boldsymbol{P}_1^{(k)} - \boldsymbol{P}_1^{*(k)} \right).
   \end{aligned}
   \tag{34}
   $$

For the gradient with respect to $P_1^{(k)}$, combining Equation (34) with Equation (31) (substituting $p = k$) we have

$$\left(\frac{\partial V_3^{(j)}}{\partial P_1^{(k)}}\right)^\top \left(V_3^{(j)} - V_3^{*(j)}\right) = \left(W_2^* \mathrm{diag}\left(Q^\top P_1^{(q)}\right) Q\right)^\top \left(W_2^* \mathrm{diag}\left(Q^\top P_1^{(q)}\right) Q \left(P_1^{(k)} - P_1^{*(k)}\right)\right)$$

$$= Q^\top \mathrm{diag}\left(Q^\top P_1^{(q)}\right) W_2^{*\top} W_2^* \mathrm{diag}\left(Q^\top P_1^{(q)}\right) Q \left(P_1^{(k)} - P_1^{*(k)}\right) \tag{35}$$

$$= Q^\top \mathrm{diag}\left(Q^\top P_1^{(q)}\right) \mathrm{diag}\left(Q^\top P_1^{(q)}\right) Q \left(P_1^{(k)} - P_1^{*(k)}\right)$$

$$= \left((\mathbf{1}_n \mathbf{1}_n^\top) \otimes I_n\right) \left(P_1^{(k)} - P_1^{*(k)}\right) \qquad \text{(by Lemma G.27)}$$

where we dropped $W_2^{*\top} W_2^*$ in the third step since $W_2^*$ is a permutation matrix and $W_2^{*\top} W_2^* = I_n$.

Now plugging in $P_1^{(k)} = e_a \otimes e_b$ and $P_1^{*(k)} = e_{a^*} \otimes e_{b^*}$ we have

$$\left(\frac{\partial V_3^{(j)}}{\partial P_1^{(k)}}\right)^\top \left(V_3^{(j)} - V_3^{*(j)}\right) = \left((\mathbf{1}_n \mathbf{1}_n^\top) \otimes I_n\right)(e_a \otimes e_b) - \left((\mathbf{1}_n \mathbf{1}_n^\top) \otimes I_n\right)(e_{a^*} \otimes e_{b^*})$$

$$= \left((\mathbf{1}_n \mathbf{1}_n^\top e_a) \otimes I_n e_b\right) - \left((\mathbf{1}_n \mathbf{1}_n^\top e_{a^*}) \otimes I_n e_{b^*}\right) \tag{36}$$

$$= \mathbf{1}_n \otimes e_b - \mathbf{1}_n \otimes e_{b^*}$$

$$= \mathbf{1}_n \otimes (e_b - e_{b^*}).$$

3. **Case 3: $\tilde{V}_1^{(j)} \neq \bar{e}_k$ and $\tilde{V}_1^{(j+1)} = \bar{e}_k$.**

This is a symmetric case with respect to case 2. Assume $\tilde{V}_1^{(j)} = \bar{e}_p$ for some $p \neq k$, from case 1 we know $V_2^{(j)} = V_2^{*(j)}$. However since $\tilde{V}_1^{(j)} = \bar{e}_k$ we have $V_2^{(j+1)} = P_1 \bar{e}_k = P_1^{(k)}$, which is not equal to $V_2^{*(j+1)} = P_1^{*(k)}$. Therefore similar to case 2 we have

$$V_3^{(j)} - V_3^{*(j)} = W_2^* \mathrm{diag}\left(Q P_1^{(p)}\right) Q^\top \left(P_1^{(k)} - P_1^{*(k)}\right). \tag{37}$$

Combining Equation (37) with Equation (32) gives

$$\left(\frac{\partial V_3^{(j)}}{\partial P_1^{(k)}}\right)^\top \left(V_3^{(j)} - V_3^{*(j)}\right) = \left(W_2^* \mathrm{diag}\left(Q P_1^{(p)}\right) Q^\top\right)^\top \left(W_2^* \mathrm{diag}\left(Q P_1^{(p)}\right) Q^\top \left(P_1^{(k)} - P_1^{*(k)}\right)\right)$$

$$= Q \mathrm{diag}\left(Q P_1^{(p)}\right) W_2^{*\top} W_2^* \mathrm{diag}\left(Q P_1^{(p)}\right) Q^\top \left(P_1^{(k)} - P_1^{*(k)}\right) \tag{38}$$

$$= Q \mathrm{diag}\left(Q P_1^{(p)}\right) \mathrm{diag}\left(Q P_1^{(p)}\right) Q^\top \left(P_1^{(k)} - P_1^{*(k)}\right)$$

$$= \left(I_n \otimes (\mathbf{1}_n \mathbf{1}_n^\top)\right) \left(P_1^{(k)} - P_1^{*(k)}\right) \qquad \text{(by Lemma G.28)}$$

Now plugging in $P_1^{(k)} = e_a \otimes e_b$ and $P_1^{*(k)} = e_{a^*} \otimes e_{b^*}$ we have

$$\left(\frac{\partial V_3^{(j)}}{\partial P_1^{(k)}}\right)^\top \left(V_3^{(j)} - V_3^{*(j)}\right) = \left(I_n \otimes (\mathbf{1}_n \mathbf{1}_n^\top)\right)(e_a \otimes e_b) - \left(I_n \otimes (\mathbf{1}_n \mathbf{1}_n^\top)\right)(e_{a^*} \otimes e_{b^*})$$

$$= \left(I_n e_a \otimes (\mathbf{1}_n \mathbf{1}_n^\top e_b)\right) - \left(I_n e_{a^*} \otimes (\mathbf{1}_n \mathbf{1}_n^\top e_{b^*})\right) \tag{39}$$

$$= e_a \otimes \mathbf{1}_n - e_{a^*} \otimes \mathbf{1}_n$$

$$= (e_a - e_{a^*}) \otimes \mathbf{1}_n.$$

4. **Case 4: $\tilde{V}_1^{(j)} = \tilde{V}_1^{(j+1)} = \bar{e}_k$.**

This is the most complicated case since the loss is contributed by two different paths. We can first decompose the negative residual as

$$
\begin{aligned}
\boldsymbol{V}_3^{(j)} - \boldsymbol{V}_3^{*(j)} &= \boldsymbol{W}_2^* \left( \boldsymbol{Q}\boldsymbol{P}_1^{(k)} \odot \boldsymbol{Q}^\top \boldsymbol{P}_1^{(k)} \right) - \boldsymbol{W}_2^* \left( \boldsymbol{Q}\boldsymbol{P}_1^{*(k)} \odot \boldsymbol{Q}^\top \boldsymbol{P}_1^{*(k)} \right) \\
&= \boldsymbol{W}_2^* \left( \boldsymbol{Q}\boldsymbol{P}_1^{(k)} \odot \boldsymbol{Q}^\top \boldsymbol{P}_1^{(k)} \right) - \boldsymbol{W}_2^* \left( \boldsymbol{Q}\boldsymbol{P}_1^{(k)} \odot \boldsymbol{Q}^\top \boldsymbol{P}_1^{*(k)} \right) \\
&\quad + \boldsymbol{W}_2^* \left( \boldsymbol{Q}\boldsymbol{P}_1^{(k)} \odot \boldsymbol{Q}^\top \boldsymbol{P}_1^{*(k)} \right) - \boldsymbol{W}_2^* \left( \boldsymbol{Q}\boldsymbol{P}_1^{*(k)} \odot \boldsymbol{Q}^\top \boldsymbol{P}_1^{*(k)} \right) \\
&= \boldsymbol{W}_2^* \mathrm{diag}\left( \boldsymbol{Q}\boldsymbol{P}_1^{(k)} \right) \boldsymbol{Q}^\top \left( \boldsymbol{P}_1^{(k)} - \boldsymbol{P}_1^{*(k)} \right) + \boldsymbol{W}_2^* \mathrm{diag}\left( \boldsymbol{Q}^\top \boldsymbol{P}_1^{*(p)} \right) \boldsymbol{Q} \left( \boldsymbol{P}_1^{(k)} - \boldsymbol{P}_1^{*(k)} \right).
\end{aligned} \tag{40}
$$

Combining with Equation (32), we have

$$
\left( \frac{\partial \boldsymbol{V}_3^{(j)}}{\partial \boldsymbol{P}_1} \right)^\top \left( \boldsymbol{V}_3^{(j)} - \boldsymbol{V}_3^{*(j)} \right)
$$

$$
= \left( \boldsymbol{W}_2^* \mathrm{diag}\left( \boldsymbol{Q}^\top \boldsymbol{P}_1^{(k)} \right) \boldsymbol{Q} + \boldsymbol{W}_2^* \mathrm{diag}\left( \boldsymbol{Q}\boldsymbol{P}_1^{(k)} \right) \boldsymbol{Q}^\top \right)^\top
$$

$$
\qquad \left( \boldsymbol{W}_2^* \mathrm{diag}\left( \boldsymbol{Q}\boldsymbol{P}_1^{(k)} \right) \boldsymbol{Q}^\top \left( \boldsymbol{P}_1^{(k)} - \boldsymbol{P}_1^{*(k)} \right) + \boldsymbol{W}_2^* \mathrm{diag}\left( \boldsymbol{Q}^\top \boldsymbol{P}_1^{*(k)} \right) \boldsymbol{Q} \left( \boldsymbol{P}_1^{(k)} - \boldsymbol{P}_1^{*(k)} \right) \right)
$$

$$
= \boldsymbol{Q}^\top \mathrm{diag}\left( \boldsymbol{Q}^\top \boldsymbol{P}_1^{(k)} \right) \mathrm{diag}\left( \boldsymbol{Q}\boldsymbol{P}_1^{(k)} \right) \boldsymbol{Q}^\top \left( \boldsymbol{P}_1^{(k)} - \boldsymbol{P}_1^{*(k)} \right) \tag{a}
$$

$$
+ \boldsymbol{Q} \mathrm{diag}\left( \boldsymbol{Q}\boldsymbol{P}_1^{(k)} \right) \mathrm{diag}\left( \boldsymbol{Q}\boldsymbol{P}_1^{(k)} \right) \boldsymbol{Q}^\top \left( \boldsymbol{P}_1^{(k)} - \boldsymbol{P}_1^{*(k)} \right) \tag{b}
$$

$$
+ \boldsymbol{Q}^\top \mathrm{diag}\left( \boldsymbol{Q}^\top \boldsymbol{P}_1^{(k)} \right) \mathrm{diag}\left( \boldsymbol{Q}^\top \boldsymbol{P}_1^{*(k)} \right) \boldsymbol{Q} \left( \boldsymbol{P}_1^{(k)} - \boldsymbol{P}_1^{*(k)} \right) \tag{c}
$$

$$
+ \boldsymbol{Q} \mathrm{diag}\left( \boldsymbol{Q}\boldsymbol{P}_1^{(k)} \right) \mathrm{diag}\left( \boldsymbol{Q}^\top \boldsymbol{P}_1^{*(k)} \right) \boldsymbol{Q} \left( \boldsymbol{P}_1^{(k)} - \boldsymbol{P}_1^{*(k)} \right). \tag{d}
$$

Now let us analyze the four cross terms term-by-term.

(a) With $\boldsymbol{P}_1^{(k)} = \boldsymbol{e}_a \otimes \boldsymbol{e}_b$, by Lemma G.26 we have $\boldsymbol{Q}^\top \boldsymbol{P}_1^{(k)} = \mathbf{1}_n \otimes \boldsymbol{e}_a$ and $\boldsymbol{Q}\boldsymbol{P}_1^{(k)} = \boldsymbol{e}_b \otimes \mathbf{1}_n$. Therefore $\boldsymbol{Q}^\top \boldsymbol{P}_1^{(k)} \odot \boldsymbol{Q}\boldsymbol{P}_1^{(k)} = \boldsymbol{e}_b \otimes \boldsymbol{e}_a$ and hence

$$
\mathrm{diag}\left( \boldsymbol{Q}^\top \boldsymbol{P}_1^{(k)} \right) \mathrm{diag}\left( \boldsymbol{Q}\boldsymbol{P}_1^{(k)} \right) = \mathrm{diag}\left( \boldsymbol{e}_b \otimes \boldsymbol{e}_a \right) = \mathrm{diag}\left( \boldsymbol{e}_b \right) \otimes \mathrm{diag}\left( \boldsymbol{e}_a \right) \tag{41}
$$

It follows that

$$
\begin{aligned}
&\boldsymbol{Q}^\top \mathrm{diag}\left( \boldsymbol{Q}^\top \boldsymbol{P}_1^{(k)} \right) \mathrm{diag}\left( \boldsymbol{Q}\boldsymbol{P}_1^{(k)} \right) \boldsymbol{Q}^\top \\
&= \left( \mathbf{1}_n \otimes I_n \right) \left( I_n^\top \otimes \mathbf{1}_n^\top \right) \left( \mathrm{diag}\left( \boldsymbol{e}_b \right) \otimes \mathrm{diag}\left( \boldsymbol{e}_a \right) \right) \left( \mathbf{1}_n \otimes I_n \right) \left( I_n^\top \otimes \mathbf{1}_n^\top \right) \\
&= \left( \mathbf{1}_n \otimes I_n \right) \left( I_n^\top \otimes \mathbf{1}_n^\top \right) \left( \boldsymbol{e}_b \otimes \mathrm{diag}\left( \boldsymbol{e}_a \right) \right) \left( I_n^\top \otimes \mathbf{1}_n^\top \right) \\
&= \left( \mathbf{1}_n \otimes I_n \right) \left( I_n^\top \otimes \mathbf{1}_n^\top \right) \left( \boldsymbol{e}_b \otimes \boldsymbol{e}_a^\top \right) \\
&= \left( \mathbf{1}_n \otimes I_n \right) \left( I_n^\top \otimes \mathbf{1}_n^\top \right) \left( \boldsymbol{e}_b \boldsymbol{e}_a^\top \otimes 1 \right) \\
&= \left( \mathbf{1}_n \otimes I_n \right) \left( \boldsymbol{e}_b \boldsymbol{e}_a^\top \otimes \mathbf{1}_n^\top \right)
\end{aligned} \tag{42}
$$

Plugging into (a) we have

$$
\begin{aligned}
(a) &= \boldsymbol{Q}^\top \mathrm{diag}\left( \boldsymbol{Q}^\top \boldsymbol{P}_1^{(k)} \right) \mathrm{diag}\left( \boldsymbol{Q}\boldsymbol{P}_1^{(k)} \right) \boldsymbol{Q}^\top \left( \boldsymbol{P}_1^{(k)} - \boldsymbol{P}_1^{*(k)} \right) \\
&= \left( \mathbf{1}_n \otimes I_n \right) \left( \boldsymbol{e}_b \boldsymbol{e}_a^\top \otimes \mathbf{1}_n^\top \right) \left( \boldsymbol{e}_a \otimes \boldsymbol{e}_b \right) - \left( \mathbf{1}_n \otimes I_n \right) \left( \boldsymbol{e}_b \boldsymbol{e}_a^\top \otimes \mathbf{1}_n^\top \right) \left( \boldsymbol{e}_{a^*} \otimes \boldsymbol{e}_{b^*} \right) \\
&= \left( \mathbf{1}_n \otimes I_n \right) \left( \boldsymbol{e}_b \boldsymbol{e}_a^\top \boldsymbol{e}_a \otimes 1 \right) - \left( \mathbf{1}_n \otimes I_n \right) \left( \boldsymbol{e}_b \boldsymbol{e}_a^\top \boldsymbol{e}_{a^*} \otimes 1 \right) \\
&= \left( \mathbf{1}_n \otimes I_n \right) \left( 1 \otimes \boldsymbol{e}_b \boldsymbol{e}_a^\top \boldsymbol{e}_a \right) - \left( \mathbf{1}_n \otimes I_n \right) \left( 1 \otimes \boldsymbol{e}_b \boldsymbol{e}_a^\top \boldsymbol{e}_{a^*} \right) \\
&= \begin{cases} \mathbf{1}_n \otimes \boldsymbol{e}_b & \text{when } a \neq a^* \\ \mathbf{0} & \text{otherwise} \end{cases}
\end{aligned} \tag{43}
$$

(b) We have seen the same term as in case 3, by Lemma G.28 we have

$$(b) = \boldsymbol{Q}\text{diag}\left(\boldsymbol{Q}\boldsymbol{P}_1^{(k)}\right)\text{diag}\left(\boldsymbol{Q}\boldsymbol{P}_1^{(k)}\right)\boldsymbol{Q}^\top\left(\boldsymbol{P}_1^{(k)} - \boldsymbol{P}_1^{*(k)}\right) = (\boldsymbol{e}_a - \boldsymbol{e}_{a^*})\otimes\mathbf{1}_n. \tag{44}$$

(c) With $\boldsymbol{P}_1^{(k)} = \boldsymbol{e}_a\otimes\boldsymbol{e}_b$ and $\boldsymbol{P}_1^{*(k)} = \boldsymbol{e}_{a^*}\otimes\boldsymbol{e}_{b^*}$, we have

$$\text{diag}\left(\boldsymbol{Q}^\top\boldsymbol{P}_1^{(k)}\right)\text{diag}\left(\boldsymbol{Q}^\top\boldsymbol{P}_1^{*(k)}\right) = \text{diag}\left(\mathbf{1}_n\otimes\boldsymbol{e}_a\right)\text{diag}\left(\mathbf{1}_n\otimes\boldsymbol{e}_{a^*}\right) = \begin{cases} \text{diag}\left(\mathbf{1}_n\otimes\boldsymbol{e}_a\right) & \text{if } a = a^* \\ \mathbf{0} & \text{otherwise} \end{cases} \tag{45}$$

Thus by Lemma G.27 we have

$$\boldsymbol{Q}^\top\text{diag}\left(\boldsymbol{Q}^\top\boldsymbol{P}_1^{(k)}\right)\text{diag}\left(\boldsymbol{Q}^\top\boldsymbol{P}_1^{*(k)}\right)\boldsymbol{Q} = \begin{cases} \left(\mathbf{1}_n\mathbf{1}_n^\top\right)\otimes I_n & \text{if } a = a^* \\ \mathbf{0} & \text{otherwise} \end{cases} \tag{46}$$

When $a = a^*$, we then have

$$\begin{aligned} &\boldsymbol{Q}^\top\text{diag}\left(\boldsymbol{Q}^\top\boldsymbol{P}_1^{(k)}\right)\text{diag}\left(\boldsymbol{Q}^\top\boldsymbol{P}_1^{*(k)}\right)\boldsymbol{Q}\left(\boldsymbol{P}_1^{(k)} - \boldsymbol{P}_1^{*(k)}\right) \\ &= \left(\left(\mathbf{1}_n\mathbf{1}_n^\top\right)\otimes I_n\right)\left(\boldsymbol{e}_a\otimes\boldsymbol{e}_b\right) - \left(\left(\mathbf{1}_n\mathbf{1}_n^\top\right)\otimes I_n\right)\left(\boldsymbol{e}_{a^*}\otimes\boldsymbol{e}_{b^*}\right) \\ &= \mathbf{1}_n\otimes\left(\boldsymbol{e}_b - \boldsymbol{e}_{b^*}\right). \end{aligned} \tag{47}$$

Thus in summary

$$(c) = \boldsymbol{Q}^\top\text{diag}\left(\boldsymbol{Q}^\top\boldsymbol{P}_1^{(k)}\right)\text{diag}\left(\boldsymbol{Q}^\top\boldsymbol{P}_1^{*(k)}\right)\boldsymbol{Q}\left(\boldsymbol{P}_1^{(k)} - \boldsymbol{P}_1^{*(k)}\right) = \begin{cases} \mathbf{1}_n\otimes\left(\boldsymbol{e}_b - \boldsymbol{e}_{b^*}\right) & \text{if } a = a^* \\ \mathbf{0} & \text{otherwise.} \end{cases} \tag{48}$$

(d) Now for the last term, note that

$$\begin{aligned} &\boldsymbol{Q}\text{diag}\left(\boldsymbol{Q}\boldsymbol{P}_1^{(k)}\right)\text{diag}\left(\boldsymbol{Q}^\top\boldsymbol{P}_1^{*(k)}\right)\boldsymbol{Q} \\ &= \boldsymbol{Q}\text{diag}\left(\boldsymbol{e}_b\otimes\mathbf{1}_n\right)\text{diag}\left(\mathbf{1}_n\otimes\boldsymbol{e}_{a^*}\right)\boldsymbol{Q} \\ &= \boldsymbol{Q}\text{diag}\left(\boldsymbol{e}_b\otimes\boldsymbol{e}_{a^*}\right)\boldsymbol{Q} \\ &= \left(I_n\otimes\mathbf{1}_n\right)\left(\mathbf{1}_n^\top\otimes I_n^\top\right)\left(\text{diag}\left(\boldsymbol{e}_b\right)\otimes\text{diag}\left(\boldsymbol{e}_{a^*}\right)\right)\left(I_n\otimes\mathbf{1}_n\right)\left(\mathbf{1}_n^\top\otimes I_n^\top\right) \\ &= \left(I_n\otimes\mathbf{1}_n\right)\left(\mathbf{1}_n^\top\otimes I_n^\top\right)\left(\text{diag}\left(\boldsymbol{e}_b\right)\otimes\boldsymbol{e}_{a^*}\right)\left(\mathbf{1}_n^\top\otimes I_n^\top\right) \\ &= \left(I_n\otimes\mathbf{1}_n\right)\left(\boldsymbol{e}_b^\top\otimes\boldsymbol{e}_{a^*}\right)\left(\mathbf{1}_n^\top\otimes I_n^\top\right) \\ &= \left(I_n\otimes\mathbf{1}_n\right)\left(\boldsymbol{e}_{a^*}\boldsymbol{e}_b^\top\otimes 1\right)\left(\mathbf{1}_n^\top\otimes I_n^\top\right) \\ &= \left(\boldsymbol{e}_{a^*}\boldsymbol{e}_b^\top\otimes\mathbf{1}_n\right)\left(\mathbf{1}_n^\top\otimes I_n^\top\right). \end{aligned} \tag{49}$$

Plugging in $\left(\boldsymbol{P}_1^{(k)} - \boldsymbol{P}_1^{*(k)}\right)$, we have that

$$\begin{aligned} &\boldsymbol{Q}\text{diag}\left(\boldsymbol{Q}\boldsymbol{P}_1^{(k)}\right)\text{diag}\left(\boldsymbol{Q}^\top\boldsymbol{P}_1^{*(k)}\right)\boldsymbol{Q}\left(\boldsymbol{P}_1^{(k)} - \boldsymbol{P}_1^{*(k)}\right) \\ &= \left(\boldsymbol{e}_{a^*}\boldsymbol{e}_b^\top\otimes\mathbf{1}_n\right)\left(\mathbf{1}_n^\top\otimes I_n^\top\right)\left(\boldsymbol{e}_a\otimes\boldsymbol{e}_b\right) - \left(\boldsymbol{e}_{a^*}\boldsymbol{e}_b^\top\otimes\mathbf{1}_n\right)\left(\mathbf{1}_n^\top\otimes I_n^\top\right)\left(\boldsymbol{e}_{a^*}\otimes\boldsymbol{e}_{b^*}\right) \\ &= \left(\boldsymbol{e}_{a^*}\boldsymbol{e}_b^\top\otimes\mathbf{1}_n\right)\left(1\otimes\boldsymbol{e}_b\right) - \left(\boldsymbol{e}_{a^*}\boldsymbol{e}_b^\top\otimes\mathbf{1}_n\right)\left(1\otimes\boldsymbol{e}_{b^*}\right) \\ &= \left(\boldsymbol{e}_{a^*}\boldsymbol{e}_b^\top\otimes\mathbf{1}_n\right)\left(\boldsymbol{e}_b\otimes 1\right) - \left(\boldsymbol{e}_{a^*}\boldsymbol{e}_b^\top\otimes\mathbf{1}_n\right)\left(\boldsymbol{e}_{b^*}\otimes 1\right) \\ &= \left(\boldsymbol{e}_{a^*}\boldsymbol{e}_b^\top\boldsymbol{e}_b\otimes\mathbf{1}_n\right) - \left(\boldsymbol{e}_{a^*}\boldsymbol{e}_b^\top\boldsymbol{e}_{b^*}\otimes\mathbf{1}_n\right) \\ &= \begin{cases} \boldsymbol{e}_{a^*}\otimes\mathbf{1}_n & \text{if } b\neq b^* \\ \mathbf{0} & \text{otherwise.} \end{cases} \end{aligned} \tag{50}$$

Summing the four terms together, we then have

$$\left(\frac{\partial\boldsymbol{V}_3^{(j)}}{\partial\boldsymbol{P}_1}\right)^\top\left(\boldsymbol{V}_3^{(j)} - \boldsymbol{V}_3^{*(j)}\right) = \begin{cases} \mathbf{0} & \text{when } a^* = a, b^* = b \\ \mathbf{1}_n\otimes\left(\boldsymbol{e}_b - \boldsymbol{e}_{b^*}\right) + \boldsymbol{e}_{a^*}\otimes\mathbf{1}_n & \text{when } a^* = a, b^*\neq b \\ \mathbf{1}_n\otimes\boldsymbol{e}_{b^*} + \left(\boldsymbol{e}_a - \boldsymbol{e}_{a^*}\right)\otimes\mathbf{1}_n & \text{when } a^*\neq a, b^* = b \\ \mathbf{1}_n\otimes\boldsymbol{e}_b + \boldsymbol{e}_a\otimes\mathbf{1}_n & \text{when } a^*\neq a, b^*\neq b. \end{cases} \tag{51}$$

Now we are ready to provide the gradient expression for the loss over the entire sequence. Observe that for every consecutive sequence of $m$ columns $\{V_1^{*(j)}, V_1^{*(j+1)}, \ldots, V_1^{*(j+m-1)}\}$ that all equals to $\bar{e}_k$, it will result in one incorrect column $V_3^{(j-1)}$ in case 2, one incorrect column $V_3^{(j+m-1)}$ in case 3, and $m-1$ incorrect columns $(V_3^{(j)}, \ldots, V_3^{(j+m-2)})$ in case 4.

*Figure 12.* Error propagation of $P_1^{(k)}$

For illustration, one can refer to the computation graph in Figure 12. In the graph, green entries agrees with the counterfactual values with correct $P_1^{*(k)}$, Red and pink entries are incorrect entries where red entries are consequence solely dependent on $P_1^{(k)}$ (case 4) and pink entries depend on other correct columns (case 2,3).

Assume that in total there are $\alpha$ columns in $V_3$ under case 2, $\alpha$ columns in $V_3$ under case 3, and $\beta$ columns in $V_3$ under case 4, then by Equation (27) the total gradient can be expressed as follows:

- When $a = a^*, b \neq b^*$:

$$
\begin{aligned}
\frac{\partial \mathcal{L}}{\partial P_1^{(k)}} &= 2\alpha \left(1_n \otimes (e_b - e_{b^*}) + 2\alpha (e_a - e_{a^*}) \otimes 1_n\right) + 2\beta \left(1_n \otimes (e_b - e_{b^*}) + e_{a^*} \otimes 1_n\right) \\
&= (2\alpha + 2\beta) \left(1_n \otimes e_b\right) - (2\alpha + 2\beta) \left(1_n \otimes e_{b^*}\right) + 2\beta \left(e_{a^*} \otimes 1_n\right).
\end{aligned}
\tag{52}
$$

- When $a \neq a^*, b = b^*$:

$$
\begin{aligned}
\frac{\partial \mathcal{L}}{\partial P_1^{(k)}} &= 2\alpha \left(1_n \otimes (e_b - e_{b^*})\right) + 2\alpha \left((e_a - e_{a^*}) \otimes 1_n\right) + 2\beta \left(1_n \otimes e_{b^*} + (e_a - e_{a^*}) \otimes 1_n\right) \\
&= (2\alpha + 2\beta) \left(e_a \otimes 1_n\right) - (2\alpha + 2\beta) \left(e_{a^*} \otimes 1_n\right) + 2\beta \left(1_n \otimes e_{b^*}\right).
\end{aligned}
\tag{53}
$$

- When $a \neq a^*, b \neq b^*$:

$$\frac{\partial \mathcal{L}}{\partial \boldsymbol{P}_1^{(k)}} = 2\alpha \left(\mathbf{1}_n \otimes (\boldsymbol{e}_b - \boldsymbol{e}_{b^*})\right) + 2\alpha \left((\boldsymbol{e}_a - \boldsymbol{e}_{a^*}) \otimes \mathbf{1}_n\right) + 2\beta \left(\mathbf{1}_n \otimes \boldsymbol{e}_b + \boldsymbol{e}_a \otimes \mathbf{1}_n\right)$$

$$= (2\alpha + 2\beta)(\boldsymbol{e}_a \otimes \mathbf{1}_n) + (2\alpha + 2\beta)(\mathbf{1}_n \otimes \boldsymbol{e}_b) - 2\alpha(\boldsymbol{e}_{a^*} \otimes \mathbf{1}_n) - 2\alpha(\mathbf{1}_n \otimes \boldsymbol{e}_{b^*}). \tag{54}$$

This gives the desired expression of gradient.⫻ □

Now with the gradient expression, we are ready to prove for the learning of a single missing column.

> **Lemma G.21** (Learning Column of $\boldsymbol{W}_1$).
> *Fix an input sequence $\boldsymbol{V}_1 \in \mathbb{R}^{n^2 \times L}$ and any column index $k \in [n^2]$, if $\texttt{HardMax}\left(\boldsymbol{C}_2 + \boldsymbol{W}_2\right) = \boldsymbol{W}_2^*$ and $\texttt{HardMax}\left(\boldsymbol{C}_{1(k)} + \boldsymbol{W}_1\right)$ equals to $\boldsymbol{W}_1^*$ everywhere except for the $k$-th column and if there exists a non-empty subset of indices $\mathcal{J} \subset [L]$ such that $\boldsymbol{V}_1^{*(j)} = \bar{\boldsymbol{e}}_k$ for all $j \in \mathcal{J}$, for any initialization $\boldsymbol{W}_{1(0)}^{(k)} \in \mathbb{R}^{n^2}$ such that $\|\boldsymbol{W}_{1(0)}^{(k)}\|_0 \leq \frac{1}{2}$. Taking two surrogate gradient updates on $\boldsymbol{W}_{1(0)}^{(k)}$ as described in Algorithm 3 gives $\boldsymbol{W}_{1(2)}^{(k)}$ such that $\texttt{HardMax}(\boldsymbol{W}_{1(2)}^{(k)}) = \boldsymbol{W}_1^{*(k)}$.*

*Proof of Lemma G.21.*

Without loss of generality, assume at the initialization $\boldsymbol{P}_{1(0)}^{(k)} = \boldsymbol{e}_{a_{(0)}} \otimes \boldsymbol{e}_{b_{(0)}}$ and $\boldsymbol{P}_1^{*(k)} = \boldsymbol{e}_{a^*} \otimes \boldsymbol{e}_{b^*}$ for some $a_{(0)}, b_{(0)}, a^*, b^* \in [n]$. We first note that the conditions specified in the lemma meets the assumptions required by Lemma G.20, namely there is only one incorrect column in $\boldsymbol{W}_1$ missing and that column is being used in the forward pass at least one time (since $\mathcal{J}$ is non-empty).

To prove $\texttt{HardMax}(\boldsymbol{W}_{1(2)}^{(k)}) = \boldsymbol{W}_1^{*(k)}$, it is sufficient to show that $\arg\max(\boldsymbol{W}_{1(2)}^{(k)}) = a^*n + b^*$.

Now we will leverage the gradient expressions in Lemma G.20. We will dive into three different cases:

- **When $a_{(0)} \neq a^*$ and $b_{(0)} \neq b^*$.**

  By Lemma G.20 we have for some $\alpha \in \mathbb{Z}_+$ and $\beta \in \mathbb{N}$ that

  $$\frac{\partial \mathcal{L}}{\partial \boldsymbol{P}_{1(0)}^{(k)}} = (2\alpha + 2\beta)\left(\boldsymbol{e}_{a_{(0)}} \otimes \mathbf{1}_n\right) + (2\alpha + 2\beta)\left(\mathbf{1}_n \otimes \boldsymbol{e}_{b_{(0)}}\right) - 2\alpha\left(\boldsymbol{e}_{a^*} \otimes \mathbf{1}_n\right) - 2\alpha\left(\mathbf{1}_n \otimes \boldsymbol{e}_{b^*}\right). \tag{55}$$

  It is not hard to verify that the unique smallest entry is of index $a^*n + b^*$ with value $-4\alpha$. This entry is contributed by the intersection of $-2\alpha(\boldsymbol{e}_{a^*} \otimes \mathbf{1}_n) - 2\alpha(\mathbf{1}_n \otimes \boldsymbol{e}_{b^*})$, the remaining smaller entries are of value $-2\alpha$ contributed by non-intersecting entries in the same expression above.

  Thus we know $\arg\min \partial \mathcal{L} / \partial \boldsymbol{P}_{1(0)}^{(k)} = a^*n + b^*$ with a margin of at least $-2$ (since $\alpha \geq 1$).

  Now we can apply the gradient step to the weight initialization. Since $\|\boldsymbol{W}_{1(0)}^{(k)}\|_0 \leq \frac{1}{2}$, the margin of $a^*n + b^*$ dominates the largest margin in the initialization (which is 1), we have

  $$\arg\max\left(\boldsymbol{W}_{1(1)}^{(k)}\right) = \arg\max\left(\boldsymbol{W}_{1(0)}^{(k)} - \frac{\partial \mathcal{L}}{\partial \boldsymbol{P}_{1(0)}^{(k)}}\right) = a^*n + b^*. \tag{56}$$

  Therefore $\boldsymbol{P}_{1(1)}^{(k)} = \boldsymbol{P}_1^{*(k)}$ with the first step. We will reach zero loss after the first step and hence the second step is static, so we have shown $\arg\max(\boldsymbol{W}_{1(2)}^{(k)}) = a^*n + b^*$ as desired.

- **When $a_{(0)} = a^*$ and $b_{(0)} \neq b^*$.**

By Lemma G.20 we have for some $\alpha \in \mathbb{Z}_+$ and $\beta \in \mathbb{N}$ that

$$\frac{\partial \mathcal{L}}{\partial \boldsymbol{P}_{1(0)}^{(k)}} = (2\alpha + 2\beta)\left(\boldsymbol{1}_n \otimes \boldsymbol{e}_{b_{(0)}}\right) - (2\alpha + 2\beta)\left(\boldsymbol{1}_n \otimes \boldsymbol{e}_{b^*}\right) + 2\beta\left(\boldsymbol{e}_{a^*} \otimes \boldsymbol{1}_n\right). \tag{57}$$

In this case we no longer have an unique smallest entry, the set of negative entries is of index $nx + b^*$ where $x \in [n]$. The values are $-2\alpha$ for the case of $x = a^*$ and $-2(\alpha + \beta)$ for other $x$'s. These negative entries are contributed by the $-(2\alpha + 2\beta)\left(\boldsymbol{1}_n \otimes \boldsymbol{e}_{b^*}\right)$, and all other entries are at least 0.

Thus we also have a negative margin of at least $-2$ since $\alpha \geq 1$. Therefore after taking the first gradient step, we know that there exists some $x \in [n]$ such that

$$\arg\max\left(\boldsymbol{W}_{1(1)}^{(k)}\right) = \arg\max\left(\boldsymbol{W}_{1(0)}^{(k)} - \frac{\partial \mathcal{L}}{\partial \boldsymbol{P}_{1(0)}^{(k)}}\right) = nx + b^*. \tag{58}$$

Let $P_{1(1)}^{(k)} = \arg\max(\boldsymbol{W}_{1(1)}^{(k)}) = \boldsymbol{e}_{a_{(1)}} \otimes \boldsymbol{e}_{b_{(1)}}$, then now we are in the case of $b_{(1)} = b^*$ since the negative margin of $-2$ dominates any entry-wise difference in the initialization as $\|\boldsymbol{W}_{1(0)}^{(k)}\|_0 \leq \frac{1}{2}$. If $\beta = 0$ and it happens that $a_{(1)} = a^*$, then we have zero loss after the first step and we are done as the second step will be static. If $b_{(1)} \neq b^*$, then by Lemma G.20 the second step gradient is of the form

$$\frac{\partial \mathcal{L}}{\partial \boldsymbol{P}_{1(1)}^{(k)}} = (2\alpha + 2\beta)\left(\boldsymbol{e}_{a_{(1)}} \otimes \boldsymbol{1}_n\right) - (2\alpha + 2\beta)\left(\boldsymbol{e}_{a^*} \otimes \boldsymbol{1}_n\right) + 2\beta\left(\boldsymbol{1}_n \otimes \boldsymbol{e}_{b^*}\right). \tag{59}$$

This is a bit tricky to analyze directly since we no longer have the small entry-wise difference from the initialization in the weights, but one may note that the sum of the two update steps is of the form

$$\begin{aligned}
&\frac{\partial \mathcal{L}}{\partial \boldsymbol{P}_{1(1)}^{(k)}} + \frac{\partial \mathcal{L}}{\partial \boldsymbol{P}_{1(0)}^{(k)}} \\
&= (2\alpha + 2\beta)\left(\boldsymbol{e}_{a_{(1)}} \otimes \boldsymbol{1}_n\right) - (2\alpha + 2\beta)\left(\boldsymbol{e}_{a^*} \otimes \boldsymbol{1}_n\right) + 2\beta\left(\boldsymbol{1}_n \otimes \boldsymbol{e}_{b^*}\right) \\
&\quad + (2\alpha + 2\beta)\left(\boldsymbol{1}_n \otimes \boldsymbol{e}_{b_{(0)}}\right) - (2\alpha + 2\beta)\left(\boldsymbol{1}_n \otimes \boldsymbol{e}_{b^*}\right) + 2\beta\left(\boldsymbol{e}_{a^*} \otimes \boldsymbol{1}_n\right) \\
&= (2\alpha + 2\beta)\left(\boldsymbol{e}_{a_{(1)}} \otimes \boldsymbol{1}_n\right) + (2\alpha + 2\beta)\left(\boldsymbol{1}_n \otimes \boldsymbol{e}_{b_{(0)}}\right) - 2\alpha\left(\boldsymbol{e}_{a^*} \otimes \boldsymbol{1}_n\right) - 2\alpha\left(\boldsymbol{1}_n \otimes \boldsymbol{e}_{b^*}\right).
\end{aligned} \tag{60}$$

This is identical to the single step gradient as in Equation (55) in the first case, so follow the identical argument we have

$$\arg\max\left(\boldsymbol{W}_{1(2)}^{(k)}\right) = \arg\max\left(\boldsymbol{W}_{1(0)}^{(k)} - \left(\frac{\partial \mathcal{L}}{\partial \boldsymbol{P}_{1(0)}^{(k)}} + \frac{\partial \mathcal{L}}{\partial \boldsymbol{P}_{1(1)}^{(k)}}\right)\right) = a^*n + b^* \tag{61}$$

as desired.

- **When $a_{(0)} \neq a^*$ and $b_{(0)} = b^*$**

  Note that all expressions are symmetric with respect to $a$ and $b$ with an additional swap of the Kronecker products, so we may follow the exact same argument as in the case of $a_{(0)} = a^*$ and $b_{(0)} \neq b^*$ and arrive at the same conclusion.

Thus for any initializations, after two surrogate gradient steps on $\boldsymbol{W}_1^{(k)}$, we have $\texttt{HardMax}(\boldsymbol{W}_{1(2)}^{(k)}) = \boldsymbol{W}_1^{*(k)}$. $\qquad\square$

### G.3.2. LEARNING THE SECOND LAYER

Now for the second layer, we will similarly first derive the gradient (which is much simpler) and show that with only one gradient step, one can learn the correct entry.

**Lemma G.22** (Gradient with respect to incorrect column in $\boldsymbol{P}_2$).

*When only the $k$-th column of translation matrix $\boldsymbol{P}_2^{(k)}$ is not equal to $\boldsymbol{P}_2^{*(k)}$. If $\boldsymbol{P}_2^{(k)}$ is used in the forward pass, there exists $\alpha \in \mathbb{Z}_+$ such that the gradient of $\mathcal{L}$ with respect to $\boldsymbol{P}_2^{(k)}$ is of the form*

$$\frac{\partial \mathcal{L}}{\partial \boldsymbol{P}_2^{(k)}} = 2\alpha\left(\boldsymbol{P}_2^{(k)} - \boldsymbol{P}_2^{*(k)}\right).$$

*Proof.* We follow the same set of notations as used in the proof for Lemma G.20. In particular, we use $\tilde{V}_1, V_2, \tilde{V}_2$ and $V_3$ to denote the intermediate sequences attained with translation matrix $P_2^{(k)}$ and $\tilde{V}_1^*, V_2^*, \tilde{V}_2^*$ and $V_3^*$ to denote the counterfactual intermediate sequences should the forward pass is done with the ground truth translations $P_2^{*(k)} = W_1^*$. Since we assume that $P_1 = P_1^*$, we have $\tilde{V}_2 = \tilde{V}_2^*$. Therefore for all column $j$ such that $V_3^{(j)} \neq V_3^{*(j)}$, it must be so that $\tilde{V}_2^{*(j)} = \bar{e}_k$, and the residual can be written as

$$V_3^{(j)} - V_3^{*(j)} = P_2 \tilde{V}_2^{*(j)} - P_2^* \tilde{V}_2^{*(j)} = P_2 \bar{e}_k - P_2^* \bar{e}_k = P_2^{(k)} - P_2^{*(k)}. \tag{62}$$

Assume that $P_2^{(k)}$ has been used $\alpha$ times in the forward pass, follow the chain rule we then have

$$\frac{\partial \mathcal{L}}{\partial P_2^{(k)}} = 2 \sum_{j=1}^{L} \left( \frac{\partial V_3^{(j)}}{\partial P_2^{(k)}} \right)^{\top} \left( V_3^{(j)} - V_3^{*(j)} \right) = 2\alpha \left( P_2^{(k)} - P_2^{*(k)} \right). \tag{63}$$

$\square$

> **Lemma G.23** (Learning $W_2$).
> *Fix an input sequence $V_1 \in \mathbb{R}^{n^2 \times L}$ and any column index $k \in [n^2]$, if $\mathtt{HardMax}\,(C_1 + W_1) = W_1^*$ and $\mathtt{HardMax}\,(C_{2(k)} + W_2)$ equals to $W_2^*$ everywhere except for the $k$-th column and if there exists a non-empty subset of indices $\mathcal{J} \subset [L]$ such that $V_2^{*(j)} = \bar{e}_k$ for all $j \in \mathcal{J}$, for any initialization $W_{2(0)}^{(k)} \in \mathbb{R}^{n^2}$ such that $\|W_{2(0)}^{(k)}\|_0 \leq \frac{1}{2}$. Taking one surrogate gradient updates on $W_{2(0)}^{(k)}$ as described in Algorithm 3 gives $W_{2(1)}^{(k)}$ such that $\mathtt{HardMax}(W_{2(1)}^{(k)}) = W_2^{*(k)}$.*

*Proof.* Without loss of generality, assume at the initialization $P_{2(0)}^{(k)} = e_{a_{(0)}} \otimes e_{b_{(0)}}$ and $P_2^{*(k)} = e_{a^*} \otimes e_{b^*}$ for some $a_{(0)}, b_{(0)}, a^*, b^* \in [n]$. We first note that the conditions specified in the lemma meets the assumptions required by Lemma G.22, namely there is only one incorrect column in $W_2$ missing and that column is being used in the forward pass at least one time (since $\mathcal{J}$ is non-empty). To prove $\mathtt{HardMax}(W_{2(1)}^{(k)}) = W_2^{*(k)}$, it is sufficient to show that $\arg\max(W_{2(1)}^{(k)}) = a^*n + b^*$. By Lemma G.22, we have

$$\frac{\partial \mathcal{L}}{\partial P_2^{(k)}} = 2 \sum_{j=1}^{L} \left( \frac{\partial V_3^{(j)}}{\partial P_2^{(k)}} \right)^{\top} \left( V_3^{(j)} - V_3^{*(j)} \right) = 2\alpha \left( P_2^{(k)} - P_2^{*(k)} \right) = 2\alpha e_{a_{(0)}} \otimes e_{b_{(0)}} - 2\alpha e_{a^*} \otimes e_{b^*}. \tag{64}$$

Since $\alpha \geq 1$, the negative margin of the $a^*n + b^*$-th entry dominates the initial difference in the initialization which is bounded by $\|W_{2(0)}^{(k)}\|_0 \leq \frac{1}{2}$. Thus we have

$$\arg\max \left( W_{2(1)}^{(k)} \right) = \arg\max \left( W_{2(0)}^{(k)} - \frac{\partial \mathcal{L}}{\partial P_{2(0)}^{(k)}} \right) = a^*n + b^* \tag{65}$$

as desired. $\square$

### G.3.3. PROOF FOR THEOREM G.24

Now we are ready to prove for the main theorem restated below:

> **Theorem G.24** (Learning $\Pi^*$ with context-enhanced surrogate GD with $\Pi^*$-coverable input).
> *For any initialization $W_{1(0)}, W_{2(0)} \in \mathbb{R}^{n^2 \times n^2}$ such that $\|W_{1(0)}\|_0 \leq \frac{1}{2}$ and $\|W_{2(0)}\|_0 \leq \frac{1}{2}$, for any target set of phrasebooks $\Pi^* = \{\pi_1^*, \pi_2^*\}$ in $\mathbf{MLT}(2, n)$, given an $\Pi^*$-coverable input $V_1$ and the corresponding ground truth label $V_3^* = \mathbf{MLT}_{\Pi^*}(V_1)$, Algorithm 2 terminates with $\mathtt{HardMax}\,(W_1) = W_1^*$ and $\mathtt{HardMax}\,(W_2) = W_2^*$.*

*Proof.* The statement can be proven by a similar induction as in the proof for Theorem G.16.

Let the induction hypothesis be such that when the enumeration goes to $k$-th column of the $i$-th layer, if $\texttt{HardMax}(W_l^{(j)} + W_l^{*(j)}) = W_l^{*(j)}$ for all $(l,j) \in [2] \times [n^2]$ and $\texttt{HardMax}\left(W_l^{(j)}\right) = W_l^{*(j)}$ for all $(l,j)$ such that $l < i$ or $l = i \wedge j < k$, then the gradient update on the $k$-th column of the $i$-th layer ends with $W_i^{(k)} = W_i^{*(k)}$ while $\texttt{HardMax}(W_l^{(j)} + W_l^{*(j)}) = W_l^{*(j)}$ for all $(l,j) \in [2] \times [n^2]$ is preserved.

The base case is satisfied as with initialization of $\|W_{1(0)}\|_0 \leq \frac{1}{2}$ and $\|W_{2(0)}\|_0 \leq \frac{1}{2}$, by Lemma G.11, we have $\texttt{HardMax}(W_l^{(j)} + W_l^{*(j)}) = \texttt{HardMax}(0 + W_l^{*(j)}) = W_l^{*(j)}$ for all $(l,j) \in [d] \times [n^2]$ and there are no requirements for $W_i^{(k)} = W_i^{*(k)}$ yet.

For the induction step, we note that with the inductive hypothesis of $\texttt{HardMax}(W_l^{(j)} + W_l^{*(j)}) = W_l^{*(j)}$ for all $(l,j) \in [2] \times [n^2]$, $(C_{1(k)}, W_2^*)$ (when $i = 1$) or $(W_1^*, C_{2(k)})$ (when $i = 2$) will correctly condition all columns of $P$'s except for the $P_i^{(k)}$ since

$$C_{i(k)}^{(k)} = W_i^* \left(I_{n^2} - \text{diag}(\bar{e}_k)\right)^{(k)} = 0. \tag{66}$$

Thus by Lemma G.21 (when $i = 1$) or Lemma G.23 (when $i = 2$), we know that after updating $W_i^{(k)}$, we have $\texttt{HardMax}(W_i^{(k)}) = W_i^{*(k)}$. The newly added column provides the correct inductive hypothesis on $\texttt{HardMax}(W_l^{(j)}) = W_l^{*(j)}$ for the next enumeration step.

By induction to $i = 2$ and $k = n^2$, we will be able to recover $\texttt{HardMax}(W_i) = W_i^*$ for all $i \in [d]$. $\square$

Similarly we may use the coupon collecting argument to generalize the input to uniformly random strings as follows:

> **Corollary G.25** (Learning $\Pi^*$ in $\textbf{MLT}(2, n)$ with context-enhanced surrogate GD with random input).
> *For any initialization $W_{1(0)}, W_{2(0)} \in \mathbb{R}^{n^2 \times n^2}$ such that $\|W_{1(0)}\|_0 \leq \frac{1}{2}$ and $\|W_{2(0)}\|_0 \leq \frac{1}{2}$, for any target set of phrasebooks $\Pi^* = \{\pi_1^*, \pi_2^*\}$ in $\textbf{MLT}(2, n)$, with probability at least $1 - \delta$ over a uniformly random input $V_1$ of length $L = 2n^2 \log \frac{2n}{\delta}$, Algorithm 2 provided with the ground truth label $V_3^* = \textbf{MLT}_{\Pi^*}(V_1)$ terminates with $\texttt{HardMax}(W_1) = W_1^*$ and $\texttt{HardMax}(W_2) = W_2^*$.*

## G.4. Auxiliary Lemmas for Learning Surrogate Models

**Lemma G.26.** *For any long one-hot vector $v = e_a \otimes e_b$, with $Q = (I_n \otimes 1_n)(1_n \otimes I_n)^\top$ we have*

$$Qv = e_b \otimes 1_n; \qquad Q^\top v = 1_n \otimes e_a. \tag{67}$$

*Proof.* Note that with $v = e_a \otimes e_b$, we have

$$
\begin{aligned}
Qv &= (I_n \otimes 1_n)(1_n^\top \otimes I_n^\top)(e_a \otimes e_b) = (I_n \otimes 1_n)(1 \otimes e_b) = (I_n \otimes 1_n)(e_b \otimes 1) = e_b \otimes 1_n; \\
Q^\top v &= (1_n \otimes I_n)(I_n^\top \otimes 1_n^\top)(e_a \otimes e_b) = (1_n \otimes I_n)(e_a \otimes 1) = (1_n \otimes I_n)(1 \otimes e_a) = 1_n \otimes e_a.
\end{aligned}
\tag{68}
$$

$\square$

**Lemma G.27.** *For any one-hot vector $v = e_a \otimes e_b \in \mathbb{R}^{n^2}$,*

$$Q^\top \text{diag}\left(Q^\top v\right) \text{diag}\left(Q^\top v\right) Q = (1_n 1_n^\top) \otimes I_n.$$

*Proof.* By Lemma G.26, $Q^\top v = 1_n \otimes e_a$. Therefore $\text{diag}\left(Q^\top v\right) = \text{diag}(1_n) \otimes \text{diag}(e_a) = I_n \otimes \text{diag}(e_a)$. Thus

$$\text{diag}\left(Q^\top v\right) Q = (I_n \otimes \text{diag}(e_a))(I_n \otimes 1_n)(1_n^\top \otimes I_n^\top) = (I_n \otimes e_a)(1_n^\top \otimes I_n^\top) \tag{69}$$

and therefore we have

$$
\begin{aligned}
\boldsymbol{Q}^\top \mathrm{diag}\left(\boldsymbol{Q}^\top \boldsymbol{v}\right) \mathrm{diag}\left(\boldsymbol{Q}^\top \boldsymbol{v}\right) \boldsymbol{Q} &= (\mathbf{1}_n \otimes I_n)\left(I_n \otimes \boldsymbol{e}_a^\top\right)(I_n \otimes \boldsymbol{e}_a)(\mathbf{1}_n^\top \otimes I_n^\top) \\
&= (\mathbf{1}_n \otimes I_n)(I_n \otimes 1)(\mathbf{1}_n^\top \otimes I_n^\top) \qquad \left(\text{since } \boldsymbol{e}_a^\top \boldsymbol{e}_a = 1\right) \\
&= (\mathbf{1}_n \otimes I_n)(1 \otimes I_n)(\mathbf{1}_n^\top \otimes I_n^\top) \\
&= (\mathbf{1}_n \otimes I_n)(\mathbf{1}_n^\top \otimes I_n^\top) \\
&= (\mathbf{1}_n \mathbf{1}_n^\top) \otimes I_n
\end{aligned}
\tag{70}
$$

$\square$

**Lemma G.28.** *For any one-hot vector $\boldsymbol{v} = \boldsymbol{e}_a \otimes \boldsymbol{e}_b \in \mathbb{R}^{n^2}$,*

$$
\boldsymbol{Q}\,\mathrm{diag}\left(\boldsymbol{Q}\boldsymbol{v}\right)\mathrm{diag}\left(\boldsymbol{Q}\boldsymbol{v}\right)\boldsymbol{Q}^\top = I_n \otimes (\mathbf{1}_n \mathbf{1}_n^\top).
$$

*Proof.* This proof is very similar to the proof for Lemma G.27. By Lemma G.26, $\boldsymbol{Q}^\top \boldsymbol{v} = \mathbf{1}_n \otimes \boldsymbol{e}_a$. Therefore $\mathrm{diag}\left(\boldsymbol{Q}\boldsymbol{v}\right) = \mathrm{diag}\left(\boldsymbol{e}_b\right) \otimes \mathrm{diag}\left(\mathbf{1}_n\right) = \mathrm{diag}\left(\boldsymbol{e}_b\right) \otimes I_n$. Thus

$$
\mathrm{diag}\left(\boldsymbol{Q}\boldsymbol{v}\right)\boldsymbol{Q}^\top = \left(\mathrm{diag}\left(\boldsymbol{e}_b\right) \otimes I_n\right)(\mathbf{1}_n \otimes I_n)\left(I_n^\top \otimes \mathbf{1}_n^\top\right) = \left(\boldsymbol{e}_b \otimes I_n\right)\left(I_n^\top \otimes \mathbf{1}_n^\top\right)
\tag{71}
$$

and therefore we have

$$
\begin{aligned}
\boldsymbol{Q}\,\mathrm{diag}\left(\boldsymbol{Q}\boldsymbol{v}\right)\mathrm{diag}\left(\boldsymbol{Q}\boldsymbol{v}\right)\boldsymbol{Q}^\top &= (I_n \otimes \mathbf{1}_n)\left(\boldsymbol{e}_b^\top \otimes I_n\right)(\boldsymbol{e}_b \otimes I_n)(I_n^\top \otimes \mathbf{1}_n^\top) \\
&= (I_n \otimes \mathbf{1}_n)(1 \otimes I_n)(I_n^\top \otimes \mathbf{1}_n^\top) \qquad \left(\text{since } \boldsymbol{e}_b^\top \boldsymbol{e}_b = 1\right) \\
&= (I_n \otimes \mathbf{1}_n)(I_n \otimes 1)(I_n^\top \otimes \mathbf{1}_n^\top) \\
&= (I_n \otimes \mathbf{1}_n)(I_n^\top \otimes \mathbf{1}_n^\top) \\
&= I_n \otimes (\mathbf{1}_n \mathbf{1}_n^\top).
\end{aligned}
\tag{72}
$$

$\square$

> **Lemma G.29** (Tail Bound for Coupon Collector Problem (Motwani, 1995))**.**
> *For a set $S$ of size $n$, with probability at least $1 - \delta$ one can cover all unique elements of $S$ in $n \log \frac{n}{\delta}$ independent uniformly random sampling trials from $S$.*

### G.5. Learning $\Pi^*$ in $\mathrm{MLT}_{\Pi^*}$ with Gradient Descent (Empirical Evidence)

In this section we provide more details on empirically optimizing the simple surrogate model $\textsc{Surr-MLT}_{\{W_i\}_{i=1}^d}$, which was only briefly discussed in the main text by the end of Section 5.1. We will first introduce the approximations we made to the surrogate model to make gradient-based optimization easy and stable, then we will present empirical results on the model learning target sets of phrasebooks $\mathbf{MLT}_{\Pi^*}$ in $\mathbf{MLT}(5, 10)$, $\mathbf{MLT}(10, 10)$, and even $\mathbf{MLT}(20, 10)$.

G.5.1. APPROXIMATED LATENT MODEL FOR GD

Recall that with input sequence represented by $\boldsymbol{V}_1 \in \mathbb{R}^{n^2 \times L}$, the surrogate model for a depth-$d$ translation is being recursively defined by the translation + shifting operations

$$
\boldsymbol{V}_{i+1} = \texttt{HardMax}(\boldsymbol{C}_i + \boldsymbol{W}_i)\,\texttt{Shift}(\boldsymbol{V}_i)
\tag{73}
$$

until we reach $\boldsymbol{V}_{d+1}$. While this model captures the essence of transition from ICL capability to memorization of specific set of phrasebooks, $\texttt{HardMax}$ is making it not directly differentiable and hard to optimize. To address this issue, we approximate it with an column-wise softmax function with very low temperature ($T = 1/25$). The recursive definition in the approximated model is then

$$
\tilde{\boldsymbol{V}}_{i+1} = \texttt{SoftMax}(25(\boldsymbol{C}_i + \boldsymbol{W}_i))\,\texttt{Shift}(\tilde{\boldsymbol{V}}_i).
\tag{74}
$$

We denote the recursive surrogate model with the softmax substitution as $\text{S\tilde{U}RR\text{-}MLT}_{\{W_i\}_{i=1}^d}(C_1, C_2, \ldots, C_d, V_1)$ where

$$
\begin{aligned}
\tilde{V}_{d+1} &= \text{S\tilde{U}RR\text{-}MLT}_{\{W_i\}_{i=1}^d}(C_1, C_2, \ldots, C_d, V_1) \\
&\triangleq \texttt{SoftMax}(25C_d + 25W_d) \\
&\quad \texttt{Shift}\Big(\texttt{SoftMax}(25C_{d-1} + 25W_{d-1})\,\texttt{Shift}\Big(\cdots\texttt{SoftMax}(25C_1 + 25W_1)\,\texttt{Shift}(\tilde{V}_1)\cdots\Big)\Big).
\end{aligned}
\tag{75}
$$

We define the objective function as the column-wise cross-entropy loss between the final output and the input. Namely for input $V_1$ with prediction $\tilde{V}_{d+1}$ and ground truth label $V_{d+1}^*$, the loss is computed as

$$
\mathcal{L} = \sum_{k=1}^{L} \text{CrossEntropy}(\tilde{V}_{d+1}^{(k)}, V_{d+1}^{*(k)}).
\tag{76}
$$

We follow the same masking (dropout) curriculum as described in Appendix G.2 and Appendix G.3, that at each step we zero-out a single column from a single context matrix $C_i$. We experiment on two gradient update schemes:

- Layer-wise Training: at each step, if we are masking a column on $C_i$, we only compute the gradient with respect to $W_i$ and update it. This training is more akin to the theoretical analysis described in Appendix G.3.

- Full Parameter Training: at any step, we compute the gradient with respect to each of the weight matrices and update all parameters. This is more akin to the real gradient-based training as we do not have the heuristics for localized update.

To allow for fast and stable training, we adopt a very large learning rate of $\eta = 100$ and apply parameter clipping between $[0, 1]$ after each update. The complete algorithm is described as follows:

---

**Algorithm 4** Layerwise Gradient Descent with Context-Enhanced Learning For Optimizing SURR-MLT

---

1: **Input:**
2: input $V_1 \in \mathbb{R}^{n^2 \times L}$, label $V_{d+1}^* \in \mathbb{R}^{n^2 \times L}$, descriptive text $W_1^*, \ldots, W_d^* \in \mathbb{R}^{n^2 \times n^2}$, learning rate $\eta$, total steps $T$
3:
4: **Initialize** $W_1, \ldots, W_d \leftarrow \mathbf{0}$          # Start with zero initialization
5: **for** $t = 1$ **to** $T$ **do**
6:     $i \leftarrow \lfloor (t-1)/n^2 \rfloor \% d + 1$          # Get the layer to be masked
7:     $k \leftarrow ((t-1)\% n^2) + 1$          # Get the column index to be masked
8:     **Initialize** $C_{i(k)} \leftarrow W_i^*\,(I_{n^2} - \text{diag}(\bar{e}_k))$          # Create masked context matrix
9:     $\tilde{V}_{d+1} \leftarrow \text{SURR\text{-}MLT}_{\{W_i\}_{i=1}^d}(W_1^* \ldots, W_{i-1}^*, C_{i(k)}, W_{i+1}^* \ldots, W_d^*, V_1)$
10:     $\mathcal{L} \leftarrow \text{CrossEntropy}(\tilde{V}_{d+1}, V_{d+1}^*)$
11:     $W_i \leftarrow W_i - \eta \nabla_{W_i} \mathcal{L}$          # Update the weight for the layer with mask
12: **end for**
13: **Return** $W_1, \ldots, W_d$.

---

---

**Algorithm 5** Full Parameter Gradient Descent with Context-Enhanced Learning For Optimizing SURR-MLT

---

1: **Input:**
2: input $\boldsymbol{V}_1 \in \mathbb{R}^{n^2 \times L}$, label $\boldsymbol{V}_{d+1}^* \in \mathbb{R}^{n^2 \times L}$, descriptive text $\boldsymbol{W}_1^*, \ldots, \boldsymbol{W}_d^* \in \mathbb{R}^{n^2 \times n^2}$, learning rate $\eta$, total steps $T$
3:
4: **Initialize** $\boldsymbol{W}_1, \ldots, W_d \leftarrow \boldsymbol{0}$          # Start with zero initialization
5: **for** $t = 1$ **to** $T$ **do**
6:     $i \leftarrow \lfloor (t-1)/n^2 \rfloor \% d + 1$          # Get the layer to be masked
7:     $k \leftarrow ((t-1)\%n^2) + 1$          # Get the column index to be masked
8:     **Initialize** $\boldsymbol{C}_{i(k)} \leftarrow \boldsymbol{W}_i^* \left( \boldsymbol{I}_{n^2} - \mathrm{diag}(\bar{\boldsymbol{e}}_k) \right)$          # Create masked context matrix
9:     $\tilde{\boldsymbol{V}}_{d+1} \leftarrow \mathrm{SURR\text{-}MLT}_{\{\boldsymbol{W}_i\}_{i=1}^d}(\boldsymbol{W}_1^* \ldots, \boldsymbol{W}_{i-1}^*, \boldsymbol{C}_{i(k)}, \boldsymbol{W}_{i+1}^* \ldots, \boldsymbol{W}_d^*, \boldsymbol{V}_1)$
10:     $\mathcal{L} \leftarrow \mathrm{CrossEntropy}(\tilde{\boldsymbol{V}}_{d+1}, \boldsymbol{V}_{d+1}^*)$
11:     **for** $l = 1$ **to** $d$ **do**
12:        $\boldsymbol{W}_l \leftarrow \boldsymbol{W}_l - \eta \nabla_{\boldsymbol{W}_l} \mathcal{L}$          # Update the weight for all layers
13:     **end for**
14: **end for**
15: **Return** $\boldsymbol{W}_1, \ldots, \boldsymbol{W}_d$.

---

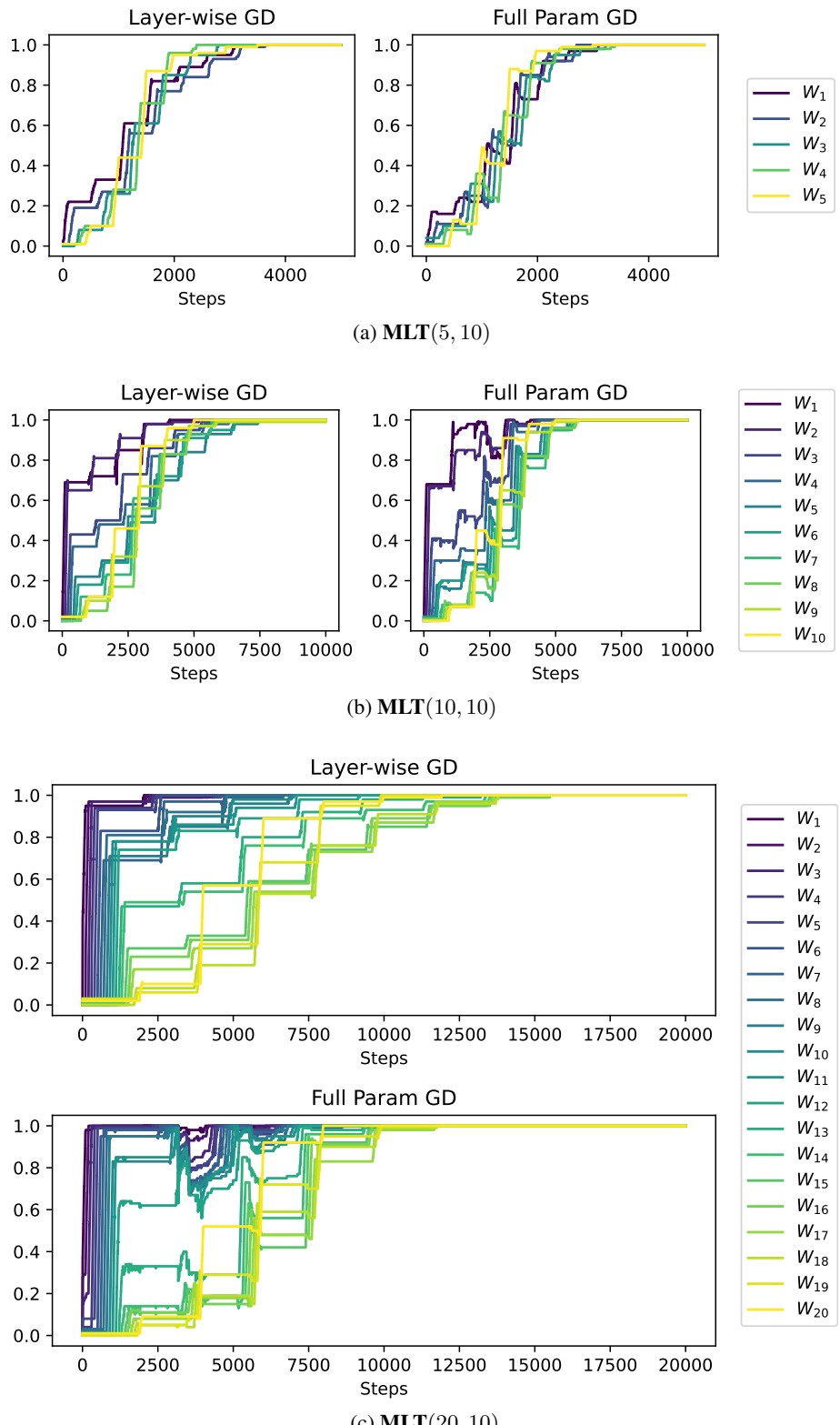

(a) **MLT**$(5, 10)$

(b) **MLT**$(10, 10)$

(c) **MLT**$(20, 10)$

*Figure 13.* We perform Layer-wise and Full Parameter Training with Gradient Descent (defined in Appendix G.5.1) on the SURR-MLT (Definition G.7) designed for **MLT**$_{\Pi^*}$ in **MLT**$(5, 10)$, **MLT**$(10, 10)$, **MLT**$(20, 10)$ respectively (alternately, depth $d = 5, 10, 20$ respectively, while number of characters is fixed at $n = 10$). Here, we report the portion of columns from the trainable parameters, which after `HardMax` application $\{\texttt{HardMax}(\boldsymbol{W}_i)\}_{i=1}^{d}$, align with the corresponding stochastic matrices of the phrasebooks $\{\texttt{Matrix}(\pi_i^*)\}_{i=1}^{d}$. We observe that under both algorithms, the trainable parameters quickly learn the relevant stochastic matrices.

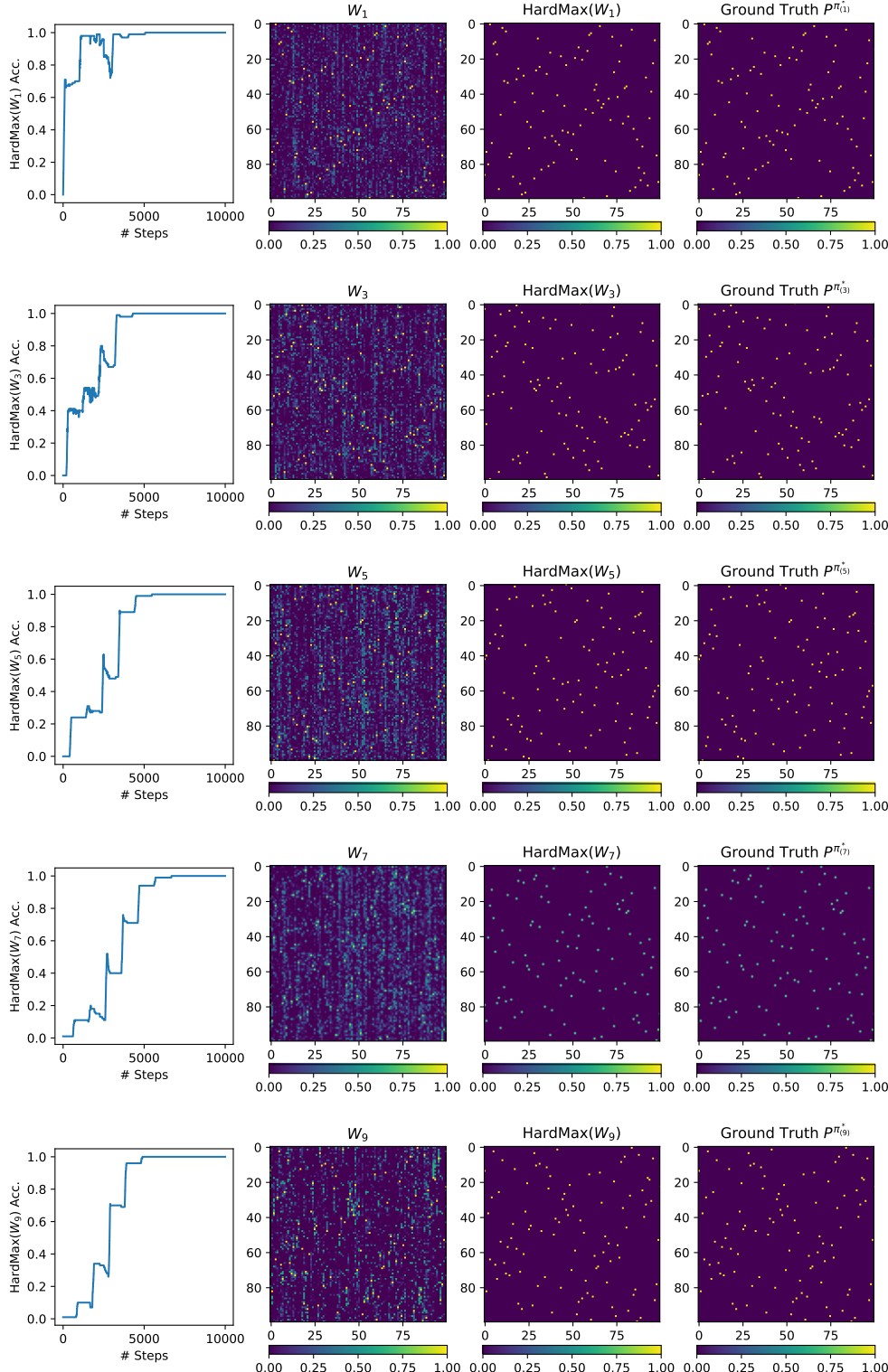

*Figure 14.* Detailed analysis on the Full parameter training behavior of the trainable parameters in SURR-MLT from Figure 13 for $\mathbf{MLT_{\Pi^*}}$ in $\mathbf{MLT}(10, 10)$ (i.e. depth $d = 10$ and number of characters $n = 10$). We report the behavior of all odd-index parameters $\boldsymbol{W}_1, \boldsymbol{W}_3, \cdots, \boldsymbol{W}_9$. (left to right) first, we show the number of columns of the trainable parameter, which after $\mathtt{HardMax}\,(\boldsymbol{W}_i)$ align with the corresponding columns of $\mathtt{Matrix}\,(\pi_i^*)$. Second, third and fourth visualize the matrices, $\boldsymbol{W}_i$, $\mathtt{HardMax}\,(\boldsymbol{W}_i)$, and $\mathtt{Matrix}\,(\pi_i^*)$ respectively. All matrices learn to match $\mathtt{Matrix}\,(\pi_i^*)$ at the end of training with $\mathtt{HardMax}$ operation.

# H. Construction of a Transformer that can Simulate the Latent Model

## H.1. Useful definitions and lemmas

**Definition H.1** (Relative self-attention with 1 head). For a set of matrices $\{W_{query}, W_{key}, W_{value}\}$ with each matrix $\in \mathbb{R}^{k \times k}$ for some $k > 0$ and a set of $(t+1)$ biases $\{b_i\}_{t \le i \le 0}$, the self-attention computation on an input sequence $x_1, \cdots, x_L$ with each $x_{p_2} \in \mathbb{R}^k$ is given by the output sequence $y_1, \cdots, y_L$, where for all $p_1 \in [1, L]$

$$y_{p_1} = \sum_{p_2=1}^{L} a_{p_1, p_2} o_j, \text{ where } a_{p_1 p_2} = \frac{e^{q_{p_1}^\top k_{p_2} + b_{p_2 - p_1}}}{\sum_{p_2' \le p_1} e^{q_{p_1}^\top k_{p_2'} + b_{p_2' - p_1}}} \text{ if } p_2 \le p_1, 0 \text{ otherwise}$$

$$q_{p_2} = W_{query} x_{p_2}, \quad k_{p_2} = W_{key} x_{p_2}, \quad o_{p_2} = W_{value} x_{p_2}, \quad \text{for all } p_2 \in [1, L].$$

For a relative self-attention with $H$ heads, we will simply add the output of the $H$ heads as the final output.

**Definition H.2** (MLP). For a set of matrices $W_{outer} \in \mathbb{R}^{k \times H}, W_{inner} \in \mathbb{R}^{H \times k}$ for some $k, H > 0$ and an activation function $\sigma$, the output of the MLP layer on an input sequence $x_1, \cdots, x_L$ with each $x_i \in \mathbb{R}^k$ is given by the output sequence $y_1, \cdots, y_L$, where for all $i$

$$y_i = W_{outer} \sigma(W_{inner} x_i).$$

**Lemma H.3.** *For GELU ([Hendrycks & Gimpel, 2016](#)) activation function, which takes $x \in \mathbb{R}$ as input and returns $x\Phi(x)$ as output, with $\Phi(x)$ representing the standard Gaussian cumulative distribution function, for any two variables $x, y \in \mathbb{R}$, the following holds true:*

$$\sqrt{\pi/2} \left(GELU(x+y) - GELU(x) - GELU(y)\right) = xy + \mathcal{O}(x^3 y^3).$$

The above lemma has been taken from [Akyurek et al. (2023)](#).

## H.2. Transformer construction

Recall that a translation task in **MLT**$(d, n)$ involves two primary operations at each step : *Circular shift* and *Translate*. We will refer to the surrogate model to use notations for different operations in the translation task. Recall from Equation (18), the surrogate model for **MLT**$(d, n)$, denoted by SURR-MLT$_{\{W_i\}_{i=1}^d}(\cdot)$ with trainable parameters $\{W_i\}_{i=1}^d = \{W_1, \cdots, W_d\}$, can be represented by the following recursive expression

$$V_{i+1} = \texttt{HardMax}(C_i + W_i)\,\texttt{Shift}(V_i) := \texttt{HardMax}(C_i + W_i)\tilde{V}_i, \tag{77}$$

for $1 \le i \le d$. Here $\texttt{Shift}$ represents *Circular shift* operation, and $\texttt{HardMax}(C_i + W_i)$ represents *Translate* operation, where the operation can be done either using the relevant in-context information $C_i$ or the in-weights memory parameters $W_i$ when in-context information isn't provided in form of $C_i$. $\texttt{HardMax}$ is the column-wise hard-max function converting $C_i + W_i$ to a binary column stochastic matrix.

---

**Lemma H.4.** *For the family of translation tasks **MLT**$(d, n)$, there exists a transformer model with embedding size $2n^2 + 2d + 4$, 2d relative self-attention layers (containing either 1 or 3 heads), and 2d MLP layers that can simulate the surrogate model,* SURR-MLT$_{\{W_i\}_{i=1}^d}(\cdot)$.

*For each input sequence $s_1$ of length $L$, the input sequence of embeddings to the transformer will be of length $n^2 d + L/2 + d$, where the last $d$ embeddings are padding tokens (`<P>`) and given as 0s, and the length of output sequence of the transformer model will be $n^2 d + d + L/2$, where the last $L/2$ output embeddings will be used for loss computation. The middle $d$ embeddings in output are represented by `<THINK>` tokens.* [a]

---

[a] In our experiments, we used output sequence length as $n^2 d + d + dL/2$, where $(d+1)L/2$ tokens in the output sequence were represented by `<THINK>` tokens. Instead, we only use $d$ `<THINK>` tokens in our output construction for simplicity, the proof can be easily modified to align with the experiments.

**Outline of the construction:** Our argument will be for any general $\mathbf{MLT}(d, n)$. To create the transformer, we will create similar transformer modules that handle $\mathtt{HardMax}(\boldsymbol{C}_i + \boldsymbol{W}_i)\,\mathtt{Shift}(\boldsymbol{V}_i)$ for each $i$. We will refer to the constructed modules as MLT-MODULE and will mention the specific step $i$ as an argument when we attempt to use the module to perform translation step $\mathtt{HardMax}(\boldsymbol{C}_i + \boldsymbol{W}_i)\,\mathtt{Shift}(\boldsymbol{V}_i)$. W.l.o.g., we will assume we are building a MLT-MODULE to perform translation step $\mathtt{HardMax}(\boldsymbol{C}_i + \boldsymbol{W}_i)\,\mathtt{Shift}(\boldsymbol{V}_i)$. MLT-MODULE will contain two modules, CIRCULAR SHIFT-MODULE and TRANSLATE-MODULE, which simulate *Circular shift* and *Translate* operations, and have been outlined in Algorithms 6 and 7.

Next, we explain the structure of the embeddings in the transformer architecture. Our embeddings will be built on the context representation matrices $\{\boldsymbol{C}_1, \cdots, \boldsymbol{C}_d\}$ and the matrix representation $\boldsymbol{V}_1$ from input sequence $\boldsymbol{s}_1 := (\boldsymbol{s}_{1,1}, \cdots, \boldsymbol{s}_{1,L})$ and subsequent intermediate representation of the translation task. Furthermore, our embeddings will contain additional information like indices of the context matrices when utilizing them in-context, segment indicators that represent whether embeddings represent context matrices, or the input sequence tokens, and start and end indicators that indicate the start and the end embeddings representing the input sequence. This information can be extracted from the input sequence using a few input processing layers, though we do not delve into the specifics.

### H.3. Structure of input embeddings to MLT-MODULE

Our embeddings will be split into 4 components: token, context matrix indicator, start and end indicator, and segment indicator. To maintain simplicity in our discussion, we will present them as 4 separate embeddings.

For a module that will represent $\mathtt{HardMax}(\boldsymbol{C}_i + \boldsymbol{W}_i)\,\mathtt{Shift}(\boldsymbol{V}_i)$, we assume the input will be provided as 2 major segments.

1. **First segment: in-context matrices** We will give the in-context information $\{\boldsymbol{C}_{p_1}\}_{p_1=1}^d$ as follows.: each context matrix $\boldsymbol{C}_{p_1}$ will be fed as $n^2$ token embeddings, $\{[\boldsymbol{e}_j; \boldsymbol{C}_{p_1}\boldsymbol{e}_j] \in \mathbb{R}^{2n^2}\}_{j=1}^{n^2}$, where $\boldsymbol{e}_j$ represents a one-hot $n^2$ dimensional vector that contains 1 in position $j$ and $[\boldsymbol{e}_j; \boldsymbol{C}_{p_1}\boldsymbol{e}_j]$ represents a concatenation of $\boldsymbol{e}_j$ and $\boldsymbol{C}_i\boldsymbol{e}_j$. Thus, the in-context information will look as follows

$$\{[\boldsymbol{e}_j; \boldsymbol{C}_1\boldsymbol{e}_j]\}_{j=1}^{n^2}, \cdots, \{[\boldsymbol{e}_j; \boldsymbol{C}_d\boldsymbol{e}_j]\}_{j=1}^{n^2}$$

   **Additional embedding: context matrix level indicator** In order to differentiate the different context matrices, we will use an additional $d$ dimensions in the embeddings to represent a one-hot vector that indicates the index of the corresponding step they will be used for. For simplicity, we will represent these dimensions separately as separate embeddings: $\boldsymbol{l}_{p_1} \in \mathbb{R}^d$, which are one-hot vectors that contain 1 in position $p_1$ and 0 otherwise.

2. **Second segment: input query** For a length-$L$ input sequence $\boldsymbol{s}_i = (\boldsymbol{s}_{i,1}, \boldsymbol{s}_{i,2}, \ldots, \boldsymbol{s}_{i,L})$, we will use the sequence of columns of its matrix representation $\boldsymbol{V}_i \in \mathbb{R}^{n^2 \times L/2}$, appended by 0s to match embedding sizes, as token embeddings for the input query. A padding embedding `<P>`, containing 0s, follows this sequence as an end of sequence embedding.

   **Additional embedding: start and end indicator embedding** We will have 2 additional dimensions representing whether a token represents the start or the end of the input query sequence (first or second dimension activated respectively). Start of the input query sequence is determined by the first embedding in the input query, while end of the input query sequence is determined by the first padding embedding containing 0s after the input query sequence. We will represent these dimensions separately as separate embeddings: $\{\boldsymbol{b}_1, \boldsymbol{b}_2, \boldsymbol{0} \in \mathbb{R}^2\}_{i \in L}$, which are one-hot vectors. $\boldsymbol{b}_1$ contains 1 in dimension 1 if the embeddings represents start of the sequence, $\boldsymbol{b}_2$ contains 1 in dimension 2 if the embeddings represent end of the sequence. Other embeddings have $\boldsymbol{0}$s.

**Additional embedding: segment embedding** We will differentiate the input embeddings in the two segments using 2 dimensions that represent one-hot vectors indicating segment indices. We will represent these dimensions separately as separate embeddings: $\boldsymbol{g}_1, \boldsymbol{g}_2 \in \mathbb{R}^2$, where both are one-hot vectors, with $\boldsymbol{g}_1$ containing 1 in dimension 1 and $\boldsymbol{g}_2$ containing 1 in dimension 2.

All the notations have been summarized in Table 8.

**Additional optional inputs:** There might be additional input embeddings, represented as `<THINK>` in the first segment, which are null inputs and are ignored during self-attention computation. As discussed next, we will right shift the sequence by $1$ at each step, in order to handle *Circular shift* operation with causal masking in transformers.

### H.4. Structure of the output embeddings from MLT-MODULE

All the embeddings in the first segment are kept intact. In the second segment, the module outputs a null output `<THINK>`, followed by $L/2$ output embeddings that represent the columns of $\boldsymbol{V}_i$. We will require `<THINK>` to represent *Circular shift* with causal self-attention in transformers. `<THINK>` will be ignored in self-attention computation and so, we will ignore their discussion for simplicity of presentation. Other embedding values are kept intact for the generated output, except start and end indicator embeddings $\boldsymbol{b}_1, \boldsymbol{b}_2$, which need to be right shifted at each step. The right shift operation can be handled similar to our computations on the token embeddings and so, we ignore them in our construction below. We summarize these in Table 9.

| Embedding Name | Dimension size | First segment values (In-context information) | Second segment values (Input sequence embeddings) |
|---|---|---|---|
| Token | $2n^2$ | $\{[\boldsymbol{e}_j; \boldsymbol{C}_1\boldsymbol{e}_j]\}_{j=1}^{n^2}, \cdots, \{[\boldsymbol{e}_j; \boldsymbol{C}_d\boldsymbol{e}_j]\}_{j=1}^{n^2}$ | $[\boldsymbol{V}_{i-1}^{(1)}; \boldsymbol{0}], \cdots, [\boldsymbol{V}_{i-1}^{(L/2)}; \boldsymbol{0}], $ `<P>` |
| Context matrix index indicator | $d$ | $\{\boldsymbol{l}_1\}_{[1,n^2]}, \{\boldsymbol{l}_2\}_{[1,n^2]}, \cdots \{\boldsymbol{l}_d\}_{[1,n^2]}$ | $\boldsymbol{0}, \cdots, \boldsymbol{0}$ |
| Start and End indicator | $2$ | $\boldsymbol{0}, \cdots, \boldsymbol{0}$ | $\boldsymbol{b}_1, \boldsymbol{0}, \cdots, \boldsymbol{0}, \boldsymbol{b}_2$ |
| Segment | $2$ | $\boldsymbol{g}_1, \cdots, \boldsymbol{g}_1$ | $\boldsymbol{g}_2, \cdots, \boldsymbol{g}_2$ |

*Table 8.* Input embeddings to MLT-MODULE that simulates $\text{HardMax}(\boldsymbol{C}_i + \boldsymbol{W}_i)\,\text{Shift}(\boldsymbol{V}_i)$. $\boldsymbol{e}_j$ indicates a one-hot $n^2$ dimensional vector that contains $1$ in dimension $j$.

| Embedding Name | Dimension size | First segment values (In-context information) | Second segment values (Input sequence embeddings) |
|---|---|---|---|
| Token | $2n^2$ | $\{[\boldsymbol{e}_j; \boldsymbol{C}_1\boldsymbol{e}_j]\}_{j=1}^{n^2}, \cdots, \{[\boldsymbol{e}_j; \boldsymbol{C}_d\boldsymbol{e}_j]\}_{j=1}^{n^2}$ | `<THINK>`, $[\boldsymbol{V}_i^{(1)}; \boldsymbol{0}], \cdots, [\boldsymbol{V}_i^{(L/2)}; \boldsymbol{0}]$ |
| Context matrix index indicator | $d$ | $\{\boldsymbol{l}_1\}_{[1,n^2]}, \{\boldsymbol{l}_2\}_{[1,n^2]}, \cdots \{\boldsymbol{l}_d\}_{[1,n^2]}$ | $\boldsymbol{0}, \boldsymbol{0}, \cdots, \boldsymbol{0}$ |
| Start and End indicator | $2$ | $\boldsymbol{0}, \cdots, \boldsymbol{0}$ | $\boldsymbol{0}, \boldsymbol{b}_1, \boldsymbol{0}, \cdots, \boldsymbol{0}, \boldsymbol{b}_2$ |
| Segment | $2$ | $\boldsymbol{g}_1, \cdots, \boldsymbol{g}_1$ | $\boldsymbol{0}, \boldsymbol{g}_2, \cdots, \boldsymbol{g}_2$ |

*Table 9.* Output of the transformer module MLT-MODULE that simulates $\text{HardMax}(\boldsymbol{C}_i + \boldsymbol{W}_i)\,\text{Shift}(\boldsymbol{V}_i)$. `<THINK>` represents a null output and won't be attended to in the future modules. We ignore this symbol for simplicity, when analyzing any module. $\boldsymbol{e}_j$ indicates a one-hot $n^2$ dimensional vector that contains $1$ in dimension $j$.

**Constructing the MLT-MODULE:** The MLT-MODULE consists of 2 self-attention layers and 2 MLP layers. We use one self-attention layer and an MLP layer to represent *Circular shift* operation, one self-attention layer to represent $\boldsymbol{C}_i\,\text{Shift}(\boldsymbol{V}_i)$, and one MLP layer to represent $\text{HardMax}(\boldsymbol{C}_i + \boldsymbol{W}_i)\,\text{Shift}(\boldsymbol{V}_i)$. We name the two modules for *Circular shift* and *Translate* as CIRCULAR SHIFT-MODULE and TRANSLATE-MODULE respectively. We have outlined their constructions in Algorithms 6 and 7.

### H.5. Step 1 (CIRCULAR SHIFT-MODULE): Represent *Circular shift* using a self-attention and an MLP layer

As *Circular shift* only focuses on the input query sequence and not the in-context matrices, we will simply focus the module's operation on embeddings in the second segment. The effect of the operation on embeddings in the first segment can be removed using a gated residual connection. From Lemma G.3, we have that the output of the *Circular shift* operation on any sequence, represented by its matrix representation $\boldsymbol{V}_i$, can be written as

$$\tilde{\boldsymbol{V}}_i^{(j)} = Q\boldsymbol{V}_i^{(j)} \odot Q^\top \boldsymbol{V}_i^{((j+1)\%L)}, \quad \text{for all } 1 \le j \le L/2.$$

where $Q = (I_n \otimes \mathbf{1}_n)(\mathbf{1}_n \otimes I_n)^\top$, $\mathbf{1}_n \in \mathbb{R}^{n \times 1}$ is the all-ones vector, and $\odot$ is the Hadamard product.

In order to represent the operation, we will first use a self-attention layer to compute $(\mathbf{1}_n \otimes I_n)^\top \boldsymbol{V}_i^{(j)}$ and $(I_n \otimes \mathbf{1}_n)\boldsymbol{V}_i^{(j+1\%L)}$ at each column $j$. Because the computation of $\tilde{\boldsymbol{V}}_i^{(j)}$ requires the model to look forward to $\boldsymbol{V}_i^{(j+1)}$, we need to shift the computation of $\tilde{\boldsymbol{V}}_i^{(j)}$ to position $j+1$, as a causal attention mask is involved in self-attention computation.

**After right shift operation,** we will represent the output of the self-attention computation as `<THINK>`, $[\mathbf{o}_2; \mathbf{0}], [\mathbf{o}_3; \mathbf{0}], \cdots, [\mathbf{o}_{L/2+1}; \mathbf{0}]$, and we will ignore the `<THINK>` embedding. Note that the second half of the output embeddings will still contain 0s and we will ignore them in the current computation. Then, the above computation can be rephrased as

$$\mathbf{o}_j = Q\mathbf{V}_i^{(j-1)} \odot Q^\top \mathbf{V}_i^{(j)}, \text{ for all } 2 \le j \le L/2.$$
$$\mathbf{o}_{L/2+1} = Q\mathbf{V}_i^{(L/2)} \odot Q^\top \mathbf{V}_i^{(1)}$$

**Self-attention layer:** The computation of $\mathbf{o}_j$, for $2 \le j \le L/2$, requires the computation of $(\mathbf{1}_n \otimes I_n)^\top \mathbf{V}_i^{(j-1)}$ and $(I_n \otimes \mathbf{1}_n) \mathbf{V}_i^{(j)}$. This will require 2 attention heads, one head that attends to itself, and another that attends to previous embedding at each position. We will outline both below. We will require one additional head, as computing $\mathbf{o}_{L/2+1}$ will require the model to compute $(I_n \otimes \mathbf{1}_n) \mathbf{V}_i^{(1)}$.

1. Attention Head 1 computes $(\mathbf{1}_n \otimes I_n)^\top \mathbf{V}_i^{(j-1)}$ at position $j$ for all $2 \le j \le L/2 + 1$. This can be done using a self-attention head (Definition H.1) that sets query and key matrices $\mathbf{W}_{query}, \mathbf{W}_{key}$, and biases $\{b_i\}_{t \le i \le 0}$ such that the attention score between embeddings at any two positions $p_1, p_2$ is given as follows:

$$a_{p_1, p_2} = 1, \text{ if } p_2 - p_1 = -1, \quad 0 \text{ otherwise}$$

$\mathbf{W}_{value}$ is set such that for any input $\boldsymbol{x}$, the output of $\mathbf{W}_{value}\boldsymbol{x}$ is given by

$$(\mathbf{W}_{value}\boldsymbol{x})_{p_1} = \sum_{j=0}^{n-1} x_{n \cdot j + p_1}.$$

In simple words, this operation simply adds up the values in dimensions $p_1, p_1 + n, p_1 + 2n, \cdots$ and stores them at position $p_1$ for all $1 \le p_1 \le n$.

2. Attention Head 2 computes $(I_n \otimes \mathbf{1}_n) \mathbf{V}_i^{(j)}$ at position $j$ for all $2 \le j \le L/2$. This can be done using a self-attention head (Definition H.1) that sets $\mathbf{W}_{query}, \mathbf{W}_{key}, \mathbf{W}_{value}$ and biases $\{b_i\}_{t \le i \le 0}$ such that the attention score between embeddings at any two positions $p_1, p_2$ is given as follows:

$$a_{p_1, p_2} = 1, \text{ if } p_2 - p_1 = 0, \quad 0 \text{ otherwise}$$

$\mathbf{W}_{value}$ is set such that for any input $\boldsymbol{x}$, the output of $\mathbf{W}_{value}\boldsymbol{x}$ is given by

$$(\mathbf{W}_{value}\boldsymbol{x})_{p_1+n} = \sum_{j=1}^{n} x_{j+p_1 n - n}.$$

In simple words, this operation simply adds up the values in dimensions $1 + (p_1 - 1)n, 2 + (p_1 - 1)n, \cdots$ and stores them at dimension $p_1 + n$ for all $1 \le p_1 \le n$.

3. Attention head 3 will compute $(I_n \otimes \mathbf{1}_n) \mathbf{V}_i^{(1)}$ and store in $\mathbf{o}_{L/2+1}$. This can be done by a self-attention layer which activates only between positions $p_1$ and $p_2$ that represent the start and the end tokens of the sequence, i.e. contain $\boldsymbol{b}_1$ and $\boldsymbol{b}_2$ as start and end indicator embeddings, and is 0 otherwise. $\mathbf{W}_{value}$ is set same as attention head 2.

The output of the three heads are simply added up. Hence, at each position $2 \le j \le L/2 + 1$, the output $\mathbf{o}_j$ has $(\mathbf{1}_n \otimes I_n)^\top \mathbf{V}_i^{(j-1)}$ in $[1, n]$ dimensions and $(I_n \otimes \mathbf{1}_n) \mathbf{V}_i^{(j\%L)}$ in $[n+1, 2n]$ dimensions.

**MLP layer:** The objective with the MLP layer (Definition H.2) will be to multiply $(\mathbf{1}_n \otimes I_n)^\top \mathbf{V}_i^{(j-1)}$ present in $[1, n]$ dimensions and $(I_n \otimes \mathbf{1}_n) \mathbf{V}_i^{(j\%L)}$ present in $[n+1, 2n]$ dimensions in each position $j$. This can be done by using an MLP layer with GELU activation by using Lemma H.3. The weights of the MLP layer are set as follows: $\mathbf{W}_{inner}$ is set such that

for all input $\boldsymbol{x}$, we have

$$(\boldsymbol{W}_{inner}\boldsymbol{x})_i = \frac{1}{N}x_i + \frac{1}{N}x_{n+i}, \quad \text{for all } 1 \le i \le n,$$

$$(\boldsymbol{W}_{inner}\boldsymbol{x})_{i+n} = \frac{1}{N}x_i, \quad \text{for all } 1 \le i \le n,$$

$$(\boldsymbol{W}_{inner}\boldsymbol{x})_{i+2n} = \frac{1}{N}x_{n+i}, \quad \text{for all } 1 \le i \le n.$$

$\boldsymbol{W}_{outer}$ is set such that for all $\boldsymbol{x} \in \mathbb{R}^{3n}$

$$(\boldsymbol{W}_{outer}\boldsymbol{x})_i = N^2(x_i - x_{i+n} - x_{i+2n}), \quad \text{for all } 1 \le i \le n.$$

All other coordinates in these matrices are set as 0s. $N$ is set as a large number (say 100). By Lemma H.3, the output of the MLP layer will be $\texttt{<THINK>}, \boldsymbol{o}_2, \cdots, \boldsymbol{o}_{L/2+1}$, with $\boldsymbol{o}_j$ containing

$$\tilde{\boldsymbol{V}}_i^{(j-1)} + \mathcal{O}(N^{-4}) := (\mathbf{1}_n \otimes I_n)^\top \boldsymbol{V}_i^{(j-1)} \odot (I_n \otimes \mathbf{1}_n)\boldsymbol{V}_i^{(j\%L)} + \mathcal{O}(N^{-4})$$

at each position $2 \le j \le L/2$.

| Embedding Name | Dimension size | First segment values
(In-context information) | Second segment values
(Input query sequence embeddings) |
|---|---|---|---|
| Token | $2n^2$ | $\{[\boldsymbol{e}_j; \boldsymbol{C}_1\boldsymbol{e}_j]\}_{j=1}^{n^2}, \cdots, \{[\boldsymbol{e}_j; \boldsymbol{C}_d\boldsymbol{e}_j]\}_{j=1}^{n^2}$ | $\texttt{<THINK>}, [\tilde{\boldsymbol{V}}_i^{(1)}; \mathbf{0}], \cdots, [\tilde{\boldsymbol{V}}_i^{(L/2)}; \mathbf{0}]$ |
| Context matrix index indicator | $d$ | $\{\boldsymbol{l}_1\}_{[1,n^2]}, \{\boldsymbol{l}_2\}_{[1,n^2]}, \cdots \{\boldsymbol{l}_d\}_{[1,n^2]}$ | $\mathbf{0}, \mathbf{0}, \cdots, \mathbf{0}$ |
| Start and End indicator | $2$ | $\mathbf{0}, \cdots, \mathbf{0}$ | $\mathbf{0}, \boldsymbol{b}_1, \mathbf{0}, \cdots, \mathbf{0}, \boldsymbol{b}_2$ |
| Segment | $2$ | $\boldsymbol{g}_1, \cdots, \boldsymbol{g}_1$ | $\mathbf{0}, \boldsymbol{g}_2, \cdots, \boldsymbol{g}_2$ |

*Table 10.* Output of CIRCULAR SHIFT-MODULE in MLT-MODULE that simulates *Circular shift*, i.e. computes $\texttt{Shift}(\boldsymbol{V}_i)$ at second segment token embeddings. $\texttt{<THINK>}$ represents a null output and won't be attended to in the future modules. We ignore this symbol for simplicity, when analyzing any module. $\boldsymbol{e}_j$ indicates a one-hot $n^2$ dimensional vector that contains 1 in dimension $j$.

## H.6. Step 2 (TRANSLATE-MODULE): *Translate* as a module containing a self-attention and an MLP layer

Our current token embeddings are given as $\texttt{<THINK>}, [\boldsymbol{o}_2; \mathbf{0}], [\boldsymbol{o}_3; \mathbf{0}], \cdots, [\boldsymbol{o}_{L/2+1}; \mathbf{0}]$, where each $\boldsymbol{o}_j$ contain $\tilde{\boldsymbol{V}}_i^{(j-1)}$. Other embeddings have been kept intact. The in-context information are given as $\{[\boldsymbol{e}_j; \boldsymbol{C}_1\boldsymbol{e}_j]\}_{j=1}^{n^2}, \cdots, \{[\boldsymbol{e}_j; \boldsymbol{C}_d\boldsymbol{e}_j]\}_{j=1}^{n^2}$. We will first use a self-attention layer to compute $[\boldsymbol{C}_i\tilde{\boldsymbol{V}}_i^{(j-1)}; \tilde{\boldsymbol{V}}_i^{(j-1)}]$ at position $2 \le j \le L/2 + 1$. We then use an MLP layer to represent $\texttt{HardMax}(\boldsymbol{C}_i + \boldsymbol{W}_i)\tilde{\boldsymbol{V}}_i^{(j-1)}$.

**Self-attention layer to represent $[\boldsymbol{C}_i\tilde{\boldsymbol{V}}_i^{(j-1)}; \tilde{\boldsymbol{V}}_i^{(j-1)}]$:** We will use two attention heads.

1. **The first attention head computes $\boldsymbol{C}_i\tilde{\boldsymbol{V}}_i^{(j-1)}$:** Matrices $\boldsymbol{W}_{query}, \boldsymbol{W}_{key}$ are set such that the attention score between an token embedding in first segment $[\boldsymbol{e}_r; \boldsymbol{C}_\ell\boldsymbol{e}_r]$ and a token embedding in second segment $\boldsymbol{o}_j$ is given by

   $$\langle \boldsymbol{e}_r, \tilde{\boldsymbol{V}}_i^{(j-1)} \rangle, \text{ if } \ell = i, \text{ and } 0 \text{ otherwise.}$$

   for any $j \in [2, L/2 + 1], r \in [1, d]$. The condition requires the model to attend to $\boldsymbol{C}_i$ and ignore other in-context information. The condition can be set using the Context matrix index indicator vectors $\boldsymbol{l}_\ell$ which is present in each in-context information embedding.

   The attention between any two input sequence embedding $\boldsymbol{o}_j$ and $\boldsymbol{o}_{j'}$ is computed as 0s. The distinction between the attention scores of pairs of embeddings in second segment, $\boldsymbol{o}_j$ and $\boldsymbol{o}_{j'}$, v/s attention scores between a token embedding in second segment and a token embedding in first segment, $\boldsymbol{o}_j$ and $[\boldsymbol{e}_r; \boldsymbol{C}_\ell\boldsymbol{e}_r]$, can be done by using the segment indicator embeddings $\boldsymbol{g}_1$ and $\boldsymbol{g}_2$ used to differentiate token embeddings in first segment and the input sequence embedding vectors.

   Matrix $\boldsymbol{W}_{value}$ is set such that the columns of each $\boldsymbol{C}_\ell$s are picked from the token embeddings in the first segment: $\{\{[\boldsymbol{e}_r; \boldsymbol{C}_\ell\boldsymbol{e}_r]\}_{r=1}^{n^2}\}_{\ell=1}^d$.

2. **The second attention head simply copies the input $\tilde{V}_i^{(j-1)}$:** This can be done with an attention head that attends to itself at each position $j$ and copies $\tilde{V}_i^{(j-1)}$ to output.

The output of the two attention heads are simply added up. The output embeddings will now look as follows: `<THINK>`, $\{[C_i\tilde{V}_i^{(j-1)}; \tilde{V}_i^{(j-1)}]\}_{j=1}^{L/2}$.

**MLP to represent `HardMax`$(C_i + W_i)\tilde{V}_i^{(j-1)}$:** Our current token embeddings at any position $j$ contain both $C_i\tilde{V}_i^{(j-1)}$ and $\tilde{V}_i^{(j-1)}$. The first layer of MLP can be used to compute $(C_i + W_i)\tilde{V}_i^{(j-1)}$ by setting the weights of the layer using $W_i$. We simulate `HardMax` operation as follows:

$$\tilde{\mathbf{o}}_j / \|\tilde{\mathbf{o}}_j\|_2, \text{ where } \tilde{\mathbf{o}}_j = \text{GELU}((C_i + W_i)\tilde{V}_i^{(j-1)})$$

The $\ell_2$ normalization is equivalent to RMSnorm operation (Zhang & Sennrich, 2019). This is an approximation of the `HardMax` function, which are equivalent only under the following conditions: for each column $j$

1. either $C_i^{(j)}$ or $W_i^{(j)}$ are all 0s.

2. $C_i^{(j)}$ and $W_i^{(j)}$ are both one-hot vectors and they match at the corresponding activated dimension.

---

**Algorithm 6** CIRCULAR SHIFT-MODULE: Self-attention and MLP layers for *Circular shift*

---

**Require:** Input embeddings (Token, Context matrix index indicator, Start and End Indicator, and Segment embeddings split into 2 segments) (Table 8). Important ones (for the current module) are

1. Token embeddings: First segment contains $\{[e_j; C_1 e_j]\}_{j=1}^{n^2}, \cdots, \{[e_j; C_d e_j]\}_{j=1}^{n^2}$ and the second segment contains $[\text{Shift}(V_{i-1}^{(1)}); \mathbf{0}], \cdots, [\text{Shift}(V_{i-1}^{(L/2)}); \mathbf{0}], \mathbf{0}$

2. Start and End indicator: First segment contains all 0s and second segments contains $b_1$ and $b_2$ at first and last embedding, while containing all 0s everywhere else.

**Step a:** Using a self-attention layer with 3 attention heads, change the token embeddings in second segment as `<THINK>`, $\{o_j\}_{j=2}^{L/2+1}$ s.t. $o_j$ has $(\mathbf{1}_n \otimes I_n)^\top V_i^{(j-1)}$ in $[1, n]$ dimensions and $(I_n \otimes \mathbf{1}_n) V_i^{(j\%L)}$ in $[n+1, 2n]$ dimensions. Primarily,

- Attention head 1: Computes attention score between any two positions $p_1, p_2$ as $a_{p_1,p_2} = 1$ iff $p_2 - p_1 = -1$ and 0 otherwise. Value matrix $W_{value}$ is set such that for any input $x$, the output of $W_{value} x$ is given by (for all $p_1 \in [1, n]$)

$$(W_{value} x)_{p_1} = \sum_{j=0}^{n-1} x_{n \cdot j + p_1}.$$

- Attention head 2: Computes attention score between any two positions $p_1, p_2$ as $a_{p_1,p_2} = 1$ iff $p_2 - p_1 = 0$ and 0 otherwise. $W_{value}$ is set such that for any input $x$, the output of $W_{value} x$ is given by (for all $p_1 \in [1, n]$)

$$(W_{value} x)_{p_1+n} = \sum_{j=1}^{n} x_{j + p_1 n - n}.$$

- Attention head 3: Computes attention score between any two positions $p_1, p_2$ as $a_{p_1,p_2} = 1$ iff $b_2$ and $b_1$ are present as at positions $p_1$ and $p_2$ respectively and 0 otherwise. $W_{value}$ is set such that for any input $x$, the output of $W_{value} x$ is given by (for all $p_1 \in [1, n]$)

$$(W_{value} x)_{p_1+n} = \sum_{j=1}^{n} x_{j + p_1 n - n}.$$

Sum the output of the three heads.

**Step b:** Use MLP layer to change the token embeddings in second segment as `<THINK>`, $\{o_j\}_{j=2}^{L/2+1}$ s.t. $o_j$ has $\tilde{V}_i^{(j-1)}$ with some small error. Primary computation at each position $j$ is given as (for a large $N$)

$$\sqrt{2/\pi} N^2 \left( \text{GELU}(\frac{1}{N} (\mathbf{1}_n \otimes I_n)^\top V_i^{(j-1)} + \frac{1}{N} (I_n \otimes \mathbf{1}_n) V_i^{(j\%L)}) \right.$$
$$\left. - \text{GELU}(\frac{1}{N} (\mathbf{1}_n \otimes I_n)^\top V_i^{(j-1)}) - \text{GELU}(\frac{1}{N} (I_n \otimes \mathbf{1}_n) V_i^{(j\%L)}) \right),$$

which will return

$$\tilde{V}_i^{(j-1)} + \mathcal{O}(N^{-4}) := (\mathbf{1}_n \otimes I_n)^\top V_i^{(j-1)} \odot (I_n \otimes \mathbf{1}_n) V_i^{(j\%L)} + \mathcal{O}(N^{-4})$$

Return the output embeddings (as given in Table 10).

---

---

**Algorithm 7** TRANSLATE-MODULE: Self-attention and MLP layers for *Translate*

---

**Require:** We will require an index, and input embeddings as input:

- Index $i$ (indicating index of the MLT-MODULE it is a part of),

- Embeddings (Token, Context matrix index indicator, Start and End Indicator, and Segment embeddings split into 2 segments) from the output of its preceding CIRCULAR SHIFT-MODULE (Table 10). Important ones (for the current module) are

    1. Token embeddings: First segment contains $\{[e_j; C_1 e_j]\}_{j=1}^{n^2}, \cdots, \{[e_j; C_d e_j]\}_{j=1}^{n^2}$ and the second segment contains $\texttt{<THINK>}, [\tilde{V}_i^{(1)}; \mathbf{0}], \cdots, [\tilde{V}_i^{(L/2)}; \mathbf{0}]$

    2. Context matrix index indicator: First segment contains $\{l_1\}_{[1,n^2]}, \{l_2\}_{[1,n^2]}, \cdots \{l_d\}_{[1,n^2]}$ and second segments contains all 0s vectors.

**Step a:** Using a self-attention layer, change token embeddings in the second segment as $\texttt{<THINK>}, \{[C_i \tilde{V}_i^{(j-1)}; \tilde{V}_i^{(j-1)}]\}_{j=1}^{L/2}$.

- Primarily, the self-attention score between embeddings that contain a second segment token embedding $[\tilde{V}_i^{(p_2)}; \mathbf{0}]$ and a first segment token embedding $[e_{p_1}; C_r e_{p_1}]$ (for any $r \in [1,d]$, $p_1 \in [1,n^2]$, $p_2 \in [1,L]$) is computed as

$$\langle e_{p_1}, \tilde{V}_i^{(p_2)} \rangle \cdot \langle l_r, l_i \rangle,$$

where $l_r$ is the corresponding context matrix index indicator for the first segment embedding under consideration and $l_i$ is constructed using the index $i$.

- $C_r e_{p_1}$ is used as value vector from each first segment embeddings.

**Step b:** Using an MLP layer, change token embeddings in the second segment to contain $\texttt{<THINK>}, \{o_j\}_{j=2}^{L/2+1}$, where

$$\mathbf{o}_j = \tilde{\mathbf{o}}_j / \|\tilde{\mathbf{o}}_j\|_2, \text{ where } \tilde{\mathbf{o}}_j = \text{GELU}((C_i + W_i)\tilde{V}_i^{(j-1)})$$

Return the output embeddings (as given in Table 9).

---

## H.7. Trainability of the constructed transformer

How does our constructed transformer perform on the **MLT** task? To do so, we hand-construct the designed transformer and train the transformer model on a random translation task in $\mathbf{MLT}(5, 10)$, that has depth 5 and number of characters 10 in each translation level. We train only parameters $\{\boldsymbol{W}_i\}_{i=1}^d$ using Layer-wise and Full Parameter optimization algorithms from Algorithms 4 and 5 respectively.

However, we make two changes. First, we use Adam optimizer instead of SGD, as we found SGD to get stuck frequently at bad minimas. Next, instead of masking in-context representation of layers in a rotating curriculum fashion, where each layer $i$'s in-context representation were masked in intervals $[j \cdot (i-1)n^2, j \cdot in^2]$ for all integers $j \geq 0$, we mix in the masking of all layers together by randomly picking a layer $i$ and mask a random column from $\boldsymbol{C}_i$. We found that mixing in masking of all layers helps train the model faster.

We vary the peak learning rate as $\{10^{-2}, 10^{-3}, 10^{-4}\}$. We use a cosine decay learning rate schedule with warmup steps 100 and minimum learning rate at end of training as $0.1\times$ the peak learning rate. We train with cross entropy loss, and use a total of 51200 training sequences of length 20 during the course of training. We use a batch size of 32 per gradient update step. We also use a small $\ell_1$ regularization, with strength $10^{-4}$, on the rows of the parameters to help optimization.

**Observations:** We report the performance of a model trained with both Layer-wise and Full Parameter optimization algorithm with peak learning rate $10^{-3}$ and $\ell_1$ regularization with strength $10^{-4}$ in Figure 15. We observe that under both algorithms, the trainable parameters quickly learn the relevant matrices that represent the true set of phrasebooks to minimize the training loss.

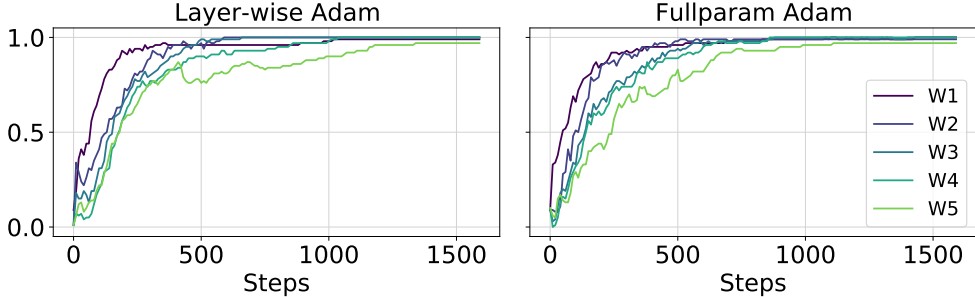

*Figure 15.* We perform Layer-wise and Full Parameter Training with Gradient Descent (defined in Appendix G.5.1) on the handconstructed transformer (Lemma H.4) that represents SURR-MLT (Definition G.7) designed for $\mathbf{MLT}_{\Pi^*}$ in $\mathbf{MLT}(5, 10)$ . Here, we report the portion of columns from the trainable parameters, which after `HardMax` application $\{\texttt{HardMax}(\boldsymbol{W}_i)\}_{i=1}^d$, align with the corresponding stochastic matrices of the phrasebooks $\{\texttt{Matrix}(\pi_i^*)\}_{i=1}^d$. We observe that under both algorithms, the trainable parameters quickly learn the relevant stochastic matrices.

# I. Additional Experiment Setups

## I.1. Format of Training Text

Here we present examples of training data used for experiments in Section 3

```
INPUT:
"<|begin_of_text|> <|begin_of_text|> <|begin_of_text|> <|start_header_id|> user <|end_header_id|>

You are performing a special translation task called language_task_2.  The subset of dictionaries used are as
follows:
Dictionary used from language 1 to language 2:
D A -> N J; A D -> J I; B C -> O I; C C -> N N; H E -> P K; F E -> M L; G H -> L N; E G -> P J; A F -> K M;
E C -> K P; B E -> K I; F D -> M N; E D -> P O; B A -> P L; B D -> I P; D G -> I I; D H -> I O; E F -> L M;
H B -> J K; E A -> K J; A H -> L J; G C -> I K; F B -> P I; F G -> J O;
Dictionary used from language 2 to language 3:
L P -> V X; M K -> R W; L I -> X R; N O -> R R; M O -> U Q; I L -> W V; O K -> Q W; P M -> R U; J I -> V S;
I M -> X W; O I -> Q U; N P -> R Q; I N -> Q X; N L -> R S; I J -> Q T; N J -> S T; L O -> U V; P J -> T X;
J L -> Q R; P K -> W R; N M -> S U; M I -> Q S; P O -> S S; K K -> U U; P L -> X U;
Dictionary used from language 3 to language 4:
X U -> Y d; R W -> a c; T Q -> Y Y; U R -> Z Y; T X -> e d; W W -> e Y; U X -> f f; X R -> b Y; Q Q -> b c;
S S -> c c; V U -> Z d; R S -> f a; V W -> Y e; R U -> f c; U S -> c a; U W -> c f; W X -> d Y; R Q -> c Z;
V Q -> Z f; W S -> b d; V R -> a a; R X -> f Z; U Q -> d c; Q V -> d Z; S W -> Y b;
Dictionary used from language 4 to language 5:
c a -> m j; b Y -> n k; c f -> k m; c b -> l n; Z d -> g h; d c -> n g; b f -> i k; f Z -> i i; Z Y -> g l;
Y a -> l l; b e -> l i; Y Y -> m g; d Z -> g g; d e -> k l; f c -> j g; f f -> k n; b Z -> n h; e a -> i m;
c e -> j h; Y d -> i h; Y b -> h i; Y f -> h k; a Y -> k j; a a -> m k; c Y -> h m;
Dictionary used from language 5 to language 6:
k g -> p s; h h -> r q; j m -> o o; i h -> q s; l m -> p o; i j -> o q; l j -> q u; k m -> v u; g h -> p p;
g n -> v p; n h -> s s; m m -> s o; m k -> v t; m i -> u t; h k -> t o; n k -> r u; j n -> t u; g l -> t s;
j g -> v v; h g -> u s; n l -> s u; l k -> q o; h l -> t p; i i -> p q; k i -> r o; l h -> u p;
Now please perform language_task_2 translation from the following sequence in language 1 to language
6.  Do not use code!  You must only reponse in the form:  "Sequence in language 1:  [the sequence in
language 1]; Sequence in language 2:  [the sequence in language 2]; Sequence in language 3:  [the
sequence in language 3]; Sequence in language 4:  [the sequence in language 4]; Sequence in language
5:  [the sequence in language 5]; Sequence in language 6:  [the sequence in language 6]".  The sequence
you need to translate from language 1 is:  C B E F E B D E C B C A H E F B C A D F G B D G H E D
E.<|eot_id|><|start_header_id|>assistant<|end_header_id|>"

LABEL:
"<|begin_of_text|>The translation result is:  Sequence in language 1:  C B E F E B D E C B C A H E F B C A D
F G B D G H E D E; Sequence in language 2:  K I M L I P K P O I L J L M O I J I J O I P L N P O K P * * * *
* * * * * * * *; Sequence in language 3:  X W X R W R S S W V Q R U Q Q T Q T Q U X U R Q Q W W R * * * * * *
* * * * * *; Sequence in language 4:  d Y a c f a Y b Z f f c b c Y Y Y Y f f Z Y b c e Y f Z * * * * * * * *
* * * *; Sequence in language 5:  l l k m k j n h k n l n h m m g h k i i h i j h h k g h * * * * * * * * * *
* *; Sequence in language 6:  q o v t t u t o s u s s s o p p r o q s o q r q p s t p;<|eot_id|>" ,
```

*Figure 16.* Input with complete context and explicit chain-of-thought. We use this data format at the begining of stage 1 training.

```
INPUT:

"<|begin_of_text|> <|begin_of_text|> <|begin_of_text|> <|start_header_id|> user <|end_header_id|>

You are performing a special translation task called language_task_2.  The subset of dictionaries used are as
follows:
Dictionary used from language 1 to language 2:
D A -> N J; A D -> J I; B C -> O I; C C -> N N; H E -> P K; F E -> M L; G H -> L N; E G -> P J; A F -> K M;
E C -> K P; B E -> K I; F D -> M N; E D -> P O; B A -> P L; B D -> I P; D G -> I I; D H -> I O; E F -> L M;
H B -> J K; E A -> K J; A H -> L J; G C -> I K; F B -> P I; F G -> J O;
Dictionary used from language 2 to language 3:
L P -> V X; M K -> R W; L I -> X R; N O -> R R; M O -> U Q; I L -> W V; O K -> Q W; P M -> R U; J I -> V S;
I M -> X W; O I -> Q U; N P -> R Q; I N -> Q X; N L -> R S; I J -> Q T; N J -> S T; L O -> U V; P J -> T X;
J L -> Q R; P K -> W R; N M -> S U; M I -> Q S; P O -> S S; K K -> U U; P L -> X U;
Dictionary used from language 3 to language 4:
X U -> Y d; R W -> a c; T Q -> Y Y; U R -> Z Y; T X -> e d; W W -> e Y; U X -> f f; X R -> b Y; Q Q -> b c;
S S -> c c; V U -> Z d; R S -> f a; V W -> Y e; R U -> f c; U S -> c a; U W -> c f; W X -> d Y; R Q -> c Z;
V Q -> Z f; W S -> b d; V R -> a a; R X -> f Z; U Q -> d c; Q V -> d Z; S W -> Y b;
Dictionary used from language 4 to language 5:
c a -> m j; b Y -> n k; c f -> k m; c b -> l n; Z d -> g h; d c -> n g; b f -> i k; f Z -> i i; Z Y -> g l;
Y a -> l l; b e -> l i; Y Y -> m g; d Z -> g g; d e -> k l; f c -> j g; f f -> k n; b Z -> n h; e a -> i m;
c e -> j h; Y d -> i h; Y b -> h i; Y f -> h k; a Y -> k j; a a -> m k; c Y -> h m;
Dictionary used from language 5 to language 6:
k g -> p s; h h -> r q; j m -> o o; i h -> q s; l m -> p o; i j -> o q; l j -> q u; k m -> v u; g h -> p p;
g n -> v p; n h -> s s; m m -> s o; m k -> v t; m i -> u t; h k -> t o; n k -> r u; j n -> t u; g l -> t s;
j g -> v v; h g -> u s; n l -> s u; l k -> q o; h l -> t p; i i -> p q; k i -> r o; l h -> u p;
Now please perform language_task_2 translation from the following sequence in language 1 to language
6.  Do not use code!  You must only reponse in the form:  "Sequence in language 1:  [the sequence in
language 1]; Sequence in language 2:  [the sequence in language 2]; Sequence in language 3:  [the
sequence in language 3]; Sequence in language 4:  [the sequence in language 4]; Sequence in language
5:  [the sequence in language 5]; Sequence in language 6:  [the sequence in language 6]".  The sequence
you need to translate from language 1 is:  C B E F E B D E C B C A H E F B C A D F G B D G H E D
E.<|eot_id|><|start_header_id|>assistant<|end_header_id|>"

LABEL:

"<|begin_of_text|>The translation result is:  Sequence in language 1:  C B E F E B D E C B C A H E F B C A D
F G B D G H E D E; Sequence in language 2:  <T> <T> <T> <T> <T> <T> <T> <T> <T> <T> <T> <T> <T> <T> <T> <T>
<T> <T> <T> <T> <T> <T> <T> <T> <T> <T> <T> <T> <T> <T> <T> <T> <T> <T> <T> <T> <T> <T> <T> <T> <T> <T>; Sequence in
language 3:  <T> <T> <T> <T> <T> <T> <T> <T> <T> <T> <T> <T> <T> <T> <T> <T> <T> <T> <T> <T> <T> <T> <T>
<T> <T> <T> <T> <T> <T> <T> <T> <T> <T> <T> <T> <T>; Sequence in language 4:  <T> <T> <T> <T>
<T> <T> <T> <T> <T> <T> <T> <T> <T> <T> <T> <T> <T> <T> <T> <T> <T> <T> <T> <T> <T> <T> <T> <T>
<T> <T> <T> <T> <T> <T> <T>; Sequence in language 5:  <T> <T> <T> <T> <T> <T> <T> <T> <T> <T> <T>
<T> <T> <T> <T> <T> <T> <T> <T> <T> <T> <T> <T> <T> <T> <T> <T> <T> <T> <T> <T> <T> <T> <T> <T> <T>
<T> <T>; Sequence in language 6:  q o v t t u t o s u s s s o p p r o q s o q r q p s t p;<|eot_id|>" ,
```

*Figure 17.* Input with complete context and internalized chain-of-thought. We use this data format by the end of stage 1 training as well as the base format for stage 2 training (before dropout)

```
INPUT:

"<|begin_of_text|> <|begin_of_text|> <|begin_of_text|> <|start_header_id|> user <|end_header_id|>

You are performing a special translation task called language_task_2.  The subset of dictionaries used are as
follows:
Dictionary used from language 1 to language 2:  ;
Dictionary used from language 2 to language 3:  ;
Dictionary used from language 3 to language 4:  ;
Dictionary used from language 4 to language 5:  ;
Dictionary used from language 5 to language 6:;
Now please perform language_task_2 translation from the following sequence in language 1 to language
6.  Do not use code!  You must only reponse in the form:  "Sequence in language 1:  [the sequence in
language 1]; Sequence in language 2:  [the sequence in language 2]; Sequence in language 3:  [the
sequence in language 3]; Sequence in language 4:  [the sequence in language 4]; Sequence in language
5:  [the sequence in language 5]; Sequence in language 6:  [the sequence in language 6]".  The sequence
you need to translate from language 1 is:  C B E F E B D E C B C A H E F B C A D F G B D G H E D
E.<|eot_id|><|start_header_id|>assistant<|end_header_id|>"

LABEL:

"<|begin_of_text|>The translation result is:  Sequence in language 1:  C B E F E B D E C B C A H E F B C A D
F G B D G H E D E; Sequence in language 2:  <T> <T> <T> <T> <T> <T> <T> <T> <T> <T> <T> <T> <T> <T> <T> <T>
<T> <T> <T> <T> <T> <T> <T> <T> <T> <T> <T> <T> <T> <T> <T> <T> <T> <T> <T> <T> <T> <T> <T> <T> <T>; Sequence in
language 3:  <T> <T> <T> <T> <T> <T> <T> <T> <T> <T> <T> <T> <T> <T> <T> <T> <T> <T> <T> <T> <T> <T> <T>
<T> <T> <T> <T> <T> <T> <T> <T> <T> <T> <T> <T> <T>; Sequence in language 4:  <T> <T> <T> <T>
<T> <T> <T> <T> <T> <T> <T> <T> <T> <T> <T> <T> <T> <T> <T> <T> <T> <T> <T> <T> <T> <T> <T>
<T> <T> <T> <T> <T> <T> <T> <T>; Sequence in language 5:  <T> <T> <T> <T> <T> <T> <T> <T> <T> <T> <T>
<T> <T> <T> <T> <T> <T> <T> <T> <T> <T> <T> <T> <T> <T> <T> <T> <T> <T> <T> <T> <T> <T> <T> <T> <T>
<T> <T>; Sequence in language 6:  q o v t t u t o s u s s s o p p r o q s o q r q p s t p;<|eot_id|>" ,
```

*Figure 18.* Input with completely masked context and internalized chain-of-thought. We use this data format to evaluate the model's capability on conducting translation without anything (useful information) in context.

