# OpenReview forum: "On the Power of Context-Enhanced Learning in LLMs"
_ICML.cc/2025/Conference — ICML 2025 spotlightposter_

### Official Review · Reviewer_PGoK · 2025-03-14

**Overall Recommendation:** 4

**Summary:**

The paper introduces *context-enhanced learning* as a training setup where an LLM is fine-tuned on a task with extra context provided, but no gradient is directly taken on that context.

The main claims of the paper are:

1. Context-enhanced learning improves training sample efficiency on reasoning tasks. This is presumably because the phrasebook rules are internalized atomically (*i.e.* the model is learning to apply intermediate steps, as opposed to memorizing end-to-end shortcuts).
    1. Context-enhanced learning only works with models that are ICL capable.
    2. Context-enhanced learning only works if you dropout parts of the context during training.
2. Phrasebooks are processed sequentially layer-by-layer.
3. It is challenging to prompt the model to regurgitate the rules of the intermediate phrasebooks without token filtering.

**Claims And Evidence:**

Overall, the claims made in this paper are well-supported by empirical evidence on **synthetic tasks**. However, in my view, the main limitation of this work is that there are no experiments on real data and even the synthetics are limited to a single synthetic task.

1. This claim is supported by the results in Figure 2 as well as the theoretical results in Section 5. It would be nice to see a demonstration of this phenomenon on real data or other synthetic tasks.
2. I found the mechanistic analysis in Figures 3 and 4 to be quite compelling for this claim — and the presentation was clear.
3. There seems to be some evidence on synthetic data supporting this claim in Section 4. However, I’m concerned that this result may not transfer when moving to real language — when dealing with real language, couldn’t there be clever prompting strategies that an adversary could use to elicit the rules? Without any results real data, the claims about the privacy preserving capabilities of context-enhanced learning feels quite speculative.

**Essential References Not Discussed:**

There are a number of prior works that study sequential reasoning tasks with synthetics and theory that could be important to cite given the mechanistic analysis of sequential reasoning. Consider mentioning them in related work.
https://arxiv.org/abs/2310.07923

https://openreview.net/forum?id=NikbrdtYvG#discussion

https://openreview.net/pdf?id=2N3CtUdoB0

**Experimental Designs Or Analyses:**

Yes see discussion of claims above.

**Methods And Evaluation Criteria:**

Overall, the authors make clever use of synthetic datasets and mechanistic analyses to illustrate their point.

As I discuss above, the paper would be greatly strengthened if it included experiments on real language data.

**Other Comments Or Suggestions:**

**Clarification and typos.**

- In Figure 2, I can’t see the pink or brown lines. Presumably they are underneath the random guess line? Is there someway this can be made more clear.

**Other Strengths And Weaknesses:**

The paper’s presentation quality is excellent.

Perhaps the most significant limitation is how well these findings translate beyond the neat confines of synthetic experiments. Real-world tasks rarely come with a perfectly curated “phrasebook” of rules that can be placed in the context. **One concern is the availability and quality of context**: The method assumes we have additional helpful text for each training example. In practice, obtaining such aligned context might require extra annotation or the output of a stronger model.

**Questions For Authors:**

None

**Relation To Broader Scientific Literature:**

The paper studies an important open problem in the literature: how do we get models to perform sequential chain of thought without explicitly supervising the model to do so. Context-enhanced learning is a creative alternative to explicit reasoning supervision that I have not seen explored in recent literature.

**Theoretical Claims:**

I did not carefully check the correctness of the proofs, but read the theoretical claims closely. The theoretical tools the authors use make sense and their theorems align nicely with their empirical claims.

---

> ### Author Rebuttal · Authors · 2025-04-01
>
> Thank you so much for your constructive comments and questions!
>
> Our responses to your questions and concerns are as follows:
>
> ---
>
> ### Applying Context-Enhanced Learning to Real World Data
>
> Context-enhanced learning (CEL) has been shown to outperform standard training or fine-tuning approaches that do not leverage in-context information across a range of real-world tasks, both during pre-training (e.g., Allen-Zhu et al., 2024; Gao et al., 2025) and fine-tuning (e.g., Liao et al., 2024; Zou et al., 2024) (see lines 036–044).
>
> Following your suggestion, we conducted fine-tuning experiments on the GSM8K dataset using two base models—Qwen-7B and LLaMA 3.1 8B—adapting the context-enhanced learning approach from prior fine-tuning works. Specifically, we included helpful in-context examples during training to enable CEL.
> We experimented with three strategies for selecting in-context learning (ICL) examples:
>
> * **Fixed ICL**: A fixed set of in-context examples is used for all training instances. We use the ones used by Qwen-2.5 models for their ICL evaluations.
>
> * **Random ICL**: Five in-context examples are randomly sampled for each training example individually.
>
> * **Skill-based ICL**: In-context examples are selected based on skill similarity, following the strategy proposed in [1].
>
> In all three cases, we find that CEL outperforms baseline fine-tuning, which only uses question–answer pairs without additional context. Evaluation is performed zero-shot on the trained models. Our fine-tuning approach for CEL follows the Annealing Dropout strategy.
>
> |Pre-trained Model| Baseline (SFT) | Fixed-ICL CEL | Random-ICL CEL| Skill-based ICL CEL|
> |:--:|:--:|:--:|:--:|:--:|
> |Qwen-7B base|85.5|87.4|87.9|88.4|
> |Llama-3.1 8B base|76.5|77.4|76.5|77.2|
>
> ---
>
> ### Difficulty of Curating In-Context Curriculum
>
> >"The method assumes we have additional helpful text for each training example. In practice, obtaining such aligned context might require extra annotation or the output of a stronger model."
>
> Thank you for raising this important point. Our primary motivation was to theoretically demonstrate that models can learn more efficiently when provided with additional supervision. In practice, such supervision could come from various sources—either via outputs from a stronger model or through retrieval of relevant information from a pre-training corpus.
>
> Prior work has offered promising examples of how such context can accelerate learning. For instance, Gao et al. (2025) and Zhu et al. (2024) show that including metadata—such as source information associated with documents—can lead to faster training and improved performance at test time.
>
> While our current work focuses on analyzing these benefits in a controlled setting, we believe that a key avenue for future research is in developing better methods to obtain and integrate high-quality supervision context for training language models more effectively.
>
> ---
>
> ### Alternative Ways for Eliciting In-Context Rules
> >"When dealing with real language, couldn’t there be clever prompting strategies that an adversary could use to elicit the rules? Without any results real data, the claims about the privacy preserving capabilities of context-enhanced learning feels quite speculative."
>
> We admit that our current evaluation is restricted to verbatim memorization of the phrasebook rules and does not consider adversarial robustness. While our results demonstrate that recovery of phrasebook information is limited in our synthetic setting, we acknowledge that this does not constitute definitive evidence that the information is completely hidden. Additionally, we do not yet provide observations on real-world data. (For the new GSM8K experiments, the context is selected from the same training set, making it difficult to distinguish memorization).
>
> That said, we view this work as the first step toward thinking about privacy preservation in context-enhanced learning settings. Memorization on real world data and in adversarial prompting settings are interesting directions to continue exploring on.
>
> ---
>
> ### Other suggestions on related work and presentation
> Thank you for bringing up the related works and the legibility issue of figure 2 (ablation data points are all overlapping around "near random"). We will surely discuss the mentioned related works and address the figure more clearly in future versions of the manuscript.
>
> ---
>
> #### References
> [1] Didolkar, Aniket, et al. "Metacognitive capabilities of llms: An exploration in mathematical problem solving." Advances in Neural Information Processing Systems 37 (2024): 19783-19812.

---

### Official Review · Reviewer_vWg8 · 2025-03-18

**Overall Recommendation:** 3

**Summary:**

This paper introduces a new concept called context-enhanced learning for LLMs, where they add context related to the task and training time step t in addition to the training data of x and y, and no loss is applied to the context text. To study how this will help LLM learning, they introduced a task called multi-level translation, which is a bijection task that maps input to output according to phrasebooks. Using this task, they prove in a simplified setting that context-enhanced learning can be exponentially more sample-efficient than standard learning when the model is capable of in-context learning. They also demonstrate mechanistically that the benefit arises from more accurate gradient learning signals and show that it's difficult to detect or recover learning materials used in the context during training.

**Claims And Evidence:**

I have concerns about the experiment in section 3.3 on sequential processing:
When replacing STR(pi i) with STR(pi hat_i), the experiment doesn't control for the positional effects of the replacement. Transformers process information based on both content and position, so the observed changes in representations could simply reflect the model's sensitivity to any change at that position, rather than specifically processing the phrasebook.
A proper control would include replacing non-phrasebook content at the same positions to test whether the observed pattern is specific to phrasebook processing or a general property of how transformers process sequential information. Without this, it's premature to conclude that "the first layer showing a significant difference is identified as the layer where the model begins processing the phrasebook."

**Essential References Not Discussed:**

no

**Experimental Designs Or Analyses:**

please see below

**Methods And Evaluation Criteria:**

please see below.

**Other Comments Or Suggestions:**

1. the "No Dropout (ablation)" curve is invisible from figure 2.
2. In the paper, they have showed "Context-enhanced learning from an ICL-capable model greatly improves training sample efficiency.", what about overall training efficiency in terms of training FLOPs? Since adding the context introduces additional computation in the attention calculations, although no loss is taken on the context.
3. What will happen if you take loss on the context, will that leads to even better sample efficiency?

**Other Strengths And Weaknesses:**

Pros:
1. The paper introduces an interesting concept and clean formalism along with an interesting synthetic task to test the idea.
2. The experiments suggest that context-enhanced learning significantly improves sample efficiency on the MLT task.
3. Extensive mechanistic experiments are conducted.
4. Strong theoretical analysis showing an exponential gap in sample complexity.
5. The paper offers privacy implications for data security and copyright through the difficulty of recovering context materials.

Cons:
1. The MLT task is artificial and synthetic with explicit rules. In real-world tasks like math or coding, there aren't many explicit rules that are universal and guarantee the mapping from a coding question to a block of code. It would greatly improve the paper if the authors had experiments on how to apply context-enhanced SFT in real-world tasks.
2. i have concern in sec 3.3 regarding proper control is needed to test whether the observed pattern is specific to phrasebook processing or a general property of how transformers process sequential information.

**Questions For Authors:**

1. In the line 165 "• Annealing Dropout: A better strategy: for s1, select the necessary rules from Π∗ plus 25% unused rules.", how are the necessary rules selected and what's the unused rules?
2. Have you considered applying this approach to more realistic tasks like math or coding problems?
3. In figure 2, i'm assuming no dropout is overlapping with no-context, why is this happening? Why is using all the useful context not improving sample efficiency. If "No Dropout" is not improving sample efficiency over "No Context," This might contradict the paper's central claim that context enhances learning.

**Relation To Broader Scientific Literature:**

The paper's concept of context-enhanced learning extends the Learning Using Privileged Information, and one of there method of annealing relates to curriculum learning.

**Theoretical Claims:**

no i didn't check fully.

---

> ### Author Rebuttal · Authors · 2025-04-01
>
> Thank you so much for your constructive comments (especially regarding section 3.3) and questions!
>
> Our responses to your questions and concerns are as follows:
>
> ---
>
> ### Definition of Necessary Rules and Unused Rules (line 165)
>
> For an input sequence $s_1$ of length 20-40 tokens, each translation step will involve 10-20 2-tuple translations specified by the phrasebook of that step. Out of the $n^2$ total rules of that phrasebook, we define the set of necessary rules for that input sequence to be the phrasebook entries involved and the set of unused rules as its complement.
>
> *We will also use the concept of “necessary rules” and “unused rules” in an additional experiment for introducing proper control on the mechanistic experiment (see below).
>
> ---
>
> ### Proper Control in Section 3.3
>
> To rule out the possibility that the affected depth is only related to the position of perturbation instead of semantic content, we introduce a new set of experiments ([see new figure](https://ibb.co/ym187Skp)). For each phrasebook $\pi_ i$, we selectively perturb tokens of 10 necessary rules (see the first row) or 10 unused rules (see the second row). Note that $STR(\pi_ i)$ is randomly shuffled, so these necessary rules and unused rules are interleaving in position.
>
> From the figure, perturbing necessary rules result in a qualitatively similar figure as in the current manuscript (with clear difference starting at certain layers), but perturbing unused rules around the same positions yields negligible representation difference across all layers. This suggests that the representation difference is reflecting the model’s processing of the **useful phrasebook information** with respect to the current translation task. We will replace fig.3 with the new figure in the next version of the manuscript.
>
> ---
>
> ### Real-World Tasks
>
> We have conducted additional experiments using context-enhanced learning to improve math capabilities. Please refer to [our response to reviewer PGoK](https://openreview.net/forum?id=Gn6L4QRKf7&noteId=qbF30B1xu4).
>
> ---
>
> ### FLOP Improvement
>
> Empirically, for $n=8, d=5$ (results reported in Figure 2), the SFT baseline with 1000k samples took around 14 hrs while the annealing dropout curriculum with 100k samples took around 2 hrs to complete on the same device.
>
> Theoretically, the full phrasebooks are of length $O(n^2d)$, so including them will lead to a slow down of $O(n^4d^2)$ in FLOPs (quadratic for attention). With large $d$, the $\exp(d)$ sample efficiency improvement will still dominate the polynomial slowdown per sample induced by in-context phrasebooks.
>
> ---
>
> ### Training with Loss on Context
>
> Following your suggestions, we reran the annealing dropout experiments with $n=8, d=5$ while explicitly computing the next-token prediction loss on the context. We report the test accuracy for various number of training samples and compare it with the context-enhanced learning case (in Figure 2).
>
> |# Samples | 10,000 | 25,000 | 50,000 | 100,000 | 250,000 |
> |---|:--:|:--:|:--:|:--:|:--:|
> | No Loss on Context | 12.7% | 20.0% | 100.0% | 100.0% | 100.0% |
> | Loss on Context (new) | 15.6% | 31.6% | 98.1% | 97.1% | 98.9% |
>
> In general there is no qualitative difference in sample efficiency when we train with loss on context.
>
> To understand what other difference does the additional loss incur, we conduct the following additional experiments:
>
> * We conducted the same verbatim memorization test as in section 4 on these models. The checkpoint trained with just 10k samples already attained a 99.8% recovery success rate, demonstrating very strong verbatim memorization of the phrasebooks despite not giving any additional benefits.
> * We [reproduced figure 4 for the model trained with loss on context](https://ibb.co/ccn6NsNk) trained with 250k samples and observe similar localized storage similar to the case of having no loss on context.
> * We also conducted the No ICL ablation with loss on context, resulting in the near random accuracy for all settings. Even if we are allowed to take loss on the context, the MLT-ICL-capability is still crucial for improved sample efficiency.
>
> The three observations above suggests that *verbatim memorizing the phrasebook* and *being able to translate without phrasebook in context* are likely to be detached. Computing loss on the context does not fundamentally change the mechanism of internalization.
>
> ---
>
> ### Bad Performance of No Dropout
>
> We would like to clarify that our central claim (both empirical and theoretical results) all require proper dropout in the helpful context.
>
> In the "No Dropout" scheme, the model would always have very low loss throughout training. This is because we start from an MLT(n,d)-ICL-capable model, which performs the task perfectly when conditioning on full context. Thus there would be little learning signal pushing the model to internalize the phrasebooks if we still always condition on full context.
> When no context is dropped, we would expect no internalization.

---

### Official Review · Reviewer_evJV · 2025-03-21

**Overall Recommendation:** 4

**Summary:**

This paper investigates the power of context-enhanced learning in large language models. The authors propose a synthetic machine translation task that utilizes phrasebooks to transform initial strings. Experimentally, the paper demonstrates that if the base model is MLT(d, n)-ICL-capable, context-enhanced learning enables more sample-efficient adaptation to MLT_{\Pi_}-capability. Furthermore, the study establishes that there is no information leakage regarding \Pi_* in the SFT model. Using a mechanistic interpretability approach, the authors analyze the trained model’s behavior to distinguish between an ICL-capable model and an MLT_{\Pi_*}-capable model. Theoretically, the paper shows that standard SFT requires an exponential number of samples to learn an MLT_{\Pi_*}-capable SURR-MLT, whereas context-enhanced learning only requires a polynomial number of samples.

**Claims And Evidence:**

The paper provides extensive experimental results, a mechanistic interpretability analysis, and theoretical justifications to support its claims. The presented evidence appears sufficient to substantiate the claims.

**Essential References Not Discussed:**

N/A

**Experimental Designs Or Analyses:**

The experimental results appear robust.

**Methods And Evaluation Criteria:**

The methodology and evaluation criteria are sound.

**Other Comments Or Suggestions:**

N/A

**Other Strengths And Weaknesses:**

The proposed framework of context-enhanced learning and the design of the synthetic MLT task is novel. The contribution is solid.

The paper was somewhat challenging to follow on the first read due to the dense mathematical formulations and technical descriptions of the task and methods. The second read was more manageable. Including a figure to illustrate the concept of context-enhanced learning, the MLT-ICL task, and the training/testing methodology for context-enhanced learning would greatly improve readability. However, I acknowledge that space limitations may make this difficult.

**Questions For Authors:**

1.	There appears to be an intriguing OOD generalization effect in the SFT and testing phases of the downstream MLT_{\Pi_*} task: the training prompt includes part of the phase codes, whereas the test prompt does not. What accounts for this OOD generalization? Standard empirical risk minimization cannot explain this behavior. Is the generalization driven by the base model’s capabilities (Llama 3.2-3B), or by its MLT(d, n)-ICL capability? In other words, if one were to train an MLT(d, n)-ICL-capable model from scratch instead of fine-tuning a pretrained Llama model, would it still be efficiently fine-tuned to become MLT_{\Pi_*}-capable?
2.	I did not fully understand how the “SQ dimension of MLT(d, n)” implies that “any algorithm attempting to learn an MLT_{\Pi_*}-capable SURR-MLT requires at least n^d sample complexity.” Could the authors provide further clarification on this reasoning?

**Relation To Broader Scientific Literature:**

The paper addresses training methods for LLMs, a topic that is closely related to the broader scientific literature.

**Theoretical Claims:**

I did not verify the proofs in detail, but they appear sound at first glance.

---

> ### Author Rebuttal · Authors · 2025-04-01
>
> Thank you so much for your comments and insightful questions!
>
> Our responses to your questions are as follows:
>
> ---
> ### What accounts for the OOD generalization to empty context?
>
> **Short Answer**: We believe this OOD generalization is fundamentally a form of compositional generalization, which is achieved by the compositional-friendly sequential translation structure in the MLT(d,n)-ICL-capable model. Following your suggestion, we have tried training an MLT(d,n)-ICL-capable model from scratch, which was proven to be difficult. The current OOD capability is likely a joint result from the base model and the ICL-capablilty training. We expect that as long as the MLT(d,n)-ICL-capable initialization has similar compositional-friendly mechanistic nature, we would see similar OOD behavior.
>
> **More details:**
>
> **Mechanism of the OOD Generalization**
>
> We believe two mechanistic properties of the reported model played an important role in this generalization phenomenon (inferred from section 3.3, with additional evidence in response to reviewer vWg8):
>
> 1. The sequential structure of translation in the model, in which each step can take information from context or the weights.
>
> 2. Phrasebook information at different translation stages are internalized in a localized manner, compensating for the corresponding dropped phrasebook information in the context.
>
> During training, when certain phrasebook entries are dropped from the context while others remain, the dropped information will be learned into the weights of certain transformer layers. When we perform sufficiently many random dropouts on the context, all phrasebook entries will be dropped and internalized at some point despite the context being never empty. Therefore the model will be robust to complete removal of the context since every phrasebook information it needs has been internalized.
>
> Our theoretical model (Surr-MLT, see Def 5.2) is built upon this rationalization. It can achieve full accuracy when conditioning on nothing at test time despite that only one rule is being dropped in each step in training.
>
> >“If one were to train an MLT(d, n)-ICL-capable model from scratch instead of fine-tuning a pretrained Llama model, would it still be efficiently fine-tuned to become $MLT_{\Pi^*}$-capable”
>
> **Response**: We tried training the same 3B model to be MLT(5, 8)-ICL-capable from scratch with various configurations. However even with a million random set of phrasebooks the model is still unable to learn any pattern. Thus unfortunately we cannot empirically answer this question. However if the model trained from scratch also shares similar mechanistic structure as discussed above, then similar sample efficiency may appear.
>
> >”Is the generalization driven by the base model’s capabilities (Llama 3.2-3B), or by its MLT(d, n)-ICL capability?”
>
> **Response**: As discussed above, it is hard to separate the effect from the base model and the MLT(d, n)-ICL capability since the base model affects how the specialized MLT(d, n)-ICL capability (i.e. the sequential translation structure) is formed in the model parameter. Whether the base model is strictly necessary remains unclear.
>
> ---
> ### SQ dimension and Sample Complexity of SGD
> **Short Answer**: We stated in the main text that “any algorithm attempting to learn an $MLT_{\Pi^*}$-capable SURR-MLT requires at least $n^d$ sample complexity.” This was an informal statement. A more precise version is: any algorithm that attempts to learn the task by querying distributional properties of $MLT_{\Pi^*}$​​ and receiving noisy estimates in return—i.e., algorithms operating within the statistical query (SQ) framework—requires at least $n^d$ sample complexity.
>
> **More details**: The formal SQ dimension (as defined in Section F.1 of the appendix) characterizes the computational difficulty for algorithms in the SQ model. In this framework, algorithms interact with an oracle by querying expectations over properties of the target task on data distribution and receive approximate (noisy) answers. A prototypical example is stochastic gradient descent (SGD), where the model estimates gradients based on average loss over a batch. The estimation noise depends on the batch size, along with potential adversarial or quantization noise. For problems with high SQ dimension (e.g., exponential in input size), it can be shown that the number of samples or gradient steps required by such algorithms becomes exponential.
>
> However, these lower bounds do not extend to algorithms that lie outside the SQ framework. For instance, if an algorithm has access to auxiliary information that directly hints at the target function (like phrasebooks in the MLT task), it can bypass the SQ limitations by exploiting this extra structure. In such cases, the learning process can be significantly more efficient due to this inductive bias.
>
> ---
> ### Presentation
> Thank you for your advice on the presentation! We will add illustrative figures in future versions of the manuscript.

---

### Official Review · Reviewer_6pqm · 2025-03-23

**Overall Recommendation:** 4

**Summary:**

The authors study the impact of augmenting context with additional data for learning in LLMs. Notably, on the added data no autoregressive gradients are computed. The authors consider a stylized problem of multi-layer text translation over finite alphabet where text from one language is translated to another in $d$ steps, where each step uses a phrasebook and is invertible.  The paper differentiates the learning difficulty between when all the phrasebooks are augmented in the context vs absence of such context enhancement. Notably, these phrasebooks are not provided at test time. Therefore, explicit CoT is avoided and the model operates in silent/internalized CoT mode adding it's own CoT before producing the results.

The authors tune the Llama 3.2B model with to expose the LLM to the specific translation task using SFT with random phrasebook. Next, they train the model using the context-enhanced learning technique with Dropout and Annealing to force the model to not memorize. The experiment show without dropout or annealing the test time performance suffers. This is augmented with interesting mechanistic observation of the learned model.

The authors next establish exponential sample complexity improvement with context-enhanced learning using a stylized learning model. They show without context enhancement the learning task has a $n^{\Omega(d)}$ with $d$-layer translation over $n$-sized alphabet (under statistical query model). However, with context enhancement a heuristics search algorithm exists with $O(d n^6 )$  sample complexity, and for $d=2$ a gradient based algorithm exists with sample complexity $O(n^4)$.

**Claims And Evidence:**

The authors claim that for multi-layer translation task context-enhanced learning boosts learning of LLMs, even when the context-enhancement is not present at test time. They prove their claims both experimentally, and on a surrogate model for learning.

**Essential References Not Discussed:**

I have basic, but not in-depth, knowledge of the literature, and I did not find any important references lacking. But I may have missed some references given I lack in-depth knowledge of the literature.

**Experimental Designs Or Analyses:**

The designed experiment is non-standard but seems suited for understanding the effectiveness of context enhancement for multi-layer translation. The loss evolution with various different annealing/dropout method shows the effectiveness of the context enhancement.

The mechanistic explanations also provide some structural insights into the learned models.

**Methods And Evaluation Criteria:**

This paper studies the utility of context enhancement in learning for LLM. The authors study this on a specific task, namely multi-layer translation. The methods seem logically sound. The mechanistic interpretation of the learning dynamics provide further evidence that context enhancement helps with multi-layer translation.

**Other Comments Or Suggestions:**

See Weakness part.

**Other Strengths And Weaknesses:**

Strengths
- The results, theoretical, mechanistic, and experimental, presented in the paper seems insightful in showcasing the effectiveness of context enhancement.
- The exponential improvement due to context enhancement in the surrogate learning model seems

Weakness/Questions
- The multi-layer translation task presented does not shed light into the generalization capability of the technique. It seems the model learns the rules of translation through context enhanced learning. But due to the assumed bijection of the translation rules (phrasebooks) the setup only shows quicker memorization (not generalization capabilities).  Does that capture the scope of the work?
- The authors consider the setting where context is not enhanced at test time. What is the motivation behind?
- In the surrogate model the phrasebooks are combined with the model weights, a more natural representation would be augmentation of these at the input space. This choice is somewhat unsatisfactory. The transformer model that mimics this surrogate learning also doe snot include the phrasebooks at the input place, so the claim that standard transformer model can replicate the learning seems misleading.

**Questions For Authors:**

N/A

**Relation To Broader Scientific Literature:**

This paper provides new insights into the utility of contexts in LLM learning, albeit in a very stylistic setup.

**Theoretical Claims:**

The authors claim that context enhancement provides exponential improvement in the learning of multi-layer translation in a surrogate learning model. They also show that the surrogate model can be a simulated using transformers.

---

> ### Author Rebuttal · Authors · 2025-04-01
>
> Thank you so much for your comments and insightful questions!
>
> Please find our responses to your questions / concerns below.
>
> ---
>
> ### Generalization in Context-Enhanced Learning
>
> >"The MLT task does not shed light into the generalization capability of the technique ... the setup only shows quicker memorization (not generalization capabilities)."
>
> **Response:** Our main message goes beyond quicker memorization of the phrasebooks. We also highlight that:
>
> 1. Even when **no next token prediction loss** is not computed on the phrasebook tokens, their simple presence in context can still accelerate learning.
>
> 2. The model does not need to verbatim memorize the phrasebooks (as shown in Section 4). This is beyond the usual notion of memorization in language model training.
>
> Within MLT there is also OOD generalization behavior: the model can perform the task with empty context at inference despite only 20% of the rules are dropped during training. This suggests that it has learned to generalize by composing rules at inference time in an OOD manner (see lines 248–249).
>
> Please also check our response to reviewer **evJV** on understanding this OOD phenomenon and response to reviewer **PGoK** demonstrating effectiveness of context-enhanced learning on real-world data.
>
> ---
>
> ### What is the Motivation of Having no Context at Test Time
>
> **Response:** Our motivation is twofold: philosophical and theoretical explanation of existing empirical works leveraging context-enhancement.
>
> First, we aim to examine whether models can internalize and benefit from additional information presented during training, even when that information is unavailable at test time. This setting mirrors the human paradigm of open-book learning followed by closed-book testing, and our framework seeks to assess whether models can exhibit similar learning behaviors.
>
> Second, we aim to offer a theoretical perspective on the empirical success of context-enhancement in LLM training. Prior works (e.g. Allen-Zhu et al. (2024) and Gao et al. (2025)) demonstrate that pre-training with auxiliary information improves both training efficiency and final performance — even when such information is absent during inference. Moreover PromptIntern (Zou et al. 2024) and SKIntern (Liao et al. 2024) suggest that context-enhancement can enable more efficient inference by reducing the prompt length. Our work provides a simple theoretical model to help explain these empirical findings and unveil the theoretical potential on sample efficiency.
>
> ---
>
> ### How Phrasebook Information are Provided to Surrogate Model
>
> >"In the surrogate model the phrasebooks are combined with the model weights, a more natural representation would be augmentation of these at the input space."
>
> **Response:** The surrogate model is motivated by our mechanistic experiments in section 3.
>
> First, the ICL-capable model utilizes the in-context phrasebooks in a sequential manner: phrasebooks of later steps are involved in the translation process in later layers (fig 3). During context-enhanced learning, the phrasebook information of a certain translation step is locally internalized to a small set of layers coupled to the layer processing the in-context phrasebooks for that step (fig 4). Thus we parameterize each translation step as a coupling of an in-context representation $C_i$ and an in-weight representation $W_i$, which is the minimal model capturing the mechanistic behavior of the model before and after the training process.
>
> >"The transformer model that mimics this surrogate learning also does not include the phrasebooks at the input place, so the claim that standard transformer model can replicate the learning seems misleading."
>
> **Response:** We would like to highlight that **the transformer model we constructed in theorem 5.3. is taking all context information from the input sequence.** (see the segment 1 of input in Algorithm 6 and 7, P70-71). Our construction of the transformer is based on an exact reparameterization of the surrogate model:$
> \def\R{\mathbb{R}}
> \def\bm#1{\begin{bmatrix} #1 \end{bmatrix}}
> \def\p#1{\left(#1\right)}
> \def\x{\times}
> $
>
> Let $e_i\in\R^d$ be the basis vector and let $0_{m\x n}$ be an $m\x n$ all-zero matrix. Let the context-augmented input be $X_1 = [C_1,\dots, C_d, V_1]\in\R^{n^2\x (dn^2+L)},$ then the reparameterized surrogate model is
>
> $\\begin{align*}
> X_{i+1} &= X_i\bm{I_{dn^2} & 0\\\\0 & 0_{L\x L}} + \p{\p{X_i\bm{e_i\otimes I_{n^2}\\\\0_{L\x n^2}} + W_i}\texttt{Shift}\p{X_i\bm{0_{n^2d\times L}\\\\I_L}}}\bm{0_{dn^2\times dn^2} & 0\\\\0 & I_{L}}\\\\
> &=[C_1,\dots, C_d, 0_{n^2\x L}] + [0,\dots, 0, \p{C_i + W_i}\texttt{Shift}\p{V_i}]\\\\
> &=[C_1,\dots, C_d, V_{i+1}].
> \\end{align*}$
>
> This model outputs $[C_1,\dots, C_d, V_{d+1}]$ with input $[C_1,\dots, C_d, V_1]$, where all phrasebooks are provided in-context. Please refer to algorithms 6 and 7 on how we modeled the computation above using self-attention, MLPs, and residual connections in a standard transformer.

---

> > ### Comment · Reviewer_6pqm · 2025-04-03
> >
> > I thank the authors for the clarifying response.
> >
> > The clarifications along OOD generalization capabilities are welcome. My concern was somewhat different, it was about generalizing to new unseen codebooks. Admittedly this is somewhat different from standard notion of distributional generalization in ML, and my phrasing could have been better.
> >
> > The explanation on Theorem 5.3  clears my doubts. The connection between surrogate models and mechanistic insights is interesting. More convincing argument that relates the mechanistic structure recovered to the $C_i$ structure used in Def G.4 is required.
> >
> > I will maintain my positive score.

---

> > > ### Author Response · Authors · 2025-04-08
> > >
> > > Thank you for your quick response. We will address your additional concerns as follows.
> > >
> > > ---
> > >
> > > ### OOD Generalization to Unseen Phrasebooks
> > >
> > > > My concern was somewhat different, it was about generalizing to new unseen codebooks
> > >
> > > **Response:** Our evaluation measures the performance of the model when not given information on the phrasebooks in-context. Because of our setup, an $MLT_{\Pi*}$-capable model will trivially fail on tasks involving unseen phrasebooks. On the other hand, our experiments with 20% dropout shows a non-trivial generalization ability with CEL. Rules in phrasebooks need not be dropped together, while the model can compose all the phrasebook rules during evaluation.
> > >
> > > There may be other evaluation strategies that can shed light on additional forms of out-of-distribution generalization, such as composition with entirely unseen phrasebooks, but we leave these directions for future work.
> > >
> > > ---
> > >
> > > ### Mechanistic Bases of Parameterizing $C_i$’s
> > >
> > > > More convincing argument that relates the mechanistic structure recovered to the
> > > $C_i$ structure used in Def G.4 is required.
> > >
> > > **Response:** Thank you for raising this point! In our theoretical analysis, we only use the fact that $C_i$’s are operators mapping 2-tuples in the current intermediate sequences to the following intermediate sequences in the representation space. In the Surr-MLT model we parameterized the sequence representations in one-hot encoding, so the operators are parameterized as permutation matrices as in Def G.4. While we do not expect the exact same parameterization to exist in the Llama models, **the identical mechanistic process (i.e. in-context representations of phrasebooks guiding the translation step for every 2-tuple) can be uncovered in the Llama models by analyzing the attention pattern from output tokens positions to in-context phrasebook token positions.**
> > >
> > > In particular, we consider the same MLT-ICL-capable model as in figure 3 and visualize the relevant attention patterns [in this new figure](https://ibb.co/pBFQ4jfv). For each translation step, we take the layers accountable to that step as discovered in section 3.3 and consider the attention pattern from the relevant output tokens positions of that step (segments of <THINK> abstract tokens or the final output sequence) to the relevant phrasebook tokens positions of that step in context. The visualization we present for each step is averaged across the attention heads in the selected layers.
> > >
> > > For all intermediate steps, we can see that the specific layers sparsely attend to the in-context token positions. Moreover, each 2-token segment of the output representation attends to the correct relevant phrasebook entry, effectively guiding the mapping from the representation of the current intermediate sequence to the next intermediate sequence. Please see the annotated attention pattern (what phrasebook entries are involved) and their exact correspondence to the next intermediate sequence as provided in the subfigure titles.
> > >
> > > We will add this discussion to the future versions of the paper to better support the parameterization of the surrogate model.

---

### Decision · Program_Chairs · 2025-05-01

**Decision:**

Accept (spotlight poster)

**Comment:**

This paper provides a systematic study of context-enhanced learning in LLMs. The paper considers a synthetic task of multi-level translation (MLT) when phrasebooks (mappings between tuples in two languages) are provided as additional context to language models. Even though the additional context is available during training, the goal is to improve the model performance on the MLT task without access to the additional context during test time. The paper empirically demonstrates that simply training with dropout (on additional context) can significantly speed up the model training compared to the SFT without additional context during training. The paper also explores the mechanistic understanding of how the model internalizes the additional context available during training. By studying a surrogate model, the paper establishes a clear separation between vanilla SFT and context-enhanced training in terms of their sample complexity.

The reviewers agreed that the paper studies an interesting problem while making significant contributions toward developing a systematic understanding of the problem. Some of the reviewers raised questions about OOD generalization which the authors successfully addressed. The reviewers also pointed out the lack of treatment of real-world tasks in the initial submission. The authors' response provided new experiments on the GSM8K dataset to showcase the value of additional context during training. The authors also addressed reviewers' questions about including loss on additional context tokens. The authors are encouraged to incorporate the additional discussion/results from the author-reviewer discussion phase in the final version of the paper.